# Optimal Threshold Labeling for Ordinal Regression Methods

**Ryoya Yamasaki**             *yamasaki@sys.i.kyoto-u.ac.jp*
*Department of Systems Science*
*Graduate School of Informatics, Kyoto University*
*36-1 Yoshida-Honmachi, Sakyo-ku, Kyoto 606-8501 JAPAN*

**Reviewed on OpenReview:** *https: // openreview. net/ forum? id= mHSAy1n65Z*

## Abstract

For an ordinal regression task, a classification task for ordinal data, one-dimensional transformation (1DT)-based methods are often employed since they are considered to capture the ordinal relation of ordinal data well. They learn a 1DT of the observation of the explanatory variables so that an observation with a larger class label tends to have a larger value of the 1DT, and classify the observation by labeling that learned 1DT. In this paper, we study the labeling procedure for 1DT-based methods, which have not been sufficiently discussed in existing studies. While regression-based methods and classical threshold methods conventionally use threshold labelings, which label a learned 1DT according to the rank of the interval to which the 1DT belongs among intervals on the real line separated by threshold parameters, we prove that likelihood-based labeling used in popular statistical 1DT-based methods is also a threshold labeling in typical usages. Moreover, we show that these threshold labelings can be sub-optimal ones depending on the learning result of the 1DT and the task under consideration. On the basis of these findings, we propose to apply empirical optimal threshold labeling, which is a threshold labeling that uses threshold parameters minimizing the empirical task risk for a learned 1DT, to those methods. In experiments with real-world datasets, changing the labeling procedure of existing 1DT-based methods to the proposed one improved the classification performance in many tried cases.

## 1 Introduction

***Ordinal regression (OR)*** (or called ordinal classification) is the classification of ***ordinal data*** in which the underlying target variable is categorical and labeled from a label set (ordinal scale) that is equipped with a natural ordinal relation for the explanatory variables; see Section 2.1 for a detailed formulation. The ordinal scale is typically formed as a graded (interval) summary of objective indicators like age groups {'0–9', '10–19', . . . , '90–99', '100–'} or graded evaluation of subjectivity like human rating {'excellent', 'good', 'average', 'bad', 'terrible'}, and ordinal data appear in various practical applications: age estimation (Niu et al., 2016; Cao et al., 2020), information retrieval (Liu, 2011), movie rating (Yu et al., 2006), and questionnaire survey in social research (Bürkner & Vuorre, 2019).

***One-dimensional transformation (1DT)-based methods*** are often applied to the OR problems as a simple way to capture the ordinal relation of ordinal data: they learn a 1DT of the observation of the explanatory variables so that an observation with a larger class label tends to have a larger value of the 1DT, and classify the observation by labeling that learned 1DT, as we will formalize in Section 2.2. For example, ***regression-based methods*** and ***classical threshold methods*** (or called threshold models) conventionally use a ***threshold labeling***, which labels a learned 1DT according to the rank of the interval to which the 1DT belongs among intervals on the real line separated by ***threshold parameters***. Regression-based methods (Kramer et al., 2001; Agarwal, 2008) learn a 1DT by solving a naïve regression task that infers a class label by the 1DT in a continuous scale, and often apply ***nearest-neighbor threshold (NNT) labeling*** that rounds the learned 1DT to its nearest label (see Section 3.1). Classical threshold methods (Shashua & Levin, 2003; Lin & Li, 2006; Chu & Keerthi, 2007; Lin & Li, 2012; Li & Lin, 2007; Pedregosa et al., 2017) learn

a 1DT using an objective function different from the empirical task risk, and commonly use **minimum threshold (MT)** and **summation threshold (ST) labelings** that apply parameters obtained incidentally in that learning procedure as threshold parameters (see Sections 3.2). Also, statistical 1DT-based methods (**statistical methods**) (McCullagh, 1980; Williams, 2006; Cao et al., 2020; Yamasaki, 2022), in which the learning procedure of the 1DT can be viewed as statistical modeling of conditional probabilities of the data, can apply **likelihood-based (LB) labeling** that is designed to minimize the task risk under the expectation that the assumed likelihood model is correctly specified to the data distribution (see Section 3.3).

We, however, have respective concerns on these existing labeling procedures. First, threshold parameters of NNT, MT, and ST labelings are generally not designed to minimize the task risk. So their threshold parameters can be sub-optimal for the minimization of the task risk, as we will demonstrate in Example 1. On the other hand, the LB labeling is designed to minimize the task risk if the assumed likelihood model is correctly specified to the distribution of the data (see Theorem 2). However, 1DT-based likelihood models have a strongly restricted representation ability and can be mis-specified to the data, and hence the LB labeling can degrade the classification performance depending on the data distribution.

Previous studies have done little theoretical work on the properties of these existing labelings. Therefore, we first study the relationship between these labelings. In particular, we show in Theorem 3 that not only the NNT, MT, and ST labelings but also the LB labeling in typical usages is a threshold labeling. This finding motivates us to search for a better labeling function among the class of threshold labelings. Then, in Section 4, we propose to apply **empirical optimal threshold (EOT) labeling**, which is a threshold labeling that uses threshold parameters minimizing the empirical task risk for the learned 1DT, to 1DT-based methods, under the expectation that the 1DT is learned successfully and the empirical (training) task risk becomes a good estimate of the (test) task risk. Here, the threshold parameters for the EOT labeling can be computed with a computational complexity of quasi-linear order $O(n \log n)$ regarding the training sample size $n$ using a dynamic programming-based algorithm (Algorithm 1) mentioned in Lin & Li (2006) (see Section 5 for the relation of our proposal to several previous studies including this reference).

In this study, we further performed numerical experiments of the OR task for real-world ordinal data to confirm the practical effectiveness of the EOT labeling (see Section 6 and Appendix D). The EOT labeling gave superior generalization performance (more exactly, smaller test task risk) than the NNT, MT, ST, and LB labelings, in many tried cases. Also, a modified 1DT-based method with the EOT labeling outperformed an existing 1DT-based method using the ST labeling that has been declared by Cao et al. (2020) to be state-of-the-art in 2020 for the age estimation from the facial image.

Therefore, in this paper, we propose to change labeling procedures of (existing) 1DT-based OR methods to the EOT labeling, on the ground of the fact (see Example 1 and Theorem 3) that the NNT, MT, and ST labelings and LB labeling in typical usages are possibly sub-optimal threshold labelings, and empirical effectiveness of the EOT labeling.

## 2 Preliminaries

### 2.1 Formulation of OR Problem

OR is the classification of ordinal data. The ordinal data have an underlying categorical **target variable** $Y \in [K] \coloneqq \{1, \ldots, K\}$ that is equipped with an ordinal relation naturally interpretable in the relationship with **explanatory variables** $\boldsymbol{X} \in \mathbb{R}^d$, where $d$ and $K$ are supposed to be integers larger than or equal to 1 and 3, respectively.[1] We here suppose that the target class labels are encoded to $1, \ldots, K$ in an order-preserving manner, like from 'excellent', ..., 'terrible' to $1, \ldots, 5$.

The task of the OR (**OR task**) is basically the same as that of the standard (including cost-sensitive) classification, to obtain a good **classifier** $f : \mathbb{R}^d \to [K]$. For a user-specified **task loss function** $\ell : [K]^2 \to [0, \infty)$, it is formulated as minimization of the **task risk** $\mathbb{E}[\ell(f(\boldsymbol{X}), Y)]$, where the expectation

---

[1] For better modeling of the ordinal data, it would be important to provide a mathematical characterization and further discussion of the natural ordinal relation. However, it would be related to learning procedure of the 1DT (defined in Section 2.2) more closely, and its necessity is not so great for the analysis and proposal of this study, so we will not mention it in this paper. Refer to, for example, OR studies (da Costa et al., 2008; Yamasaki, 2022) for the discussion on such characterizations.

value $\mathbb{E}[\cdot]$ is taken for all random variables in its argument (here $\boldsymbol{X}$ and $Y$). Popular task losses for OR tasks include not only the ***zero-one task loss*** $\ell_{\mathrm{zo}}(j,k) \coloneqq \mathbb{1}(j \neq k)$, where $\mathbb{1}(c)$ takes 1 if a condition $c$ is true and 0 otherwise, but also V-shaped losses (for cost-sensitive tasks) reflecting one's preference of smaller prediction errors over larger ones such as the ***absolute deviation task loss*** $\ell_{\mathrm{ad}}(j,k) \coloneqq |j-k|$, ***squared task loss*** $\ell_{\mathrm{sq}}(j,k) \coloneqq (j-k)^2$, and $\ell_{\mathrm{zo},c}(j,k) \coloneqq \mathbb{1}(|j-k| > c)$ with $c \geq 0$.

For the evaluation in the OR task, one may use criteria that cannot decomposed into a sum of losses for each sample point: for example, quadratic weighted kappa (Cohen, 1960; 1968). Our discussion in this paper does not cover such criteria.

## 2.2 Formulation of One-Dimensional Transformation (1DT)-Based Methods and Threshold Methods

In this paper, we discuss only 1DT-based OR methods. We here provide notations and terminologies common for all the 1DT-based OR methods.

Suppose that one has the sample $\mathcal{D}_n \coloneqq \{(\boldsymbol{x}_i, y_i)\}_{i=1}^n$, each of which is drawn independently from an identical distribution of $(\boldsymbol{X}, Y)$. First, 1DT-based methods learn a ***1DT*** $a : \mathbb{R}^d \to \mathbb{R}$ of an observation of the explanatory variables from a user-specified class $\mathcal{A} \subseteq \{a : \mathbb{R}^d \to \mathbb{R}\}$ (***1DT class***) (e.g., a neural network with a fixed network architecture and learnable weight and bias parameters) possibly together with other objects so that an observation with a larger class label tends to have a larger value of the 1DT; we call this procedure the ***learning procedure*** (of the 1DT). Next, 1DT-based methods build up a classifier as $f = h \circ \bar{a}$ with a learned 1DT $\bar{a}$ and a ***labeling function*** $h : \mathbb{R} \to [K]$; we call this procedure the ***labeling procedure***. Most existing 1DT-based methods can be seen as adopting one of the NNT, MT, ST, and LB labelings, depending on the properties of their learning procedure, as we review later in Section 3.

In particular, we call a labeling function $h$ that can be represented as

$$h(u) = h_{\mathrm{thr}}(u; \boldsymbol{t}) \coloneqq 1 + \sum_{k=1}^{K-1} \mathbb{1}(u \geq t_k) \tag{1}$$

with parameters $\boldsymbol{t} = (t_1, \ldots, t_{K-1}) \in \mathbb{R}^{K-1}$ as the ***threshold labeling***. Also, we call the parameters $\boldsymbol{t}$ as the ***threshold parameters***, and a 1DT-based method using a threshold labeling as the ***threshold method***, in this paper. Note that this formulation of the threshold method is a generalization of the one employed in most previous studies on conventional threshold methods that we will review latter in Section 3.2. The threshold labeling $h_{\mathrm{thr}}(u; \boldsymbol{t})$ has the following properties:

**Proposition 1.** *The threshold labeling $h_{\mathrm{thr}}(u; \boldsymbol{t})$ is non-decreasing and right-continuous in $u \in \mathbb{R}$ and invariant regarding the permutation of the threshold parameters $t_1, \ldots, t_{K-1}$. Conversely, an arbitrary non-decreasing right-continuous function $h : \mathbb{R} \to [K]$ can be represented by a threshold labeling $h_{\mathrm{thr}}(\cdot; \boldsymbol{t})$ with certain threshold parameters $\boldsymbol{t} \in \mathbb{R}^{K-1}$ (i.e., there exist $\boldsymbol{t} \in \mathbb{R}^{K-1}$ such that $h(u) = h_{\mathrm{thr}}(u; \boldsymbol{t})$ for any $u \in \mathbb{R}$) or their permutation. Also, if $t_1, \ldots, t_{K-1}$ take only $L$ different values, then $h_{\mathrm{thr}}(u; \boldsymbol{t})$ has $L$ change points $u = u_1, \ldots, u_L$ such that $h_{\mathrm{thr}}(u_l - \epsilon; \boldsymbol{t}) \neq h_{\mathrm{thr}}(u_l; \boldsymbol{t})$ with a sufficiently small $\epsilon > 0$ for $l = 1, \ldots, L$.*

The last result implies that the threshold labeling is the simplest as the labeling function in the sense that the resulting classifier has only $(K-1)$ decision boundaries for the learned 1DT at most.

Note that, in the learning procedure, since the ***empirical task risk*** $\frac{1}{n} \sum_{i=1}^n \ell(h(a(\boldsymbol{x}_i)), y_i)$ is discontinuous with respect to the 1DT $a$ and hence difficult to optimize numerically, many methods depend on another objective function. In this paper, we call that objective function the ***empirical surrogate risk***, its population version the ***surrogate risk***, and its data-dependent minimal component the ***surrogate loss (function)***.

## 2.3 Formulation of Policy of This Study

The goal of this study is to improve the labeling procedure of 1DT-based methods. Thus, regarding the learning procedure of the 1DT, this paper adopts those by existing studies, and we do not discuss the goodness of the learning procedure. Assuming that a learned 1DT $\bar{a}$ and task loss $\ell$ are given, we will discuss the goodness of the labeling function $h$ with the task risk $\mathbb{E}[\ell(h(\bar{a}(\boldsymbol{X})), Y)]$ or the empirical task risk $\frac{1}{n} \sum_{i=1}^n \ell(h(\bar{a}(\boldsymbol{x}_i)), y_i)$ as a criterion, since the aim of the OR task is to minimize the task risk.

## 3 Review and Analysis of Existing 1DT-Based Methods and Labeling Functions

### 3.1 Regression-based Method with Nearest-Neighbor Threshold (NNT) Labeling

Regression-based methods (Kramer et al., 2001; Agarwal, 2008) learn a 1DT $a$ by solving a naïve regression task that infers a class label by the 1DT: for a regression loss function $\phi : \mathbb{R} \times [K] \to [0, \infty)$,

$$\min_{a \in \mathcal{A}} \frac{1}{n} \sum_{i=1}^{n} \phi(a(\boldsymbol{x}_i), y_i). \tag{2}$$

For example, Kramer et al. (2001) used the **squared (SQ) loss** $\phi_{\mathrm{sq}}(u, y) \coloneqq (u - y)^2$, and Agarwal (2008) used the **absolute-deviation (AD) loss** $\phi_{\mathrm{ad}}(u, y) \coloneqq |u - y|$, as a regression loss function $\phi$.

As a labeling function $h$, regression-based methods often use the NNT labeling

$$h_{\mathrm{nnt}}(u) \coloneqq h_{\mathrm{thr}}(u; (1.5, 2.5, \ldots, K - 0.5)) \tag{3}$$

that is a threshold labeling $h_{\mathrm{thr}}(\cdot; \boldsymbol{t})$ with the threshold parameters $t_k = k + 0.5$, $k = 1, \ldots, K - 1$. The NNT labeling rounds the learned 1DT to its nearest label.

The regression-based methods using the above two surrogate loss functions and NNT labeling have the following optimality guarantee for an OR task with a respective specific task loss:

**Theorem 1** (Pedregosa et al. (2017, Theorems 9 and 11)). *Let $(\ell, \phi)$ be $(\ell_{\mathrm{ad}}, \phi_{\mathrm{ad}})$ or $(\ell_{\mathrm{sq}}, \phi_{\mathrm{sq}})$. Then, for the surrogate risk minimizer $\bar{a} \in \arg\min_{a:\mathbb{R}^d \to \mathbb{R}} \mathbb{E}[\phi(a(\boldsymbol{X}), Y)]$ based on the surrogate loss $\phi$, the classifier $f_{\mathrm{nnt}} = h_{\mathrm{nnt}} \circ \bar{a}$ minimizes the task risk $\mathbb{E}[\ell(f(\boldsymbol{X}), Y)]$ for the task loss $\ell$: $\mathbb{E}[\ell(f_{\mathrm{nnt}}(\boldsymbol{X}), Y)] = \min_{f:\mathbb{R}^d \to [K]} \mathbb{E}[\ell(f(\boldsymbol{X}), Y)]$.*

According to this theorem, the regression-based methods (Kramer et al., 2001; Agarwal, 2008) would be expectable to work well if the sample size $n$ and 1DT class $\mathcal{A}$ are sufficiently large. On the other hand, if the 1DT class $\mathcal{A}$ is strongly restricted, a classifier based on the NNT labeling may be sub-optimal for the OR task (because, generally, $\mathbb{E}[\ell(f_{\mathrm{nnt}}(\boldsymbol{X}), Y)] \neq \mathbb{E}[\ell(\bar{h}(\bar{a}(\boldsymbol{X})), Y)]$ for $f_{\mathrm{nnt}} = h_{\mathrm{nnt}} \circ \bar{a}$ with $\bar{a} \in \arg\min_{a \in \mathcal{A}} \mathbb{E}[\phi(a(\boldsymbol{X}), Y)]$ and $\bar{h} \in \arg\min_{h:\mathbb{R} \to [K]} \mathbb{E}[\ell(h(\bar{a}(\boldsymbol{X})), Y)]$).

### 3.2 Classical Threshold Method with Minimum and Summation Threshold (MT and ST) Labelings

Classical threshold methods have been studied actively in the machine learning and statistical literature (Shashua & Levin, 2003; Chu & Keerthi, 2007; Li & Lin, 2007; Lin & Li, 2012; Pedregosa et al., 2017; Cao et al., 2020). Many of these methods are formulated with a learning procedure that simultaneously learns $(K - 1)$ parameters in addition to the 1DT $a$:

$$\min_{a \in \mathcal{A}, \boldsymbol{b} \in \mathcal{B}} \frac{1}{n} \sum_{i=1}^{n} \phi(a(\boldsymbol{x}_i), \boldsymbol{b}, y_i), \tag{4}$$

where we call $\boldsymbol{b} = (b_1, \ldots, b_{K-1}) \in \mathbb{R}^{K-1}$ the **bias parameters**, $\mathcal{B} \subseteq \mathbb{R}^{K-1}$ is a user-specified class for the bias parameters (**BPs class**), and $\phi : \mathbb{R} \times \mathbb{R}^{K-1} \times [K] \to [0, \infty)$ is a surrogate loss function. As the labeling function $h$, several early works (Shashua & Levin, 2003; Chu & Keerthi, 2007) use the MT labeling

$$h_{\mathrm{mt}}(u; \bar{\boldsymbol{b}}) \coloneqq \min\{k \in [K] \mid u < \bar{b}_k\} \text{ with } \bar{b}_K \coloneqq \infty \tag{5}$$

with learned bias parameters $\bar{\boldsymbol{b}} = (\bar{b}_1, \ldots, \bar{b}_{K-1})$, and more recent works (Pedregosa et al., 2017; Cao et al., 2020) use the ST labeling

$$h_{\mathrm{st}}(u; \bar{\boldsymbol{b}}) \coloneqq h_{\mathrm{thr}}(u; \bar{\boldsymbol{b}}). \tag{6}$$

The MT and ST labelings are threshold labelings and have the following relationship:

**Proposition 2.** *Given $\bar{\boldsymbol{b}} \in \mathbb{R}^{K-1}$ together with $\bar{b}_0 := -\infty$, let $t_k$ be $\bar{b}_{i_k}$ with $i_k := \min\{j \in \{0, \ldots, k\} \mid \bar{b}_k \leq \bar{b}_j\}$ for each $k = 1, \ldots, K-1$. Then, one has that $h_{\mathrm{mt}}(u; \bar{\boldsymbol{b}}) = h_{\mathrm{thr}}(u; \boldsymbol{t})$ with $\boldsymbol{t} = (t_1, \ldots, t_{K-1})$. Also, $h_{\mathrm{mt}}(u; \bar{\boldsymbol{b}}) = h_{\mathrm{thr}}(u; \bar{\boldsymbol{b}})$ if $\bar{b}_1 \leq \cdots \leq \bar{b}_{K-1}$.*

We remark that the MT labeling $h_{\mathrm{mt}}(u; \bar{\boldsymbol{b}})$ and ST labeling $h_{\mathrm{st}}(u; \bar{\boldsymbol{b}})$ are different when the learned bias parameters $\bar{\boldsymbol{b}}$ are not ordered.

A surrogate loss function $\phi(u, \boldsymbol{b}, k)$ for these threshold methods is often built by replacing step functions $\mathbb{1}(l \geq \cdot)$ and $\mathbb{1}(l+1 \leq \cdot)$ in the expression of the task loss $\ell(\cdot, k) = \ell(k, k) + \sum_{l=1}^{k-1}\{\ell(l, k) - \ell(l+1, k)\}\mathbb{1}(l \geq \cdot) + \sum_{l=k}^{K-1}\{\ell(l+1, k) - \ell(l, k)\}\mathbb{1}(l+1 \leq \cdot)$ by convex surrogates widely-used in binary classification (Lin & Li, 2012; Pedregosa et al., 2017) such as the hinge, logistic, and exponential losses, so that $\phi(\cdot, \boldsymbol{b}, k)$ becomes a continuous convex upper bound of $\ell(h_{\mathrm{thr}}(\cdot; \boldsymbol{b}), k)$. For example, in the terminology of Pedregosa et al. (2017), the ***immediate threshold (IT) loss function***

$$\phi(a(\boldsymbol{x}), \boldsymbol{b}, y) = \begin{cases} \varphi(b_1 - a(\boldsymbol{x})), & \text{if } y = 1, \\ \varphi(a(\boldsymbol{x}) - b_{K-1}), & \text{if } y = K, \\ \varphi(a(\boldsymbol{x}) - b_{y-1}) + \varphi(b_y - a(\boldsymbol{x})), & \text{otherwise} \end{cases} \tag{7}$$

is an upper bound of zero-one task loss $\ell_{\mathrm{zo}}$, and the ***all threshold (AT) loss function***

$$\phi(a(\boldsymbol{x}), \boldsymbol{b}, y) = \begin{cases} \sum_{k=1}^{K-1} \varphi(b_k - a(\boldsymbol{x})), & \text{if } y = 1, \\ \sum_{k=1}^{K-1} \varphi(a(\boldsymbol{x}) - b_k), & \text{if } y = K, \\ \sum_{k=1}^{y-1} \varphi(a(\boldsymbol{x}) - b_k) + \sum_{k=y}^{K-1} \varphi(b_k - a(\boldsymbol{x})), & \text{otherwise} \end{cases} \tag{8}$$

is an upper bound of absolute deviation task loss $\ell_{\mathrm{ad}}$. For instance, SVOR-IMC (Chu & Keerthi, 2007) uses the IT loss with $\varphi(u) = \min\{0, 1-u\}$, fixed margin strategy (Shashua & Levin, 2003) and SVOR-EXC (Chu & Keerthi, 2007) use the AT loss with $\varphi(u) = \min\{0, 1-u\}$, ORBoost-LR (Lin & Li, 2006) uses the IT loss with $\varphi(u) = e^{-u}$, ORBoost-ALL (Lin & Li, 2006) uses the AT loss with $\varphi(u) = e^{-u}$, and CORAL (Cao et al., 2020) uses the AT loss with $\varphi(u) = \log(1 + e^{-u})$.

As the BPs class $\mathcal{B}$, the non-restricted (non-ordered) class $\mathbb{R}^{K-1}$ and ordered class $\{\boldsymbol{b} \in \mathbb{R}^{K-1} \mid b_1 \leq \cdots \leq b_{K-1}\}$ are often applied. As a simple implementation of the ordered class, Franses & Paap (2001) mentioned to parametrize the bias parameters $\boldsymbol{b}$ as

$$b_1 = b'_1, \text{ and } b_k = b_{k-1} + {b'_k}^2 \text{ for } k = 2, \ldots, K-1, \tag{9}$$

with other parameters $b'_1, \ldots, b'_{K-1} \in \mathbb{R}$. When the ordered BPs class is applied, the MT and ST labelings bring the same classification results. Also, for many AT loss functions, bias parameters $\bar{\boldsymbol{b}}$ of the surrogate risk minimizer $(\bar{a}, \bar{\boldsymbol{b}}) \in \arg\min_{a \in \mathcal{A}, \boldsymbol{b} \in \mathbb{R}^{K-1}} \mathbb{E}[\phi(a(\boldsymbol{X}), \boldsymbol{b}, Y)]$ are ensured to be ordered ($\bar{b}_1 \leq \cdots \leq \bar{b}_{K-1}$) (see Chu & Keerthi (2005, Lemma 1) and Li & Lin (2007, Theorem 2)), and hence the use of the ordered BPs class will be justified.

The use of the surrogate loss $\varphi$ and the MT or ST labeling $h$, which make $\phi(\cdot, \bar{\boldsymbol{b}}, y)$ (with ordered $\bar{\boldsymbol{b}}$) to be an upper bound of the task loss $\ell(h(\cdot), y)$, is almost like a convention and may facilitate generalization analysis (Li & Lin, 2007; Lin & Li, 2012), but the goodness of selecting the MT or ST labeling is not supported by theoretical discussion. We demonstrate in the following example that the MT or ST labeling may be a negative factor that degrades the classification performance of threshold methods:

**Example 1.** *We here consider the IT loss (7) with $\varphi(u) = \min\{0, 1-u\}$, which we denote $\phi_{\text{hinge-it}}(u, \boldsymbol{b}, k)$ and call Hinge-IT loss. The Hinge-IT loss is a continuous convex upper bound of the zero-one task loss with the MT labeling $\ell_{\mathrm{zo}}(h_{\mathrm{mt}}(\cdot; \boldsymbol{b}), k)$ (and ST labeling $\ell_{\mathrm{zo}}(h_{\mathrm{st}}(\cdot; \boldsymbol{b}), k)$) when $\boldsymbol{b}$ is ordered (i.e., $b_1 \leq \cdots \leq b_{K-1}$): $\phi_{\text{hinge-it}}(\cdot, \boldsymbol{b}, k)$ is a convex function and $\phi_{\text{hinge-it}}(\cdot, \boldsymbol{b}, k) \geq \ell_{\mathrm{zo}}(h_{\mathrm{mt}}(\cdot; \boldsymbol{b}), k)$. So one may think that the Hinge-IT loss and the task with the zero-one task loss (**Task-Z**) have friendly compatibility, from an analogy of a well-known result, classification calibration (Bartlett et al., 2006), in binary classification. However, the following demonstration shows that the MT labeling may be sub-optimal in minimizing the task risk as a labeling function for the combination of the Hinge-IT loss and Task-Z.*

*We consider a 4-class OR problem (let $K = 4$), and suppose that the data appear only on 4 different points $\boldsymbol{x}^{[1]}, \ldots, \boldsymbol{x}^{[4]}$ in $\mathbb{R}^d$ and follow the probability distribution, $\Pr(\boldsymbol{x}^{[i]}) = 0.25$ and $(\Pr(1|\boldsymbol{x}^{[i]}), \ldots, \Pr(K|\boldsymbol{x}^{[i]})) = (0.5, 0.4, 0, 0.1), (0.3, 0.5, 0, 0.2), (0.2, 0, 0.5, 0.3), (0.1, 0, 0.4, 0.5)$ for $i = 1, \ldots, 4$.[2]*

*It can be shown that the surrogate risk minimizer $(\bar{a}, \bar{\boldsymbol{b}}) \in \arg\min_{a:\mathbb{R}^d \to \mathbb{R}, \boldsymbol{b} \in \mathbb{R}^{K-1}} \mathbb{E}[\phi_{\text{hinge-it}}(a(\boldsymbol{X}), \boldsymbol{b}, Y)]$ satisfies $\bar{b}_1 = \bar{b}_2 = \bar{b}_3 = 0$, $\bar{a}(\boldsymbol{x}^{[1]}) = \bar{a}(\boldsymbol{x}^{[2]}) = -1$, and $\bar{a}(\boldsymbol{x}^{[3]}) = \bar{a}(\boldsymbol{x}^{[4]}) = 1$ (ignore the translation invariance) by several simple calculations. The MT labeling predicts a label of the data on $\boldsymbol{x}^{[i]}$ as $h_{\text{mt}}(\bar{a}(\boldsymbol{x}^{[i]}); \bar{\boldsymbol{b}}) = 1, 1, 4, 4$ for $i = 1, \ldots, 4$ ($\mathbb{E}[\ell_{\text{zo}}(h_{\text{mt}}(\bar{a}(\boldsymbol{X}); \bar{\boldsymbol{b}}), Y)] = 0.6$), despite that using different threshold parameters (say $\boldsymbol{t} = (-2, 0, 2)$) one can predict it as $h_{\text{mt}}(\bar{a}(\boldsymbol{x}^{[i]}); \boldsymbol{t}) = 2, 2, 3, 3$ for $i = 1, \ldots, 4$ and yield a smaller value of the task risk ($\mathbb{E}[\ell_{\text{zo}}(h_{\text{mt}}(\bar{a}(\boldsymbol{X}); \boldsymbol{t}), Y)] = 0.55$).* $\square$

### 3.3 Statistical Methods with Likelihood-Based (LB) Labeling

In the OR research in statistics, several methods have been developed according to the statistical modeling of the conditional probabilities of the data through a 1DT (McCullagh, 1980; Williams, 2006; Chu & Ghahramani, 2005; Yamasaki, 2022). For example, the cumulative link model (McCullagh, 1980), which is popular in the OR research, models the conditional probabilities $\Pr(y|\boldsymbol{x})$, $(\boldsymbol{x}, y) \in \mathbb{R}^d \times [K]$ by

$$P(y; \sigma, \tilde{a}(\boldsymbol{x}), \tilde{\boldsymbol{b}}) := \begin{cases} \sigma(\tilde{b}_y - \tilde{a}(\boldsymbol{x})), & \text{if } y = 1, \\ 1 - \sigma(\tilde{b}_{y-1} - \tilde{a}(\boldsymbol{x})), & \text{if } y = K, \\ \sigma(\tilde{b}_y - \tilde{a}(\boldsymbol{x})) - \sigma(\tilde{b}_{y-1} - \tilde{a}(\boldsymbol{x})), & \text{otherwise}, \end{cases} \tag{10}$$

where $\sigma : \mathbb{R} \to [0, 1]$ is a link function that is non-decreasing and satisfies $\sigma(-\infty) = 0$ and $\sigma(+\infty) = 1$ (i.e., $\sigma$ is a cumulative distribution (CPD) function), $\tilde{a} : \mathbb{R}^d \to \mathbb{R}$ is an assumed 1DT, and $\tilde{\boldsymbol{b}} = (\tilde{b}_1, \ldots, \tilde{b}_{K-1}) \in \mathbb{R}^{K-1}$ are assumed bias parameters that satisfy $\tilde{b}_1 \leq \cdots \leq \tilde{b}_{K-1}$. As a link function $\sigma$, ordinal logistic regression (OLR) (McCullagh, 1980) applies the CPD function of the logistic distribution (a.k.a. the sigmoid function) $\sigma_{\text{logistic}}(u) := 1/(1 + e^{-u})$, and cumulative probit model (Agresti, 2010, Section 5.2) and Gaussian process OR (GPOR) proposed by Chu & Ghahramani (2005) use the CPD function of the standard Gaussian distribution (a.k.a. the inverse function of the probit function) $\sigma_{\text{gauss}}(u) := \int_{-\infty}^{u} (2\pi)^{-1/2} e^{-v^2/2} \, dv$.

In the learning procedure of the 1DT, statistical methods apply surrogate loss functions associated with their statistical modeling. For the cumulative link model, a surrogate loss function $\phi$ and the BPs class $\mathcal{B}$ for the learning procedure same as (4) should satisfy

$$(\tilde{u}, \tilde{\boldsymbol{b}}) = \arg\min_{u \in \mathbb{R}, \boldsymbol{b} \in \mathcal{B}} \sum_{y=1}^{K} P(y; \sigma, \tilde{u}, \tilde{\boldsymbol{b}}) \phi(u, \boldsymbol{b}, y). \tag{11}$$

A popular instance of the surrogate loss function is the ***negative log likelihood (NLL) loss function*** $\phi_{\text{nll}}(u, \boldsymbol{b}, y; \sigma) := -\log\{P(y; \sigma, u, \boldsymbol{b})\}$, for which the learning procedure amounts to the maximum likelihood estimation for the model (10). Also, Cao et al. (2020) used the AT loss (8) with $\varphi(u) = -\log\{\sigma(u)\}$, which we call the ***all negative log cumulative likelihoods (ANLCL) loss function*** and denote $\phi_{\text{anlcl}}(u, \boldsymbol{b}, y; \sigma)$, for $\sigma = \sigma_{\text{logistic}}$. The learning procedure for this loss function can be characterized as the minimization of the sum of the NLLs of the models of cumulative conditional probabilities $\Pr(Y \leq k | \boldsymbol{X} = \boldsymbol{x})$ for binary classification problems, '$k$ or less' vs. 'more than $k$', $k = 1, \ldots, K - 1$.

The above interpretation on using the surrogate losses $\phi_{\text{nll}}$ and $\phi_{\text{anlcl}}$ under the statistical model (10) can be mathematically understood as follows:

**Theorem 2.** *Assume that the random variable $(\boldsymbol{X}, Y)$ underlying the data has conditional probabilities that can be represented as (10): $\Pr(y|\boldsymbol{x}) = P(y; \sigma, \tilde{a}(\boldsymbol{x}), \tilde{\boldsymbol{b}})$ for every $y \in [K]$ and any $\boldsymbol{x} \in \mathbb{R}^d$ in the support of the distribution of $\boldsymbol{X}$ with $\sigma$ that is non-decreasing and satisfies $\sigma(-\infty) = 0$ and $\sigma(+\infty) = 1$ (and $\sigma(-\cdot) = 1 - \sigma(\cdot)$ for $\phi = \phi_{\text{anlcl}}$) such as $\sigma_{\text{logistic}}$ and $\sigma_{\text{gauss}}$, $\tilde{a} : \mathbb{R}^d \to \mathbb{R}$, and $\tilde{\boldsymbol{b}} \in \mathbb{R}^{K-1}$ satisfying $\tilde{b}_1 \leq \cdots \leq \tilde{b}_{K-1}$. Let $\mathcal{A}$ be $\{g : \mathbb{R}^d \to \mathbb{R}\}$, and $(\phi, \mathcal{B})$ be $(\phi_{\text{nll}}, \{\boldsymbol{b} \in \mathbb{R}^{K-1} \mid b_1 \leq \cdots \leq b_{K-1}\})$, $(\phi_{\text{anlcl}}, \mathbb{R}^{K-1})$, or $(\phi_{\text{anlcl}}, \{\boldsymbol{b} \in \mathbb{R}^{K-1} \mid b_1 \leq \cdots \leq b_{K-1}\})$. Then, any surrogate risk minimizer $(\bar{a}, \bar{\boldsymbol{b}}) \in \arg\min_{a \in \mathcal{A}, \boldsymbol{b} \in \mathcal{B}} \mathbb{E}[\phi(a(\boldsymbol{X}), \boldsymbol{b}, Y)]$ satisfies $P(y; \sigma, \bar{a}(\boldsymbol{x}), \bar{\boldsymbol{b}}) = \Pr(y|\boldsymbol{x})$ for any $\boldsymbol{x} \in \mathbb{R}^d$ in the support of the distribution of $\boldsymbol{X}$.*

---

[2]We abbreviate the marginal probability $\Pr(\boldsymbol{X} = \boldsymbol{x})$ to $\Pr(\boldsymbol{x})$ and the conditional probability $\Pr(Y = y | \boldsymbol{X} = \boldsymbol{x})$ to $\Pr(y|\boldsymbol{x})$ (this abbreviation applies to an estimate $\hat{\Pr}$ as well).

For such methods, not only the threshold labelings but also the LB labeling that grounds on the assumed statistical model is a widely-used option for the labeling function. Considering Theorem 2 and the equality $\mathbb{E}[\ell(f(\boldsymbol{x}), Y)] = \sum_{y=1}^{K} \Pr(y|\boldsymbol{x})\ell(f(\boldsymbol{x}), y)$, and aiming to minimize the task risk, $\min_{f:\mathbb{R}^d \to [K]} \mathbb{E}[\ell(f(X), Y)]$, these methods can predict a label with the LB labeling

$$h_{\mathrm{lb}}(u; \sigma, \bar{\boldsymbol{b}}, \ell) := \min\left(\underset{k \in [K]}{\arg\min} \sum_{y=1}^{K} P(y; \sigma, u, \bar{\boldsymbol{b}})\ell(k, y)\right) \tag{12}$$

with learned bias parameters $\bar{\boldsymbol{b}}$, under the expectation that the assumed statistical model (10) correctly represents the actual statistical behavior of the data and it is learned successfully. Note that there can be a tie situation where objective functions with different $k$ of (12) take the same value, and $\arg\min_{k \in [K]}$ operation outputs multiple labels. The overlaid min operation in (12) avoids such a tie situation.

These methods tend to perform better when their assumed statistical model represents the actual statistical behavior of the data well. One can, however, find that the condition in Theorem 2 is very restrictive. Therefore, in many practical situations, their statistical model would deviate from the actual statistical behavior of the data, and then their 1DT model may not be learned appropriately, and the LB labeling $h_{\mathrm{lb}}(\cdot; \sigma, \bar{\boldsymbol{b}}, \ell)$ may be sub-optimal for the learned 1DT model $\bar{a}$.

One may still consider that the LB labeling is more flexible, in that it is generally not restricted within the class of non-decreasing threshold labelings, and superior to threshold labelings. However, we found that the LB labeling takes the form of the threshold labeling, for typical statistical models such as ones in OLR and GPOR (i.e., the link function $\sigma$ such as $\sigma_{\mathrm{logistic}}, \sigma_{\mathrm{gauss}}$) and for typical task losses such as $\ell = \ell_{\mathrm{zo}}, \ell_{\mathrm{zo},c}, \ell_{\mathrm{ad}}, \ell_{\mathrm{sq}}$.

**Theorem 3.** *Suppose that $\sigma$ is non-decreasing and satisfies $\sigma(-\infty) = 0$ and $\sigma(+\infty) = 1$ and that $\bar{b}_1 \leq \cdots \leq \bar{b}_{K-1}$. Then, the LB labeling $h_{\mathrm{lb}}(u; \sigma, \bar{\boldsymbol{b}}, \ell)$ is*

(i) *a certain threshold labeling $h_{\mathrm{thr}}(u; \boldsymbol{t})$ for some $\boldsymbol{t} \in \mathbb{R}^{K-1}$, if $\ell(k, l)$ at each fixed $k \in [K]$ is non-increasing in $l$ for $l \leq k$ and non-decreasing in $l$ for $l \geq k$, and $\ell_{k,l}(j) := \ell(k, j) - \ell(k, j+1) - \ell(l, j) + \ell(l, j+1)$ at each fixed different $k, l \in [K]$ is non-positive (resp. non-negative) for all $j \in [K-1]$ respectively when $k < l$ (resp. $k > l$), for example, $\ell = \ell_{\mathrm{ad}}, \ell_{\mathrm{sq}}$,*

(ii) *a certain threshold labeling $h_{\mathrm{thr}}(u; \boldsymbol{t})$ for some $\boldsymbol{t} \in \mathbb{R}^{K-1}$, if $\ell = \ell_{\mathrm{zo}}, \ell_{\mathrm{zo},c}$ with $c \in [0, \lfloor K/2 \rfloor)$, $\sigma$ is differentiable, $\sigma'(v) := \frac{d}{dv}\sigma(v)$ is even and non-increasing in $v$ if $v > 0$, and $\frac{\sigma'(v_1) - \sigma'(v_2)}{\sigma(v_1) - \sigma(v_2)}$ is non-increasing in $v_1$ with fixed $v_2$ (and in $v_2$ with fixed $v_1$) if $v_1 < v_2$, for example, $\sigma = \sigma_{\mathrm{logistic}}, \sigma_{\mathrm{gauss}}$, where $\lfloor v \rfloor$ is the greatest integer less than or equal to $v$,*

(iii) *the threshold labeling $h_{\mathrm{thr}}(u; \bar{\boldsymbol{b}})$ that is same as the MT labeling $h_{\mathrm{mt}}(u; \bar{\boldsymbol{b}})$ and ST labeling $h_{\mathrm{st}}(u; \bar{\boldsymbol{b}})$, if $\ell = \ell_{\mathrm{ad}}$ and $\sigma(0) = 0.5$.*

Here, Theorem 3 (i) assumes that the task loss $\ell$ is V-shaped, and the condition on $\ell_{k,l}$ in Theorem 3 (i) holds under the convexity of $\ell$ defined below:

**Theorem 4.** *$\ell_{k,l}(j)$ at each fixed different $k, l \in [K]$ is non-positive (resp. non-negative) for all $j \in [K-1]$ respectively when $k < l$ (resp. $k > l$), if the task risk $\ell$ is convex in the difference of the two arguments:*

$$\ell(j_3, k_3) \leq \frac{(j_3 - k_3) - (j_1 - k_1)}{(j_2 - k_2) - (j_1 - k_1)}\ell(j_1, k_1) + \frac{(j_2 - k_2) - (j_3 - k_3)}{(j_2 - k_2) - (j_1 - k_1)}\ell(j_2, k_2) \tag{13}$$

*for all $j_1, \ldots, k_3 \in [K]$ such that $j_1 - k_1 \neq j_2 - k_2$ and $j_1 - k_1 \leq j_3 - k_3 \leq j_2 - k_2$.*

The condition on $\sigma$ in Theorem 3 (ii) comes from the consideration for non-convex task losses.

A result similar to Theorem 3 also holds for 1DT-based likelihood models other than the CL model (10); refer to Theorem 5 in Section B. For 1DT-based statistical methods, it may be common that their LB labeling is a threshold labeling.

## 4    Proposal of 1DT-based Methods with Empirical Optimal Threshold (EOT) Labeling

In typical usages, not only the NNT, MT, and ST labelings, but also the LB labeling is a threshold labeling, as we confirmed in Theorem 3. Thus, we consider that it would be meaningful to aim for a better threshold labeling for improving the classification performance of existing 1DT-based methods. Recalling that the final goal is to make the task risk $\mathbb{E}[\ell(f(\boldsymbol{X}), Y)]$ small, and expecting that the 1DT was learned successfully and the empirical (training) task risk becomes a good estimate of the (test) task risk, we propose to apply the EOT labeling

$$h(u; \boldsymbol{t}_{\text{eot}}) \text{ with } \boldsymbol{t}_{\text{eot}} \in \underset{\boldsymbol{t} \in \mathbb{R}^{K-1}}{\arg\min} \frac{1}{n} \sum_{i=1}^{n} \ell(h_{\text{thr}}(\bar{a}(\boldsymbol{x}_i); \boldsymbol{t}), y_i) \tag{14}$$

that uses threshold parameters minimizing the empirical task risk for a given learned 1DT model $\bar{a}$.

The threshold parameters for the EOT labeling can be computed with a dynamic programming-based algorithm (Algorithm 1) mentioned in Lin & Li (2006); see also a researchers' site (`https://home.work.caltech.edu/~htlin/program/orensemble/`) of Lin & Li (2006), and Section C of our paper for its optimality guarantee. This algorithm first sorts unique elements of $\{\bar{a}(\boldsymbol{x}_i)\}_{i=1}^{n}$ to $\{\bar{a}_j\}$, and takes advantage of the recurrence relation (46) of the minimizer of the empirical task risk for sample points $i$'s s.t. $\bar{a}(\boldsymbol{x}_i) \leq \bar{a}_j$ along the ascending order of $\{\bar{a}_j\}$ to calculate threshold parameters minimizing the empirical task risk. It costs a computational complexity of quasi-linear order $O(n \log n)$ regarding the training sample size $n$, which stems from the sorting operation in Line 2 while the rest operation in Lines 3–15 costs a computational complexity of $O(nK)$.

The NNT labeling for regression-based methods (reviewed in Section 3.1) and LB labeling for statistical methods (reviewed in Section 3.3) have optimality guarantees for the task risk minimization under ideal situations; refer to Theorems 1 and 2. Also, many threshold methods employ $(K-1)$ bias parameters, and those bias parameters can be directly used in their labeling procedure like MT and ST labelings, as reviewed in Section 3.2. Presumably, for these reasons, a formulation that allows a threshold labeling with variable threshold parameters has not been considered for these methods. Our formulation in Section 2.2 introduces the threshold labeling with variable threshold parameters and clearly distinguishes the bias and threshold parameters. This is also an important contribution of this paper: due to this formulation, it becomes natural to consider the application of the threshold labeling to 1DT-based methods with a different number of bias parameters than $(K-1)$ such as PO-VS-SL (Yamasaki, 2022). Furthermore, this led to the EOT labeling that has a potential to improve the classification performance.

We, however, have to provide a remark on the additional learning of the decision boundaries (here threshold parameters) after the learning of the learner model (here a 1DT): the additional learning generally has a risk of enlarging the generalization gap. One can adjust the labeling function $h$ so that $h(\bar{g}(\boldsymbol{x}_i)) = y_i$ for every training example $i = 1, \ldots, n$ if allowing arbitrary formats and $y_{i_1} = y_{i_2}$ for any $i_1, i_2$ s.t. $\boldsymbol{x}_{i_1} = \boldsymbol{x}_{i_2}$, but the resulting classifier $h \circ \bar{g}$ would have quite low generalization performance. On the other hand, we here consider the additional learning of the labeling function among the class of threshold labelings. A threshold labeling has up to $(K-1)$ decision boundaries, that is, it is strictly restricted, and we expect that the degree of the generalization gap will not differ much with any threshold labelings.

## 5    Relationship to Other Previous Studies

Most existing studies on 1DT-based methods discuss the learning procedure. In Section 3, we reviewed ***point-wise 1DT-based methods*** in which the learning procedure is formulated with a surrogate loss function defined for each point $(a(\boldsymbol{x}_i), y_i)$, but some 1DT-based methods employ different formulations. For example, ***pair-wise 1DT-based methods*** including RankSVM (Herbrich et al., 1999) and RankBoost (Lin & Li, 2006) are formulated with a surrogate loss function that is defined for each pair of two points $(a(\boldsymbol{x}_i), y_i)$ and $(a(\boldsymbol{x}_j), y_j)$; see also Lin & Li (2012); Gutierrez et al. (2015) for survey of such methods. The EOT labeling can be applied to those 1DT-based methods as well.

---

**Algorithm 1:** Calculation of the threshold parameters for the EOT labeling

---

**Input:** Task loss $\ell$, learned 1DT $\bar{a}$, and training data $\mathcal{D}_n = \{(\boldsymbol{x}_i, y_i)\}_{i=1}^n$.
/* Preparation of $\{\bar{a}_j\}_{j=1}^N$ and $\mathcal{Y}_j$ for $j = 1, \ldots, N$.       */

**1** Let $\{\bar{a}_j'\}_{j=1}^N$ be unique elements of $\{\bar{a}(\boldsymbol{x}_i)\}_{i=1}^n$: $\bar{a}_{j_1}' \neq \bar{a}_{j_2}'$ for all $j_1, j_2 \in [N]$ s.t. $j_1 \neq j_2$, and
  $\bar{a}(\boldsymbol{x}_i) \in \{\bar{a}_j'\}_{j=1}^N$ for all $i \in [n]$.

**2** Sort $\{\bar{a}_j'\}_{j=1}^N$ in the ascending order, and represent the result as $\{\bar{a}_j\}_{j=1}^N$: $\bar{a}_1 \leq \cdots \leq \bar{a}_N$.

**3** Create sets $\mathcal{Y}_j = \{y_i \mid \bar{a}(\boldsymbol{x}_i) = \bar{a}_j\}_{i=1}^n$, $j = 1, \ldots, N$.
  /* Calculate matrix $L \in \mathbb{R}^{N \times K}$.       */

**4** **for** $k = 1, \ldots, K$ **do**

**5**     $L_{1,k} = \sum_{y_i \in \mathcal{Y}_1} \ell(k, y_i)$.

**6** **for** $j = 2, \ldots, N$ **do**

**7**     **for** $k = 1, \ldots, K$ **do**
    /* An efficient implementation of $L_{j,k} = \min_{l \in [k]} L_{j-1,l} + \sum_{y_i \in \mathcal{Y}_j} \ell(k, y_i)$.       */

**8**        $L_{\mathrm{tmp}} \leftarrow L_{j-1,1}$ (if $k = 1$), $\min\{L_{\mathrm{tmp}}, L_{j-1,k}\}$ (otherwise).

**9**        $L_{j,k} = L_{\mathrm{tmp}} + \sum_{y_i \in \mathcal{Y}_j} \ell(k, y_i)$.
  /* Calculate threshold parameters $\bar{t}$.       */

**10** $I \leftarrow \min(\arg\min_{l \in [K]} L_{N,l})$.

**11** **if** $I \neq K$ **then**
  Let $\bar{t}_k$ be a value larger than $\bar{a}_N$ (e.g., $+\infty$) for $k = I, \ldots, K-1$.

**12** **for** $j = N-1, \ldots, 1$ **do**

**13**     $J \leftarrow \min(\arg\min_{l \in [I]} L_{j,l})$.

**14**     **if** $I \neq J$ **then**
    $\bar{t}_k = (\bar{a}_j + \bar{a}_{j+1})/2$ for $k = J, \ldots, I-1$, and $I \leftarrow J$.

**15** **if** $I \neq 1$ **then**
  Let $\bar{t}_k$ be a value smaller than $\bar{a}_1$ (e.g., $-\infty$) for $k = 1, \ldots, I-1$.

**Output:** Threshold parameters $\bar{t} = (\bar{t}_1, \ldots, \bar{t}_{K-1})$.

---

On the other hand, there are several previous works that have studied the labeling procedures different from NNT, MT, ST, LB, and EOT labelings. We review the discussion of those previous works, and describe the relationship between their methods and the EOT labeling, in the following.

Herbrich et al. (1999) considered a pairwise 1DT-based method based on a hinge-type surrogate loss, for the task with the zero-one task loss. Their method (Herbrich et al., 1999, (12)) adopts a threshold labeling with threshold parameters determined by minimizing a unique criterion that emphasizes the shape of the used hinge-type loss and is different from the (empirical) surrogate risk, and has no optimality guarantee in the task risk minimization.

Lin & Li (2006) considered three methods; see RankBoost (4), ORBoost-LR (7), and ORBoost-ALL (8) of this reference. The third method applies the AT loss with $\varphi(u) = e^{-u}$ (ORBoost-ALL), and its labeling procedure is the same as the MT and ST labelings in this paper. Also, one can understand that the second method, ORBoost-LR (7), tries to minimize (an upper bound of) the empirical surrogate risk based on the IT loss function with $\varphi(u) = e^{-u}$ and ordered BPs class in its learning procedure, and its labeling procedure also can be seen to follow the idea of the MT and ST labelings. Finally, the first method, RankBoost, is a pairwise 1DT-based method, and its objective function (4) for the labeling procedure is defined as the empirical task risk with the absolute deviation task loss. This labeling procedure is similar to the EOT labeling, but Lin & Li (2006) did not mention the relevance between that objective function and the task under consideration. Although Lin & Li (2006) presented a description on the threshold parameters determination based on the zero-one task loss at the part following (4), they tried only the threshold parameters determined by minimizing the empirical task risk with the absolute deviation task loss even when they considered the task with the zero-one task loss in their experiments. The main claim of our consideration in Section 4 is that the threshold parameters should be determined via minimizing the (empirical) task risk for the task under consideration. Therefore, the EOT labeling in Section 4 can be interpreted as a variant of

Table 1: Dataset properties, the total sample size $n_{\text{tot}}$, the dimension $d$ of the explanatory variables, and the number $K$ of classes of the target variable, of the RW datasets used for the experiments in Section 6.

| | COL | PAS | SQ1 | SQ2 | BON | TAE | AUT | NEW | TOY | ESL |
|---|---|---|---|---|---|---|---|---|---|---|
| $n_{\text{tot}}$ | 24 | 36 | 52 | 52 | 57 | 151 | 205 | 215 | 300 | 488 |
| $d$ | 6 | 25 | 51 | 52 | 37 | 54 | 71 | 5 | 2 | 4 |
| $K$ | 3 | 3 | 3 | 3 | 5 | 3 | 6 | 3 | 5 | 9 |

| | BAS | EUQ | LEV | ERA | SWD | WQR | CAR | MORPH | CACD | AFAD |
|---|---|---|---|---|---|---|---|---|---|---|
| $n_{\text{tot}}$ | 625 | 736 | 1000 | 1000 | 1000 | 1599 | 1728 | 55013 | 159402 | 164418 |
| $d$ | 4 | 91 | 4 | 4 | 10 | 11 | 21 | $128^2 \times 3$ | $128^2 \times 3$ | $128^2 \times 3$ |
| $K$ | 3 | 5 | 5 | 9 | 4 | 6 | 4 | 55 | 49 | 26 |

those for Lin & Li (2006, (4)) with respect to the relevance between the objective function to determine the threshold parameters and the task under consideration. Our contributions, namely, are to expand the scope of applicability of the EOT labeling to regression-based, threshold, and statistical methods reviewed in Section 3, and to modify the EOT labeling to be performed based on the task loss function corresponding to the task under consideration, not the development of Algorithm 1 as a method for the minimization of the empirical task risk with a given task loss function.

## 6  Numerical Experiments with Real-World Datasets

**Purpose**  We performed numerical experiments using real-world (RW) datasets to answer the question, whether a modified 1DT-based method with the EOT labeling can yield better classification performance (i.e., smaller test task risk) for the OR task than existing 1DT-based methods using other labeling functions.

**Datasets and Preprocessing**  In the experiments, we used the 17 ***various-domain datasets***, COL (contact-lenses), PAS (pasture), SQ1 (squash-stored), SQ2 (squash-unstored), BON (bondrate), TAE (tae), AUT (automobile), NEW (newthyroid), TOY (da Costa et al., 2008), ESL (employee selection), BAS (balance-scale), EUQ (eucalyptus), LEV (lectures evaluation), ERA (employee rejection/acceptance), SWD (social workers decision), WQR (winequality-red), CAR (car evaluation) datasets, and the 3 ***face-age datasets***, MORPH (MORPH Album2), CACD, and AFAD datasets (Ricanek & Tesafaye, 2006; Chen et al., 2014; Niu et al., 2016). The main reason why we used the various-domain and face-age datasets is respectively to experiment with many real-world datasets in various domains and to confirm whether the proposed method achieves the classification performance competitive to the state-of-the-art method in a modern application. For most of the experimental settings, we referred to those of the previous studies (Cao et al., 2020; Gutierrez et al., 2015) with few modifications.[3]

For the various-domain datasets, we used datasets, which Gutierrez et al. (2015) used as OR datasets, following the setting of the explanatory and target variables in Gutierrez et al. (2015). One can obtain the various-domain datasets from a researchers' site (`http://www.uco.es/grupos/ayrna/orreview`) of Gutierrez et al. (2015) or our GitHub repository (`https://github.com/yamasakiryoya/OTL`). We purchased the MORPH dataset at `https://ebill.uncw.edu/C20231_ustores/web/` and preprocessed it so that the face spanned the whole image with the nose tip, which was located by facial landmark detection (Sagonas et al., 2016), at the center by using `EyepadAlign` function by Raschka (2018). While this dataset contains 55,134 facial images with ages from 16 to 77, we used 55,013 images with ages from 16 to 70. The CACD dataset can be downloaded from `https://bcsiriuschen.github.io/CARC/`. We preprocessed this dataset similarly to the MORPH dataset. Since the CACD dataset collects images from the Internet using computer vision techniques, it includes some facial images inappropriate for our consideration. Excluding images, in which no face or more than two faces were detected in the preprocessing, from the original 163,446 images, we used 159,402 facial images in the age range of 14–62 years. For the AFAD dataset obtainable at `https://github.com/afad-dataset/tarball`, because faces in its images were already centered, we took no further

---

[3]For the face-age datasets, we used a part of program codes published in `https://github.com/Raschka-research-group/coral-cnn` by Cao et al. (2020), but results of our reproduction of their method differ from theirs mainly because we changed the learning rate from $5 \times 10^{-5}$ to $10^{-2.5}$. See `https://github.com/yamasakiryoya/OTL` for program codes that we used.

preprocessing, and used its 164,418 images with ages 15–40. For these face-age datasets, we treated the age rank as the target variable. Table 1 summarizes basic dataset properties, the total sample size $n_{\text{tot}}$, the dimension $d$ of the explanatory variables, and the number $K$ of classes of the target variable, of all the used datasets.

For the various-domain datasets, we randomly divided each dataset into 72 % training, 8 % validation, and 20 % test sets. For the face-age datasets, we resized all images to $128 \times 128 \times 3$ pixels (3 stems from RGB channels) and randomly divided each dataset into 72 % training, 8 % validation, and 20 % test sets, and the training phase used images randomly cropped with the size of $120 \times 120 \times 3$ pixels as input to improve the stability of the model against the difference of facial positions, and validation and test phases used images center-cropped to the same size, following procedures by Cao et al. (2020).

**Tasks**  We considered the three popular OR tasks: minimization of the task risk for the zero-one task loss $\ell_{\text{zo}}(j, k) = \mathbb{1}(j \neq k)$ (***Task-Z***), that for the absolute deviation task loss $\ell_{\text{ad}}(j, k) = |j - k|$ (***Task-A***), and that for the squared task loss $\ell_{\text{sq}}(j, k) = (j - k)^2$ (***Task-S***).

**Methods**  For the various-domain datasets, we applied a 1DT class based on a 4-layer fully-connected neural network, in which every hidden layer has 100 nodes activated with the sigmoid function in addition to bias nodes. Also, for the face-age datasets, we applied a 1DT class based on ResNet-34 (He et al., 2016), a modern CNN architecture, following (Cao et al., 2020)'s implementation. It modifies a fully-connected (the number of classes)-output final layer of the conventional ResNet-34 to a fully-connected 1-output layer.

As the surrogate loss function, we tried the AD loss $\varphi_{\text{ad}}$; the IT loss (7) with $\varphi(u) = \min\{0, 1 - u\}$ (***Hinge-IT***), $\log(1 + e^{-u})$ (***Logistic-IT***); the AT loss (8) with $\varphi(u) = \min\{0, 1 - u\}$ (***Hinge-AT***), $\log(1 + e^{-u})$ (***Logistic-AT***); the NLL loss $\phi_{\text{nll}}$ for the statistical model (10) with $\sigma = \sigma_{\text{logistic}}$ (***OLR-NLL***).[4]

As the BPs class, we tried the non-ordered class and the ordered class for the IT losses; the ordered class for the AT and OLR-NLL losses. Note that the AD loss has no bias parameters.

As the labeling procedure, we tried the NNT and EOT labeling for the AD loss; the MT, ST, and EOT labelings for the Hinge-IT, Hinge-AT and Logistic-IT losses; the MT, ST, LB, and EOT labelings for the Logistic-AT and OLR-NLL losses. When using the ordered BPs class, the MT and ST labeling yield the same result (see Proposition 2). Also, for the Logistic-AT and OLR-NLL losses with the ordered BPs class, MT, ST, and LB labelings yield the same result under the Task-A (see Theorem 3).

Cao et al. (2020) declare that their method, which is a combination of the Logistic-AT loss and ST labeling, is the state-of-the-art method for the face-age datasets in 2020. Although they used the non-ordered BPs class, bias parameters of the surrogate risk minimizer are guaranteed to be ordered, and hence using the ordered BPs class would have made little difference to the result. Thus, we treated results for the method with the Logistic-AT loss, ordered BPs class, and ST labeling as their results, in the comparison.

**Training and Evaluation**  During the validation and test phases, models are evaluated based on the mean zero-one error (***MZE***), the mean absolute deviation error (***MAE***), and the root of the mean squared error (***RMSE***), which are defined for a classifier $f$ and $m$ used data points as $\frac{1}{m} \sum_{i=1}^{m} \ell_{\text{zo}}(f(\boldsymbol{x}_i), y_i)$, $\frac{1}{m} \sum_{i=1}^{m} \ell_{\text{ad}}(f(\boldsymbol{x}_i), y_i)$, and $\{\frac{1}{m} \sum_{i=1}^{m} \ell_{\text{sq}}(f(\boldsymbol{x}_i), y_i)\}^{1/2}$, for the Task-Z, Task-A, and Task-S, respectively. Here, the root operation of the RMSE is only for adjusting the scale of the error and does not affect our discussion related to the optimality of the EOT labeling.

We ran 50 trials for the various-domain datasets and 10 trials for the face-age datasets, with randomly-set different divisions of training, validation, and test sets and initial parameters of the network. In each trial, we trained the network using Adam of the learning rate $10^{-2.5}$ and mini-batch size 256 (or 16 when the training sample size is less than 256) as an optimization procedure for 500 epochs when $n_{\text{tot}} \leq 2000$ (i.e., for the various-domain datasets) or 100 epochs otherwise (i.e., for the face-age datasets). Additionally, for methods using the EOT labeling, we calculated the threshold parameters according to Algorithm 1 at the end of every training epoch. The above errors were evaluated on the validation set at the end of every

---

[4]For numerical stability (to avoid $\log(0)$), we used an approximation of the NLL loss in which the logarithmic function $\log(\cdot)$ of $\phi_{\text{nll}}$ is replaced to $\log(\cdot + 10^{-8})$ in the learning procedure.

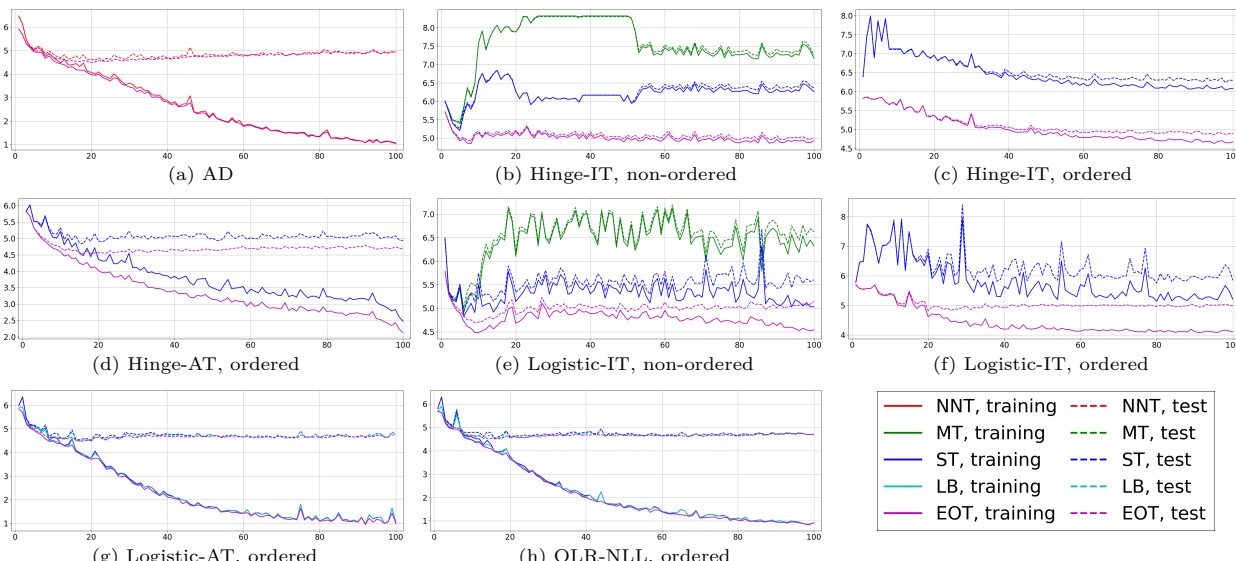

Figure 1: The learning curves of the training and test errors over 100 training epochs in a certain trial. These are for a special case for the AFAD dataset and the RMSE for the Task-S. In Figures (c), (d), and (f)–(h), the ST and MT labelings yield the same curves.

training epoch, and then we adopted the model at the timing with the smallest error among the obtained validation error sequences as the test model.

We judge the significance on the classification performance of the labeling function by the one-sided Wilcoxon rank sum test with $p$-value 0.05 based on errors for all the trials of methods using different labeling functions, in each combination of the dataset, error, surrogate loss function, and BPs class.

**Results**   Figure 1 shows the learning curves of the training and test errors. The EOT labeling results in smaller training errors as its design to do, which appears to in turn result in smaller test errors. Also, we can find that the EOT labeling may stabilize (or smooth) the learning curve from Figures (e) and (f). Tables 2, 3, and 4 show the mean and standard deviation of the errors, for the test model, evaluated on the test set. In many tried cases, the EOT labeling outperformed the NNT, MT, ST, and LB labelings regarding the test task risk. In particular, for the face-age datasets, modified 1DT-based methods using the EOT labeling provided better performance than the method (Logistic-AT, Ordered, ST) in Cao et al. (2020) that was the state-of-the-art in 2020. These results suggest the success of the EOT labeling in the subject (aiming for a better labeling procedure) of our research.

# 7   Conclusion and Future Prospect

The NNT, MT, and ST labelings and the LB labeling in typical usages are threshold labelings that may be sub-optimal depending on the learning result of the 1DT, task under consideration, and data distribution. In this study we propose to change the labeling procedure of the existing 1DT-based methods to the EOT labeling that applies the threshold parameters minimizing the empirical task risk for a given 1DT, in order to obtain higher classification performance. Experiments in this paper showed the usefulness of this proposal.

We are also interested in the design of the learning procedure, especially the selection of the surrogate loss function, for the threshold method. One may be able to undertake systematic discussion on the goodness of the loss function by fixing components of the threshold method other than the loss function to the optimal ones. In such discussion, the EOT labeling will serve as the optimal other component. This is a future prospect.

Table 2: Mean (M) and standard deviation (S) of the test MZE for Task-Z and RW datasets in the form 'M$_S$'. The smaller the error, the better that method is for that dataset and that task. In each block specified with the dataset, error, and learning procedure, we highlighted in bold font the best results that were tied with each other and superior to all other results with respect to the one-sided Wilcoxon rank sum test with a significance level of 0.05, if they exist. Also, we colored the best results in red for each combination of dataset and error.

| Learning | Labeling | COL | PAS | SQ1 | SQ2 | BON | TAE | AUT | NEW | TOY | ESL |
|---|---|---|---|---|---|---|---|---|---|---|---|
| AD | NNT | $.364_{207}$ | $.345_{126}$ | $.360_{162}$ | $.365_{141}$ | $.497_{163}$ | $.427_{085}$ | $.280_{082}$ | $.035_{024}$ | $.102_{045}$ | $.311_{046}$ |
|  | EOT | $.380_{197}$ | $.333_{134}$ | $.376_{154}$ | $.353_{132}$ | $.503_{159}$ | $.432_{089}$ | $.290_{077}$ | $.037_{024}$ | $\mathbf{.085_{042}}$ | $\mathbf{.295_{038}}$ |
| Hinge-IT non-ordered | MT | $.364_{177}$ | $.385_{148}$ | $.395_{161}$ | $.356_{133}$ | $.512_{154}$ | $.461_{096}$ | $.291_{095}$ | $.039_{024}$ | $.080_{038}$ | $.320_{048}$ |
|  | ST | $.364_{177}$ | $.385_{148}$ | $.395_{161}$ | $.356_{133}$ | $.512_{154}$ | $.461_{096}$ | $.290_{093}$ | $.039_{024}$ | $.080_{038}$ | $.320_{048}$ |
|  | EOT | $.380_{180}$ | $.335_{138}$ | $.367_{137}$ | $.345_{127}$ | $.477_{152}$ | $.431_{087}$ | $.280_{074}$ | $.034_{025}$ | $\mathbf{.069_{041}}$ | $\mathbf{.285_{045}}$ |
| Hinge-IT ordered | MT,ST | $.356_{185}$ | $.395_{157}$ | $.420_{157}$ | $.389_{135}$ | $.537_{166}$ | $.451_{098}$ | $.293_{082}$ | $.033_{028}$ | $.065_{037}$ | $.303_{042}$ |
|  | EOT | $.364_{182}$ | $\mathbf{.333_{143}}$ | $.384_{129}$ | $.376_{140}$ | $.520_{159}$ | $\mathbf{.414_{078}}$ | $.287_{080}$ | $.035_{030}$ | $.064_{032}$ | $\mathbf{.289_{040}}$ |
| Hinge-AT ordered | MT,ST | $.388_{206}$ | $.367_{159}$ | $.396_{155}$ | $.373_{133}$ | $.503_{146}$ | $.448_{084}$ | $.288_{091}$ | $.032_{025}$ | $.074_{036}$ | $.303_{037}$ |
|  | EOT | $.332_{177}$ | $.320_{150}$ | $.396_{158}$ | $.351_{132}$ | $.485_{128}$ | $\mathbf{.419_{089}}$ | $.293_{082}$ | $.034_{027}$ | $.072_{034}$ | $\mathbf{.288_{040}}$ |
| Logistic-IT non-ordered | MT | $.364_{186}$ | $.398_{171}$ | $.442_{154}$ | $.433_{143}$ | $.560_{183}$ | $.415_{093}$ | $.338_{079}$ | $.038_{029}$ | $.071_{033}$ | $.313_{053}$ |
|  | ST | $.364_{186}$ | $.398_{171}$ | $.442_{154}$ | $.433_{143}$ | $.560_{183}$ | $.415_{093}$ | $.309_{072}$ | $.038_{029}$ | $.071_{033}$ | $.312_{053}$ |
|  | EOT | $.352_{194}$ | $.355_{128}$ | $.391_{143}$ | $\mathbf{.356_{128}}$ | $.513_{142}$ | $.404_{088}$ | $\mathbf{.297_{085}}$ | $.034_{028}$ | $.068_{034}$ | $\mathbf{.294_{040}}$ |
| Logistic-IT ordered | MT,ST | $.364_{207}$ | $.395_{142}$ | $.444_{152}$ | $.418_{141}$ | $.552_{169}$ | $.410_{083}$ | $.307_{092}$ | $.034_{025}$ | $.064_{038}$ | $.297_{039}$ |
|  | EOT | $.344_{200}$ | $\mathbf{.345_{140}}$ | $.413_{149}$ | $.373_{139}$ | $.528_{128}$ | $.397_{078}$ | $.299_{084}$ | $.032_{022}$ | $.061_{039}$ | $.292_{037}$ |
| Logistic-AT ordered | MT,ST | $.360_{230}$ | $.375_{125}$ | $.476_{137}$ | $.427_{140}$ | $.633_{167}$ | $.422_{080}$ | $.281_{075}$ | $.033_{024}$ | $.060_{035}$ | $.303_{042}$ |
|  | LB | $.348_{203}$ | $.385_{152}$ | $.451_{149}$ | $.415_{134}$ | $.522_{158}$ | $.420_{081}$ | $.291_{097}$ | $.043_{040}$ | $.060_{035}$ | $.303_{043}$ |
|  | EOT | $.308_{197}$ | $\mathbf{.355_{133}}$ | $.389_{130}$ | $\mathbf{.376_{138}}$ | $.520_{143}$ | $.408_{086}$ | $.284_{084}$ | $.034_{026}$ | $.057_{031}$ | $.298_{040}$ |
| OLR-NLL ordered | MT,ST | $.384_{243}$ | $.390_{129}$ | $.467_{132}$ | $.424_{140}$ | $.577_{167}$ | $.406_{079}$ | $.294_{084}$ | $.043_{040}$ | $.063_{034}$ | $.301_{045}$ |
|  | LB | $.372_{212}$ | $.393_{170}$ | $.455_{144}$ | $.418_{150}$ | $.548_{146}$ | $.422_{081}$ | $.281_{074}$ | $.043_{036}$ | $.063_{034}$ | $.299_{042}$ |
|  | EOT | $.340_{193}$ | $.352_{132}$ | $.393_{148}$ | $.371_{132}$ | $\mathbf{.490_{149}}$ | $.410_{089}$ | $.291_{070}$ | $.031_{025}$ | $.058_{028}$ | $.294_{040}$ |

| Learning | Labeling | BAS | EUQ | LEV | ERA | SWD | WQR | CAR | MORPH | CACD | AFAD |
|---|---|---|---|---|---|---|---|---|---|---|---|
| AD | NNT | $.089_{025}$ | $.385_{044}$ | $.385_{026}$ | $.754_{027}$ | $.411_{028}$ | $.392_{026}$ | $.014_{009}$ | $.875_{005}$ | $.927_{002}$ | $.882_{002}$ |
|  | EOT | $\mathbf{.067_{029}}$ | $.388_{044}$ | $.384_{027}$ | $\mathbf{.733_{032}}$ | $.412_{025}$ | $.395_{023}$ | $.013_{008}$ | $.874_{003}$ | $\mathbf{.925_{001}}$ | $\mathbf{.878_{002}}$ |
| Hinge-IT non-ordered | MT | $.031_{021}$ | $.402_{042}$ | $.402_{028}$ | $.739_{034}$ | $.413_{031}$ | $.395_{027}$ | $.009_{005}$ | $.882_{003}$ | $.939_{002}$ | $.902_{003}$ |
|  | ST | $.031_{021}$ | $.402_{042}$ | $.402_{028}$ | $.739_{034}$ | $.413_{031}$ | $.395_{027}$ | $.009_{005}$ | $.881_{003}$ | $.939_{002}$ | $.909_{002}$ |
|  | EOT | $.027_{017}$ | $\mathbf{.385_{046}}$ | $\mathbf{.387_{025}}$ | $.732_{036}$ | $.416_{027}$ | $.395_{026}$ | $\mathbf{.008_{005}}$ | $.882_{003}$ | $.937_{003}$ | $\mathbf{.889_{002}}$ |
| Hinge-IT ordered | MT,ST | $.032_{022}$ | $.389_{048}$ | $.382_{024}$ | $.734_{035}$ | $.417_{031}$ | $.409_{023}$ | $.007_{005}$ | $.882_{003}$ | $.958_{002}$ | $.905_{006}$ |
|  | EOT | $.025_{013}$ | $.375_{043}$ | $.385_{025}$ | $.732_{030}$ | $.414_{026}$ | $\mathbf{.399_{025}}$ | $.007_{005}$ | $.880_{003}$ | $\mathbf{.949_{002}}$ | $\mathbf{.886_{003}}$ |
| Hinge-AT ordered | MT,ST | $.055_{036}$ | $.395_{047}$ | $.387_{029}$ | $.750_{027}$ | $.413_{032}$ | $.396_{027}$ | $.007_{004}$ | $.873_{005}$ | $.928_{002}$ | $.882_{002}$ |
|  | EOT | $\mathbf{.031_{016}}$ | $.386_{047}$ | $.387_{026}$ | $\mathbf{.728_{029}}$ | $.414_{027}$ | $.402_{024}$ | $.007_{005}$ | $.874_{004}$ | $\mathbf{.926_{001}}$ | $\mathbf{.878_{002}}$ |
| Logistic-IT non-ordered | MT | $.024_{020}$ | $.420_{051}$ | $.390_{030}$ | $.744_{029}$ | $.414_{036}$ | $.393_{023}$ | $.017_{007}$ | $.884_{003}$ | $.937_{001}$ | $.887_{004}$ |
|  | ST | $.024_{020}$ | $.420_{051}$ | $.390_{030}$ | $.744_{029}$ | $.414_{036}$ | $.394_{021}$ | $.018_{008}$ | $.885_{003}$ | $.936_{002}$ | $.895_{002}$ |
|  | EOT | $.015_{014}$ | $\mathbf{.388_{048}}$ | $.387_{026}$ | $.734_{033}$ | $.414_{031}$ | $\mathbf{.384_{021}}$ | $\mathbf{.010_{006}}$ | $.883_{004}$ | $.936_{002}$ | $\mathbf{.882_{002}}$ |
| Logistic-IT ordered | MT,ST | $.024_{023}$ | $.407_{043}$ | $.385_{029}$ | $.725_{031}$ | $.416_{033}$ | $.390_{024}$ | $.011_{006}$ | $.893_{004}$ | $.947_{002}$ | $.881_{002}$ |
|  | EOT | $\mathbf{.016_{013}}$ | $.389_{050}$ | $.386_{026}$ | $.729_{031}$ | $.412_{030}$ | $.389_{025}$ | $.010_{006}$ | $.890_{002}$ | $.946_{002}$ | $.882_{002}$ |
| Logistic-AT ordered | MT,ST | $.019_{017}$ | $.414_{039}$ | $.383_{028}$ | $.731_{026}$ | $\mathbf{.413_{034}}$ | $.397_{022}$ | $.012_{006}$ | $.874_{005}$ | $.929_{004}$ | $.883_{002}$ |
|  | LB | $.018_{012}$ | $.399_{047}$ | $.386_{029}$ | $.758_{026}$ | $.418_{031}$ | $.397_{022}$ | $.011_{006}$ | $.874_{006}$ | $.930_{005}$ | $\mathbf{.880_{002}}$ |
|  | EOT | $.016_{011}$ | $.383_{046}$ | $.389_{025}$ | $.732_{030}$ | $\mathbf{.412_{028}}$ | $.393_{020}$ | $.011_{006}$ | $.872_{006}$ | $.929_{004}$ | $\mathbf{.879_{003}}$ |
| OLR-NLL ordered | MT,ST | $.018_{015}$ | $.414_{051}$ | $.380_{025}$ | $\mathbf{.725_{035}}$ | $.411_{033}$ | $.394_{025}$ | $.010_{005}$ | $.871_{005}$ | $.926_{002}$ | $.878_{002}$ |
|  | LB | $.017_{012}$ | $.395_{045}$ | $.383_{029}$ | $.747_{027}$ | $.419_{030}$ | $.392_{024}$ | $.010_{005}$ | $.870_{005}$ | $.926_{003}$ | $\mathbf{.876_{002}}$ |
|  | EOT | $.015_{012}$ | $\mathbf{.376_{047}}$ | $.384_{026}$ | $\mathbf{.733_{032}}$ | $.410_{031}$ | $.394_{026}$ | $.010_{005}$ | $.871_{004}$ | $.925_{002}$ | $\mathbf{.876_{003}}$ |

## Acknowledgments

This work was supported by Grant-in-Aid for JSPS Fellows, Number 20J23367.

## References

Shivani Agarwal. Generalization bounds for some ordinal regression algorithms. In *Algorithmic Learning Theory*, pp. 7–21, 2008.

Alan Agresti. *Analysis of Ordinal Categorical Data*, volume 656. John Wiley & Sons, 2010.

Peter L Bartlett, Michael I Jordan, and Jon D McAuliffe. Convexity, classification, and risk bounds. *Journal of the American Statistical Association*, 101(473):138–156, 2006.

Table 3: A counterpart of Table 2 regarding MAE for Task-A and RW datasets.

| Learning | Labeling | COL | PAS | SQ1 | SQ2 | BON | TAE | AUT | NEW | TOY | ESL |
|---|---|---|---|---|---|---|---|---|---|---|---|
| AD | NNT | .492$_{272}$ | .352$_{141}$ | .373$_{171}$ | .382$_{154}$ | .555$_{176}$ | .587$_{122}$ | .365$_{129}$ | .035$_{024}$ | .102$_{045}$ | .321$_{048}$ |
|  | EOT | .512$_{272}$ | .338$_{142}$ | .391$_{166}$ | .369$_{145}$ | .535$_{170}$ | .550$_{117}$ | .369$_{105}$ | .037$_{024}$ | **.085$_{042}$** | **.303$_{037}$** |
| Hinge-IT non-ordered | MT | .500$_{266}$ | .407$_{195}$ | .418$_{187}$ | .371$_{149}$ | .580$_{189}$ | .601$_{138}$ | .406$_{168}$ | .039$_{024}$ | .080$_{038}$ | .333$_{052}$ |
|  | ST | .500$_{266}$ | .407$_{195}$ | .418$_{187}$ | .371$_{149}$ | .580$_{189}$ | .601$_{138}$ | .403$_{168}$ | .039$_{024}$ | .080$_{038}$ | .333$_{052}$ |
|  | EOT | .464$_{257}$ | .345$_{153}$ | .378$_{150}$ | .360$_{142}$ | .527$_{211}$ | .572$_{120}$ | .391$_{151}$ | .034$_{025}$ | **.069$_{041}$** | **.294$_{038}$** |
| Hinge-IT ordered | MT,ST | .492$_{269}$ | .400$_{164}$ | .440$_{184}$ | .409$_{151}$ | .593$_{203}$ | .582$_{137}$ | .380$_{130}$ | .033$_{028}$ | .065$_{037}$ | .320$_{048}$ |
|  | EOT | .468$_{261}$ | .338$_{146}$ | .398$_{143}$ | .387$_{152}$ | .555$_{165}$ | .545$_{121}$ | .384$_{156}$ | .035$_{030}$ | .064$_{032}$ | **.300$_{038}$** |
| Hinge-AT ordered | MT,ST | .504$_{281}$ | .372$_{174}$ | .407$_{170}$ | .389$_{143}$ | .558$_{187}$ | .567$_{119}$ | .372$_{130}$ | .032$_{025}$ | .074$_{036}$ | .317$_{040}$ |
|  | EOT | .448$_{273}$ | .338$_{168}$ | .398$_{168}$ | .364$_{147}$ | .520$_{163}$ | .552$_{131}$ | .379$_{122}$ | .034$_{027}$ | .072$_{034}$ | **.304$_{040}$** |
| Logistic-IT non-ordered | MT | .516$_{274}$ | .420$_{215}$ | .484$_{192}$ | .456$_{156}$ | .630$_{244}$ | .544$_{153}$ | .420$_{109}$ | .039$_{029}$ | .071$_{033}$ | .322$_{061}$ |
|  | ST | .516$_{274}$ | .420$_{215}$ | .484$_{192}$ | .456$_{156}$ | .627$_{236}$ | .544$_{153}$ | .384$_{098}$ | .039$_{029}$ | .071$_{033}$ | .320$_{060}$ |
|  | EOT | .468$_{290}$ | .360$_{151}$ | **.409$_{165}$** | **.369$_{140}$** | .575$_{195}$ | .508$_{119}$ | .380$_{121}$ | .034$_{028}$ | .068$_{034}$ | .303$_{040}$ |
| Logistic-IT ordered | MT,ST | .508$_{303}$ | .403$_{172}$ | .484$_{188}$ | .440$_{156}$ | .600$_{192}$ | .521$_{129}$ | .390$_{125}$ | .034$_{025}$ | .064$_{038}$ | .312$_{043}$ |
|  | EOT | .464$_{290}$ | .365$_{171}$ | .425$_{174}$ | **.385$_{152}$** | .565$_{163}$ | .519$_{126}$ | .389$_{130}$ | .032$_{022}$ | .061$_{039}$ | .304$_{039}$ |
| Logistic-AT ordered | MT,ST,LB | .484$_{291}$ | .410$_{198}$ | .484$_{184}$ | .436$_{144}$ | .573$_{200}$ | .538$_{124}$ | .369$_{110}$ | .043$_{040}$ | .060$_{035}$ | .314$_{044}$ |
|  | EOT | .420$_{281}$ | .360$_{155}$ | **.407$_{149}$** | .389$_{151}$ | .563$_{190}$ | .521$_{114}$ | .364$_{117}$ | .034$_{026}$ | .057$_{031}$ | .307$_{042}$ |
| OLR-NLL ordered | MT,ST,LB | .516$_{300}$ | .417$_{210}$ | .489$_{185}$ | .440$_{161}$ | .597$_{184}$ | .519$_{119}$ | .366$_{110}$ | .043$_{036}$ | .063$_{034}$ | .310$_{043}$ |
|  | EOT | .464$_{284}$ | **.352$_{149}$** | **.413$_{168}$** | **.384$_{146}$** | .555$_{185}$ | .509$_{111}$ | .380$_{111}$ | .031$_{025}$ | .058$_{028}$ | .304$_{037}$ |

| Learning | Labeling | BAS | EUQ | LEV | ERA | SWD | WQR | CAR | MORPH | CACD | AFAD |
|---|---|---|---|---|---|---|---|---|---|---|---|
| AD | NNT | .109$_{039}$ | .424$_{058}$ | .413$_{031}$ | 1.236$_{069}$ | .429$_{035}$ | .430$_{029}$ | .014$_{009}$ | 2.931$_{025}$ | 5.243$_{029}$ | 3.352$_{024}$ |
|  | EOT | **.080$_{032}$** | .423$_{054}$ | .413$_{030}$ | 1.231$_{072}$ | .428$_{031}$ | .432$_{028}$ | .013$_{009}$ | 2.918$_{033}$ | 5.238$_{029}$ | 3.349$_{020}$ |
| Hinge-IT non-ordered | MT | .035$_{026}$ | .453$_{051}$ | .435$_{035}$ | 1.243$_{073}$ | .432$_{034}$ | .428$_{031}$ | .009$_{005}$ | 3.183$_{022}$ | 5.937$_{215}$ | 4.029$_{057}$ |
|  | ST | .035$_{026}$ | .453$_{051}$ | .435$_{035}$ | 1.243$_{073}$ | .432$_{034}$ | .428$_{031}$ | .009$_{005}$ | 3.173$_{017}$ | 5.939$_{217}$ | 3.979$_{063}$ |
|  | EOT | .033$_{023}$ | **.424$_{061}$** | .412$_{029}$ | 1.232$_{068}$ | .431$_{031}$ | .429$_{031}$ | **.008$_{005}$** | **3.099$_{021}$** | **5.742$_{065}$** | **3.740$_{027}$** |
| Hinge-IT ordered | MT,ST | .035$_{024}$ | .431$_{058}$ | .414$_{029}$ | 1.245$_{074}$ | .437$_{037}$ | .446$_{027}$ | .007$_{005}$ | 3.349$_{040}$ | 8.052$_{279}$ | 4.694$_{116}$ |
|  | EOT | .029$_{019}$ | .417$_{053}$ | .413$_{030}$ | 1.229$_{075}$ | .430$_{027}$ | .437$_{027}$ | .007$_{005}$ | **3.204$_{032}$** | **6.581$_{057}$** | **3.761$_{107}$** |
| Hinge-AT ordered | MT,ST | .066$_{046}$ | .435$_{059}$ | .417$_{031}$ | 1.222$_{062}$ | .433$_{038}$ | .433$_{031}$ | .007$_{004}$ | 2.881$_{033}$ | 5.264$_{030}$ | 3.582$_{090}$ |
|  | EOT | **.043$_{023}$** | .422$_{062}$ | .413$_{030}$ | 1.226$_{063}$ | .431$_{031}$ | .435$_{029}$ | .007$_{005}$ | 2.863$_{022}$ | **5.242$_{034}$** | **3.390$_{018}$** |
| Logistic-IT non-ordered | MT | .026$_{024}$ | .463$_{057}$ | .425$_{036}$ | 1.262$_{073}$ | .438$_{037}$ | .430$_{026}$ | .017$_{007}$ | 3.189$_{049}$ | 5.758$_{101}$ | 3.796$_{080}$ |
|  | ST | .026$_{024}$ | .463$_{057}$ | .425$_{036}$ | 1.262$_{073}$ | .438$_{037}$ | .429$_{026}$ | .018$_{009}$ | 3.188$_{049}$ | 5.757$_{103}$ | 3.732$_{038}$ |
|  | EOT | .017$_{018}$ | **.425$_{064}$** | .414$_{031}$ | **1.228$_{070}$** | .436$_{034}$ | .421$_{026}$ | **.010$_{006}$** | 3.124$_{033}$ | 5.631$_{093}$ | 3.538$_{030}$ |
| Logistic-IT ordered | MT,ST | .027$_{027}$ | .456$_{052}$ | .416$_{030}$ | 1.239$_{076}$ | .440$_{033}$ | .426$_{026}$ | .011$_{006}$ | 3.742$_{084}$ | 6.685$_{131}$ | 4.082$_{043}$ |
|  | EOT | .019$_{016}$ | **.432$_{062}$** | .413$_{026}$ | 1.226$_{069}$ | .436$_{031}$ | .428$_{027}$ | .010$_{006}$ | **3.540$_{038}$** | **6.359$_{042}$** | **3.619$_{023}$** |
| Logistic-AT ordered | MT,ST,LB | .017$_{015}$ | **.409$_{056}$** | .412$_{029}$ | 1.225$_{068}$ | .433$_{032}$ | .429$_{027}$ | .010$_{005}$ | 2.874$_{037}$ | 5.381$_{138}$ | 3.375$_{033}$ |
|  | EOT | .020$_{014}$ | .438$_{052}$ | .416$_{032}$ | 1.233$_{063}$ | .437$_{034}$ | .431$_{023}$ | .011$_{006}$ | 2.856$_{030}$ | 5.342$_{128}$ | 3.367$_{025}$ |
| OLR-NLL ordered | MT,ST,LB | .018$_{015}$ | .407$_{055}$ | .416$_{030}$ | 1.230$_{069}$ | .434$_{035}$ | .432$_{025}$ | .011$_{006}$ | 2.873$_{043}$ | 5.243$_{034}$ | 3.345$_{020}$ |
|  | EOT | .019$_{015}$ | .433$_{057}$ | .417$_{029}$ | 1.234$_{067}$ | .436$_{035}$ | .427$_{026}$ | .010$_{005}$ | 2.845$_{037}$ | 5.228$_{041}$ | 3.343$_{017}$ |

Paul-Christian Bürkner and Matti Vuorre. Ordinal regression models in psychology: A tutorial. *Advances in Methods and Practices in Psychological Science*, 2(1):77–101, 2019.

Wenzhi Cao, Vahid Mirjalili, and Sebastian Raschka. Rank consistent ordinal regression for neural networks with application to age estimation. *Pattern Recognition Letters*, 140:325–331, 2020.

Bor-Chun Chen, Chu-Song Chen, and Winston H Hsu. Cross-age reference coding for age-invariant face recognition and retrieval. In *Proceedings of the European Conference on Computer Vision*, pp. 768–783, 2014.

Wei Chu and Zoubin Ghahramani. Gaussian processes for ordinal regression. *Journal of Machine Learning Research*, 6(Jul):1019–1041, 2005.

Wei Chu and S Sathiya Keerthi. New approaches to support vector ordinal regression. In *Proceedings of the International Conference on Machine Learning*, pp. 145–152, 2005.

Wei Chu and S Sathiya Keerthi. Support vector ordinal regression. *Neural Computation*, 19(3):792–815, 2007.

Jacob Cohen. A coefficient of agreement for nominal scales. *Educational and Psychological Measurement*, 20 (1):37–46, 1960.

Table 4: A counterpart of Table 2 regarding RMSE for Task-S and RW datasets.

| Learning | Labeling | COL | PAS | SQ1 | SQ2 | BON | TAE | AUT | NEW | TOY | ESL |
|---|---|---|---|---|---|---|---|---|---|---|---|
| AD | NNT | $.779_{343}$ | $.593_{144}$ | $.613_{193}$ | $.630_{145}$ | $.821_{194}$ | $.883_{116}$ | $.730_{164}$ | $.167_{085}$ | $.310_{075}$ | $.589_{056}$ |
| | EOT | $.827_{324}$ | $.576_{161}$ | $.633_{167}$ | $.617_{145}$ | $.797_{171}$ | $\mathbf{.828_{109}}$ | $.748_{149}$ | $.173_{084}$ | $\mathbf{.281_{075}}$ | $.574_{045}$ |
| Hinge-IT non-ordered | MT | $.799_{319}$ | $.652_{194}$ | $.676_{190}$ | $.609_{171}$ | $.866_{214}$ | $.884_{124}$ | $.767_{199}$ | $.179_{084}$ | $.275_{068}$ | $.604_{067}$ |
| | ST | $.799_{319}$ | $.652_{194}$ | $.676_{190}$ | $.609_{171}$ | $.866_{214}$ | $.884_{124}$ | $.765_{200}$ | $.179_{084}$ | $.275_{068}$ | $.604_{067}$ |
| | EOT | $.760_{326}$ | $.584_{171}$ | $.614_{169}$ | $.601_{168}$ | $\mathbf{.793_{210}}$ | $.857_{110}$ | $.753_{161}$ | $.160_{091}$ | $\mathbf{.253_{074}}$ | $\mathbf{.570_{046}}$ |
| Hinge-IT ordered | MT,ST | $.789_{331}$ | $.633_{156}$ | $.694_{186}$ | $.649_{156}$ | $.837_{204}$ | $.885_{140}$ | $.745_{164}$ | $.154_{095}$ | $.243_{078}$ | $.591_{060}$ |
| | EOT | $.763_{325}$ | $.571_{178}$ | $.642_{149}$ | $.618_{166}$ | $.791_{147}$ | $.848_{105}$ | $.736_{151}$ | $.159_{101}$ | $.242_{071}$ | $.570_{051}$ |
| Hinge-AT ordered | MT,ST | $.786_{307}$ | $.596_{194}$ | $.646_{172}$ | $.628_{164}$ | $.817_{199}$ | $.866_{116}$ | $.741_{171}$ | $.153_{092}$ | $.264_{066}$ | $.598_{052}$ |
| | EOT | $.752_{343}$ | $.569_{209}$ | $.638_{165}$ | $.601_{166}$ | $.757_{160}$ | $.834_{097}$ | $.750_{156}$ | $.157_{095}$ | $.260_{064}$ | $\mathbf{.569_{043}}$ |
| Logistic-IT non-ordered | MT | $.812_{330}$ | $.656_{222}$ | $.734_{193}$ | $.695_{142}$ | $.875_{242}$ | $.846_{132}$ | $.789_{186}$ | $.175_{094}$ | $.258_{064}$ | $.589_{065}$ |
| | ST | $.812_{330}$ | $.656_{222}$ | $.734_{193}$ | $.695_{142}$ | $.870_{233}$ | $.846_{132}$ | $.774_{192}$ | $.175_{094}$ | $.258_{064}$ | $.586_{064}$ |
| | EOT | $.754_{345}$ | $.597_{169}$ | $\mathbf{.650_{173}}$ | $\mathbf{.608_{158}}$ | $.806_{194}$ | $.807_{100}$ | $.745_{146}$ | $.161_{091}$ | $.252_{068}$ | $.566_{049}$ |
| Logistic-IT ordered | MT,ST | $.790_{352}$ | $.644_{182}$ | $.723_{188}$ | $.680_{144}$ | $.842_{186}$ | $.837_{124}$ | $.753_{159}$ | $.159_{091}$ | $.239_{082}$ | $.583_{055}$ |
| | EOT | $.779_{342}$ | $.611_{199}$ | $\mathbf{.663_{173}}$ | $\mathbf{.624_{145}}$ | $.793_{160}$ | $.809_{110}$ | $.742_{164}$ | $.160_{082}$ | $.230_{091}$ | $.572_{047}$ |
| Logistic-AT ordered | MT,ST | $.722_{368}$ | $.614_{187}$ | $.685_{176}$ | $.639_{144}$ | $.797_{205}$ | $.812_{112}$ | $.736_{150}$ | $.182_{094}$ | $.234_{080}$ | $.585_{046}$ |
| | LB | $.757_{361}$ | $.655_{202}$ | $.729_{181}$ | $.680_{134}$ | $.815_{197}$ | $.825_{117}$ | $.726_{153}$ | $.182_{098}$ | $.230_{082}$ | $.582_{049}$ |
| | EOT | $.732_{346}$ | $.601_{178}$ | $.651_{160}$ | $.623_{163}$ | $.802_{199}$ | $.813_{110}$ | $.720_{141}$ | $.162_{086}$ | $.224_{080}$ | $\mathbf{.565_{047}}$ |
| OLR-NLL ordered | MT,ST | $.795_{347}$ | $.634_{158}$ | $.706_{177}$ | $.640_{153}$ | $.800_{180}$ | $.821_{118}$ | $.741_{157}$ | $.166_{096}$ | $.239_{071}$ | $.582_{051}$ |
| | LB | $.795_{351}$ | $.651_{221}$ | $.735_{182}$ | $.680_{146}$ | $.830_{168}$ | $.833_{111}$ | $.736_{153}$ | $.184_{097}$ | $.240_{073}$ | $.578_{049}$ |
| | EOT | $.754_{351}$ | $.587_{161}$ | $.655_{174}$ | $.620_{159}$ | $.787_{181}$ | $.812_{107}$ | $.751_{156}$ | $.153_{089}$ | $.231_{066}$ | $.569_{048}$ |

| Learning | Labeling | BAS | EUQ | LEV | ERA | SWD | WQR | CAR | MORPH | CACD | AFAD |
|---|---|---|---|---|---|---|---|---|---|---|---|
| AD | NNT | $.391_{075}$ | $.712_{062}$ | $.697_{036}$ | $1.623_{094}$ | $.684_{035}$ | $.710_{030}$ | $.113_{044}$ | $3.962_{040}$ | $7.395_{027}$ | $4.571_{032}$ |
| | EOT | $\mathbf{.341_{069}}$ | $.719_{057}$ | $.698_{036}$ | $1.589_{077}$ | $.676_{030}$ | $.721_{033}$ | $.108_{044}$ | $3.942_{039}$ | $7.410_{035}$ | $\mathbf{4.534_{027}}$ |
| Hinge-IT non-ordered | MT | $.187_{097}$ | $.748_{058}$ | $.711_{039}$ | $1.657_{092}$ | $.689_{036}$ | $.708_{028}$ | $.088_{033}$ | $4.273_{028}$ | $8.009_{277}$ | $5.435_{086}$ |
| | ST | $.187_{097}$ | $.748_{058}$ | $.711_{039}$ | $1.657_{092}$ | $.689_{036}$ | $.708_{028}$ | $.088_{033}$ | $4.272_{024}$ | $8.003_{276}$ | $5.301_{072}$ |
| | EOT | $.190_{097}$ | $\mathbf{.711_{068}}$ | $\mathbf{.695_{035}}$ | $\mathbf{1.586_{072}}$ | $.681_{031}$ | $.712_{030}$ | $\mathbf{.081_{033}}$ | $\mathbf{4.137_{043}}$ | $\mathbf{7.721_{060}}$ | $\mathbf{4.927_{024}}$ |
| Hinge-IT ordered | MT,ST | $.188_{088}$ | $.727_{068}$ | $.697_{037}$ | $1.644_{083}$ | $.692_{037}$ | $.722_{025}$ | $.078_{036}$ | $4.529_{041}$ | $10.098_{276}$ | $6.374_{183}$ |
| | EOT | $.180_{092}$ | $.713_{062}$ | $.693_{036}$ | $\mathbf{1.588_{075}}$ | $\mathbf{.679_{029}}$ | $.716_{029}$ | $.077_{036}$ | $\mathbf{4.288_{038}}$ | $\mathbf{8.475_{060}}$ | $\mathbf{4.958_{116}}$ |
| Hinge-AT ordered | MT,ST | $.277_{118}$ | $.726_{072}$ | $.698_{036}$ | $1.607_{086}$ | $.685_{034}$ | $.707_{027}$ | $.079_{033}$ | $3.874_{046}$ | $7.443_{044}$ | $4.902_{130}$ |
| | EOT | $.260_{101}$ | $.714_{071}$ | $.693_{034}$ | $1.591_{077}$ | $.676_{027}$ | $.711_{025}$ | $.077_{036}$ | $3.867_{031}$ | $7.442_{048}$ | $\mathbf{4.550_{024}}$ |
| Logistic-IT non-ordered | MT | $.151_{103}$ | $.756_{074}$ | $.706_{041}$ | $1.655_{085}$ | $.696_{035}$ | $.713_{032}$ | $.133_{032}$ | $4.257_{070}$ | $7.870_{109}$ | $5.151_{078}$ |
| | ST | $.151_{103}$ | $.756_{074}$ | $.706_{041}$ | $1.655_{085}$ | $.696_{035}$ | $.710_{039}$ | $.135_{036}$ | $4.255_{068}$ | $7.869_{110}$ | $5.058_{061}$ |
| | EOT | $.119_{101}$ | $\mathbf{.708_{062}}$ | $.694_{034}$ | $1.585_{079}$ | $.684_{030}$ | $.702_{026}$ | $\mathbf{.094_{034}}$ | $\mathbf{4.172_{045}}$ | $\mathbf{7.620_{087}}$ | $\mathbf{4.706_{036}}$ |
| Logistic-IT ordered | MT,ST | $.148_{112}$ | $.748_{062}$ | $.698_{037}$ | $1.644_{090}$ | $.694_{033}$ | $.707_{029}$ | $.102_{032}$ | $5.016_{100}$ | $8.902_{182}$ | $5.720_{062}$ |
| | EOT | $.134_{096}$ | $\mathbf{.708_{065}}$ | $.693_{036}$ | $\mathbf{1.592_{078}}$ | $.684_{027}$ | $.709_{031}$ | $.095_{036}$ | $\mathbf{4.726_{064}}$ | $\mathbf{8.320_{055}}$ | $\mathbf{4.798_{035}}$ |
| Logistic-AT ordered | MT,ST | $.140_{065}$ | $\mathbf{.706_{060}}$ | $.696_{035}$ | $1.585_{077}$ | $.686_{031}$ | $.708_{027}$ | $.102_{033}$ | $3.859_{043}$ | $7.520_{096}$ | $4.589_{053}$ |
| | LB | $.137_{071}$ | $.731_{066}$ | $.696_{038}$ | $1.604_{081}$ | $.689_{034}$ | $.710_{025}$ | $.102_{033}$ | $3.858_{043}$ | $7.497_{080}$ | $\mathbf{4.517_{040}}$ |
| | EOT | $.130_{095}$ | $.693_{067}$ | $.694_{036}$ | $1.587_{080}$ | $.682_{032}$ | $.713_{029}$ | $.099_{037}$ | $3.841_{034}$ | $7.494_{053}$ | $\mathbf{4.517_{022}}$ |
| OLR-NLL ordered | MT,ST | $.134_{067}$ | $.706_{064}$ | $.697_{033}$ | $1.592_{081}$ | $.683_{024}$ | $.710_{028}$ | $.100_{025}$ | $3.870_{037}$ | $7.437_{061}$ | $4.574_{031}$ |
| | LB | $.129_{074}$ | $.724_{061}$ | $.696_{032}$ | $1.616_{085}$ | $.690_{034}$ | $.710_{029}$ | $.099_{025}$ | $3.869_{036}$ | $7.423_{060}$ | $4.508_{029}$ |
| | EOT | $.128_{091}$ | $.699_{066}$ | $.693_{035}$ | $1.588_{079}$ | $.682_{026}$ | $.708_{028}$ | $.096_{030}$ | $3.852_{044}$ | $7.417_{059}$ | $\mathbf{4.510_{025}}$ |

Jacob Cohen. Weighted kappa: nominal scale agreement provision for scaled disagreement or partial credit. *Psychological Bulletin*, 70(4):213, 1968.

Joaquim F Pinto da Costa, Hugo Alonso, and Jaime S Cardoso. The unimodal model for the classification of ordinal data. *Neural Networks*, 21(1):78–91, 2008.

Stephen E Fienberg and William M Mason. Identification and estimation of age-period-cohort models in the analysis of discrete archival data. *Sociological Methodology*, 10:1–67, 1979.

Eibe Frank and Mark Hall. A simple approach to ordinal classification. In *Proceedings of the European Conference on Machine Learning*, pp. 145–156, 2001.

Philip Hans Franses and Richard Paap. *Quantitative Models in Marketing Research*. Cambridge University Press, 2001.

Pedro Antonio Gutierrez, Maria Perez-Ortiz, Javier Sanchez-Monedero, Francisco Fernandez-Navarro, and Cesar Hervas-Martinez. Ordinal regression methods: survey and experimental study. *IEEE Transactions on Knowledge and Data Engineering*, 28(1):127–146, 2015.

Kaiming He, Xiangyu Zhang, Shaoqing Ren, and Jian Sun. Deep residual learning for image recognition. In *Proceedings of the IEEE Conference on Computer Vision and Pattern Recognition*, pp. 770–778, 2016.

R Herbrich, T Graepel, and K Obermayer. Support vector learning for ordinal regression. In *International Conference on Artificial Neural Networks*, volume 1, pp. 97–102, 1999.

Stefan Kramer, Gerhard Widmer, Bernhard Pfahringer, and Michael De Groeve. Prediction of ordinal classes using regression trees. *Fundamenta Informaticae*, 47(1-2):1–13, 2001.

Ling Li and Hsuan-Tien Lin. Ordinal regression by extended binary classification. In *Advances in Neural Information Processing Systems*, pp. 865–872, 2007.

Hsuan-Tien Lin and Ling Li. Large-margin thresholded ensembles for ordinal regression: Theory and practice. In *Algorithmic Learning Theory*, pp. 319–333. Springer, 2006.

Hsuan-Tien Lin and Ling Li. Reduction from cost-sensitive ordinal ranking to weighted binary classification. *Neural Computation*, 24(5):1329–1367, 2012.

Tie-Yan Liu. *Learning to Rank for Information Retrieval*. Springer Science & Business Media, 2011.

Peter McCullagh. Regression models for ordinal data. *Journal of the Royal Statistical Society: Series B (Methodological)*, 42(2):109–127, 1980.

Zhenxing Niu, Mo Zhou, Le Wang, Xinbo Gao, and Gang Hua. Ordinal regression with multiple output cnn for age estimation. In *Proceedings of the IEEE Conference on Computer Vision and Pattern Recognition*, pp. 4920–4928, 2016.

Fabian Pedregosa, Francis Bach, and Alexandre Gramfort. On the consistency of ordinal regression methods. *Journal of Machine Learning Research*, 18(Jan):1769–1803, 2017.

Sebastian Raschka. Mlxtend: Providing machine learning and data science utilities and extensions to python's scientific computing stack. *Journal of Open Source Software*, 3(24):638, 2018.

Karl Ricanek and Tamirat Tesafaye. Morph: A longitudinal image database of normal adult age-progression. In *Proceedings of the IEEE International Conference on Automatic Face and Gesture Recognition*, pp. 341–345, 2006.

Christos Sagonas, Epameinondas Antonakos, Georgios Tzimiropoulos, Stefanos Zafeiriou, and Maja Pantic. 300 faces in-the-wild challenge: Database and results. *Image and Vision Computing*, 47:3–18, 2016.

Amnon Shashua and Anat Levin. Ranking with large margin principle: Two approaches. In *Advances in Neural Information Processing Systems*, pp. 961–968, 2003.

WA Thompson Jr. On the treatment of grouped observations in life studies. *Biometrics*, pp. 463–470, 1977.

Richard Williams. Generalized ordered logit/partial proportional odds models for ordinal dependent variables. *The Stata Journal*, 6(1):58–82, 2006.

Ryoya Yamasaki. Unimodal likelihood models for ordinal data. *Transactions on Machine Learning Research*, 2022. URL https://openreview.net/forum?id=1lOsClLiPc.

Shipeng Yu, Kai Yu, Volker Tresp, and Hans-Peter Kriegel. Collaborative ordinal regression. In *Proceedings of the International Conference on Machine Learning*, pp. 1089–1096, 2006.

# A    Proof of Consistency of Statistical Methods

We here give proof of the theorem on the interpretation of the learning procedure for statistical methods.

*Proof of Theorem 2.* We can characterize the surrogate risk minimization for the NLL loss as maximum likelihood estimation for the statistical model (10) for multi-class classification problem through the equation

$$
\min_{a \in \mathcal{A}, b \in \mathcal{B}} \mathbb{E}[\phi_{\mathrm{nll}}(a(\boldsymbol{X}), \boldsymbol{b}, Y; \sigma)] = \min_{a \in \mathcal{A}, b \in \mathcal{B}} \mathbb{E}\left[\sum_{y=1}^{K} \Pr(y|\boldsymbol{X})\phi_{\mathrm{nll}}(a(\boldsymbol{X}), \boldsymbol{b}, y; \sigma)\right]
$$

$$
= \min_{a \in \mathcal{A}, b \in \mathcal{B}} \mathbb{E}\left[-\sum_{y=1}^{K} \Pr(y|\boldsymbol{X}) \log P(y; \sigma, a(\boldsymbol{x}), \boldsymbol{b})\right]. \tag{15}
$$

According to the method of Lagrange multiplier, one solution of a point-wise (at each $\boldsymbol{X} = \boldsymbol{x}$) minimization problem

$$
\min_{\{\hat{\Pr}(k|\boldsymbol{x})\}_k} -\sum_{y=1}^{K} \Pr(y|\boldsymbol{x}) \log \hat{\Pr}(y|\boldsymbol{x}), \quad \text{subject to } \sum_{y=1}^{K} \hat{\Pr}(y|\boldsymbol{x}) = 1 \tag{16}
$$

is $\hat{\Pr}(y|\boldsymbol{x}) = \Pr(y|\boldsymbol{x}) = P(y; \sigma, \tilde{a}(\boldsymbol{x}), \tilde{\boldsymbol{b}})$, $y = 1, \dots, K$, where the existence of such $\{\tilde{a}(\boldsymbol{x}), \tilde{\boldsymbol{b}}\}$ is assumed in the statement of the theorem. This solution applies for any $\boldsymbol{x} \in \mathbb{R}^d$, and one can see that a solution of (15) is $\{\tilde{a}, \tilde{\boldsymbol{b}}\}$, which completes the proof of the statement for the NLL loss.

Also, for the ANLCL loss, we can provide the following characterization:

$$
\mathbb{E}[\phi_{\mathrm{anlcl}}(a(\boldsymbol{X}), \boldsymbol{b}, Y)]
$$

$$
= \mathbb{E}\left[-\sum_{y=1}^{K} \Pr(y|\boldsymbol{X})\left\{\sum_{k=1}^{y-1} \log\{1 - Q(k; \sigma, a(\boldsymbol{X}), \boldsymbol{b})\} + \sum_{k=y}^{K-1} \log Q(k; \sigma, a(\boldsymbol{X}), \boldsymbol{b})\right\}\right]
$$

$$
= -\sum_{y=1}^{K-1} \mathbb{E}\left[\Pr(Y \le y|\boldsymbol{X}) \log Q(k; \sigma, a(\boldsymbol{X}), \boldsymbol{b}) + \{1 - \Pr(Y \le y|\boldsymbol{X})\} \log\{1 - Q(y; \sigma, a(\boldsymbol{X}), \boldsymbol{b})\}\right], \tag{17}
$$

where $Q(y; \sigma, a(\boldsymbol{x}), \boldsymbol{b}) := \sum_{k=1}^{y} P(k; \sigma, a(\boldsymbol{x}), \boldsymbol{b})$ and the expectation value $\mathbb{E}[\cdot]$ is taken for $\boldsymbol{X}$. On the ground of the binary version, '$y$ or less' vs. 'more than $y$' ($y = 1, \dots, K-1$), of (16), one can prove the statement similarly. $\qquad \square$

One may consider the IT loss (7) with $\varphi(u) = -\log\{\sigma(u)\}$, which we call ***immediate negative log cumulative likelihoods (INLCL) loss function***. However, it is difficult to characterize the surrogate risk minimization with the INLCL loss as a problem with a known solution unlike those for the NLL and ANLCL losses, and the optimality condition for the INLCL loss is unknown.

# B    Proof of Relationships between Labeling Functions

This section provides proofs of Theorems 3 and 4 regarding the relationships between the LB and threshold labelings. Propositions 1 and 2 would be trivial, so we omit proofs of them.

First, we prove Theorem 3.

*Proof of Theorem 3.* We introduce the functions

$$
R_j(u) := \sum_{k=1}^{K} \{\sigma(\bar{b}_k - u) - \sigma(\bar{b}_{k-1} - u)\}\ell(j, k) = \ell(j, K) + \sum_{k=1}^{K-1} \sigma(\bar{b}_k - u)\{\ell(j, k) - \ell(j, k+1)\} \text{ for } j = 1, \dots, K,
$$

$$
\tag{18}
$$

with $\bar{b}_0 = -\infty$ and $\bar{b}_K = +\infty$, where the equation holds, since $\sigma(-\infty) = 0$ and $\sigma(+\infty) = 1$. The classifier based on the LB labeling, $f(\boldsymbol{x}) = \arg\min_{j \in [K]} \sum_{k=1}^{K} P(k; \sigma, \bar{a}(\boldsymbol{x}), \bar{\boldsymbol{b}}) \ell(j, k)$, is equal to $\arg\min_{j \in [K]} R_j(\bar{a}(\boldsymbol{x}))$. According to Proposition 1, the LB labeling is a certain threshold labeling if and only if $\arg\min_{j \in [K]} \{R_j(u_1)\}_{j=1}^{K} \le \arg\min_{j \in [K]} \{R_j(u_2)\}_{j=1}^{K}$ for any $u_1, u_2 \in \mathbb{R}$ such that $u_1 \le u_2$. The latter condition holds if the situation

$$R_k(u) > R_l(u) \text{ for } u \in (s_1, s_2) \text{ and } R_k(u) < R_l(u) \text{ for } u \in (s_2, s_3) \text{ with } k < l, \ s_1 < s_2 < s_3 \qquad (19)$$

does not occur. In the following we assume $k < l$ for the indices $k, l \in [K]$.

**Proof of (i).** Under the assumption described in the statement of the theorem, the difference

$$R_k(u) - R_l(u) = \underbrace{\{\ell(k, K) - \ell(l, K)\}}_{\substack{\text{non-negative} \\ \text{constant}}} + \sum_{j=1}^{K-1} \underbrace{\sigma(\bar{b}_j - u)}_{\substack{\text{non-negative} \\ \text{non-increasing}}} \underbrace{\{\ell(k, j) - \ell(k, j+1) - \ell(l, j) + \ell(l, j+1)\}}_{\substack{\text{non-positive} \\ \text{constant}}} \qquad (20)$$

is non-decreasing with respect to $u$. Thus, $R_k(u) \le R_l(u)$ for $u \le p$ and $R_k(u) \ge R_l(u)$ for $u \ge p$ for some point $p$, $R_k(u) \le R_l(u)$ for any $u$, or $R_k(u) \ge R_l(u)$ for any $u$, which implies that the above-mentioned situation (19) does not occur. Note that, for the instances $\ell = \ell_{\mathrm{ad}}, \ell_{\mathrm{sq}}$, one has that

$$\ell_{k,l}(j) = \ell(k, j) - \ell(k, j+1) - \ell(l, j) + \ell(l, j+1) = \begin{cases} -2 \cdot \mathbb{1}(k \le j \le l - 1) & \text{for } \ell = \ell_{\mathrm{ad}}, \\ 2(k - l), & \text{for } \ell = \ell_{\mathrm{sq}}. \end{cases} \qquad (21)$$

This completes the proof of the statement (i).

**Proof of (ii).** For $\ell = \ell_{\mathrm{zo},c}$ with $c \in [0, \lfloor K/2 \rfloor)$ where $\ell_{\mathrm{zo}} = \ell_{\mathrm{zo},0}$, the function $R_j(u)$ reduces to

$$R_j(u) = 1 - \{\sigma(b_j - u) - \sigma(a_j - u)\}, \qquad (22)$$

with $a_j := \bar{b}_{\max\{0, j-c\}}$ and $b_j := \bar{b}_{\min\{j+c, K\}}$, where $a_j < b_j$. Lemma 1 (described after the proof of Theorem 3) shows the shape of the function $R_j(u)$: Under the assumption of Theorem 3 (ii), $R_j(u)$ is minimized at $u = (a_j + b_j)/2 := c_j$, symmetric in $u$ around $u = c_j$, non-increasing in $u$ for $u < c_j$, and non-decreasing in $u$ when $u > c_j$, from Lemma 1 (i) and (ii). Also, assuming that $c_j$ is fixed, then $R_j(u)$ is non-decreasing in $b_j - a_j$, from Lemma 1 (iii).

When $b_k - a_k = b_l - a_l$, the translated two curves $R_k(u)$ and $R_l(u)$ have just one intersection point at $u = (c_k + c_l)/2$, and it holds that $R_k(u) \le R_l(u)$ for $u \le (c_k + c_l)/2$ and $R_k(u) \ge R_l(u)$ for $u \ge (c_k + c_l)/2$. Therefore, the situation (19) does not occur if $b_k - a_k = b_l - a_l$.

Then, assume $b_k - a_k < b_l - a_l$ (the following proof strategy for this setting can be applied to the other setting $b_k - a_k > b_l - a_l$). In this setting, $R_k(u) > R_l(u)$ for $u \ge c_l$ due to the shape of the functions $R_k$ and $R_l$. Also, within $[c_k, c_l]$, they can have one intersection point $p$ at most such that $R_k(u) \le R_l(u)$ for $u \in [c_k, p]$ and $R_k(u) \ge R_l(u)$ for $u \in [p, c_l]$, since $R_k(u)$ and $R_l(u)$ are respectively non-decreasing and non-increasing in $u$. Therefore, the situation (19) can be satisfied only in such a situation that there exists a point $p$ satisfying

$$R_k(p) = R_l(p), \ R_k'(p) < R_l'(p), \text{ and } p \le c_k. \qquad (23)$$

The existence of such a point $p$ implies that

$$\frac{\sigma'(a_k - p) - \sigma'(b_k - p)}{\sigma(a_k - p) - \sigma(b_k - p)} < \frac{\sigma'(a_l - p) - \sigma'(b_l - p)}{\sigma(a_l - p) - \sigma(b_l - p)} \text{ with } a_k \le a_l, \ b_k \le b_l, \ a_k \le b_k, \ a_l \le b_l, \ p \le c_k. \qquad (24)$$

However, the assumption that $\frac{\sigma'(v_1) - \sigma'(v_2)}{\sigma(v_1) - \sigma(v_2)}$ is non-increasing in $v_1$ with fixed $v_2$ and in $v_2$ with fixed $v_1$ when $v_1 < v_2$ shows that

$$\frac{\sigma'(a_k - p) - \sigma'(b_k - p)}{\sigma(a_k - p) - \sigma(b_k - p)} \ge \frac{\sigma'(a_k - p) - \sigma'(b_l - p)}{\sigma(a_k - p) - \sigma(b_l - p)} \ge \frac{\sigma'(a_l - p) - \sigma'(b_l - p)}{\sigma(a_l - p) - \sigma(b_l - p)}, \qquad (25)$$

which contradicts to equation (24). Therefore, the situation (19) does not occur also when $b_k - a_k < b_l - a_l$. Note that, especially when $\sigma = \sigma_{\text{logistic}}$, one can show that

$$
\begin{aligned}
\frac{\sigma'(v_1) - \sigma'(v_2)}{\sigma(v_1) - \sigma(v_2)} &= \frac{\sigma_{\text{logistic}}(v_1)(1 - \sigma_{\text{logistic}}(v_1)) - \sigma_{\text{logistic}}(v_2)(1 - \sigma_{\text{logistic}}(v_2))}{\sigma_{\text{logistic}}(v_1) - \sigma_{\text{logistic}}(v_2)} \\
&= 1 - \{\sigma_{\text{logistic}}(v_1) + \sigma_{\text{logistic}}(v_2)\},
\end{aligned}
\tag{26}
$$

is decreasing in $v_1$ with fixed $v_2$ and in $v_2$ with fixed $v_1$. Moreover, when $\sigma = \sigma_{\text{gauss}}$, one has that

$$
\frac{\sigma'(v_1) - \sigma'(v_2)}{\sigma(v_1) - \sigma(v_2)} \propto \frac{e^{-v_1^2/2} - e^{-v_2^2/2}}{\sigma_{\text{gauss}}(v_1) - \sigma_{\text{gauss}}(v_2)} := f_1(v_1, v_2),
\tag{27}
$$

that the derivative of $f_1(v_1, v_2)$ with respect to $v_1$,

$$
\frac{\partial}{\partial v_1} f_1(v_1, v_2) = \frac{-v_1 e^{-v_1^2/2}\{\sigma_{\text{gauss}}(v_1) - \sigma_{\text{gauss}}(v_2)\} - \left(e^{-v_1^2/2} - e^{-v_2^2/2}\right)\frac{1}{\sqrt{2\pi}} e^{-v_1^2/2}}{\{\sigma_{\text{gauss}}(v_1) - \sigma_{\text{gauss}}(v_2)\}^2}
\tag{28}
$$

has the same sign as

$$
f_2(v_1, v_2) := -v_1\{\sigma_{\text{gauss}}(v_1) - \sigma_{\text{gauss}}(v_2)\} - \left(\frac{1}{\sqrt{2\pi}} e^{-v_1^2/2} - \frac{1}{\sqrt{2\pi}} e^{-v_2^2/2}\right),
\tag{29}
$$

and that the derivative of $f_2(v_1, v_2)$ with respect to $v_2$ is

$$
\frac{\partial}{\partial v_2} f_2(v_1, v_2) = (v_1 - v_2)\frac{1}{\sqrt{2\pi}} e^{-v_2^2/2}.
\tag{30}
$$

Since $\frac{\partial}{\partial v_2} f_2(v_1, v_2) < 0$ when $v_1 < v_2$ and $f_2(v_1, v_1) = 0$, it holds that $f_2(v_1, v_2)$, which has the same sign as $\frac{\partial}{\partial v_1}\frac{\sigma'(v_1) - \sigma'(v_2)}{\sigma(v_1) - \sigma(v_2)}$, is negative when $v_1 < v_2$, that is, $\frac{\sigma'(v_1) - \sigma'(v_2)}{\sigma(v_1) - \sigma(v_2)}$ is decreasing in $v_1$ with fixed $v_2$ when $v_1 < v_2$; monotonicity in $v_2$ with fixed $v_1$ can be proved by the same discussion.

**Proof of (iii).** Regarding the MT and ST labelings, let $y = h_{\text{thr}}(u; \bar{\boldsymbol{b}})$ under the assumption $\bar{b}_1 \le \cdots \le \bar{b}_{K-1}$, which implies that $\bar{b}_1 \le \cdots \le \bar{b}_{y-1} \le u \le \bar{b}_y \le \cdots \le \bar{b}_{K-1}$. Regarding the LB labeling for the likelihood model (10), one has that, with the abbreviations $\sigma_k := \sigma(\bar{b}_k - u)$ for $k = 1, \ldots, K$,

$$
\begin{aligned}
R_j(u) &= \sum_{k=1}^{K} \{\sigma_k - \sigma_{k-1}\}|j - k|, \\
&= |j - 1|\{\sigma_1 - \sigma_0\} + |j - 2|\{\sigma_2 - \sigma_1\} + \cdots + 2\{\sigma_{j-2} - \sigma_{j-3}\} + \{\sigma_{j-1} - \sigma_{j-2}\} \\
&\quad + \{\sigma_{j+1} - \sigma_j\} + 2\{\sigma_{j+2} - \sigma_{j+1}\} + \cdots + |j - K + 1|\{\sigma_{K-1} - \sigma_{K-2}\} + |j - K|\{\sigma_K - \sigma_{K-1}\} \\
&= -|j - 1|\underbrace{\sigma_0}_{0} + \left\{\sum_{k=1}^{j-1} \sigma_k\right\} - \left\{\sum_{k=j}^{K-1} \sigma_k\right\} + |j - K|\underbrace{\sigma_K}_{1} \\
&= \sum_{k=1}^{j-1} \sigma(\bar{b}_k - u) + \sum_{k=j}^{K-1} \{1 - \sigma(\bar{b}_k - u)\},
\end{aligned}
\tag{31}
$$

for every $j \in [K]$. Simple calculations show that $\sigma(\bar{b}_k - u) \le 0.5$ for $k = 1, \ldots, y - 1$ and $\{1 - \sigma(\bar{b}_k - u)\} \le 0.5$ for $k = y, \ldots, K - 1$, from $\bar{b}_1 \le \cdots \le \bar{b}_{y-1} \le u \le \bar{b}_y \le \cdots \le \bar{b}_{K-1}$ and the assumption on the shape of $\sigma$. One would see that objective function (31) is minimized at $j = y$ because some summands are replaced by ones of 0.5 or more if $j$ deviates from $y$, which concludes the proof. □

The following is an auxiliary lemma for the above-described proof of Theorem 3.

**Lemma 1.** *Suppose that $\sigma$ is non-decreasing and satisfies $\sigma(-\infty) = 0$ and $\sigma(+\infty) = 1$. Define $S(u; a, b) := \sigma(b - u) - \sigma(a - u)$ for $a < b$. Then, one has that*

    (i) $S(u; a, b)$ with fixed $a$ and $b$ is symmetric in $u$ around $u = \frac{a+b}{2}$, if $\sigma(-\cdot) = 1 - \sigma(\cdot)$, or if $\sigma$ is differentiable and $\sigma'$ is even.

    (ii) $S(u; a, b)$ with fixed $a$ and $b$ is maximized with respect to $u$ at $u = \frac{a+b}{2}$, non-decreasing in $u$ for $u < \frac{a+b}{2}$, and non-increasing in $u$ for $u > \frac{a+b}{2}$, if $\sigma$ is differentiable and $\sigma'(u)$ is even and non-increasing in $u$ if $u > 0$.

    (iii) $S(u; a, b)$ with fixed $u$ and $\frac{a+b}{2}$ is increasing with respect to $(b - a)$.

*Proof of Lemma 1.* **Proof of (i).** The assumptions that $\sigma(-\infty) = 0$, $\sigma(+\infty) = 1$, and $\sigma'$ is even imply that $\sigma(-\cdot) = 1 - \sigma(\cdot)$. On the basis of this result, one then has that

$$S\left(u + \frac{a+b}{2}; a, b\right) = \sigma\left(b - \left\{u + \frac{a+b}{2}\right\}\right) - \sigma\left(a - \left\{u + \frac{a+b}{2}\right\}\right)$$
$$= \sigma\left(\frac{b-a}{2} - u\right) - \sigma\left(-\frac{b-a}{2} - u\right) = \sigma\left(\frac{b-a}{2} - u\right) - 1 + \sigma\left(\frac{b-a}{2} + u\right), \tag{32}$$

which implies that

$$S\left(u + \frac{a+b}{2}; a, b\right) = S\left(-u + \frac{a+b}{2}; a, b\right). \tag{33}$$

**Proof of (ii).** The above equation (32) shows that

$$\frac{\partial}{\partial u} S\left(u + \frac{a+b}{2}; a, b\right) = \sigma'\left(\frac{b-a}{2} + u\right) - \sigma'\left(\frac{b-a}{2} - u\right) = 0, \quad \text{at } u = 0. \tag{34}$$

Also, one can show that

$$\frac{\partial}{\partial u} S\left(u + \frac{a+b}{2}; a, b\right) = \sigma'\left(\frac{b-a}{2} + u\right) - \sigma'\left(\frac{b-a}{2} - u\right)$$
$$= \begin{cases} \sigma'\left(|\frac{b-a}{2} + u|\right) - \sigma'\left(\frac{b-a}{2} - u\right) \geq 0, & \text{for } u < 0, \\ \sigma'\left(\frac{b-a}{2} + u\right) - \sigma'\left(|\frac{b-a}{2} - u|\right) \leq 0, & \text{for } u > 0. \end{cases} \tag{35}$$

Here, for $u < 0$, we used the fact that $\sigma'$ is even, which implies that $\sigma'(\frac{b-a}{2} + u) = \sigma'(|\frac{b-a}{2} + u|)$, and $\sigma'(v)$ is non-increasing in $v$ for $v > 0$ and $\frac{b-a}{2} - u > |\frac{b-a}{2} + u| > 0$; for $u > 0$, we used the fact that $\sigma'$ is even, which implies that $\sigma'(\frac{b-a}{2} - u) = \sigma'(|\frac{b-a}{2} - u|)$, and $\sigma'(v)$ is non-increasing in $v$ for $v > 0$ and $\frac{b-a}{2} + u > |\frac{b-a}{2} - u| > 0$.

**Proof of (iii).** With change of variables $t = \frac{b-a}{2}, v = \frac{a+b}{2}$, we introduce a function

$$T(t; u, v) = S(u; v - t, v + t) = \sigma(v - u + t) - \sigma(v - u - t). \tag{36}$$

For this function, one has that

$$\frac{\partial}{\partial t} T(t; u, v) = \sigma'(v - u + t) + \sigma'(v - u - t) \geq 0, \tag{37}$$

since $\sigma$ is non-decreasing (i.e., $\sigma'(u) \geq 0$ for any $u$). $\qquad\square$

Next, we give a proof of Theorem 4.

*Proof of Theorem 4.* If $k < l$, the convexity shows that

$$\ell(k, j) \leq \frac{\{k - j\} - \{k - (j + 1)\}}{\{l - j\} - \{k - (j + 1)\}} \ell(k, j + 1) + \frac{\{l - j\} - \{k - j\}}{\{l - j\} - \{k - (j + 1)\}} \ell(l, j)$$
$$= \frac{1}{l - k + 1} \ell(k, j + 1) + \frac{l - k}{l - k + 1} \ell(l, j), \tag{38}$$

and that

$$\ell(l, j+1) \le \frac{\{l-(j+1)\} - \{k-(j+1)\}}{\{l-j\} - \{k-(j+1)\}} \ell(k, j+1) + \frac{\{l-j\} - \{l-(j+1)\}}{\{l-j\} - \{k-(j+1)\}} \ell(l, j)$$
$$= \frac{l-k}{l-k+1} \ell(k, j+1) + \frac{1}{l-k+1} \ell(l, j). \tag{39}$$

These inequalities imply that $\ell_{k,l}$ is non-positive:

$$\ell_{k,l}(j)$$
$$= \{\ell(k, j) + \ell(l, j+1)\} - \{\ell(k, j+1) + \ell(l, j)\}$$
$$= \{\ell(k, j) + \ell(l, j+1)\} - \left[\left\{\frac{1}{l-k+1} \ell(k, j+1) + \frac{l-k}{l-k+1} \ell(l, j)\right\} + \left\{\frac{l-k}{l-k+1} \ell(k, j+1) + \frac{1}{l-k+1} \ell(l, j)\right\}\right]$$
$$= \left[\ell(k, j) - \left\{\frac{1}{l-k+1} \ell(k, j+1) + \frac{l-k}{l-k+1} \ell(l, j)\right\}\right] + \left[\ell(l, j+1) - \left\{\frac{l-k}{l-k+1} \ell(k, j+1) + \frac{1}{l-k+1} \ell(l, j)\right\}\right]$$
$$\le 0. \tag{40}$$

Similarly, one can show that $\ell_{k,l}$ is non-negative if $k > l$. □

McCullagh (1980, Section 6.1) has proposed the heteroscedastic extension of the cmulative link model (10),

$$P_2(y; \sigma, a(\boldsymbol{x}), \boldsymbol{b}, s(\boldsymbol{x})) := P(y; \sigma, a(\boldsymbol{x})/s(\boldsymbol{x}), \boldsymbol{b}/s(\boldsymbol{x})) \tag{41}$$

with the scale model $s : \mathbb{R}^d \to (0, \infty)$, and statistical OR studies, Thompson Jr (1977); Fienberg & Mason (1979) and Agresti (2010, Section 4.2), have also considered another model

$$P_3(y; \sigma, a(\boldsymbol{x}), \boldsymbol{b}) := \sigma(b_y - a(\boldsymbol{x})) \prod_{k=1}^{y-1} \{1 - \sigma(b_{k-1} - a(\boldsymbol{x}))\}. \tag{42}$$

We obtain the following theorem that is similar to Theorem 3 and suggests the efficiency of the EOT labeling for statistical methods adopting these other likelihood models:

**Theorem 5.** *Suppose that $\sigma$ is non-decreasing and satisfies $\sigma(-\infty) = 0$ and $\sigma(+\infty) = 1$ and that $\bar{a} : \mathbb{R}^d \to \mathbb{R}$, $\bar{\boldsymbol{b}} \in \mathbb{R}^{K-1}$, and $\bar{s} : \mathbb{R}^d \to (0, \infty)$.*

(i) $\arg\min_{j \in [K]} \sum_{k=1}^{K} P_2(k; \sigma, \bar{a}(\boldsymbol{x}), \bar{\boldsymbol{b}}, \bar{s}(\boldsymbol{x}))\ell(j, k) = h_{\mathrm{thr}}(\bar{a}(\boldsymbol{x}); \bar{\boldsymbol{b}})$ *if $\ell = \ell_{\mathrm{ad}}$, $\sigma(0) = 0.5$, and $\bar{b}_1 \le \cdots \le \bar{b}_{K-1}$.*

(ii) $\arg\min_{j \in [K]} \sum_{k=1}^{K} P_3(k; \sigma, \bar{a}(\boldsymbol{x}), \bar{\boldsymbol{b}})\ell(j, k) = h_{\mathrm{thr}}(\bar{a}(\boldsymbol{x}); \boldsymbol{t})$ *for some $\boldsymbol{t} \in \mathbb{R}^{K-1}$ if $\ell = \ell_{\mathrm{ad}}$.*

*Proof of Theorem 5.* **Proof of (i).** The statement (i) of Theorem 5 is trivial from the statement (iii) of Theorem 3.

**Proof of (ii).** Regarding the LB labeling for the likelihood model (42), one has that, with the abbreviations $\dot{\sigma}_k := 1 - \sigma(\bar{b}_k - \bar{a}(\boldsymbol{x}))$ for $k = 1, \ldots, K$,

$$R_j(\bar{a}(\boldsymbol{x})) := \sum_{k=1}^{K} P_3(k; \sigma, \bar{a}(\boldsymbol{x}), \bar{\boldsymbol{b}})\ell_{\mathrm{ad}}(j, k)$$
$$= \sum_{k=1}^{K} \left((1 - \dot{\sigma}_k) \prod_{l=1}^{k-1} \dot{\sigma}_{l-1}\right) |j - k|,$$
$$= |j - 1|(1 - \dot{\sigma}_1) + |j - 2|\dot{\sigma}_1(1 - \dot{\sigma}_2) + \cdots + \dot{\sigma}_1 \cdots \dot{\sigma}_{j-2}(1 - \dot{\sigma}_{j-1})$$
$$+ \dot{\sigma}_1 \cdots \dot{\sigma}_j(1 - \dot{\sigma}_{j+1}) + \cdots + |j - K + 1|\dot{\sigma}_1 \cdots \dot{\sigma}_{K-2}(1 - \dot{\sigma}_{K-1}) + |j - K|\dot{\sigma}_1 \cdots \dot{\sigma}_{K-1}$$
$$= (j - 1) - \left(\sum_{k=1}^{j-1} \prod_{l=1}^{k} \{1 - \sigma(\bar{b}_l - \bar{a}(\boldsymbol{x}))\}\right) + \left(\sum_{k=j}^{K-1} \prod_{l=1}^{k} \{1 - \sigma(\bar{b}_l - \bar{a}(\boldsymbol{x}))\}\right), \tag{43}$$

for every $j \in [K]$. One has that

$$R_{j+1}(\bar{a}(\boldsymbol{x})) - R_j(\bar{a}(\boldsymbol{x})) = 1 - 2 \prod_{l=1}^{j} \{1 - \sigma(\bar{b}_l - \bar{a}(\boldsymbol{x}))\}, \tag{44}$$

is non-decreasing in $j$ with fixed $\bar{a}(\boldsymbol{x})$. Therefore, $\arg\min_{j\in[K]} \sum_{k=1}^{K} P_3(k; \sigma, \bar{a}(\boldsymbol{x}), \bar{\boldsymbol{b}}) \ell_{\mathrm{ad}}(j,k)$ is the first index $l$ such that $R_{l+1}(\bar{a}(\boldsymbol{x})) - R_l(\bar{a}(\boldsymbol{x})) \leq 0$, or $K$ if $R_{l+1}(\bar{a}(\boldsymbol{x})) - R_l(\bar{a}(\boldsymbol{x})) > 0$ for all $l = 1, \ldots, K-1$. Also, $R_{l+1}(\bar{a}(\boldsymbol{x})) - R_l(\bar{a}(\boldsymbol{x}))$ is non-increasing in $\bar{a}(\boldsymbol{x})$, for each $l = 1, \ldots, K-1$. These facts show that $\arg\min_{j\in[K]} \sum_{k=1}^{K} P_3(k; \sigma, \bar{a}(\boldsymbol{x}), \bar{\boldsymbol{b}}) \ell_{\mathrm{ad}}(j,k) = h_{\mathrm{thr}}(\bar{a}(\boldsymbol{x}); \boldsymbol{t})$ with the threshold parameters $t_k$, $k = 1, \ldots, K-1$ satisfying $R_{k+1}(t_k) - R_k(t_k) = 0$. □

## C  Optimality Guarantee of Algorithm for Empirical Optimal Threshold Labeling

Lin & Li (2006) do not describe the optimality guarantee of Algorithm 1 in their paper. As a supplement to their development, we write here the optimality guarantee of Algorithm 1.

**Theorem 6.** *For any task loss $\ell : [K]^2 \to [0, \infty)$, 1DT $\bar{a}$, and training data $\mathcal{D}_n = \{(\boldsymbol{x}_i, y_i)\}_{i=1}^{n}$, the threshold parameters $\bar{\boldsymbol{t}}$ obtained by Algorithm 1 minimize the empirical task risk for a classifier based on the threshold labeling: $\bar{\boldsymbol{t}} \in \arg\min_{\boldsymbol{t} \in \mathbb{R}^{K-1}} \frac{1}{n} \sum_{i=1}^{n} \ell(h_{\mathrm{thr}}(\bar{a}(\boldsymbol{x}_i); \boldsymbol{t}), y_i).$*

*Proof of Theorem 6.* First, we prove 'statement($j$)' that, for each $k \in [K]$, $L_{j,k}$ is the minimum task risk for a task such that 1DTs $\{\bar{a}(\boldsymbol{x}_i) \mid \bar{a}(\boldsymbol{x}_i) = \bar{a}_1\}, \ldots, \{\bar{a}(\boldsymbol{x}_i) \mid \bar{a}(\boldsymbol{x}_i) = \bar{a}_{j-1}\}$ are predicted as any of $1, \ldots, k$ in a non-decreasing manner, and 1DTs $\{\bar{a}(\boldsymbol{x}_i) \mid \bar{a}(\boldsymbol{x}_i) = \bar{a}_j\}$ are predicted as $k$:

$$L_{j,k} = \min_{\substack{h_1,\ldots,h_j \in [k] \\ \mathrm{s.t.}\ h_1 \leq \cdots \leq h_j = k}} \sum_{l \in [j]} \sum_{y_m \in \mathcal{Y}_l} \ell(h_l, y_m) \tag{45}$$

The statement(1), which is the starting point for mathematical induction, is trivial. Also, according to the equation,

$$
\begin{aligned}
L_{j+1,k} &= \min_{l \in [k]} L_{j,l} + \sum_{y_i \in \mathcal{Y}_{j+1}} \ell(k, y_i) \\
&= \min_{l \in [k]} \left( \min_{\substack{h_1,\ldots,h_j \in [l] \\ \mathrm{s.t.}\ h_1 \leq \cdots \leq h_j = l}} \sum_{l \in [j]} \sum_{y_m \in \mathcal{Y}_l} \ell(h_l, y_m) \right) + \left( \sum_{y_i \in \mathcal{Y}_{j+1}} \ell(k, y_i) \right) \\
&= \min_{\substack{h_1,\ldots,h_{j+1} \in [k] \\ \mathrm{s.t.}\ h_1 \leq \cdots \leq h_j = l^* \leq h_{j+1} = k}} \sum_{l \in [j+1]} \sum_{y_m \in \mathcal{Y}_l} \ell(h_l, y_m) \text{ with } l^* = \arg\min_{l \in [k]} \left( \min_{\substack{h_1,\ldots,h_j \in [l] \\ \mathrm{s.t.}\ h_1 \leq \cdots \leq h_j = l}} \sum_{l \in [j]} \sum_{y_m \in \mathcal{Y}_l} \ell(h_l, y_m) \right) \\
&= \min_{\substack{h_1,\ldots,h_{j+1} \in [k] \\ \mathrm{s.t.}\ h_1 \leq \cdots \leq h_{j+1} = k}} \sum_{l \in [j+1]} \sum_{y_m \in \mathcal{Y}_l} \ell(h_l, y_m),
\end{aligned}
\tag{46}
$$

one can find that the statement($j+1$) holds with $j \geq 1$ as well.

The statement($N$) shows that 1DTs $\{\bar{a}(\boldsymbol{x}_i) \mid \bar{a}(\boldsymbol{x}_i) = \bar{a}_N\}$ should be labeled as $\min(\arg\min_{l\in[K]} L_{N,l}) \coloneqq M$. Also, for

$$(\bar{h}_1, \ldots, \bar{h}_N) \in \arg\min_{\substack{h_1,\ldots,h_N \in [M] \\ \mathrm{s.t.}\ h_1 \leq \cdots \leq h_N = M}} \sum_{l \in [N]} \sum_{y_m \in \mathcal{Y}_l} \ell(h_l, y_m), \tag{47}$$

it will also be clear that 1DTs $\{\bar{a}(\boldsymbol{x}_i) \mid \bar{a}(\boldsymbol{x}_i) = \bar{a}_1\}, \ldots, \{\bar{a}(\boldsymbol{x}_i) \mid \bar{a}(\boldsymbol{x}_i) = \bar{a}_{N-1}\}$ should be labeled as $\bar{h}_1, \ldots, \bar{h}_{N-1}$. The index $I$ or $J$ in Lines 9–15 tracks $\bar{h}_N (= M), \bar{h}_{N-1}, \ldots, \bar{h}_1$. Therefore, it can be found that the obtained threshold parameters are optimal. □

Table 5: Dataset properties, the total sample size $n_{\text{tot}}$, and the dimension $d$ of the explanatory variables, of classes of the target variable, of the benchmark datasets used for the experiments in Appendix D. Note that AMP originally has 6 missing values and we excluded them.

| | DIA | PYR | APR | SER | TRI | WBC | CPU | AMP | BOS | STO |
|---|---|---|---|---|---|---|---|---|---|---|
| $n_{\text{tot}}$ | 43 | 74 | 159 | 167 | 186 | 194 | 209 | 392 | 506 | 950 |
| $d$ | 2 | 27 | 15 | 4 | 60 | 32 | 6 | 7 | 13 | 9 |

| | ABA | AI2 | KRA | CO1 | PU1 | BA1 | CO2 | PU2 | BA2 | EL2 |
|---|---|---|---|---|---|---|---|---|---|---|
| $n_{\text{tot}}$ | 4177 | 7129 | 8192 | 8192 | 8192 | 8912 | 8912 | 8192 | 8192 | 9517 |
| $d$ | 8 | 5 | 8 | 8 | 8 | 12 | 21 | 32 | 32 | 6 |

| | POT | AI1 | EL1 | CAL | CE1 | CE2 | 2DP | FRA | MVA |
|---|---|---|---|---|---|---|---|---|---|
| $n_{\text{tot}}$ | 13750 | 15000 | 16599 | 20640 | 22784 | 22784 | 40768 | 40768 | 40768 |
| $d$ | 40 | 48 | 18 | 8 | 8 | 16 | 10 | 10 | 10 |

## D   Additional Experiments with Benchmark Datasets

**Purpose**   Many OR studies use datasets, which are generated by discretizing a real-valued target of benchmark datasets commonly used for evaluation in a regression task, in their experiments. For example, Frank & Hall (2001) applied regression benchmark datasets summarized in `https://www.dcc.fc.up.pt/~ltorgo/Regression/DataSets.html`, and experimented with datasets generated via the discretization into 3/5/10 equal-frequency bins. Also, Chu & Ghahramani (2005) tried the discretization into 5/10 equal-length bins in addition to the 5/10 equal-frequency discretization. Therefore, we performed similar experiments additionally to enforce our claim, and this section describes results of these experiments.

**Settings**   In the experiments, as regression benchmark datasets, we use 29 datasets, DIA (Diabetes), PYR (Pyrimidines), APR (Auto Price), SER (Servo), TRI (Triazines), WBC (Wisconsin Breast Cancer), CPU (Machine CPU), AMP (Auto MPG), BOS (Boston Housing), STO (Stocks Domain), ABA (Abalone), AI2 (Delta Ailerons), KRA (Kinematics of Robot Arm), CO1 (Computer Activity (1)), PU1 (Pumadyn Domain (1)), BA1 (Bank Domain (1)), CO2 (Computer Activity (2)), PU2 (Pumadyn Domain (2)), BA2 (Bank Domain (2)), EL2 (Delta Elevators), POT (Pole Telecomm), AI1 (Ailerons), EL1 (Elevators), CAL (California Housing), CE1 (Census (1)), CE2 (Census (2)), 2DP (2D Planes), FRA (Friedman Artificial), MVA (MV Artificial), which are obtainable in `https://www.dcc.fc.up.pt/~ltorgo/Regression/DataSets.html`, a researchers' site of Chu & Ghahramani (2005) (`http://www.gatsby.ucl.ac.uk/~chuwei/ordinalregression.html`), our GitHub repository (`https://github.com/yamasakiryoya/OTL`); see also Table 5. As the discretization manner, we tried 3/5/10 equal-frequency/length discretization; we denote these generated datasets, for example, as EF3 and EL10 datasets. We adopted the same neural network model applied for the RW datasets. All other settings are the same as those in Section 6.

**Results**   Tables 6–23 show the mean and standard deviation of the test errors. Table 24 summarizes all the results: the column 'SUM' shows that the EOT labeling was superior to existing other labelings for all learning methods and tasks, and reinforces our claim regarding the usefulness of the EOT labeling.

Table 6: A counterpart of Table 2 regarding MZE for Task-Z and EF3 datasets.

| Learning | Labeling | DIA | PYR | APR | SER | TRI | WBC | CPU | AMP | BOS | STO |
|---|---|---|---|---|---|---|---|---|---|---|---|
| AD | NNT | $.558_{163}$ | $.388_{119}$ | $.150_{060}$ | $.206_{076}$ | $.513_{090}$ | $.596_{082}$ | $.261_{062}$ | $.204_{045}$ | $.250_{034}$ | $.079_{017}$ |
| | EOT | $.576_{161}$ | $.372_{129}$ | $.145_{065}$ | $.198_{070}$ | $.518_{076}$ | $.589_{080}$ | $.260_{062}$ | $\mathbf{.185_{041}}$ | $.248_{038}$ | $.080_{023}$ |
| Hinge-IT non-ordered | MT | $.534_{161}$ | $.388_{107}$ | $.159_{060}$ | $.198_{070}$ | $.501_{089}$ | $.580_{081}$ | $.264_{063}$ | $.206_{048}$ | $.245_{038}$ | $.079_{018}$ |
| | ST | $.534_{161}$ | $.388_{107}$ | $.159_{066}$ | $.198_{070}$ | $.501_{089}$ | $.580_{081}$ | $.264_{063}$ | $.206_{048}$ | $.245_{036}$ | $.079_{018}$ |
| | EOT | $.568_{135}$ | $.379_{110}$ | $.154_{048}$ | $.208_{067}$ | $.488_{088}$ | $.579_{077}$ | $.254_{055}$ | $\mathbf{.184_{041}}$ | $.247_{038}$ | $.075_{018}$ |
| Hinge-IT ordered | MT,ST | $.536_{153}$ | $.391_{102}$ | $.152_{052}$ | $.211_{070}$ | $.511_{081}$ | $.583_{090}$ | $.256_{065}$ | $.197_{043}$ | $.244_{039}$ | $.079_{017}$ |
| | EOT | $.544_{127}$ | $.365_{116}$ | $.148_{052}$ | $.199_{061}$ | $.510_{083}$ | $.575_{082}$ | $.256_{060}$ | $.190_{037}$ | $.250_{036}$ | $.076_{019}$ |
| Hinge-AT ordered | MT,ST | $.582_{155}$ | $.388_{100}$ | $.152_{060}$ | $.207_{068}$ | $.497_{093}$ | $.595_{086}$ | $.252_{063}$ | $.201_{045}$ | $.243_{037}$ | $.077_{017}$ |
| | EOT | $.554_{140}$ | $.363_{108}$ | $.147_{053}$ | $.212_{065}$ | $.505_{089}$ | $.584_{099}$ | $.265_{062}$ | $.193_{040}$ | $.245_{037}$ | $.077_{020}$ |
| Logistic-IT non-ordered | MT | $.538_{167}$ | $.384_{114}$ | $.158_{066}$ | $.217_{060}$ | $.483_{092}$ | $.587_{094}$ | $.242_{063}$ | $.196_{047}$ | $.241_{036}$ | $.077_{018}$ |
| | ST | $.538_{167}$ | $.384_{114}$ | $.158_{066}$ | $.217_{060}$ | $.483_{092}$ | $.587_{094}$ | $.242_{063}$ | $.196_{047}$ | $.241_{036}$ | $.077_{018}$ |
| | EOT | $.538_{171}$ | $.385_{119}$ | $.146_{047}$ | $.215_{047}$ | $.481_{079}$ | $.568_{088}$ | $.247_{061}$ | $\mathbf{.178_{039}}$ | $.234_{037}$ | $.081_{022}$ |
| Logistic-IT ordered | MT,ST | $.536_{156}$ | $.400_{099}$ | $.153_{068}$ | $.229_{082}$ | $.475_{081}$ | $.571_{092}$ | $.255_{070}$ | $.195_{044}$ | $.232_{038}$ | $.073_{018}$ |
| | EOT | $.550_{158}$ | $.377_{095}$ | $.134_{044}$ | $.219_{060}$ | $.486_{082}$ | $.559_{082}$ | $.252_{065}$ | $.183_{033}$ | $.229_{040}$ | $.074_{019}$ |
| Logistic-AT ordered | MT,ST | $.542_{148}$ | $.397_{106}$ | $.149_{063}$ | $.221_{071}$ | $.487_{079}$ | $.584_{085}$ | $.249_{061}$ | $.191_{045}$ | $.220_{032}$ | $.074_{018}$ |
| | LB | $.580_{159}$ | $.395_{104}$ | $.148_{064}$ | $.218_{064}$ | $.491_{081}$ | $.575_{088}$ | $.245_{063}$ | $.192_{038}$ | $.221_{034}$ | $.075_{019}$ |
| | EOT | $.536_{167}$ | $.385_{115}$ | $.136_{046}$ | $.215_{063}$ | $.503_{082}$ | $.580_{088}$ | $.252_{065}$ | $.181_{038}$ | $.227_{040}$ | $.076_{016}$ |
| OLR-NLL ordered | MT,ST | $.552_{175}$ | $.415_{104}$ | $.155_{064}$ | $.214_{063}$ | $.489_{089}$ | $.576_{095}$ | $.250_{067}$ | $.190_{040}$ | $.228_{036}$ | $.073_{020}$ |
| | LB | $.590_{136}$ | $.388_{105}$ | $.146_{064}$ | $.218_{067}$ | $.487_{096}$ | $.591_{095}$ | $.245_{071}$ | $.197_{049}$ | $.223_{035}$ | $.073_{020}$ |
| | EOT | $.560_{172}$ | $.392_{105}$ | $.136_{049}$ | $.212_{065}$ | $.486_{088}$ | $.569_{103}$ | $.253_{069}$ | $.177_{037}$ | $.227_{039}$ | $.076_{018}$ |

| Learning | Labeling | ABA | AI2 | KRA | CO1 | PU1 | BA1 | CO2 | PU2 | BA2 | EL2 |
|---|---|---|---|---|---|---|---|---|---|---|---|
| AD | NNT | $.351_{018}$ | $.350_{012}$ | $.179_{008}$ | $.171_{008}$ | $.321_{012}$ | $.107_{008}$ | $.200_{010}$ | $.167_{008}$ | $.339_{011}$ | $.350_{010}$ |
| | EOT | $.351_{016}$ | $.350_{011}$ | $\mathbf{.176_{009}}$ | $.171_{008}$ | $.323_{013}$ | $.107_{008}$ | $.199_{009}$ | $.166_{008}$ | $.340_{010}$ | $.348_{010}$ |
| Hinge-IT non-ordered | MT | $.347_{015}$ | $.353_{011}$ | $.157_{010}$ | $.168_{008}$ | $.318_{011}$ | $.102_{008}$ | $.201_{011}$ | $.163_{009}$ | $.340_{010}$ | $.347_{011}$ |
| | ST | $.347_{015}$ | $.353_{011}$ | $.157_{010}$ | $.168_{008}$ | $.318_{011}$ | $.102_{008}$ | $.201_{011}$ | $.163_{009}$ | $.340_{010}$ | $.347_{011}$ |
| | EOT | $.350_{017}$ | $.350_{010}$ | $.156_{009}$ | $.167_{009}$ | $.318_{010}$ | $.102_{008}$ | $.198_{010}$ | $\mathbf{.160_{009}}$ | $.339_{010}$ | $.347_{010}$ |
| Hinge-IT ordered | MT,ST | $.349_{017}$ | $.354_{011}$ | $.157_{010}$ | $.169_{008}$ | $.321_{011}$ | $.101_{008}$ | $.200_{009}$ | $.159_{009}$ | $.340_{010}$ | $.348_{011}$ |
| | EOT | $.349_{016}$ | $\mathbf{.350_{011}}$ | $.157_{009}$ | $.168_{008}$ | $.318_{010}$ | $.101_{008}$ | $.198_{010}$ | $.158_{009}$ | $.338_{010}$ | $.348_{010}$ |
| Hinge-AT ordered | MT,ST | $.358_{017}$ | $.352_{011}$ | $.156_{010}$ | $.169_{008}$ | $.322_{012}$ | $.101_{008}$ | $.199_{010}$ | $.160_{008}$ | $.341_{011}$ | $.350_{010}$ |
| | EOT | $\mathbf{.350_{017}}$ | $.352_{011}$ | $.158_{011}$ | $.168_{008}$ | $.320_{010}$ | $.100_{008}$ | $.198_{009}$ | $.158_{009}$ | $.339_{010}$ | $.347_{011}$ |
| Logistic-IT non-ordered | MT | $.347_{017}$ | $.348_{010}$ | $.154_{009}$ | $.166_{009}$ | $.319_{011}$ | $.101_{007}$ | $.195_{008}$ | $.174_{009}$ | $.342_{011}$ | $.348_{011}$ |
| | ST | $.347_{017}$ | $.348_{010}$ | $.154_{009}$ | $.166_{009}$ | $.319_{011}$ | $.101_{007}$ | $.195_{008}$ | $.174_{009}$ | $.342_{011}$ | $.348_{011}$ |
| | EOT | $.347_{015}$ | $.347_{010}$ | $.153_{008}$ | $.166_{008}$ | $.319_{009}$ | $.101_{008}$ | $.195_{008}$ | $\mathbf{.157_{008}}$ | $.340_{011}$ | $.347_{010}$ |
| Logistic-IT ordered | MT,ST | $.349_{017}$ | $.347_{010}$ | $.153_{010}$ | $.167_{009}$ | $.318_{010}$ | $.101_{007}$ | $.196_{009}$ | $.160_{009}$ | $.342_{011}$ | $.349_{012}$ |
| | EOT | $.348_{016}$ | $.348_{010}$ | $.152_{008}$ | $.167_{008}$ | $.319_{010}$ | $.100_{008}$ | $.194_{009}$ | $.159_{009}$ | $.339_{011}$ | $.346_{011}$ |
| Logistic-AT ordered | MT,ST | $.348_{017}$ | $.347_{010}$ | $.154_{009}$ | $.167_{009}$ | $.319_{010}$ | $.101_{008}$ | $.197_{007}$ | $.158_{009}$ | $.342_{012}$ | $.349_{010}$ |
| | LB | $.351_{017}$ | $.348_{010}$ | $.154_{010}$ | $.167_{008}$ | $.319_{009}$ | $.101_{008}$ | $.197_{009}$ | $.158_{010}$ | $.342_{013}$ | $.351_{011}$ |
| | EOT | $.350_{016}$ | $.349_{010}$ | $.153_{010}$ | $.168_{009}$ | $.319_{009}$ | $.100_{008}$ | $.196_{008}$ | $.156_{009}$ | $.343_{015}$ | $.348_{011}$ |
| OLR-NLL ordered | MT,ST | $.348_{017}$ | $.348_{010}$ | $.152_{010}$ | $.167_{009}$ | $.318_{010}$ | $.101_{007}$ | $.195_{008}$ | $.158_{008}$ | $.341_{012}$ | $.349_{011}$ |
| | LB | $.351_{017}$ | $.347_{010}$ | $.152_{009}$ | $.166_{009}$ | $.318_{010}$ | $.101_{007}$ | $.196_{008}$ | $.158_{009}$ | $.340_{011}$ | $.350_{011}$ |
| | EOT | $.349_{015}$ | $.349_{010}$ | $.153_{010}$ | $.167_{008}$ | $.318_{011}$ | $.100_{008}$ | $.194_{007}$ | $.156_{009}$ | $.341_{012}$ | $.347_{011}$ |

| Learning | Labeling | POT | AI1 | EL1 | CAL | CE1 | CE2 | 2DP | FRA | MVA |
|---|---|---|---|---|---|---|---|---|---|---|
| AD | NNT | $.315_{006}$ | $.243_{007}$ | $.300_{009}$ | $.258_{006}$ | $.292_{007}$ | $.267_{006}$ | $.134_{003}$ | $.119_{003}$ | $.010_{008}$ |
| | EOT | $.315_{008}$ | $.243_{008}$ | $.299_{008}$ | $.257_{006}$ | $.290_{007}$ | $.266_{006}$ | $.134_{003}$ | $.118_{003}$ | $.009_{008}$ |
| Hinge-IT non-ordered | MT | $.365_{015}$ | $.239_{008}$ | $.298_{007}$ | $.248_{006}$ | $.325_{006}$ | $.301_{008}$ | $.149_{008}$ | $.121_{004}$ | $.007_{001}$ |
| | ST | $.365_{015}$ | $.239_{008}$ | $.298_{007}$ | $.248_{006}$ | $.325_{006}$ | $.301_{008}$ | $.149_{008}$ | $.121_{004}$ | $.007_{001}$ |
| | EOT | $\mathbf{.359_{012}}$ | $.241_{008}$ | $.297_{008}$ | $.247_{006}$ | $\mathbf{.321_{006}}$ | $\mathbf{.296_{007}}$ | $\mathbf{.137_{003}}$ | $\mathbf{.120_{004}}$ | $\mathbf{.007_{001}}$ |
| Hinge-IT ordered | MT,ST | $.311_{008}$ | $.239_{008}$ | $.298_{008}$ | $.248_{007}$ | $.288_{008}$ | $.266_{006}$ | $.134_{003}$ | $.120_{004}$ | $.007_{001}$ |
| | EOT | $.311_{007}$ | $.240_{008}$ | $.297_{007}$ | $.247_{007}$ | $.287_{007}$ | $.265_{006}$ | $.133_{003}$ | $\mathbf{.119_{004}}$ | $\mathbf{.007_{001}}$ |
| Hinge-AT ordered | MT,ST | $.314_{008}$ | $.239_{008}$ | $.299_{008}$ | $.252_{007}$ | $.290_{007}$ | $.269_{007}$ | $.134_{003}$ | $.120_{004}$ | $.009_{008}$ |
| | EOT | $.314_{008}$ | $.239_{007}$ | $.297_{007}$ | $.252_{007}$ | $.288_{007}$ | $\mathbf{.267_{007}}$ | $.134_{003}$ | $.119_{004}$ | $.008_{007}$ |
| Logistic-IT non-ordered | MT | $.313_{008}$ | $.235_{007}$ | $.296_{008}$ | $.238_{007}$ | $.282_{006}$ | $.290_{006}$ | $.133_{003}$ | $.121_{004}$ | $.006_{001}$ |
| | ST | $.313_{008}$ | $.235_{007}$ | $.296_{008}$ | $.238_{007}$ | $.282_{006}$ | $.290_{006}$ | $.133_{003}$ | $.121_{004}$ | $.006_{001}$ |
| | EOT | $.311_{007}$ | $.235_{008}$ | $.296_{008}$ | $.237_{007}$ | $.281_{007}$ | $\mathbf{.286_{006}}$ | $.133_{003}$ | $\mathbf{.119_{004}}$ | $.005_{001}$ |
| Logistic-IT ordered | MT,ST | $.313_{008}$ | $.235_{008}$ | $.297_{007}$ | $.239_{007}$ | $.282_{006}$ | $.263_{007}$ | $.133_{004}$ | $.116_{004}$ | $.009_{012}$ |
| | EOT | $.311_{007}$ | $.235_{008}$ | $.296_{008}$ | $.238_{007}$ | $.281_{007}$ | $.262_{007}$ | $.133_{003}$ | $.116_{004}$ | $\mathbf{.008_{012}}$ |
| Logistic-AT ordered | MT,ST | $.312_{007}$ | $.235_{008}$ | $.298_{008}$ | $.241_{008}$ | $.282_{006}$ | $.297_{008}$ | $.133_{003}$ | $.119_{004}$ | $.021_{024}$ |
| | LB | $.312_{007}$ | $.236_{008}$ | $.298_{008}$ | $.241_{007}$ | $.283_{006}$ | $.297_{008}$ | $.133_{003}$ | $.120_{004}$ | $.021_{024}$ |
| | EOT | $.310_{007}$ | $.235_{007}$ | $.298_{008}$ | $.240_{007}$ | $.280_{006}$ | $.295_{007}$ | $.133_{003}$ | $.119_{004}$ | $.020_{023}$ |
| OLR-NLL ordered | MT,ST | $.313_{007}$ | $.235_{007}$ | $.298_{008}$ | $.239_{007}$ | $.281_{007}$ | $.290_{007}$ | $.134_{003}$ | $.119_{004}$ | $.017_{022}$ |
| | LB | $.312_{007}$ | $.235_{008}$ | $.297_{008}$ | $.239_{007}$ | $.282_{006}$ | $.290_{007}$ | $.134_{003}$ | $.119_{004}$ | $.017_{022}$ |
| | EOT | $.311_{007}$ | $.236_{008}$ | $.298_{008}$ | $.239_{007}$ | $.281_{007}$ | $.288_{006}$ | $.133_{003}$ | $.118_{004}$ | $\mathbf{.016_{021}}$ |

Table 7: A counterpart of Table 2 regarding MZE for Task-Z and EF5 datasets.

| Learning | Labeling | DIA | PYR | APR | SER | TRI | WBC | CPU | AMP | BOS | STO |
|---|---|---|---|---|---|---|---|---|---|---|---|
| AD | NNT | $.712_{146}$ | $.568_{132}$ | $.397_{089}$ | $.326_{079}$ | $.677_{065}$ | $.757_{061}$ | $.445_{072}$ | $.319_{048}$ | $.347_{043}$ | $.179_{025}$ |
| | EOT | $.688_{126}$ | $.547_{152}$ | $.369_{068}$ | $.322_{075}$ | $.685_{062}$ | $.741_{062}$ | $.445_{065}$ | $.318_{044}$ | $.343_{042}$ | $.176_{023}$ |
| Hinge-IT non-ordered | MT | $.694_{155}$ | $.597_{114}$ | $.418_{087}$ | $.355_{110}$ | $.675_{069}$ | $.751_{080}$ | $.451_{087}$ | $.311_{051}$ | $.349_{047}$ | $.207_{039}$ |
| | ST | $.696_{154}$ | $.597_{114}$ | $.418_{087}$ | $.355_{110}$ | $.675_{069}$ | $.751_{080}$ | $.451_{087}$ | $.311_{051}$ | $.349_{047}$ | $.207_{039}$ |
| | EOT | $.668_{127}$ | $.587_{119}$ | $\mathbf{.385_{070}}$ | $.338_{085}$ | $.675_{064}$ | $.760_{073}$ | $.448_{076}$ | $.315_{051}$ | $.353_{043}$ | $\mathbf{.158_{026}}$ |
| Hinge-IT ordered | MT,ST | $.666_{153}$ | $.588_{134}$ | $.403_{077}$ | $.323_{078}$ | $.685_{068}$ | $.751_{079}$ | $.431_{075}$ | $.307_{047}$ | $.333_{042}$ | $.155_{026}$ |
| | EOT | $.658_{131}$ | $.571_{133}$ | $.379_{083}$ | $.328_{075}$ | $.679_{074}$ | $.749_{061}$ | $.447_{079}$ | $.315_{051}$ | $.345_{044}$ | $.154_{026}$ |
| Hinge-AT ordered | MT,ST | $.728_{159}$ | $.569_{135}$ | $.397_{091}$ | $.330_{088}$ | $.676_{079}$ | $.780_{063}$ | $.422_{078}$ | $.310_{051}$ | $.340_{046}$ | $.156_{025}$ |
| | EOT | $.662_{132}$ | $.557_{129}$ | $.383_{081}$ | $.325_{091}$ | $.685_{083}$ | $\mathbf{.750_{073}}$ | $.443_{078}$ | $.320_{047}$ | $.345_{042}$ | $.151_{025}$ |
| Logistic-IT non-ordered | MT | $.690_{157}$ | $.593_{127}$ | $.417_{082}$ | $.366_{117}$ | $.662_{078}$ | $.765_{070}$ | $.447_{070}$ | $.323_{047}$ | $.353_{040}$ | $.164_{028}$ |
| | ST | $.692_{151}$ | $.593_{127}$ | $.417_{082}$ | $.366_{117}$ | $.662_{078}$ | $.765_{070}$ | $.447_{070}$ | $.323_{047}$ | $.353_{040}$ | $.164_{028}$ |
| | EOT | $.668_{138}$ | $.575_{112}$ | $\mathbf{.385_{075}}$ | $\mathbf{.320_{078}}$ | $.672_{080}$ | $.751_{063}$ | $.454_{077}$ | $.305_{049}$ | $.352_{049}$ | $\mathbf{.152_{026}}$ |
| Logistic-IT ordered | MT,ST | $.656_{142}$ | $.581_{119}$ | $.405_{082}$ | $.307_{082}$ | $.683_{080}$ | $.761_{076}$ | $.440_{069}$ | $.308_{056}$ | $.345_{049}$ | $.136_{024}$ |
| | EOT | $.664_{120}$ | $.577_{118}$ | $.385_{086}$ | $.311_{071}$ | $.668_{071}$ | $.746_{076}$ | $.451_{072}$ | $.310_{046}$ | $.341_{048}$ | $.136_{025}$ |
| Logistic-AT ordered | MT,ST | $\mathbf{.656_{149}}$ | $.547_{137}$ | $.423_{085}$ | $.309_{065}$ | $.677_{070}$ | $\mathbf{.737_{075}}$ | $.443_{083}$ | $.298_{045}$ | $.330_{044}$ | $.136_{022}$ |
| | LB | $.730_{146}$ | $.565_{150}$ | $.418_{085}$ | $.304_{063}$ | $.671_{074}$ | $.765_{065}$ | $.422_{077}$ | $.305_{048}$ | $.335_{051}$ | $.136_{023}$ |
| | EOT | $\mathbf{.660_{131}}$ | $.557_{142}$ | $.399_{072}$ | $.299_{070}$ | $.687_{074}$ | $\mathbf{.735_{069}}$ | $.455_{085}$ | $.308_{041}$ | $.342_{051}$ | $.139_{025}$ |
| OLR-NLL ordered | MT,ST | $.664_{135}$ | $.569_{116}$ | $.404_{089}$ | $.329_{084}$ | $.663_{072}$ | $\mathbf{.743_{075}}$ | $.443_{081}$ | $.299_{048}$ | $.331_{043}$ | $.134_{022}$ |
| | LB | $.718_{142}$ | $.585_{106}$ | $.396_{088}$ | $.308_{066}$ | $.652_{067}$ | $.773_{061}$ | $.439_{083}$ | $.303_{046}$ | $.337_{042}$ | $.136_{022}$ |
| | EOT | $.684_{125}$ | $.575_{129}$ | $.382_{080}$ | $.308_{078}$ | $.664_{075}$ | $\mathbf{.740_{068}}$ | $.448_{080}$ | $.309_{044}$ | $.342_{044}$ | $.134_{023}$ |

| Learning | Labeling | ABA | AI2 | KRA | CO1 | PU1 | BA1 | CO2 | PU2 | BA2 | EL2 |
|---|---|---|---|---|---|---|---|---|---|---|---|
| AD | NNT | $.533_{017}$ | $.489_{014}$ | $.292_{011}$ | $.277_{011}$ | $.483_{010}$ | $.206_{010}$ | $.331_{011}$ | $.271_{009}$ | $.526_{012}$ | $.555_{012}$ |
| | EOT | $\mathbf{.517_{017}}$ | $\mathbf{.479_{011}}$ | $.292_{013}$ | $.277_{011}$ | $.482_{008}$ | $.204_{011}$ | $.331_{011}$ | $.268_{012}$ | $\mathbf{.522_{009}}$ | $\mathbf{.547_{011}}$ |
| Hinge-IT non-ordered | MT | $.517_{016}$ | $.484_{012}$ | $.279_{012}$ | $.277_{011}$ | $.495_{013}$ | $.205_{009}$ | $.331_{011}$ | $.274_{010}$ | $.520_{013}$ | $.551_{011}$ |
| | ST | $.517_{016}$ | $.484_{012}$ | $.279_{012}$ | $.277_{011}$ | $.495_{013}$ | $.205_{009}$ | $.331_{011}$ | $.274_{010}$ | $.520_{013}$ | $.551_{011}$ |
| | EOT | $.519_{018}$ | $.481_{010}$ | $.277_{011}$ | $.277_{013}$ | $\mathbf{.485_{010}}$ | $.203_{009}$ | $.329_{011}$ | $\mathbf{.262_{010}}$ | $.521_{011}$ | $\mathbf{.545_{011}}$ |
| Hinge-IT ordered | MT,ST | $.521_{017}$ | $.484_{012}$ | $.279_{013}$ | $.275_{010}$ | $.491_{010}$ | $.203_{009}$ | $.329_{011}$ | $.268_{011}$ | $.524_{013}$ | $.554_{014}$ |
| | EOT | $.516_{016}$ | $\mathbf{.480_{010}}$ | $.279_{013}$ | $.276_{011}$ | $\mathbf{.485_{009}}$ | $.205_{009}$ | $.328_{011}$ | $.265_{010}$ | $.521_{012}$ | $\mathbf{.546_{011}}$ |
| Hinge-AT ordered | MT,ST | $.531_{017}$ | $.503_{013}$ | $.279_{011}$ | $.276_{011}$ | $.489_{010}$ | $.204_{008}$ | $.330_{011}$ | $.263_{010}$ | $.526_{012}$ | $.554_{013}$ |
| | EOT | $\mathbf{.515_{018}}$ | $.478_{010}$ | $.277_{010}$ | $.276_{011}$ | $\mathbf{.486_{010}}$ | $.205_{010}$ | $.328_{012}$ | $.263_{010}$ | $\mathbf{.523_{012}}$ | $\mathbf{.548_{010}}$ |
| Logistic-IT non-ordered | MT | $.514_{017}$ | $.494_{014}$ | $.281_{013}$ | $.278_{013}$ | $.486_{010}$ | $.204_{010}$ | $.330_{010}$ | $.300_{012}$ | $.523_{014}$ | $.549_{013}$ |
| | ST | $.514_{017}$ | $.494_{014}$ | $.281_{013}$ | $.278_{013}$ | $.486_{010}$ | $.204_{010}$ | $.330_{010}$ | $.300_{012}$ | $.523_{014}$ | $.549_{013}$ |
| | EOT | $.514_{019}$ | $\mathbf{.482_{011}}$ | $\mathbf{.274_{012}}$ | $.275_{011}$ | $.486_{010}$ | $.203_{010}$ | $.328_{012}$ | $\mathbf{.268_{010}}$ | $.520_{013}$ | $.546_{011}$ |
| Logistic-IT ordered | MT,ST | $.515_{018}$ | $.493_{011}$ | $.271_{010}$ | $.275_{012}$ | $.484_{009}$ | $.202_{009}$ | $.327_{010}$ | $.268_{012}$ | $.522_{016}$ | $.551_{012}$ |
| | EOT | $.514_{018}$ | $\mathbf{.483_{012}}$ | $.272_{011}$ | $.274_{010}$ | $.485_{009}$ | $.204_{010}$ | $.326_{011}$ | $.268_{010}$ | $.519_{011}$ | $\mathbf{.546_{011}}$ |
| Logistic-AT ordered | MT,ST | $\mathbf{.521_{017}}$ | $.486_{010}$ | $.274_{011}$ | $.275_{011}$ | $.488_{010}$ | $.203_{009}$ | $.329_{011}$ | $.259_{011}$ | $.524_{012}$ | $.551_{011}$ |
| | LB | $.532_{017}$ | $.493_{013}$ | $.273_{011}$ | $.275_{011}$ | $.489_{010}$ | $.203_{009}$ | $.328_{011}$ | $.258_{009}$ | $.527_{011}$ | $.554_{013}$ |
| | EOT | $\mathbf{.521_{016}}$ | $\mathbf{.482_{013}}$ | $.271_{011}$ | $.275_{011}$ | $.491_{011}$ | $.203_{009}$ | $.326_{010}$ | $.258_{009}$ | $.525_{012}$ | $.549_{011}$ |
| OLR-NLL ordered | MT,ST | $\mathbf{.512_{015}}$ | $.485_{013}$ | $.271_{012}$ | $.275_{011}$ | $.486_{008}$ | $.202_{010}$ | $.327_{009}$ | $.260_{011}$ | $.520_{014}$ | $.548_{012}$ |
| | LB | $.526_{016}$ | $.490_{012}$ | $.272_{012}$ | $.275_{010}$ | $.487_{009}$ | $.202_{009}$ | $.328_{010}$ | $.259_{011}$ | $.527_{014}$ | $.553_{014}$ |
| | EOT | $.511_{017}$ | $\mathbf{.479_{012}}$ | $.271_{010}$ | $.274_{011}$ | $.487_{009}$ | $.202_{009}$ | $.325_{011}$ | $.261_{011}$ | $\mathbf{.521_{012}}$ | $.547_{011}$ |

| Learning | Labeling | POT | AI1 | EL1 | CAL | CE1 | CE2 | 2DP | FRA | MVA |
|---|---|---|---|---|---|---|---|---|---|---|
| AD | NNT | $.513_{011}$ | $.397_{009}$ | $.486_{010}$ | $.389_{009}$ | $.502_{007}$ | $.485_{008}$ | $.242_{005}$ | $.220_{005}$ | $.021_{004}$ |
| | EOT | $\mathbf{.486_{013}}$ | $.398_{008}$ | $\mathbf{.479_{008}}$ | $.388_{009}$ | $\mathbf{.497_{007}}$ | $\mathbf{.479_{007}}$ | $\mathbf{.241_{004}}$ | $.219_{005}$ | $\mathbf{.017_{005}}$ |
| Hinge-IT non-ordered | MT | $.474_{010}$ | $.412_{009}$ | $.476_{009}$ | $.448_{008}$ | $.496_{007}$ | $.476_{008}$ | $.245_{005}$ | $.217_{005}$ | $.089_{018}$ |
| | ST | $.474_{010}$ | $.412_{009}$ | $.476_{009}$ | $.448_{008}$ | $.496_{007}$ | $.476_{008}$ | $.245_{005}$ | $.217_{005}$ | $.089_{018}$ |
| | EOT | $.471_{013}$ | $.412_{007}$ | $\mathbf{.472_{009}}$ | $.448_{008}$ | $.492_{007}$ | $\mathbf{.473_{009}}$ | $\mathbf{.241_{005}}$ | $.214_{004}$ | $\mathbf{.083_{014}}$ |
| Hinge-IT ordered | MT,ST | $.469_{008}$ | $.397_{009}$ | $.474_{007}$ | $.447_{008}$ | $.494_{007}$ | $.474_{008}$ | $.246_{004}$ | $.214_{004}$ | $.015_{003}$ |
| | EOT | $.466_{011}$ | $.395_{009}$ | $.471_{008}$ | $.447_{009}$ | $.492_{008}$ | $.472_{008}$ | $\mathbf{.241_{004}}$ | $\mathbf{.213_{004}}$ | $.013_{003}$ |
| Hinge-AT ordered | MT,ST | $.488_{011}$ | $.412_{009}$ | $.482_{010}$ | $.451_{009}$ | $.505_{008}$ | $.484_{008}$ | $.245_{005}$ | $.216_{004}$ | $.072_{022}$ |
| | EOT | $\mathbf{.482_{012}}$ | $.410_{008}$ | $\mathbf{.476_{009}}$ | $.450_{008}$ | $\mathbf{.497_{007}}$ | $\mathbf{.477_{007}}$ | $.240_{004}$ | $\mathbf{.214_{004}}$ | $\mathbf{.065_{020}}$ |
| Logistic-IT non-ordered | MT | $.469_{009}$ | $.414_{008}$ | $.479_{010}$ | $.449_{009}$ | $.497_{008}$ | $.473_{010}$ | $.244_{005}$ | $.218_{006}$ | $.076_{007}$ |
| | ST | $.470_{010}$ | $.414_{008}$ | $.479_{010}$ | $.449_{009}$ | $.497_{008}$ | $.473_{010}$ | $.244_{005}$ | $.218_{006}$ | $.076_{007}$ |
| | EOT | $\mathbf{.465_{009}}$ | $.412_{007}$ | $.477_{010}$ | $.448_{007}$ | $.496_{007}$ | $\mathbf{.470_{007}}$ | $\mathbf{.241_{004}}$ | $.211_{005}$ | $\mathbf{.066_{005}}$ |
| Logistic-IT ordered | MT,ST | $.465_{007}$ | $.395_{009}$ | $.478_{009}$ | $.447_{008}$ | $.497_{008}$ | $.472_{009}$ | $.244_{004}$ | $.212_{004}$ | $.015_{003}$ |
| | EOT | $\mathbf{.462_{010}}$ | $.395_{008}$ | $.475_{009}$ | $.447_{008}$ | $.496_{007}$ | $.469_{008}$ | $\mathbf{.241_{004}}$ | $.211_{004}$ | $\mathbf{.013_{003}}$ |
| Logistic-AT ordered | MT,ST | $.468_{008}$ | $.412_{009}$ | $\mathbf{.480_{010}}$ | $\mathbf{.451_{009}}$ | $\mathbf{.501_{007}}$ | $\mathbf{.484_{009}}$ | $.245_{004}$ | $.211_{005}$ | $.086_{044}$ |
| | LB | $.470_{011}$ | $.412_{008}$ | $.483_{010}$ | $.454_{008}$ | $.505_{008}$ | $.488_{009}$ | $.244_{005}$ | $.212_{006}$ | $.087_{045}$ |
| | EOT | $.460_{010}$ | $.412_{008}$ | $\mathbf{.478_{010}}$ | $.450_{007}$ | $.499_{007}$ | $\mathbf{.482_{007}}$ | $\mathbf{.242_{005}}$ | $.210_{005}$ | $\mathbf{.081_{044}}$ |
| OLR-NLL ordered | MT,ST | $.463_{009}$ | $.411_{009}$ | $.475_{010}$ | $.449_{010}$ | $\mathbf{.496_{006}}$ | $\mathbf{.472_{008}}$ | $.245_{005}$ | $.212_{005}$ | $.071_{036}$ |
| | LB | $.465_{011}$ | $.410_{009}$ | $.477_{008}$ | $.450_{008}$ | $.501_{007}$ | $.478_{008}$ | $.245_{005}$ | $.211_{005}$ | $.071_{036}$ |
| | EOT | $\mathbf{.460_{009}}$ | $.410_{008}$ | $.474_{009}$ | $.448_{009}$ | $\mathbf{.495_{007}}$ | $\mathbf{.471_{007}}$ | $\mathbf{.241_{004}}$ | $.211_{005}$ | $\mathbf{.066_{035}}$ |

Table 8: A counterpart of Table 2 regarding MZE for Task-Z and EF10 datasets.

| Learning | Labeling | DIA | PYR | APR | SER | TRI | WBC | CPU | AMP | BOS | STO |
|---|---|---|---|---|---|---|---|---|---|---|---|
| AD | NNT | $.892_{096}$ | $.728_{110}$ | $.647_{069}$ | $.508_{119}$ | $.843_{057}$ | $.875_{058}$ | $.647_{072}$ | $.581_{066}$ | $.537_{043}$ | $.340_{035}$ |
| | EOT | $.868_{095}$ | $.713_{115}$ | $\mathbf{.622_{073}}$ | $.506_{081}$ | $.838_{063}$ | $.859_{049}$ | $.660_{068}$ | $.596_{051}$ | $.543_{048}$ | $.337_{036}$ |
| Hinge-IT non-ordered | MT | $.920_{077}$ | $.756_{103}$ | $.649_{070}$ | $.568_{070}$ | $.841_{059}$ | $.884_{057}$ | $.673_{074}$ | $.594_{067}$ | $.549_{048}$ | $.372_{044}$ |
| | ST | $.912_{084}$ | $.756_{103}$ | $.641_{076}$ | $.551_{082}$ | $.841_{059}$ | $.879_{050}$ | $.665_{064}$ | $.594_{067}$ | $.549_{048}$ | $.368_{044}$ |
| | EOT | $\mathbf{.866_{091}}$ | $.731_{131}$ | $.639_{073}$ | $.546_{084}$ | $.851_{058}$ | $.876_{058}$ | $.666_{062}$ | $.598_{056}$ | $.561_{045}$ | $\mathbf{.349_{034}}$ |
| Hinge-IT ordered | MT,ST | $.860_{094}$ | $.727_{103}$ | $.636_{074}$ | $.538_{098}$ | $.837_{047}$ | $.868_{053}$ | $.676_{065}$ | $.597_{058}$ | $.553_{040}$ | $.335_{041}$ |
| | EOT | $.868_{076}$ | $.731_{129}$ | $.641_{082}$ | $\mathbf{.504_{086}}$ | $.839_{063}$ | $.860_{045}$ | $.656_{067}$ | $.597_{058}$ | $.563_{047}$ | $.329_{040}$ |
| Hinge-AT ordered | MT,ST | $.894_{114}$ | $.725_{118}$ | $.632_{073}$ | $.506_{104}$ | $.823_{065}$ | $.877_{053}$ | $.643_{073}$ | $.574_{072}$ | $.557_{051}$ | $.319_{033}$ |
| | EOT | $.872_{106}$ | $.727_{122}$ | $.637_{072}$ | $.495_{095}$ | $.831_{061}$ | $.868_{055}$ | $.656_{077}$ | $.586_{066}$ | $.561_{044}$ | $.316_{035}$ |
| Logistic-IT non-ordered | MT | $.906_{099}$ | $.737_{122}$ | $.653_{086}$ | $.559_{104}$ | $.828_{059}$ | $.882_{041}$ | $.661_{083}$ | $.601_{063}$ | $.565_{046}$ | $.355_{048}$ |
| | ST | $.908_{093}$ | $.732_{119}$ | $.658_{081}$ | $.555_{100}$ | $.824_{057}$ | $.880_{038}$ | $.658_{081}$ | $.601_{063}$ | $.565_{046}$ | $.353_{045}$ |
| | EOT | $\mathbf{.868_{101}}$ | $.752_{106}$ | $.641_{082}$ | $.536_{099}$ | $.827_{059}$ | $.871_{045}$ | $.656_{069}$ | $.594_{062}$ | $.564_{044}$ | $.343_{039}$ |
| Logistic-IT ordered | MT,ST | $.856_{098}$ | $.727_{130}$ | $.635_{073}$ | $.552_{103}$ | $.826_{060}$ | $.869_{048}$ | $.659_{071}$ | $.593_{065}$ | $\mathbf{.556_{043}}$ | $.315_{032}$ |
| | EOT | $.872_{083}$ | $.720_{145}$ | $.636_{076}$ | $.535_{095}$ | $.823_{068}$ | $.863_{068}$ | $.658_{069}$ | $.595_{057}$ | $.573_{043}$ | $.320_{038}$ |
| Logistic-AT ordered | MT,ST | $.866_{099}$ | $.739_{119}$ | $.626_{089}$ | $.505_{080}$ | $.836_{063}$ | $.867_{053}$ | $.669_{076}$ | $.585_{055}$ | $.574_{052}$ | $.301_{031}$ |
| | LB | $.878_{090}$ | $.719_{110}$ | $.655_{077}$ | $.491_{088}$ | $.821_{078}$ | $.888_{046}$ | $.636_{074}$ | $.585_{070}$ | $\mathbf{.552_{045}}$ | $.301_{037}$ |
| | EOT | $.862_{106}$ | $.732_{128}$ | $.639_{080}$ | $.492_{073}$ | $.831_{056}$ | $.868_{054}$ | $.651_{063}$ | $.590_{065}$ | $.575_{050}$ | $.311_{038}$ |
| OLR-NLL ordered | MT,ST | $.878_{106}$ | $.751_{105}$ | $.631_{071}$ | $.495_{082}$ | $.827_{060}$ | $.869_{058}$ | $.680_{069}$ | $.577_{059}$ | $.567_{046}$ | $.309_{037}$ |
| | LB | $.910_{078}$ | $.717_{122}$ | $.637_{077}$ | $.492_{084}$ | $.826_{057}$ | $.884_{048}$ | $.651_{071}$ | $.573_{061}$ | $.553_{048}$ | $.308_{035}$ |
| | EOT | $.866_{097}$ | $.736_{118}$ | $.635_{078}$ | $.476_{071}$ | $.829_{069}$ | $.867_{052}$ | $.659_{076}$ | $.585_{057}$ | $.558_{053}$ | $.303_{031}$ |

| Learning | Labeling | ABA | AI2 | KRA | CO1 | PU1 | BA1 | CO2 | PU2 | BA2 | EL2 |
|---|---|---|---|---|---|---|---|---|---|---|---|
| AD | NNT | $.736_{014}$ | $.701_{013}$ | $.494_{013}$ | $.468_{013}$ | $.721_{010}$ | $.403_{014}$ | $.528_{011}$ | $.465_{017}$ | $.743_{011}$ | $.764_{011}$ |
| | EOT | $\mathbf{.722_{017}}$ | $.686_{012}$ | $.493_{015}$ | $.464_{011}$ | $\mathbf{.711_{008}}$ | $.405_{013}$ | $.527_{012}$ | $\mathbf{.458_{016}}$ | $.741_{010}$ | $\mathbf{.736_{010}}$ |
| Hinge-IT non-ordered | MT | $.720_{016}$ | $.702_{013}$ | $.492_{014}$ | $.475_{012}$ | $.713_{009}$ | $.404_{014}$ | $.534_{011}$ | $.457_{011}$ | $.742_{010}$ | $.751_{012}$ |
| | ST | $.719_{016}$ | $.702_{013}$ | $.492_{014}$ | $.475_{012}$ | $.713_{009}$ | $.404_{014}$ | $.534_{011}$ | $.457_{011}$ | $.742_{010}$ | $.751_{012}$ |
| | EOT | $.721_{015}$ | $\mathbf{.690_{013}}$ | $\mathbf{.486_{014}}$ | $.471_{012}$ | $\mathbf{.711_{009}}$ | $.400_{011}$ | $.533_{013}$ | $.456_{010}$ | $.742_{010}$ | $.736_{008}$ |
| Hinge-IT ordered | MT,ST | $.719_{017}$ | $.695_{011}$ | $.490_{015}$ | $.467_{010}$ | $.719_{011}$ | $.399_{013}$ | $.529_{011}$ | $.458_{015}$ | $.741_{009}$ | $.742_{011}$ |
| | EOT | $.724_{019}$ | $\mathbf{.691_{012}}$ | $.490_{013}$ | $.467_{012}$ | $\mathbf{.709_{009}}$ | $.401_{011}$ | $.531_{013}$ | $.458_{012}$ | $.741_{011}$ | $\mathbf{.734_{008}}$ |
| Hinge-AT ordered | MT,ST | $.732_{013}$ | $.716_{013}$ | $.487_{015}$ | $.476_{013}$ | $.725_{011}$ | $.401_{015}$ | $.532_{013}$ | $.449_{014}$ | $.748_{011}$ | $.782_{012}$ |
| | EOT | $\mathbf{.722_{015}}$ | $\mathbf{.686_{013}}$ | $.488_{013}$ | $.473_{012}$ | $\mathbf{.715_{009}}$ | $.400_{012}$ | $.530_{011}$ | $.446_{013}$ | $\mathbf{.743_{010}}$ | $\mathbf{.739_{010}}$ |
| Logistic-IT non-ordered | MT | $.722_{014}$ | $.711_{011}$ | $.487_{012}$ | $.478_{012}$ | $.714_{009}$ | $.404_{012}$ | $.536_{012}$ | $.459_{016}$ | $.743_{011}$ | $.752_{013}$ |
| | ST | $.722_{014}$ | $.711_{011}$ | $.487_{012}$ | $.478_{012}$ | $.714_{009}$ | $.404_{012}$ | $.536_{012}$ | $.459_{016}$ | $.743_{011}$ | $.752_{013}$ |
| | EOT | $.721_{016}$ | $\mathbf{.686_{012}}$ | $.485_{015}$ | $\mathbf{.472_{011}}$ | $\mathbf{.710_{010}}$ | $\mathbf{.400_{011}}$ | $.535_{011}$ | $.456_{013}$ | $\mathbf{.740_{011}}$ | $\mathbf{.738_{009}}$ |
| Logistic-IT ordered | MT,ST | $\mathbf{.717_{016}}$ | $.701_{011}$ | $.494_{014}$ | $.471_{013}$ | $.709_{009}$ | $.397_{013}$ | $.532_{012}$ | $.464_{013}$ | $.742_{012}$ | $.744_{009}$ |
| | EOT | $.725_{016}$ | $\mathbf{.692_{012}}$ | $.490_{014}$ | $.467_{012}$ | $.708_{009}$ | $.399_{013}$ | $.535_{011}$ | $.464_{012}$ | $.741_{010}$ | $\mathbf{.735_{010}}$ |
| Logistic-AT ordered | MT,ST | $\mathbf{.720_{014}}$ | $.696_{011}$ | $.489_{014}$ | $.477_{012}$ | $\mathbf{.711_{008}}$ | $.397_{011}$ | $\mathbf{.531_{009}}$ | $.451_{014}$ | $\mathbf{.743_{014}}$ | $.743_{009}$ |
| | LB | $.734_{015}$ | $.712_{013}$ | $.492_{014}$ | $.480_{012}$ | $.720_{008}$ | $.398_{011}$ | $.538_{013}$ | $.450_{014}$ | $.750_{013}$ | $.767_{010}$ |
| | EOT | $\mathbf{.723_{015}}$ | $\mathbf{.687_{012}}$ | $.489_{015}$ | $.473_{011}$ | $.715_{010}$ | $.399_{011}$ | $\mathbf{.532_{011}}$ | $.447_{014}$ | $\mathbf{.743_{010}}$ | $\mathbf{.738_{009}}$ |
| OLR-NLL ordered | MT,ST | $\mathbf{.718_{016}}$ | $.699_{013}$ | $.482_{015}$ | $.465_{011}$ | $\mathbf{.709_{008}}$ | $.398_{013}$ | $.527_{012}$ | $.450_{012}$ | $\mathbf{.737_{010}}$ | $\mathbf{.739_{009}}$ |
| | LB | $.734_{015}$ | $.708_{012}$ | $.486_{014}$ | $.469_{012}$ | $.717_{011}$ | $.397_{013}$ | $.527_{010}$ | $.449_{014}$ | $.747_{012}$ | $.764_{011}$ |
| | EOT | $\mathbf{.721_{017}}$ | $\mathbf{.687_{012}}$ | $.483_{013}$ | $.466_{014}$ | $\mathbf{.710_{010}}$ | $.398_{012}$ | $.524_{010}$ | $.447_{012}$ | $\mathbf{.742_{011}}$ | $\mathbf{.738_{009}}$ |

| Learning | Labeling | POT | AI1 | EL1 | CAL | CE1 | CE2 | 2DP | FRA | MVA |
|---|---|---|---|---|---|---|---|---|---|---|
| AD | NNT | $.662_{010}$ | $.612_{010}$ | $.676_{008}$ | $.598_{009}$ | $.685_{007}$ | $.665_{008}$ | $.447_{006}$ | $.423_{007}$ | $.189_{069}$ |
| | EOT | $\mathbf{.652_{007}}$ | $.609_{009}$ | $\mathbf{.655_{008}}$ | $.596_{009}$ | $\mathbf{.678_{008}}$ | $\mathbf{.659_{007}}$ | $.446_{005}$ | $.420_{006}$ | $\mathbf{.155_{059}}$ |
| Hinge-IT non-ordered | MT | $.682_{010}$ | $.629_{008}$ | $.661_{008}$ | $.656_{009}$ | $.719_{007}$ | $.705_{007}$ | $.451_{005}$ | $.420_{005}$ | $.300_{072}$ |
| | ST | $.682_{010}$ | $.629_{008}$ | $.661_{008}$ | $.656_{009}$ | $.719_{007}$ | $.705_{007}$ | $.451_{005}$ | $.420_{005}$ | $.300_{072}$ |
| | EOT | $\mathbf{.675_{010}}$ | $.627_{007}$ | $.647_{009}$ | $.652_{008}$ | $\mathbf{.711_{006}}$ | $.698_{006}$ | $\mathbf{.447_{005}}$ | $.416_{006}$ | $\mathbf{.220_{068}}$ |
| Hinge-IT ordered | MT,ST | $.643_{007}$ | $.612_{009}$ | $.650_{009}$ | $.655_{008}$ | $.721_{007}$ | $.710_{009}$ | $.452_{006}$ | $.418_{006}$ | $.140_{059}$ |
| | EOT | $.643_{009}$ | $.614_{009}$ | $.649_{009}$ | $.654_{008}$ | $\mathbf{.715_{007}}$ | $.703_{005}$ | $\mathbf{.447_{005}}$ | $.416_{006}$ | $\mathbf{.115_{040}}$ |
| Hinge-AT ordered | MT,ST | $.650_{009}$ | $.637_{009}$ | $.675_{008}$ | $.667_{008}$ | $.721_{007}$ | $.711_{008}$ | $.450_{006}$ | $.417_{006}$ | $.179_{069}$ |
| | EOT | $.646_{009}$ | $\mathbf{.629_{009}}$ | $\mathbf{.656_{008}}$ | $\mathbf{.655_{008}}$ | $\mathbf{.717_{007}}$ | $\mathbf{.704_{008}}$ | $.446_{005}$ | $.414_{005}$ | $\mathbf{.138_{033}}$ |
| Logistic-IT non-ordered | MT | $.689_{010}$ | $.631_{008}$ | $.670_{009}$ | $.664_{008}$ | $.720_{006}$ | $.709_{010}$ | $.448_{006}$ | $.424_{007}$ | $.222_{070}$ |
| | ST | $.688_{010}$ | $.631_{008}$ | $.670_{009}$ | $.664_{008}$ | $.720_{006}$ | $.709_{010}$ | $.448_{006}$ | $.424_{007}$ | $.222_{070}$ |
| | EOT | $\mathbf{.673_{010}}$ | $.627_{006}$ | $.648_{008}$ | $.653_{008}$ | $\mathbf{.712_{006}}$ | $.700_{006}$ | $\mathbf{.447_{005}}$ | $.417_{006}$ | $\mathbf{.182_{058}}$ |
| Logistic-IT ordered | MT,ST | $.647_{008}$ | $.614_{009}$ | $.656_{010}$ | $.607_{008}$ | $.687_{008}$ | $.661_{008}$ | $.448_{006}$ | $.418_{007}$ | $.128_{032}$ |
| | EOT | $\mathbf{.644_{009}}$ | $.614_{009}$ | $.654_{010}$ | $.604_{009}$ | $.686_{007}$ | $.660_{007}$ | $\mathbf{.447_{005}}$ | $\mathbf{.415_{006}}$ | $.114_{030}$ |
| Logistic-AT ordered | MT,ST | $.655_{009}$ | $.634_{008}$ | $.660_{010}$ | $\mathbf{.658_{008}}$ | $\mathbf{.720_{007}}$ | $\mathbf{.707_{010}}$ | $.449_{006}$ | $.418_{006}$ | $.187_{056}$ |
| | LB | $.654_{009}$ | $.638_{011}$ | $.676_{009}$ | $.668_{008}$ | $.723_{007}$ | $.714_{007}$ | $.449_{006}$ | $.417_{006}$ | $.194_{064}$ |
| | EOT | $\mathbf{.649_{009}}$ | $\mathbf{.629_{008}}$ | $\mathbf{.657_{008}}$ | $.656_{007}$ | $\mathbf{.719_{006}}$ | $.706_{007}$ | $\mathbf{.446_{005}}$ | $.415_{005}$ | $\mathbf{.156_{045}}$ |
| OLR-NLL ordered | MT,ST | $.675_{009}$ | $\mathbf{.628_{008}}$ | $\mathbf{.647_{009}}$ | $.651_{008}$ | $\mathbf{.714_{007}}$ | $\mathbf{.695_{007}}$ | $.448_{006}$ | $.416_{005}$ | $.164_{068}$ |
| | LB | $.688_{009}$ | $.634_{009}$ | $.670_{008}$ | $.664_{009}$ | $.719_{007}$ | $.703_{007}$ | $.448_{006}$ | $.416_{005}$ | $.166_{069}$ |
| | EOT | $\mathbf{.666_{009}}$ | $\mathbf{.628_{008}}$ | $\mathbf{.648_{007}}$ | $.652_{008}$ | $.714_{006}$ | $\mathbf{.695_{006}}$ | $.447_{005}$ | $.415_{005}$ | $\mathbf{.145_{059}}$ |

Table 9: A counterpart of Table 2 regarding MZE for Task-Z and EL3 datasets.

| Learning | Labeling | DIA | PYR | APR | SER | TRI | WBC | CPU | AMP | BOS | STO |
|---|---|---|---|---|---|---|---|---|---|---|---|
| AD | NNT | $\mathbf{.330_{140}}$ | $.183_{121}$ | $.139_{065}$ | $.070_{038}$ | $.326_{076}$ | $.527_{076}$ | $.046_{033}$ | $.199_{041}$ | $.222_{053}$ | $.041_{017}$ |
| | EOT | $.386_{144}$ | $.175_{101}$ | $.122_{060}$ | $.067_{042}$ | $.319_{074}$ | $.523_{085}$ | $.045_{033}$ | $\mathbf{.177_{042}}$ | $\mathbf{.200_{039}}$ | $.043_{016}$ |
| Hinge-IT non-ordered | MT | $.346_{147}$ | $.185_{123}$ | $.147_{057}$ | $.067_{048}$ | $.312_{071}$ | $.510_{062}$ | $.047_{034}$ | $.190_{047}$ | $.197_{041}$ | $.032_{012}$ |
| | ST | $.346_{147}$ | $.185_{123}$ | $.147_{057}$ | $.067_{048}$ | $.312_{071}$ | $.510_{062}$ | $.047_{034}$ | $.190_{047}$ | $.197_{041}$ | $.032_{012}$ |
| | EOT | $.380_{144}$ | $.181_{094}$ | $\mathbf{.122_{048}}$ | $.068_{042}$ | $.336_{082}$ | $.523_{068}$ | $.042_{032}$ | $\mathbf{.167_{043}}$ | $.189_{034}$ | $.036_{012}$ |
| Hinge-IT ordered | MT,ST | $\mathbf{.324_{156}}$ | $.180_{105}$ | $.144_{051}$ | $.070_{040}$ | $.311_{066}$ | $.516_{063}$ | $.048_{034}$ | $.195_{040}$ | $.198_{041}$ | $\mathbf{.031_{012}}$ |
| | EOT | $.382_{149}$ | $.176_{110}$ | $\mathbf{.119_{055}}$ | $.071_{044}$ | $.330_{078}$ | $.527_{074}$ | $.045_{031}$ | $\mathbf{.165_{040}}$ | $.192_{036}$ | $.036_{013}$ |
| Hinge-AT ordered | MT,ST | $.340_{156}$ | $.172_{098}$ | $.147_{059}$ | $.077_{043}$ | $.305_{072}$ | $.526_{085}$ | $.048_{034}$ | $.191_{043}$ | $.207_{051}$ | $\mathbf{.030_{014}}$ |
| | EOT | $.376_{148}$ | $.179_{094}$ | $\mathbf{.119_{048}}$ | $.074_{040}$ | $.318_{065}$ | $.541_{094}$ | $.044_{034}$ | $\mathbf{.163_{042}}$ | $.201_{044}$ | $.034_{011}$ |
| Logistic-IT non-ordered | MT | $.390_{145}$ | $.209_{112}$ | $.142_{065}$ | $.065_{042}$ | $.319_{071}$ | $.547_{065}$ | $.052_{029}$ | $.168_{041}$ | $.185_{046}$ | $.029_{011}$ |
| | ST | $.390_{145}$ | $.209_{112}$ | $.142_{065}$ | $.065_{042}$ | $.319_{071}$ | $.547_{065}$ | $.052_{029}$ | $.168_{041}$ | $.185_{046}$ | $.029_{011}$ |
| | EOT | $.404_{152}$ | $.187_{106}$ | $\mathbf{.120_{059}}$ | $.069_{043}$ | $.319_{073}$ | $.544_{083}$ | $.048_{031}$ | $.160_{043}$ | $.182_{045}$ | $.030_{011}$ |
| Logistic-IT ordered | MT,ST | $.392_{140}$ | $.215_{128}$ | $.142_{055}$ | $.063_{041}$ | $.313_{055}$ | $.522_{078}$ | $.053_{029}$ | $.166_{038}$ | $.162_{034}$ | $.029_{011}$ |
| | EOT | $.426_{137}$ | $.203_{114}$ | $\mathbf{.119_{054}}$ | $.073_{045}$ | $.311_{071}$ | $.548_{070}$ | $.052_{029}$ | $.165_{042}$ | $.173_{042}$ | $.031_{012}$ |
| Logistic-AT ordered | MT,ST | $.438_{143}$ | $.211_{121}$ | $.148_{052}$ | $.069_{040}$ | $.308_{083}$ | $.525_{081}$ | $.055_{029}$ | $.162_{034}$ | $.167_{039}$ | $.030_{009}$ |
| | LB | $.408_{128}$ | $.205_{115}$ | $.143_{056}$ | $.069_{039}$ | $.304_{084}$ | $.540_{082}$ | $.055_{030}$ | $.167_{043}$ | $.171_{039}$ | $.030_{009}$ |
| | EOT | $.408_{141}$ | $.184_{106}$ | $\mathbf{.119_{044}}$ | $.068_{041}$ | $.307_{075}$ | $.544_{070}$ | $.052_{030}$ | $.160_{040}$ | $.178_{039}$ | $.030_{010}$ |
| OLR-NLL ordered | MT,ST | $.426_{148}$ | $.204_{128}$ | $.140_{059}$ | $.067_{044}$ | $.320_{073}$ | $.528_{076}$ | $.054_{029}$ | $.166_{037}$ | $.166_{033}$ | $.031_{013}$ |
| | LB | $.396_{134}$ | $.200_{119}$ | $.135_{057}$ | $.068_{044}$ | $.319_{071}$ | $.537_{074}$ | $.054_{030}$ | $.166_{038}$ | $.174_{046}$ | $.031_{012}$ |
| | EOT | $.418_{134}$ | $.195_{112}$ | $.115_{044}$ | $.072_{045}$ | $.317_{080}$ | $.535_{080}$ | $.051_{030}$ | $.164_{039}$ | $.182_{044}$ | $.030_{010}$ |

| Learning | Labeling | ABA | AI2 | KRA | CO1 | PU1 | BA1 | CO2 | PU2 | BA2 | EL2 |
|---|---|---|---|---|---|---|---|---|---|---|---|
| AD | NNT | $.237_{013}$ | $.030_{004}$ | $.198_{011}$ | $.012_{004}$ | $.276_{010}$ | $.065_{010}$ | $.015_{003}$ | $.126_{008}$ | $.088_{007}$ | $.062_{004}$ |
| | EOT | $.236_{012}$ | $.030_{005}$ | $\mathbf{.193_{014}}$ | $.011_{003}$ | $.276_{010}$ | $\mathbf{.055_{005}}$ | $.015_{003}$ | $\mathbf{.120_{007}}$ | $.082_{005}$ | $.061_{004}$ |
| Hinge-IT non-ordered | MT | $.236_{014}$ | $.030_{004}$ | $.150_{010}$ | $.010_{002}$ | $.270_{010}$ | $.056_{007}$ | $.013_{003}$ | $\mathbf{.110_{009}}$ | $.085_{006}$ | $.062_{004}$ |
| | ST | $.236_{014}$ | $.030_{004}$ | $.150_{010}$ | $.010_{002}$ | $.270_{010}$ | $.056_{007}$ | $.013_{003}$ | $\mathbf{.110_{009}}$ | $.085_{006}$ | $.062_{004}$ |
| | EOT | $.235_{013}$ | $.030_{004}$ | $.149_{011}$ | $.011_{003}$ | $.269_{011}$ | $.055_{006}$ | $.013_{003}$ | $.113_{010}$ | $.083_{005}$ | $.061_{004}$ |
| Hinge-IT ordered | MT,ST | $.236_{014}$ | $.030_{004}$ | $.147_{012}$ | $.011_{002}$ | $.269_{009}$ | $.054_{006}$ | $.013_{002}$ | $\mathbf{.110_{008}}$ | $.085_{005}$ | $.062_{004}$ |
| | EOT | $.235_{012}$ | $.030_{004}$ | $.145_{011}$ | $.011_{002}$ | $.269_{010}$ | $.054_{006}$ | $.013_{003}$ | $.114_{008}$ | $.084_{006}$ | $.061_{004}$ |
| Hinge-AT ordered | MT,ST | $.235_{014}$ | $.030_{004}$ | $.144_{009}$ | $.010_{003}$ | $.269_{010}$ | $.055_{006}$ | $.013_{002}$ | $.115_{010}$ | $.084_{005}$ | $.062_{004}$ |
| | EOT | $.235_{012}$ | $.030_{004}$ | $.143_{010}$ | $.010_{003}$ | $.268_{011}$ | $.054_{005}$ | $.013_{003}$ | $.117_{010}$ | $\mathbf{.083_{006}}$ | $.061_{004}$ |
| Logistic-IT non-ordered | MT | $.232_{016}$ | $.029_{004}$ | $.138_{008}$ | $.011_{002}$ | $.265_{009}$ | $.053_{005}$ | $.014_{003}$ | $.098_{008}$ | $.083_{007}$ | $.061_{004}$ |
| | ST | $.232_{016}$ | $.029_{004}$ | $.138_{008}$ | $.011_{002}$ | $.265_{009}$ | $.053_{005}$ | $.014_{002}$ | $.098_{008}$ | $.083_{007}$ | $.061_{004}$ |
| | EOT | $.230_{013}$ | $.030_{004}$ | $.138_{009}$ | $.011_{003}$ | $.267_{010}$ | $.054_{005}$ | $.013_{002}$ | $.098_{008}$ | $.083_{006}$ | $.061_{004}$ |
| Logistic-IT ordered | MT,ST | $.230_{012}$ | $.029_{004}$ | $.133_{008}$ | $.011_{002}$ | $.267_{009}$ | $.053_{004}$ | $.013_{002}$ | $.092_{007}$ | $.083_{006}$ | $.061_{004}$ |
| | EOT | $.229_{012}$ | $.030_{005}$ | $.131_{009}$ | $.010_{002}$ | $.265_{009}$ | $.054_{005}$ | $.013_{002}$ | $.091_{007}$ | $.083_{006}$ | $.061_{004}$ |
| Logistic-AT ordered | MT,ST | $.232_{016}$ | $.029_{004}$ | $.133_{009}$ | $.011_{002}$ | $.265_{010}$ | $.053_{005}$ | $.014_{003}$ | $.092_{006}$ | $.084_{007}$ | $.061_{004}$ |
| | LB | $.230_{014}$ | $.029_{004}$ | $.133_{009}$ | $.011_{002}$ | $.265_{010}$ | $.053_{005}$ | $.014_{003}$ | $.092_{006}$ | $.084_{006}$ | $.061_{004}$ |
| | EOT | $.229_{014}$ | $.030_{004}$ | $.131_{008}$ | $.010_{002}$ | $.264_{010}$ | $.054_{005}$ | $.014_{003}$ | $.092_{006}$ | $.083_{007}$ | $.061_{004}$ |
| OLR-NLL ordered | MT,ST | $.230_{013}$ | $.029_{004}$ | $.132_{007}$ | $.010_{002}$ | $.267_{009}$ | $.053_{005}$ | $.013_{002}$ | $.093_{007}$ | $.084_{006}$ | $.061_{004}$ |
| | LB | $.230_{013}$ | $.029_{004}$ | $.132_{007}$ | $.011_{003}$ | $.267_{010}$ | $.054_{005}$ | $.013_{002}$ | $.093_{007}$ | $.084_{007}$ | $.061_{004}$ |
| | EOT | $.230_{014}$ | $.030_{004}$ | $.132_{008}$ | $.011_{002}$ | $.265_{010}$ | $.053_{005}$ | $.013_{003}$ | $.093_{007}$ | $.083_{007}$ | $.061_{004}$ |

| Learning | Labeling | POT | AI1 | EL1 | CAL | CE1 | CE2 | 2DP | FRA | MVA |
|---|---|---|---|---|---|---|---|---|---|---|
| AD | NNT | $.061_{004}$ | $.068_{005}$ | $.019_{002}$ | $.224_{005}$ | $.033_{002}$ | $.039_{003}$ | $.094_{003}$ | $.125_{011}$ | $.002_{001}$ |
| | EOT | $.060_{004}$ | $.069_{004}$ | $\mathbf{.018_{002}}$ | $.224_{006}$ | $.033_{002}$ | $.039_{003}$ | $.095_{003}$ | $.123_{008}$ | $.001_{000}$ |
| Hinge-IT non-ordered | MT | $.070_{004}$ | $.068_{005}$ | $.019_{002}$ | $.214_{006}$ | $.031_{002}$ | $.039_{002}$ | $.095_{003}$ | $.094_{007}$ | $.002_{001}$ |
| | ST | $.070_{004}$ | $.068_{005}$ | $.019_{002}$ | $.214_{006}$ | $.031_{002}$ | $.039_{002}$ | $.095_{003}$ | $.094_{007}$ | $.002_{001}$ |
| | EOT | $\mathbf{.061_{004}}$ | $.068_{005}$ | $.018_{002}$ | $.213_{006}$ | $.031_{002}$ | $.039_{003}$ | $.095_{003}$ | $\mathbf{.092_{006}}$ | $\mathbf{.001_{000}}$ |
| Hinge-IT ordered | MT,ST | $.071_{003}$ | $.069_{005}$ | $.019_{002}$ | $.214_{006}$ | $.032_{003}$ | $.038_{003}$ | $.095_{003}$ | $.091_{005}$ | $.002_{001}$ |
| | EOT | $\mathbf{.062_{004}}$ | $.068_{005}$ | $\mathbf{.018_{002}}$ | $.212_{006}$ | $.031_{002}$ | $\mathbf{.035_{002}}$ | $.095_{003}$ | $.090_{006}$ | $.002_{001}$ |
| Hinge-AT ordered | MT,ST | $.068_{003}$ | $.068_{005}$ | $.019_{002}$ | $.218_{007}$ | $.032_{002}$ | $.034_{002}$ | $.094_{003}$ | $.091_{005}$ | $.002_{001}$ |
| | EOT | $.062_{004}$ | $.069_{004}$ | $\mathbf{.018_{002}}$ | $.217_{007}$ | $.032_{002}$ | $.035_{003}$ | $.095_{003}$ | $.090_{005}$ | $\mathbf{.001_{000}}$ |
| Logistic-IT non-ordered | MT | $.049_{004}$ | $.071_{004}$ | $.019_{002}$ | $.200_{006}$ | $.031_{002}$ | $.038_{003}$ | $.095_{003}$ | $.084_{003}$ | $.002_{001}$ |
| | ST | $.049_{004}$ | $.071_{004}$ | $.019_{002}$ | $.200_{006}$ | $.031_{002}$ | $.038_{003}$ | $.095_{003}$ | $.084_{003}$ | $.002_{001}$ |
| | EOT | $\mathbf{.036_{004}}$ | $.069_{005}$ | $\mathbf{.018_{002}}$ | $.199_{006}$ | $.032_{002}$ | $.038_{003}$ | $.095_{003}$ | $.083_{003}$ | $\mathbf{.001_{000}}$ |
| Logistic-IT ordered | MT,ST | $.050_{004}$ | $.070_{005}$ | $.018_{002}$ | $.200_{006}$ | $.032_{002}$ | $.035_{003}$ | $.094_{003}$ | $.081_{004}$ | $.002_{000}$ |
| | EOT | $\mathbf{.035_{004}}$ | $.069_{005}$ | $.017_{002}$ | $.199_{006}$ | $.032_{002}$ | $.035_{003}$ | $.095_{003}$ | $.081_{003}$ | $\mathbf{.001_{000}}$ |
| Logistic-AT ordered | MT,ST | $.041_{004}$ | $.070_{005}$ | $.019_{002}$ | $.200_{005}$ | $.031_{002}$ | $.035_{003}$ | $.095_{003}$ | $.082_{003}$ | $.002_{000}$ |
| | LB | $.038_{004}$ | $.069_{005}$ | $.019_{002}$ | $.201_{005}$ | $.032_{002}$ | $.035_{002}$ | $.095_{003}$ | $.082_{003}$ | $.002_{000}$ |
| | EOT | $\mathbf{.034_{003}}$ | $.069_{004}$ | $\mathbf{.018_{002}}$ | $.199_{005}$ | $.032_{002}$ | $.035_{002}$ | $.095_{003}$ | $.081_{004}$ | $\mathbf{.001_{000}}$ |
| OLR-NLL ordered | MT,ST | $.038_{004}$ | $.069_{005}$ | $.019_{002}$ | $.201_{006}$ | $.032_{002}$ | $.038_{003}$ | $.095_{003}$ | $.082_{003}$ | $.002_{000}$ |
| | LB | $.037_{004}$ | $.070_{005}$ | $.019_{002}$ | $.200_{005}$ | $.032_{002}$ | $.037_{003}$ | $.095_{003}$ | $.082_{003}$ | $.002_{000}$ |
| | EOT | $.034_{004}$ | $.069_{005}$ | $\mathbf{.018_{002}}$ | $.199_{006}$ | $.031_{002}$ | $.037_{003}$ | $.095_{003}$ | $.081_{003}$ | $\mathbf{.001_{000}}$ |

Table 10: A counterpart of Table 2 regarding MZE for Task-Z and EL5 datasets.

| Learning | Labeling | DIA | PYR | APR | SER | TRI | WBC | CPU | AMP | BOS | STO |
|---|---|---|---|---|---|---|---|---|---|---|---|
| AD | NNT | $.582_{113}$ | $.463_{137}$ | $.172_{066}$ | $.114_{055}$ | $.528_{071}$ | $.714_{070}$ | $.087_{040}$ | $.265_{050}$ | $.268_{046}$ | $.169_{025}$ |
|  | EOT | $.566_{138}$ | $.483_{128}$ | $.191_{070}$ | $.118_{060}$ | $.549_{083}$ | $.694_{073}$ | $.090_{044}$ | $.261_{052}$ | $.264_{049}$ | $.167_{031}$ |
| Hinge-IT non-ordered | MT | $.518_{179}$ | $.468_{171}$ | $.183_{072}$ | $.119_{056}$ | $.529_{073}$ | $.707_{069}$ | $.092_{046}$ | $.279_{051}$ | $.288_{040}$ | $.209_{034}$ |
|  | ST | $.518_{179}$ | $.468_{171}$ | $.185_{074}$ | $.119_{055}$ | $.530_{073}$ | $.706_{069}$ | $.099_{050}$ | $.279_{051}$ | $.288_{040}$ | $.209_{034}$ |
|  | EOT | $.554_{142}$ | $.476_{142}$ | $.195_{064}$ | $.124_{050}$ | $.532_{069}$ | $.702_{074}$ | $.097_{047}$ | $\mathbf{.252_{043}}$ | $\mathbf{.268_{040}}$ | $\mathbf{.164_{024}}$ |
| Hinge-IT ordered | MT,ST | $.542_{170}$ | $.471_{130}$ | $.176_{077}$ | $.130_{050}$ | $.530_{079}$ | $.704_{070}$ | $.098_{049}$ | $.266_{044}$ | $.260_{040}$ | $.147_{024}$ |
|  | EOT | $.548_{143}$ | $.489_{124}$ | $.179_{067}$ | $.129_{052}$ | $.547_{078}$ | $.699_{075}$ | $.095_{053}$ | $\mathbf{.253_{049}}$ | $.260_{041}$ | $.146_{026}$ |
| Hinge-AT ordered | MT,ST | $.588_{142}$ | $.452_{152}$ | $.168_{075}$ | $.121_{050}$ | $.536_{083}$ | $.740_{067}$ | $.094_{048}$ | $.266_{045}$ | $.285_{052}$ | $.153_{030}$ |
|  | EOT | $.538_{148}$ | $.481_{121}$ | $.181_{070}$ | $.128_{050}$ | $.553_{076}$ | $\mathbf{.705_{081}}$ | $.100_{048}$ | $.262_{057}$ | $.272_{042}$ | $.148_{024}$ |
| Logistic-IT non-ordered | MT | $.532_{142}$ | $.469_{133}$ | $.201_{071}$ | $.126_{052}$ | $.546_{079}$ | $.715_{078}$ | $.105_{044}$ | $.258_{045}$ | $.283_{045}$ | $.145_{023}$ |
|  | ST | $.532_{148}$ | $.469_{133}$ | $.204_{074}$ | $.125_{049}$ | $.546_{077}$ | $.715_{078}$ | $.107_{046}$ | $.258_{045}$ | $.283_{045}$ | $.145_{023}$ |
|  | EOT | $.532_{163}$ | $.467_{119}$ | $.196_{066}$ | $.124_{044}$ | $.545_{079}$ | $.713_{067}$ | $.106_{043}$ | $.257_{054}$ | $\mathbf{.264_{039}}$ | $.140_{027}$ |
| Logistic-IT ordered | MT,ST | $.556_{151}$ | $.463_{127}$ | $.193_{067}$ | $.122_{052}$ | $.532_{077}$ | $.706_{081}$ | $.105_{041}$ | $.245_{043}$ | $.255_{039}$ | $.128_{021}$ |
|  | EOT | $.538_{159}$ | $.471_{131}$ | $.192_{068}$ | $.128_{052}$ | $.537_{076}$ | $.701_{076}$ | $.110_{041}$ | $.247_{056}$ | $.265_{044}$ | $.127_{023}$ |
| Logistic-AT ordered | MT,ST | $\mathbf{.536_{128}}$ | $.457_{147}$ | $.187_{065}$ | $.121_{053}$ | $.551_{086}$ | $.708_{065}$ | $.108_{041}$ | $.248_{044}$ | $.265_{033}$ | $.131_{018}$ |
|  | LB | $.592_{135}$ | $.484_{131}$ | $.195_{071}$ | $.120_{058}$ | $.549_{073}$ | $.722_{061}$ | $.107_{044}$ | $.250_{052}$ | $.261_{038}$ | $.130_{019}$ |
|  | EOT | $\mathbf{.538_{157}}$ | $.472_{107}$ | $.205_{071}$ | $.124_{042}$ | $.557_{083}$ | $.701_{076}$ | $.106_{045}$ | $.253_{057}$ | $.274_{040}$ | $.130_{020}$ |
| OLR-NLL ordered | MT,ST | $.562_{135}$ | $.485_{133}$ | $.189_{064}$ | $.124_{051}$ | $.544_{077}$ | $.714_{077}$ | $.108_{043}$ | $.249_{040}$ | $.257_{039}$ | $.129_{021}$ |
|  | LB | $.560_{128}$ | $.481_{120}$ | $.194_{074}$ | $.122_{053}$ | $.547_{077}$ | $.728_{077}$ | $.108_{041}$ | $.250_{044}$ | $.255_{039}$ | $.129_{021}$ |
|  | EOT | $.534_{152}$ | $.471_{120}$ | $.191_{072}$ | $.121_{054}$ | $.548_{080}$ | $.708_{088}$ | $.110_{042}$ | $.250_{054}$ | $.265_{044}$ | $.129_{019}$ |

| Learning | Labeling | ABA | AI2 | KRA | CO1 | PU1 | BA1 | CO2 | PU2 | BA2 | EL2 |
|---|---|---|---|---|---|---|---|---|---|---|---|
| AD | NNT | $.249_{026}$ | $.154_{009}$ | $.223_{013}$ | $.052_{006}$ | $.470_{011}$ | $.113_{009}$ | $.069_{006}$ | $.164_{009}$ | $.176_{008}$ | $.174_{008}$ |
|  | EOT | $\mathbf{.223_{015}}$ | $\mathbf{.140_{009}}$ | $.220_{012}$ | $\mathbf{.049_{006}}$ | $.469_{012}$ | $\mathbf{.110_{007}}$ | $.068_{006}$ | $\mathbf{.160_{010}}$ | $.175_{008}$ | $\mathbf{.166_{008}}$ |
| Hinge-IT non-ordered | MT | $.232_{012}$ | $.152_{009}$ | $.233_{016}$ | $.053_{005}$ | $.496_{015}$ | $.111_{008}$ | $.068_{007}$ | $.202_{011}$ | $.178_{009}$ | $.174_{008}$ |
|  | ST | $.232_{012}$ | $.152_{009}$ | $.233_{016}$ | $.053_{005}$ | $.496_{015}$ | $.111_{008}$ | $.068_{007}$ | $.202_{011}$ | $.178_{009}$ | $.174_{008}$ |
|  | EOT | $\mathbf{.223_{015}}$ | $\mathbf{.138_{009}}$ | $.229_{014}$ | $\mathbf{.050_{005}}$ | $\mathbf{.471_{012}}$ | $\mathbf{.108_{008}}$ | $.066_{006}$ | $\mathbf{.164_{011}}$ | $.177_{008}$ | $\mathbf{.166_{008}}$ |
| Hinge-IT ordered | MT,ST | $.233_{014}$ | $.145_{010}$ | $.220_{014}$ | $.052_{006}$ | $.483_{011}$ | $.111_{008}$ | $.068_{006}$ | $.169_{009}$ | $.179_{009}$ | $.174_{008}$ |
|  | EOT | $\mathbf{.221_{014}}$ | $\mathbf{.137_{009}}$ | $.217_{014}$ | $\mathbf{.050_{006}}$ | $\mathbf{.470_{012}}$ | $\mathbf{.108_{009}}$ | $.067_{006}$ | $\mathbf{.161_{009}}$ | $.177_{008}$ | $\mathbf{.166_{008}}$ |
| Hinge-AT ordered | MT,ST | $.230_{014}$ | $.146_{010}$ | $.228_{013}$ | $.052_{005}$ | $.476_{011}$ | $.111_{008}$ | $.068_{007}$ | $.164_{010}$ | $.177_{008}$ | $.173_{008}$ |
|  | EOT | $\mathbf{.223_{013}}$ | $\mathbf{.137_{008}}$ | $\mathbf{.223_{012}}$ | $\mathbf{.050_{006}}$ | $.469_{012}$ | $.109_{008}$ | $\mathbf{.066_{006}}$ | $\mathbf{.158_{009}}$ | $.177_{009}$ | $\mathbf{.166_{008}}$ |
| Logistic-IT non-ordered | MT | $.217_{013}$ | $.148_{008}$ | $.216_{013}$ | $.054_{006}$ | $.474_{011}$ | $.106_{008}$ | $.068_{006}$ | $.209_{010}$ | $.176_{009}$ | $.168_{008}$ |
|  | ST | $.217_{013}$ | $.137_{009}$ | $.216_{013}$ | $.054_{005}$ | $.474_{011}$ | $.106_{008}$ | $.067_{006}$ | $.209_{010}$ | $.176_{009}$ | $.166_{008}$ |
|  | EOT | $.218_{014}$ | $.138_{009}$ | $.212_{012}$ | $\mathbf{.049_{005}}$ | $\mathbf{.470_{010}}$ | $.107_{007}$ | $\mathbf{.064_{006}}$ | $\mathbf{.160_{011}}$ | $.178_{010}$ | $.166_{008}$ |
| Logistic-IT ordered | MT,ST | $.216_{013}$ | $.137_{010}$ | $.212_{014}$ | $.053_{005}$ | $.473_{011}$ | $.107_{007}$ | $.065_{006}$ | $.161_{012}$ | $.176_{009}$ | $.167_{008}$ |
|  | EOT | $.218_{014}$ | $.136_{008}$ | $.210_{013}$ | $\mathbf{.050_{005}}$ | $.472_{012}$ | $.107_{007}$ | $.063_{006}$ | $.157_{010}$ | $.179_{009}$ | $.166_{008}$ |
| Logistic-AT ordered | MT,ST | $.215_{014}$ | $.136_{008}$ | $.217_{014}$ | $.053_{005}$ | $.474_{012}$ | $.108_{007}$ | $.064_{007}$ | $.159_{012}$ | $.176_{008}$ | $.166_{009}$ |
|  | LB | $.216_{013}$ | $.137_{008}$ | $.218_{014}$ | $.052_{006}$ | $.474_{012}$ | $.108_{007}$ | $.064_{006}$ | $.160_{011}$ | $.178_{008}$ | $.166_{008}$ |
|  | EOT | $.216_{012}$ | $.136_{009}$ | $.215_{013}$ | $.049_{005}$ | $.474_{012}$ | $.107_{008}$ | $.063_{007}$ | $.157_{011}$ | $.177_{007}$ | $.165_{008}$ |
| OLR-NLL ordered | MT,ST | $.215_{014}$ | $.137_{008}$ | $.212_{014}$ | $.052_{005}$ | $.471_{012}$ | $.106_{007}$ | $.065_{006}$ | $.160_{010}$ | $.178_{010}$ | $.166_{008}$ |
|  | LB | $.215_{013}$ | $.137_{010}$ | $.212_{014}$ | $.051_{005}$ | $.471_{010}$ | $.106_{007}$ | $.064_{006}$ | $.160_{010}$ | $.179_{010}$ | $.166_{009}$ |
|  | EOT | $.217_{012}$ | $.136_{008}$ | $.208_{013}$ | $.050_{005}$ | $.473_{012}$ | $.106_{007}$ | $.063_{006}$ | $.158_{011}$ | $.179_{008}$ | $.165_{008}$ |

| Learning | Labeling | POT | AI1 | EL1 | CAL | CE1 | CE2 | 2DP | FRA | MVA |
|---|---|---|---|---|---|---|---|---|---|---|
| AD | NNT | $.043_{004}$ | $.173_{006}$ | $.045_{003}$ | $.349_{008}$ | $.090_{004}$ | $.104_{004}$ | $.172_{006}$ | $.142_{006}$ | $.004_{001}$ |
|  | EOT | $\mathbf{.041_{003}}$ | $\mathbf{.170_{006}}$ | $.046_{004}$ | $.347_{009}$ | $.090_{004}$ | $.105_{004}$ | $\mathbf{.167_{005}}$ | $.140_{004}$ | $\mathbf{.003_{001}}$ |
| Hinge-IT non-ordered | MT | $.108_{006}$ | $.173_{006}$ | $.061_{007}$ | $.386_{009}$ | $.092_{005}$ | $.104_{004}$ | $.177_{003}$ | $.163_{009}$ | $.032_{005}$ |
|  | ST | $.107_{006}$ | $.173_{006}$ | $.061_{007}$ | $.386_{009}$ | $.093_{005}$ | $.104_{004}$ | $.177_{003}$ | $.163_{009}$ | $.032_{005}$ |
|  | EOT | $\mathbf{.095_{007}}$ | $\mathbf{.170_{007}}$ | $\mathbf{.051_{004}}$ | $\mathbf{.375_{007}}$ | $\mathbf{.089_{004}}$ | $\mathbf{.102_{004}}$ | $.177_{004}$ | $\mathbf{.152_{009}}$ | $\mathbf{.006_{001}}$ |
| Hinge-IT ordered | MT,ST | $.035_{003}$ | $.172_{006}$ | $.050_{004}$ | $.379_{008}$ | $.101_{007}$ | $.104_{004}$ | $.172_{006}$ | $.143_{007}$ | $.004_{001}$ |
|  | EOT | $\mathbf{.034_{003}}$ | $\mathbf{.169_{006}}$ | $\mathbf{.047_{004}}$ | $\mathbf{.375_{007}}$ | $\mathbf{.089_{004}}$ | $\mathbf{.103_{004}}$ | $\mathbf{.165_{005}}$ | $\mathbf{.137_{004}}$ | $\mathbf{.003_{001}}$ |
| Hinge-AT ordered | MT,ST | $.030_{003}$ | $.172_{006}$ | $.054_{005}$ | $.376_{008}$ | $.091_{004}$ | $.104_{004}$ | $.175_{005}$ | $.146_{006}$ | $.004_{001}$ |
|  | EOT | $.029_{003}$ | $\mathbf{.169_{006}}$ | $\mathbf{.049_{004}}$ | $.375_{008}$ | $\mathbf{.090_{004}}$ | $\mathbf{.098_{004}}$ | $\mathbf{.167_{005}}$ | $\mathbf{.138_{004}}$ | $\mathbf{.003_{001}}$ |
| Logistic-IT non-ordered | MT | $.083_{004}$ | $.175_{007}$ | $.056_{005}$ | $.382_{009}$ | $.089_{004}$ | $.092_{004}$ | $.172_{005}$ | $.144_{005}$ | $.043_{008}$ |
|  | ST | $.083_{004}$ | $.173_{006}$ | $.056_{005}$ | $.382_{009}$ | $.088_{004}$ | $.093_{004}$ | $.172_{005}$ | $.144_{005}$ | $.043_{008}$ |
|  | EOT | $\mathbf{.064_{006}}$ | $\mathbf{.169_{007}}$ | $\mathbf{.050_{004}}$ | $\mathbf{.378_{006}}$ | $.088_{004}$ | $.092_{004}$ | $\mathbf{.163_{003}}$ | $.136_{004}$ | $\mathbf{.007_{002}}$ |
| Logistic-IT ordered | MT,ST | $.029_{003}$ | $.167_{006}$ | $.046_{003}$ | $.380_{007}$ | $.089_{004}$ | $.092_{004}$ | $.163_{004}$ | $.137_{004}$ | $.003_{001}$ |
|  | EOT | $\mathbf{.028_{003}}$ | $.166_{006}$ | $.046_{003}$ | $\mathbf{.377_{008}}$ | $.088_{004}$ | $.092_{004}$ | $.163_{004}$ | $\mathbf{.135_{004}}$ | $\mathbf{.003_{001}}$ |
| Logistic-AT ordered | MT,ST | $.028_{003}$ | $.172_{006}$ | $.052_{004}$ | $.380_{008}$ | $.089_{004}$ | $.094_{004}$ | $.163_{004}$ | $.137_{004}$ | $.004_{001}$ |
|  | LB | $.028_{003}$ | $\mathbf{.169_{006}}$ | $.052_{004}$ | $.379_{009}$ | $.089_{004}$ | $.094_{004}$ | $.163_{004}$ | $.137_{004}$ | $.004_{001}$ |
|  | EOT | $.028_{002}$ | $\mathbf{.169_{008}}$ | $\mathbf{.049_{003}}$ | $.378_{008}$ | $.089_{004}$ | $.094_{004}$ | $.162_{003}$ | $\mathbf{.135_{003}}$ | $\mathbf{.003_{001}}$ |
| OLR-NLL ordered | MT,ST | $.028_{003}$ | $.172_{007}$ | $.050_{003}$ | $.376_{007}$ | $.088_{004}$ | $.092_{004}$ | $.163_{004}$ | $.136_{004}$ | $.010_{002}$ |
|  | LB | $.028_{003}$ | $\mathbf{.170_{006}}$ | $.049_{004}$ | $.376_{008}$ | $.088_{004}$ | $.091_{004}$ | $.163_{004}$ | $.136_{004}$ | $.010_{003}$ |
|  | EOT | $.028_{004}$ | $\mathbf{.168_{007}}$ | $\mathbf{.048_{003}}$ | $.376_{008}$ | $.088_{004}$ | $.091_{004}$ | $.162_{004}$ | $.136_{004}$ | $\mathbf{.007_{001}}$ |

Table 11: A counterpart of Table 2 regarding MZE for Task-Z and EL10 datasets.

| Learning | Labeling | DIA | PYR | APR | SER | TRI | WBC | CPU | AMP | BOS | STO |
|---|---|---|---|---|---|---|---|---|---|---|---|
| AD | NNT | $.756_{146}$ | $.617_{101}$ | $.387_{069}$ | $.272_{064}$ | $.711_{067}$ | $.853_{059}$ | $.189_{058}$ | $.473_{073}$ | $\mathbf{.442_{048}}$ | $.284_{029}$ |
| | EOT | $.768_{117}$ | $.632_{113}$ | $.404_{071}$ | $.274_{089}$ | $.707_{071}$ | $\mathbf{.833_{059}}$ | $.182_{056}$ | $.470_{064}$ | $.460_{054}$ | $.286_{025}$ |
| Hinge-IT non-ordered | MT | $.712_{144}$ | $.620_{113}$ | $.393_{076}$ | $.315_{069}$ | $.708_{074}$ | $.843_{050}$ | $.190_{059}$ | $.486_{066}$ | $.463_{049}$ | $.376_{027}$ |
| | ST | $.712_{144}$ | $.623_{110}$ | $.387_{076}$ | $.345_{070}$ | $.708_{070}$ | $.850_{050}$ | $.194_{058}$ | $.490_{067}$ | $.462_{050}$ | $.376_{027}$ |
| | EOT | $.738_{131}$ | $.633_{103}$ | $.397_{084}$ | $\mathbf{.257_{087}}$ | $.720_{070}$ | $.829_{065}$ | $.183_{050}$ | $\mathbf{.454_{063}}$ | $.447_{049}$ | $\mathbf{.289_{029}}$ |
| Hinge-IT ordered | MT,ST | $.698_{146}$ | $.599_{116}$ | $.392_{082}$ | $.244_{065}$ | $.713_{086}$ | $.826_{064}$ | $.188_{059}$ | $.488_{064}$ | $.441_{044}$ | $.281_{029}$ |
| | EOT | $.744_{130}$ | $.608_{113}$ | $.398_{074}$ | $.263_{082}$ | $.716_{074}$ | $.826_{071}$ | $.184_{055}$ | $.447_{061}$ | $.443_{053}$ | $.279_{028}$ |
| Hinge-AT ordered | MT,ST | $.750_{143}$ | $.616_{118}$ | $.395_{077}$ | $.291_{079}$ | $.725_{063}$ | $.852_{058}$ | $.197_{058}$ | $.471_{060}$ | $.435_{049}$ | $.284_{028}$ |
| | EOT | $.730_{157}$ | $.612_{112}$ | $.390_{075}$ | $.284_{082}$ | $.710_{063}$ | $.837_{067}$ | $.186_{057}$ | $.461_{062}$ | $.444_{042}$ | $\mathbf{.273_{026}}$ |
| Logistic-IT non-ordered | MT | $.742_{137}$ | $.631_{111}$ | $.405_{078}$ | $.296_{073}$ | $.707_{066}$ | $.848_{061}$ | $.197_{054}$ | $.483_{069}$ | $.441_{043}$ | $.306_{031}$ |
| | ST | $.742_{137}$ | $.629_{103}$ | $.403_{075}$ | $.358_{072}$ | $.706_{072}$ | $.849_{055}$ | $.206_{062}$ | $.485_{067}$ | $.441_{042}$ | $.306_{031}$ |
| | EOT | $.752_{145}$ | $.607_{098}$ | $.384_{077}$ | $\mathbf{.269_{074}}$ | $.711_{070}$ | $.829_{066}$ | $.194_{054}$ | $\mathbf{.456_{053}}$ | $.441_{044}$ | $\mathbf{.279_{027}}$ |
| Logistic-IT ordered | MT,ST | $.740_{147}$ | $.616_{122}$ | $.396_{069}$ | $.232_{067}$ | $.705_{077}$ | $.829_{048}$ | $.189_{054}$ | $.465_{062}$ | $.446_{051}$ | $.273_{027}$ |
| | EOT | $.744_{120}$ | $.612_{117}$ | $.395_{086}$ | $.239_{081}$ | $.722_{069}$ | $.838_{068}$ | $.188_{062}$ | $.456_{053}$ | $.443_{049}$ | $.273_{027}$ |
| Logistic-AT ordered | MT,ST | $.718_{149}$ | $.593_{115}$ | $.387_{079}$ | $.248_{076}$ | $.704_{075}$ | $\mathbf{.822_{060}}$ | $.187_{053}$ | $.477_{073}$ | $.444_{047}$ | $.266_{024}$ |
| | LB | $.748_{149}$ | $.611_{130}$ | $.393_{072}$ | $.259_{075}$ | $.703_{082}$ | $.852_{061}$ | $.193_{055}$ | $.465_{071}$ | $.446_{056}$ | $.266_{026}$ |
| | EOT | $.718_{138}$ | $.601_{105}$ | $.396_{084}$ | $.245_{069}$ | $.713_{078}$ | $\mathbf{.823_{069}}$ | $.184_{060}$ | $.456_{055}$ | $.455_{046}$ | $.263_{024}$ |
| OLR-NLL ordered | MT,ST | $.716_{141}$ | $.604_{132}$ | $.378_{080}$ | $.235_{070}$ | $.700_{061}$ | $.835_{058}$ | $.191_{055}$ | $.468_{068}$ | $.436_{044}$ | $.266_{027}$ |
| | LB | $.770_{124}$ | $.615_{111}$ | $.396_{091}$ | $.229_{077}$ | $.701_{061}$ | $.853_{050}$ | $.188_{050}$ | $.466_{066}$ | $.444_{043}$ | $.262_{024}$ |
| | EOT | $.748_{146}$ | $.617_{101}$ | $.392_{086}$ | $.229_{082}$ | $.717_{064}$ | $.837_{056}$ | $.189_{059}$ | $.453_{055}$ | $.454_{050}$ | $.264_{025}$ |

| Learning | Labeling | ABA | AI2 | KRA | CO1 | PU1 | BA1 | CO2 | PU2 | BA2 | EL2 |
|---|---|---|---|---|---|---|---|---|---|---|---|
| AD | NNT | $.434_{017}$ | $.212_{011}$ | $.399_{013}$ | $.166_{009}$ | $.652_{011}$ | $.248_{010}$ | $.212_{011}$ | $.325_{014}$ | $.313_{011}$ | $.294_{009}$ |
| | EOT | $.433_{015}$ | $\mathbf{.205_{011}}$ | $.400_{014}$ | $.166_{009}$ | $.649_{013}$ | $.250_{012}$ | $.209_{009}$ | $.321_{013}$ | $\mathbf{.307_{011}}$ | $.294_{008}$ |
| Hinge-IT non-ordered | MT | $.470_{016}$ | $.212_{011}$ | $.407_{015}$ | $.164_{007}$ | $.662_{014}$ | $.249_{013}$ | $.208_{011}$ | $.347_{016}$ | $.305_{012}$ | $.294_{009}$ |
| | ST | $.471_{016}$ | $.212_{011}$ | $.407_{015}$ | $.164_{008}$ | $.662_{014}$ | $.249_{013}$ | $.208_{011}$ | $.348_{014}$ | $.305_{012}$ | $.294_{009}$ |
| | EOT | $\mathbf{.437_{015}}$ | $.203_{011}$ | $\mathbf{.396_{016}}$ | $\mathbf{.161_{008}}$ | $\mathbf{.655_{011}}$ | $.245_{011}$ | $\mathbf{.204_{010}}$ | $.327_{014}$ | $.306_{011}$ | $.295_{009}$ |
| Hinge-IT ordered | MT,ST | $.432_{017}$ | $.212_{010}$ | $.410_{016}$ | $.160_{009}$ | $.670_{012}$ | $.245_{011}$ | $.205_{010}$ | $.340_{012}$ | $.308_{011}$ | $.293_{008}$ |
| | EOT | $.432_{016}$ | $\mathbf{.202_{011}}$ | $\mathbf{.404_{012}}$ | $.159_{009}$ | $\mathbf{.656_{010}}$ | $.245_{011}$ | $.204_{010}$ | $.338_{014}$ | $.304_{010}$ | $.294_{009}$ |
| Hinge-AT ordered | MT,ST | $.431_{016}$ | $.209_{012}$ | $.406_{015}$ | $.162_{008}$ | $.655_{012}$ | $.246_{011}$ | $.206_{012}$ | $.324_{013}$ | $.311_{011}$ | $.294_{008}$ |
| | EOT | $.432_{014}$ | $\mathbf{.201_{011}}$ | $.403_{014}$ | $.162_{009}$ | $.654_{012}$ | $.246_{011}$ | $.205_{011}$ | $.321_{011}$ | $.308_{014}$ | $.294_{009}$ |
| Logistic-IT non-ordered | MT | $.444_{018}$ | $.202_{012}$ | $.408_{014}$ | $.163_{007}$ | $.652_{011}$ | $.245_{012}$ | $.204_{009}$ | $.351_{017}$ | $.310_{012}$ | $.291_{008}$ |
| | ST | $.453_{020}$ | $.206_{010}$ | $.407_{015}$ | $.163_{007}$ | $.652_{011}$ | $.245_{012}$ | $.204_{010}$ | $.351_{016}$ | $.310_{012}$ | $.291_{008}$ |
| | EOT | $\mathbf{.430_{017}}$ | $.199_{011}$ | $\mathbf{.391_{016}}$ | $\mathbf{.158_{010}}$ | $.653_{011}$ | $.243_{011}$ | $\mathbf{.196_{010}}$ | $.325_{014}$ | $.308_{012}$ | $.293_{009}$ |
| Logistic-IT ordered | MT,ST | $.426_{013}$ | $.198_{011}$ | $.398_{013}$ | $.158_{009}$ | $.654_{013}$ | $.243_{011}$ | $.196_{011}$ | $.337_{013}$ | $.306_{012}$ | $.291_{008}$ |
| | EOT | $.433_{016}$ | $.199_{012}$ | $.396_{013}$ | $.156_{007}$ | $.652_{012}$ | $.242_{010}$ | $.196_{011}$ | $.333_{013}$ | $.309_{013}$ | $.292_{009}$ |
| Logistic-AT ordered | MT,ST | $.429_{013}$ | $.198_{010}$ | $.398_{017}$ | $.161_{006}$ | $.655_{011}$ | $.244_{011}$ | $.200_{010}$ | $.322_{012}$ | $\mathbf{.306_{011}}$ | $.293_{010}$ |
| | LB | $.433_{013}$ | $.199_{011}$ | $.396_{017}$ | $.160_{007}$ | $.653_{012}$ | $.244_{011}$ | $.198_{010}$ | $.320_{012}$ | $.315_{013}$ | $.294_{010}$ |
| | EOT | $.437_{016}$ | $.199_{011}$ | $.394_{015}$ | $\mathbf{.157_{008}}$ | $.654_{012}$ | $.245_{013}$ | $.197_{010}$ | $.317_{013}$ | $\mathbf{.308_{012}}$ | $.293_{009}$ |
| OLR-NLL ordered | MT,ST | $.426_{013}$ | $.198_{010}$ | $.392_{015}$ | $.157_{007}$ | $.652_{011}$ | $.242_{011}$ | $.197_{011}$ | $.321_{013}$ | $.307_{012}$ | $.292_{009}$ |
| | LB | $.430_{015}$ | $.199_{011}$ | $.389_{015}$ | $.157_{008}$ | $.652_{011}$ | $.242_{011}$ | $.197_{012}$ | $.320_{012}$ | $.315_{012}$ | $.293_{009}$ |
| | EOT | $.434_{016}$ | $.199_{013}$ | $.389_{014}$ | $.157_{008}$ | $.654_{012}$ | $.243_{011}$ | $.197_{010}$ | $.320_{011}$ | $.308_{012}$ | $.293_{010}$ |

| Learning | Labeling | POT | AI1 | EL1 | CAL | CE1 | CE2 | 2DP | FRA | MVA |
|---|---|---|---|---|---|---|---|---|---|---|
| AD | NNT | $.141_{008}$ | $.286_{007}$ | $.267_{005}$ | $.539_{013}$ | $.218_{007}$ | $.211_{006}$ | $.318_{005}$ | $.263_{005}$ | $.050_{004}$ |
| | EOT | $\mathbf{.130_{006}}$ | $\mathbf{.283_{007}}$ | $.268_{005}$ | $\mathbf{.528_{009}}$ | $.216_{006}$ | $.210_{005}$ | $\mathbf{.315_{005}}$ | $.262_{005}$ | $\mathbf{.041_{004}}$ |
| Hinge-IT non-ordered | MT | $.119_{009}$ | $.305_{011}$ | $.310_{010}$ | $.587_{008}$ | $.226_{007}$ | $.226_{005}$ | $.344_{005}$ | $.303_{007}$ | $.074_{006}$ |
| | ST | $.119_{009}$ | $.305_{011}$ | $.310_{010}$ | $.588_{008}$ | $.226_{007}$ | $.226_{005}$ | $.330_{006}$ | $.288_{005}$ | $.066_{006}$ |
| | EOT | $\mathbf{.114_{008}}$ | $\mathbf{.286_{007}}$ | $\mathbf{.289_{008}}$ | $\mathbf{.583_{008}}$ | $\mathbf{.222_{006}}$ | $.227_{006}$ | $\mathbf{.316_{005}}$ | $\mathbf{.269_{005}}$ | $\mathbf{.039_{003}}$ |
| Hinge-IT ordered | MT,ST | $.151_{005}$ | $.285_{007}$ | $.260_{005}$ | $.521_{010}$ | $.227_{006}$ | $.239_{018}$ | $.323_{005}$ | $.269_{004}$ | $.047_{005}$ |
| | EOT | $\mathbf{.129_{005}}$ | $\mathbf{.283_{007}}$ | $.260_{006}$ | $\mathbf{.517_{008}}$ | $.225_{006}$ | $\mathbf{.229_{006}}$ | $\mathbf{.315_{005}}$ | $.268_{005}$ | $\mathbf{.038_{004}}$ |
| Hinge-AT ordered | MT,ST | $.120_{008}$ | $.284_{007}$ | $.259_{006}$ | $.535_{011}$ | $.235_{005}$ | $.237_{007}$ | $.324_{006}$ | $.270_{005}$ | $.051_{006}$ |
| | EOT | $\mathbf{.106_{008}}$ | $\mathbf{.283_{007}}$ | $.260_{007}$ | $\mathbf{.526_{008}}$ | $\mathbf{.223_{006}}$ | $\mathbf{.230_{006}}$ | $\mathbf{.315_{004}}$ | $\mathbf{.268_{005}}$ | $\mathbf{.043_{004}}$ |
| Logistic-IT non-ordered | MT | $.104_{007}$ | $.292_{008}$ | $.303_{009}$ | $.586_{008}$ | $.225_{006}$ | $.225_{006}$ | $.342_{006}$ | $.304_{009}$ | $.077_{005}$ |
| | ST | $.104_{007}$ | $.291_{008}$ | $.303_{009}$ | $.586_{008}$ | $.225_{006}$ | $.225_{006}$ | $.329_{006}$ | $.293_{008}$ | $.071_{006}$ |
| | EOT | $.097_{008}$ | $\mathbf{.283_{008}}$ | $\mathbf{.285_{006}}$ | $\mathbf{.582_{008}}$ | $\mathbf{.222_{006}}$ | $.224_{006}$ | $\mathbf{.315_{004}}$ | $\mathbf{.268_{005}}$ | $\mathbf{.038_{003}}$ |
| Logistic-IT ordered | MT,ST | $.135_{007}$ | $.278_{007}$ | $.258_{007}$ | $.515_{008}$ | $.207_{007}$ | $.226_{006}$ | $.318_{004}$ | $.260_{004}$ | $.038_{004}$ |
| | EOT | $\mathbf{.115_{007}}$ | $.277_{006}$ | $.258_{007}$ | $\mathbf{.514_{007}}$ | $.207_{006}$ | $.225_{006}$ | $\mathbf{.316_{005}}$ | $.260_{004}$ | $\mathbf{.033_{003}}$ |
| Logistic-AT ordered | MT,ST | $.137_{006}$ | $.287_{007}$ | $.281_{007}$ | $.591_{007}$ | $.229_{007}$ | $.234_{006}$ | $.321_{004}$ | $.269_{005}$ | $.052_{004}$ |
| | LB | $.124_{007}$ | $.285_{008}$ | $.277_{006}$ | $.594_{008}$ | $.231_{006}$ | $.232_{006}$ | $.320_{005}$ | $.268_{005}$ | $.043_{004}$ |
| | EOT | $\mathbf{.107_{007}}$ | $\mathbf{.283_{007}}$ | $.272_{006}$ | $.587_{007}$ | $.225_{006}$ | $.231_{006}$ | $.315_{005}$ | $.266_{005}$ | $.037_{003}$ |
| OLR-NLL ordered | MT,ST | $.126_{007}$ | $.285_{007}$ | $.275_{007}$ | $\mathbf{.584_{007}}$ | $\mathbf{.223_{007}}$ | $.225_{006}$ | $.319_{004}$ | $.268_{006}$ | $.042_{004}$ |
| | LB | $.110_{007}$ | $.285_{008}$ | $.275_{007}$ | $.589_{008}$ | $.226_{007}$ | $.225_{007}$ | $.319_{005}$ | $.267_{005}$ | $.040_{003}$ |
| | EOT | $\mathbf{.100_{008}}$ | $\mathbf{.282_{007}}$ | $\mathbf{.272_{006}}$ | $\mathbf{.582_{008}}$ | $\mathbf{.222_{006}}$ | $.223_{006}$ | $\mathbf{.315_{005}}$ | $\mathbf{.266_{005}}$ | $\mathbf{.034_{003}}$ |

Table 12: A counterpart of Table 2 regarding MAE for Task-A and EF3 datasets.

| Learning | Labeling | DIA | PYR | APR | SER | TRI | WBC | CPU | AMP | BOS | STO |
|---|---|---|---|---|---|---|---|---|---|---|---|
| AD | NNT | $.658_{183}$ | $.404_{125}$ | $.150_{060}$ | $.214_{075}$ | $.631_{115}$ | $.721_{105}$ | $.271_{066}$ | $.209_{048}$ | $.263_{037}$ | $.080_{019}$ |
|  | EOT | $.668_{189}$ | $.373_{135}$ | $.145_{065}$ | $.226_{091}$ | $.637_{109}$ | $.748_{102}$ | $.268_{065}$ | $\mathbf{.183_{039}}$ | $.256_{037}$ | $.079_{021}$ |
| Hinge-IT non-ordered | MT | $.670_{219}$ | $.396_{115}$ | $.159_{060}$ | $.211_{079}$ | $.644_{123}$ | $.730_{104}$ | $.271_{067}$ | $.209_{051}$ | $.255_{041}$ | $.080_{018}$ |
|  | ST | $.670_{219}$ | $.396_{115}$ | $.159_{060}$ | $.211_{079}$ | $.644_{123}$ | $.730_{104}$ | $.271_{067}$ | $.209_{051}$ | $.255_{041}$ | $.080_{018}$ |
|  | EOT | $.654_{184}$ | $.392_{123}$ | $.154_{048}$ | $.215_{077}$ | $.616_{125}$ | $.711_{112}$ | $.267_{057}$ | $\mathbf{.184_{041}}$ | $.254_{039}$ | $.075_{018}$ |
| Hinge-IT ordered | MT,ST | $.666_{204}$ | $.401_{114}$ | $.152_{052}$ | $.218_{071}$ | $.648_{109}$ | $.710_{109}$ | $.262_{066}$ | $.199_{045}$ | $.250_{041}$ | $.079_{018}$ |
|  | EOT | $.656_{171}$ | $.375_{135}$ | $.148_{052}$ | $.204_{061}$ | $.618_{109}$ | $.706_{111}$ | $.264_{063}$ | $.190_{037}$ | $.256_{036}$ | $.076_{019}$ |
| Hinge-AT ordered | MT,ST | $.696_{178}$ | $.396_{104}$ | $.152_{060}$ | $.218_{075}$ | $.621_{124}$ | $.706_{113}$ | $.256_{065}$ | $.203_{046}$ | $.251_{039}$ | $.077_{018}$ |
|  | EOT | $.666_{194}$ | $.365_{109}$ | $.147_{053}$ | $.210_{057}$ | $.629_{119}$ | $.717_{122}$ | $.267_{066}$ | $.194_{040}$ | $.248_{038}$ | $.077_{020}$ |
| Logistic-IT non-ordered | MT | $.622_{194}$ | $.416_{134}$ | $.158_{066}$ | $.228_{066}$ | $.625_{099}$ | $.725_{123}$ | $.249_{065}$ | $.197_{047}$ | $.251_{040}$ | $.077_{019}$ |
|  | ST | $.622_{194}$ | $.416_{134}$ | $.158_{066}$ | $.228_{066}$ | $.625_{099}$ | $.725_{123}$ | $.249_{065}$ | $.197_{047}$ | $.251_{040}$ | $.077_{019}$ |
|  | EOT | $.666_{181}$ | $.391_{115}$ | $.146_{047}$ | $.226_{052}$ | $.604_{094}$ | $.714_{112}$ | $.250_{066}$ | $\mathbf{.179_{038}}$ | $.241_{038}$ | $.081_{022}$ |
| Logistic-IT ordered | MT,ST | $.636_{199}$ | $.423_{114}$ | $.153_{068}$ | $.237_{083}$ | $.606_{108}$ | $.721_{131}$ | $.263_{070}$ | $.194_{043}$ | $.239_{040}$ | $.074_{018}$ |
|  | EOT | $.672_{180}$ | $.385_{096}$ | $.134_{044}$ | $.224_{059}$ | $.611_{106}$ | $.721_{119}$ | $.260_{066}$ | $.182_{033}$ | $.238_{046}$ | $.074_{018}$ |
| Logistic-AT ordered | MT,ST | $.638_{208}$ | $.411_{115}$ | $.148_{064}$ | $.223_{067}$ | $.616_{102}$ | $.715_{122}$ | $.256_{066}$ | $.193_{040}$ | $.230_{037}$ | $.075_{019}$ |
|  | LB | $.638_{208}$ | $.411_{115}$ | $.148_{064}$ | $.223_{067}$ | $.616_{102}$ | $.715_{122}$ | $.256_{066}$ | $.193_{040}$ | $.230_{037}$ | $.075_{019}$ |
|  | EOT | $.690_{192}$ | $.381_{111}$ | $.136_{046}$ | $.221_{065}$ | $.616_{110}$ | $.715_{122}$ | $.261_{070}$ | $.182_{038}$ | $.233_{042}$ | $.076_{016}$ |
| OLR-NLL ordered | MT,ST | $.658_{188}$ | $.409_{120}$ | $.146_{064}$ | $.234_{083}$ | $.611_{112}$ | $.713_{137}$ | $.251_{071}$ | $.196_{050}$ | $.230_{037}$ | $.074_{021}$ |
|  | LB | $.658_{188}$ | $.409_{120}$ | $.146_{064}$ | $.234_{083}$ | $.611_{112}$ | $.713_{137}$ | $.251_{071}$ | $.196_{050}$ | $.230_{037}$ | $.074_{021}$ |
|  | EOT | $.678_{186}$ | $.393_{101}$ | $.136_{049}$ | $.217_{058}$ | $.605_{104}$ | $.733_{118}$ | $.264_{073}$ | $\mathbf{.177_{039}}$ | $.234_{042}$ | $.076_{017}$ |

| Learning | Labeling | ABA | AI2 | KRA | CO1 | PU1 | BA1 | CO2 | PU2 | BA2 | EL2 |
|---|---|---|---|---|---|---|---|---|---|---|---|
| AD | NNT | $.375_{020}$ | $.360_{012}$ | $.180_{008}$ | $.173_{008}$ | $.332_{012}$ | $.107_{008}$ | $.202_{009}$ | $.167_{009}$ | $.361_{013}$ | $.365_{012}$ |
|  | EOT | $.378_{019}$ | $.358_{010}$ | $.177_{010}$ | $.173_{008}$ | $.335_{013}$ | $.107_{008}$ | $.202_{010}$ | $.166_{008}$ | $.360_{013}$ | $.365_{012}$ |
| Hinge-IT non-ordered | MT | $.375_{020}$ | $.369_{012}$ | $.157_{010}$ | $.170_{009}$ | $.330_{012}$ | $.102_{008}$ | $.203_{011}$ | $.163_{009}$ | $.364_{013}$ | $.364_{011}$ |
|  | ST | $.375_{020}$ | $.369_{012}$ | $.157_{010}$ | $.170_{009}$ | $.330_{012}$ | $.102_{008}$ | $.203_{011}$ | $.163_{009}$ | $.364_{013}$ | $.364_{011}$ |
|  | EOT | $.374_{015}$ | $\mathbf{.359_{011}}$ | $.157_{010}$ | $.169_{009}$ | $.332_{011}$ | $.102_{008}$ | $.201_{010}$ | $\mathbf{.160_{009}}$ | $\mathbf{.359_{014}}$ | $.365_{011}$ |
| Hinge-IT ordered | MT,ST | $.376_{020}$ | $.370_{013}$ | $.158_{010}$ | $.170_{008}$ | $.332_{012}$ | $.101_{008}$ | $.202_{009}$ | $.159_{009}$ | $.364_{012}$ | $.363_{011}$ |
|  | EOT | $.373_{017}$ | $\mathbf{.358_{011}}$ | $.157_{009}$ | $.170_{008}$ | $.332_{011}$ | $.101_{008}$ | $.200_{009}$ | $.158_{009}$ | $.359_{012}$ | $.363_{010}$ |
| Hinge-AT ordered | MT,ST | $.376_{018}$ | $.364_{015}$ | $.157_{010}$ | $.170_{008}$ | $.332_{012}$ | $.101_{008}$ | $.201_{009}$ | $.160_{008}$ | $.361_{013}$ | $.365_{009}$ |
|  | EOT | $.376_{021}$ | $\mathbf{.359_{011}}$ | $.158_{011}$ | $.170_{009}$ | $.332_{011}$ | $.100_{008}$ | $.200_{009}$ | $.158_{009}$ | $.359_{013}$ | $.365_{011}$ |
| Logistic-IT non-ordered | MT | $.368_{019}$ | $.355_{011}$ | $.154_{009}$ | $.167_{009}$ | $.329_{010}$ | $.101_{007}$ | $.197_{008}$ | $.174_{009}$ | $.364_{013}$ | $.363_{012}$ |
|  | ST | $.368_{019}$ | $.355_{011}$ | $.154_{009}$ | $.167_{009}$ | $.329_{010}$ | $.101_{007}$ | $.197_{008}$ | $.174_{009}$ | $.364_{013}$ | $.363_{012}$ |
|  | EOT | $.368_{018}$ | $.352_{012}$ | $.153_{008}$ | $.167_{009}$ | $.329_{009}$ | $.101_{008}$ | $.197_{008}$ | $\mathbf{.157_{008}}$ | $.359_{014}$ | $.364_{011}$ |
| Logistic-IT ordered | MT,ST | $.369_{019}$ | $.355_{011}$ | $.153_{010}$ | $.168_{009}$ | $.329_{011}$ | $.101_{007}$ | $.198_{008}$ | $.160_{009}$ | $.363_{014}$ | $.363_{011}$ |
|  | EOT | $.371_{017}$ | $.352_{011}$ | $.152_{008}$ | $.168_{009}$ | $.330_{010}$ | $.100_{008}$ | $.196_{009}$ | $.159_{009}$ | $.361_{015}$ | $.364_{011}$ |
| Logistic-AT ordered | MT,ST | $.368_{019}$ | $.353_{010}$ | $.154_{010}$ | $.168_{008}$ | $.328_{011}$ | $.101_{008}$ | $.199_{009}$ | $.158_{009}$ | $.361_{014}$ | $.364_{012}$ |
|  | LB | $.368_{019}$ | $.353_{010}$ | $.154_{010}$ | $.168_{008}$ | $.328_{011}$ | $.101_{008}$ | $.199_{009}$ | $.158_{009}$ | $.361_{014}$ | $.364_{012}$ |
|  | EOT | $.370_{018}$ | $.353_{010}$ | $.153_{010}$ | $.168_{009}$ | $.329_{010}$ | $.100_{008}$ | $.197_{008}$ | $.156_{009}$ | $.360_{014}$ | $.363_{011}$ |
| OLR-NLL ordered | MT,ST | $.367_{017}$ | $.353_{012}$ | $.152_{009}$ | $.167_{008}$ | $.329_{011}$ | $.101_{007}$ | $.197_{008}$ | $.158_{009}$ | $.361_{013}$ | $.363_{012}$ |
|  | LB | $.367_{017}$ | $.353_{012}$ | $.152_{009}$ | $.167_{008}$ | $.329_{011}$ | $.101_{007}$ | $.197_{008}$ | $.158_{009}$ | $.361_{013}$ | $.363_{012}$ |
|  | EOT | $.369_{018}$ | $.353_{010}$ | $.153_{010}$ | $.169_{009}$ | $.330_{009}$ | $.100_{008}$ | $.197_{008}$ | $.157_{009}$ | $.360_{015}$ | $.363_{011}$ |

| Learning | Labeling | POT | AI1 | EL1 | CAL | CE1 | CE2 | 2DP | FRA | MVA |
|---|---|---|---|---|---|---|---|---|---|---|
| AD | NNT | $.316_{007}$ | $.246_{007}$ | $.311_{008}$ | $.268_{006}$ | $.308_{008}$ | $.283_{007}$ | $.134_{003}$ | $.119_{003}$ | $.010_{008}$ |
|  | EOT | $.316_{009}$ | $.246_{008}$ | $.311_{009}$ | $.268_{007}$ | $.308_{008}$ | $.282_{006}$ | $.134_{003}$ | $.118_{003}$ | $.009_{008}$ |
| Hinge-IT non-ordered | MT | $.393_{026}$ | $.242_{009}$ | $.311_{008}$ | $.258_{007}$ | $.350_{008}$ | $.324_{009}$ | $.149_{008}$ | $.121_{004}$ | $.007_{001}$ |
|  | ST | $.393_{026}$ | $.242_{009}$ | $.311_{008}$ | $.258_{007}$ | $.350_{008}$ | $.324_{009}$ | $.149_{008}$ | $.121_{004}$ | $.007_{001}$ |
|  | EOT | $\mathbf{.363_{014}}$ | $.243_{008}$ | $.311_{008}$ | $.257_{008}$ | $\mathbf{.338_{007}}$ | $\mathbf{.314_{008}}$ | $\mathbf{.137_{003}}$ | $.120_{004}$ | $\mathbf{.007_{001}}$ |
| Hinge-IT ordered | MT,ST | $.312_{007}$ | $.242_{008}$ | $.311_{008}$ | $.258_{007}$ | $.303_{008}$ | $.282_{006}$ | $.134_{003}$ | $.120_{004}$ | $.007_{001}$ |
|  | EOT | $.312_{008}$ | $.242_{008}$ | $.310_{008}$ | $.256_{007}$ | $.303_{008}$ | $.280_{006}$ | $.133_{003}$ | $\mathbf{.119_{004}}$ | $\mathbf{.007_{001}}$ |
| Hinge-AT ordered | MT,ST | $.314_{008}$ | $.242_{008}$ | $.311_{008}$ | $.260_{007}$ | $.302_{007}$ | $.282_{007}$ | $.134_{003}$ | $.120_{004}$ | $.009_{008}$ |
|  | EOT | $.314_{008}$ | $.241_{008}$ | $.309_{008}$ | $.259_{007}$ | $.301_{008}$ | $.280_{007}$ | $.134_{003}$ | $.119_{004}$ | $.008_{007}$ |
| Logistic-IT non-ordered | MT | $.313_{009}$ | $.238_{007}$ | $.307_{008}$ | $.245_{007}$ | $.293_{007}$ | $.306_{007}$ | $.133_{003}$ | $.121_{004}$ | $.006_{001}$ |
|  | ST | $.313_{009}$ | $.238_{007}$ | $.307_{008}$ | $.245_{007}$ | $.293_{007}$ | $.306_{007}$ | $.133_{003}$ | $.121_{004}$ | $.006_{001}$ |
|  | EOT | $.312_{009}$ | $.239_{008}$ | $.307_{009}$ | $.244_{008}$ | $.292_{007}$ | $\mathbf{.299_{007}}$ | $.133_{003}$ | $\mathbf{.119_{004}}$ | $\mathbf{.005_{001}}$ |
| Logistic-IT ordered | MT,ST | $.313_{008}$ | $.238_{008}$ | $.308_{008}$ | $.246_{007}$ | $.294_{007}$ | $.275_{007}$ | $.133_{004}$ | $.116_{004}$ | $.009_{012}$ |
|  | EOT | $.312_{007}$ | $.238_{008}$ | $\mathbf{.306_{009}}$ | $.244_{007}$ | $.292_{007}$ | $.274_{006}$ | $.133_{003}$ | $.116_{004}$ | $\mathbf{.008_{012}}$ |
| Logistic-AT ordered | MT,ST,LB | $.313_{008}$ | $.239_{008}$ | $.310_{008}$ | $.247_{008}$ | $.292_{007}$ | $.310_{009}$ | $.133_{003}$ | $.120_{004}$ | $.021_{024}$ |
|  | EOT | $.311_{007}$ | $.238_{009}$ | $.309_{010}$ | $.246_{007}$ | $.292_{006}$ | $.308_{008}$ | $.133_{003}$ | $.119_{004}$ | $.020_{023}$ |
| OLR-NLL ordered | MT,ST,LB | $.313_{007}$ | $.238_{008}$ | $.309_{009}$ | $.246_{008}$ | $.293_{007}$ | $.304_{007}$ | $.134_{003}$ | $.119_{004}$ | $.017_{022}$ |
|  | EOT | $.312_{008}$ | $.238_{009}$ | $.308_{009}$ | $.245_{007}$ | $.291_{007}$ | $\mathbf{.301_{007}}$ | $.133_{003}$ | $.118_{004}$ | $\mathbf{.016_{021}}$ |

Table 13: A counterpart of Table 2 regarding MAE for Task-A and EF5 datasets

| Learning | Labeling | DIA | PYR | APR | SER | TRI | WBC | CPU | AMP | BOS | STO |
|---|---|---|---|---|---|---|---|---|---|---|---|
| AD | NNT | $1.146_{279}$ | $.755_{246}$ | $.414_{100}$ | $.423_{107}$ | $1.049_{147}$ | $1.225_{189}$ | $.496_{093}$ | $.328_{048}$ | $.394_{055}$ | $.180_{024}$ |
|  | EOT | $1.204_{251}$ | $.740_{239}$ | $.401_{081}$ | $.438_{152}$ | $1.038_{166}$ | $1.220_{195}$ | $\mathbf{.468_{073}}$ | $.328_{044}$ | $.390_{055}$ | $.176_{023}$ |
| Hinge-IT non-ordered | MT | $1.104_{321}$ | $.781_{242}$ | $.437_{098}$ | $.471_{156}$ | $1.084_{171}$ | $1.263_{219}$ | $.507_{106}$ | $.320_{052}$ | $.394_{057}$ | $.207_{039}$ |
|  | ST | $1.090_{316}$ | $.781_{242}$ | $.437_{098}$ | $.471_{156}$ | $1.084_{171}$ | $1.259_{216}$ | $.507_{106}$ | $.320_{052}$ | $.394_{057}$ | $.207_{039}$ |
|  | EOT | $1.126_{278}$ | $.795_{242}$ | $\mathbf{.399_{085}}$ | $\mathbf{.424_{133}}$ | $1.042_{176}$ | $1.243_{192}$ | $\mathbf{.459_{074}}$ | $.320_{049}$ | $.390_{058}$ | $\mathbf{.158_{026}}$ |
| Hinge-IT ordered | MT,ST | $1.092_{292}$ | $.805_{252}$ | $.430_{085}$ | $.425_{126}$ | $1.095_{158}$ | $1.287_{221}$ | $.504_{113}$ | $.317_{050}$ | $.379_{054}$ | $.155_{026}$ |
|  | EOT | $1.154_{298}$ | $.792_{266}$ | $\mathbf{.396_{088}}$ | $.415_{122}$ | $1.045_{151}$ | $1.238_{196}$ | $.480_{086}$ | $.325_{055}$ | $.385_{053}$ | $.154_{027}$ |
| Hinge-AT ordered | MT,ST | $1.170_{310}$ | $.749_{231}$ | $.427_{103}$ | $.411_{124}$ | $1.075_{144}$ | $1.217_{169}$ | $.471_{098}$ | $.318_{052}$ | $.377_{057}$ | $.157_{025}$ |
|  | EOT | $1.152_{251}$ | $.764_{264}$ | $\mathbf{.398_{084}}$ | $.406_{131}$ | $1.035_{131}$ | $1.178_{186}$ | $\mathbf{.451_{085}}$ | $.331_{052}$ | $.387_{054}$ | $.152_{027}$ |
| Logistic-IT non-ordered | MT | $1.114_{310}$ | $.813_{253}$ | $.452_{097}$ | $.459_{159}$ | $1.087_{157}$ | $1.230_{232}$ | $.495_{109}$ | $.333_{050}$ | $.405_{056}$ | $.165_{030}$ |
|  | ST | $1.108_{301}$ | $.813_{253}$ | $.452_{097}$ | $.459_{159}$ | $1.087_{157}$ | $1.230_{232}$ | $.495_{109}$ | $.333_{050}$ | $.405_{056}$ | $.165_{030}$ |
|  | EOT | $1.110_{266}$ | $.759_{239}$ | $\mathbf{.408_{082}}$ | $\mathbf{.422_{147}}$ | $1.052_{175}$ | $1.240_{195}$ | $.478_{082}$ | $.316_{053}$ | $.392_{056}$ | $\mathbf{.151_{024}}$ |
| Logistic-IT ordered | MT,ST | $1.092_{293}$ | $.816_{235}$ | $.432_{099}$ | $.379_{119}$ | $1.084_{153}$ | $1.240_{200}$ | $.493_{094}$ | $.310_{055}$ | $.378_{061}$ | $.136_{025}$ |
|  | EOT | $1.136_{256}$ | $.763_{245}$ | $.404_{097}$ | $.392_{156}$ | $1.049_{144}$ | $1.259_{213}$ | $.485_{085}$ | $.314_{050}$ | $.385_{053}$ | $.137_{026}$ |
| Logistic-AT ordered MT,ST,LB | MT,ST,LB | $1.118_{270}$ | $.768_{257}$ | $.453_{101}$ | $.361_{088}$ | $1.075_{150}$ | $1.188_{189}$ | $.467_{087}$ | $.308_{048}$ | $.365_{059}$ | $.137_{024}$ |
|  | EOT | $1.154_{265}$ | $.757_{256}$ | $.420_{088}$ | $.368_{118}$ | $1.046_{148}$ | $1.196_{180}$ | $.459_{081}$ | $.318_{048}$ | $.367_{053}$ | $.139_{025}$ |
| OLR-NLL ordered MT,ST,LB | MT,ST,LB | $1.092_{275}$ | $.772_{210}$ | $.435_{110}$ | $.380_{115}$ | $1.073_{137}$ | $1.205_{193}$ | $.479_{088}$ | $.310_{049}$ | $.367_{046}$ | $.136_{022}$ |
|  | EOT | $1.140_{262}$ | $.739_{230}$ | $.407_{095}$ | $.381_{149}$ | $1.058_{148}$ | $1.211_{179}$ | $.467_{070}$ | $.318_{048}$ | $.373_{054}$ | $.136_{024}$ |

| Learning | Labeling | ABA | AI2 | KRA | CO1 | PU1 | BA1 | CO2 | PU2 | BA2 | EL2 |
|---|---|---|---|---|---|---|---|---|---|---|---|
| AD | NNT | $.668_{021}$ | $.559_{017}$ | $.302_{012}$ | $.297_{013}$ | $.589_{013}$ | $.206_{010}$ | $.361_{013}$ | $.274_{010}$ | $.659_{017}$ | $.666_{017}$ |
|  | EOT | $.662_{024}$ | $.560_{014}$ | $.300_{013}$ | $.296_{011}$ | $.592_{014}$ | $.204_{011}$ | $.362_{013}$ | $.274_{013}$ | $.658_{016}$ | $.665_{016}$ |
| Hinge-IT non-ordered | MT | $.688_{028}$ | $.560_{016}$ | $.287_{012}$ | $.296_{012}$ | $.623_{019}$ | $.205_{009}$ | $.361_{012}$ | $.280_{011}$ | $.672_{018}$ | $.709_{020}$ |
|  | ST | $.688_{028}$ | $.560_{016}$ | $.287_{012}$ | $.296_{012}$ | $.623_{019}$ | $.205_{009}$ | $.361_{012}$ | $.280_{011}$ | $.672_{018}$ | $.709_{020}$ |
|  | EOT | $\mathbf{.661_{024}}$ | $.560_{016}$ | $.284_{013}$ | $.292_{013}$ | $\mathbf{.589_{012}}$ | $.204_{010}$ | $.356_{012}$ | $\mathbf{.266_{010}}$ | $.656_{018}$ | $\mathbf{.660_{015}}$ |
| Hinge-IT ordered | MT,ST | $.688_{029}$ | $.559_{015}$ | $.286_{014}$ | $.294_{011}$ | $.617_{018}$ | $.203_{009}$ | $.357_{012}$ | $.273_{011}$ | $.673_{020}$ | $.722_{020}$ |
|  | EOT | $\mathbf{.662_{024}}$ | $.560_{015}$ | $.286_{013}$ | $.293_{012}$ | $\mathbf{.590_{014}}$ | $.205_{008}$ | $.354_{012}$ | $.270_{011}$ | $\mathbf{.658_{017}}$ | $\mathbf{.662_{015}}$ |
| Hinge-AT ordered | MT,ST | $.659_{023}$ | $.565_{017}$ | $.284_{010}$ | $.292_{011}$ | $.591_{013}$ | $.205_{009}$ | $.355_{013}$ | $.267_{011}$ | $.658_{018}$ | $.668_{015}$ |
|  | EOT | $.658_{023}$ | $\mathbf{.559_{015}}$ | $.285_{011}$ | $.291_{009}$ | $.588_{014}$ | $.204_{009}$ | $.355_{012}$ | $.267_{010}$ | $.656_{018}$ | $\mathbf{.660_{016}}$ |
| Logistic-IT non-ordered | MT | $.678_{024}$ | $.584_{017}$ | $.290_{014}$ | $.294_{012}$ | $.594_{013}$ | $.204_{010}$ | $.357_{013}$ | $.309_{013}$ | $.665_{017}$ | $.683_{016}$ |
|  | ST | $.678_{024}$ | $.584_{017}$ | $.290_{014}$ | $.294_{012}$ | $.594_{013}$ | $.204_{010}$ | $.357_{013}$ | $.309_{013}$ | $.665_{017}$ | $.683_{016}$ |
|  | EOT | $\mathbf{.659_{024}}$ | $\mathbf{.559_{013}}$ | $\mathbf{.281_{012}}$ | $.293_{011}$ | $.589_{014}$ | $.203_{009}$ | $.354_{012}$ | $\mathbf{.273_{011}}$ | $\mathbf{.657_{017}}$ | $\mathbf{.661_{016}}$ |
| Logistic-IT ordered | MT,ST | $.679_{027}$ | $.582_{015}$ | $.278_{011}$ | $.291_{013}$ | $.595_{013}$ | $.202_{010}$ | $.353_{012}$ | $.273_{014}$ | $.667_{020}$ | $.686_{015}$ |
|  | EOT | $\mathbf{.660_{024}}$ | $\mathbf{.561_{015}}$ | $.278_{011}$ | $.290_{011}$ | $\mathbf{.589_{012}}$ | $.204_{010}$ | $.351_{012}$ | $.273_{011}$ | $\mathbf{.657_{019}}$ | $\mathbf{.662_{016}}$ |
| Logistic-AT ordered MT,ST,LB | MT,ST,LB | $.658_{023}$ | $.565_{015}$ | $.278_{012}$ | $.291_{012}$ | $.590_{013}$ | $.203_{009}$ | $.353_{012}$ | $.261_{010}$ | $.656_{019}$ | $.663_{016}$ |
|  | EOT | $.658_{021}$ | $.558_{013}$ | $.279_{012}$ | $.290_{010}$ | $.589_{013}$ | $.204_{008}$ | $.351_{011}$ | $.261_{011}$ | $.656_{016}$ | $.663_{015}$ |
| OLR-NLL ordered MT,ST,LB | MT,ST,LB | $.655_{021}$ | $.562_{016}$ | $.277_{013}$ | $.291_{010}$ | $.589_{013}$ | $.202_{009}$ | $.353_{012}$ | $.264_{013}$ | $.655_{019}$ | $.662_{015}$ |
|  | EOT | $.657_{021}$ | $.558_{013}$ | $.278_{012}$ | $.290_{010}$ | $.588_{016}$ | $.202_{009}$ | $\mathbf{.349_{011}}$ | $.264_{012}$ | $.656_{018}$ | $.663_{016}$ |

| Learning | Labeling | POT | AI1 | EL1 | CAL | CE1 | CE2 | 2DP | FRA | MVA |
|---|---|---|---|---|---|---|---|---|---|---|
| AD | NNT | $.577_{014}$ | $.441_{012}$ | $.581_{013}$ | $.450_{011}$ | $.617_{010}$ | $.592_{013}$ | $.243_{005}$ | $.221_{005}$ | $.022_{005}$ |
|  | EOT | $\mathbf{.549_{013}}$ | $.439_{011}$ | $.582_{013}$ | $.448_{011}$ | $.616_{011}$ | $.592_{009}$ | $.241_{005}$ | $\mathbf{.219_{005}}$ | $\mathbf{.019_{005}}$ |
| Hinge-IT non-ordered | MT | $.606_{024}$ | $.464_{011}$ | $.616_{015}$ | $.539_{012}$ | $.634_{015}$ | $.596_{012}$ | $.246_{005}$ | $.217_{005}$ | $.097_{018}$ |
|  | ST | $.606_{024}$ | $.464_{011}$ | $.616_{015}$ | $.539_{012}$ | $.634_{015}$ | $.596_{012}$ | $.246_{005}$ | $.217_{005}$ | $.097_{018}$ |
|  | EOT | $\mathbf{.528_{015}}$ | $\mathbf{.454_{010}}$ | $\mathbf{.572_{012}}$ | $\mathbf{.527_{009}}$ | $\mathbf{.612_{011}}$ | $.575_{011}$ | $\mathbf{.242_{004}}$ | $\mathbf{.214_{004}}$ | $\mathbf{.091_{014}}$ |
| Hinge-IT ordered | MT,ST | $.664_{025}$ | $.441_{011}$ | $.617_{013}$ | $.535_{010}$ | $.632_{013}$ | $.589_{013}$ | $.247_{004}$ | $.214_{004}$ | $.015_{004}$ |
|  | EOT | $\mathbf{.518_{012}}$ | $.437_{011}$ | $.571_{012}$ | $.527_{010}$ | $.610_{011}$ | $.575_{011}$ | $\mathbf{.241_{004}}$ | $.213_{004}$ | $\mathbf{.014_{003}}$ |
| Hinge-AT ordered | MT,ST | $.557_{019}$ | $.456_{012}$ | $.577_{012}$ | $.530_{010}$ | $.615_{010}$ | $.579_{012}$ | $.246_{005}$ | $.216_{004}$ | $.073_{023}$ |
|  | EOT | $\mathbf{.549_{018}}$ | $.454_{011}$ | $.576_{012}$ | $.529_{009}$ | $.613_{010}$ | $.577_{011}$ | $.241_{004}$ | $.215_{004}$ | $\mathbf{.067_{021}}$ |
| Logistic-IT non-ordered | MT | $.662_{020}$ | $.464_{012}$ | $.617_{015}$ | $.536_{012}$ | $.630_{011}$ | $.583_{014}$ | $.245_{005}$ | $.219_{006}$ | $.078_{008}$ |
|  | ST | $.662_{020}$ | $.464_{012}$ | $.617_{015}$ | $.536_{012}$ | $.630_{011}$ | $.583_{014}$ | $.245_{005}$ | $.219_{006}$ | $.078_{008}$ |
|  | EOT | $\mathbf{.519_{012}}$ | $\mathbf{.454_{009}}$ | $\mathbf{.576_{013}}$ | $\mathbf{.527_{009}}$ | $\mathbf{.614_{011}}$ | $.568_{011}$ | $\mathbf{.242_{004}}$ | $\mathbf{.211_{005}}$ | $\mathbf{.068_{005}}$ |
| Logistic-IT ordered | MT,ST | $.660_{030}$ | $.437_{011}$ | $.616_{014}$ | $.533_{010}$ | $.630_{012}$ | $.578_{012}$ | $.244_{004}$ | $.212_{004}$ | $.015_{003}$ |
|  | EOT | $\mathbf{.515_{013}}$ | $.434_{010}$ | $\mathbf{.577_{012}}$ | $.526_{010}$ | $\mathbf{.614_{010}}$ | $.566_{009}$ | $\mathbf{.242_{004}}$ | $.211_{004}$ | $.014_{003}$ |
| Logistic-AT ordered MT,ST,LB | MT,ST,LB | $.523_{013}$ | $.455_{009}$ | $.581_{011}$ | $.531_{009}$ | $.614_{011}$ | $.581_{013}$ | $.245_{005}$ | $.212_{005}$ | $.088_{046}$ |
|  | EOT | $.509_{011}$ | $.453_{010}$ | $\mathbf{.577_{012}}$ | $.531_{009}$ | $.613_{011}$ | $.579_{011}$ | $\mathbf{.242_{005}}$ | $.211_{005}$ | $\mathbf{.082_{045}}$ |
| OLR-NLL ordered MT,ST,LB | MT,ST,LB | $.520_{012}$ | $.454_{010}$ | $.575_{012}$ | $.527_{010}$ | $.612_{011}$ | $.570_{011}$ | $.245_{005}$ | $.212_{005}$ | $.072_{037}$ |
|  | EOT | $\mathbf{.506_{011}}$ | $.452_{010}$ | $.573_{013}$ | $.526_{010}$ | $.611_{010}$ | $.567_{009}$ | $\mathbf{.241_{004}}$ | $.211_{005}$ | $\mathbf{.068_{036}}$ |

Table 14: A counterpart of Table 2 regarding MAE for Task-A and EF10 datasets

| Learning | Labeling | DIA | PYR | APR | SER | TRI | WBC | CPU | AMP | BOS | STO |
|---|---|---|---|---|---|---|---|---|---|---|---|
| AD | NNT | $2.466_{569}$ | $1.397_{353}$ | $.852_{134}$ | $.783_{216}$ | $2.297_{347}$ | $2.524_{323}$ | $.995_{157}$ | $.721_{094}$ | $.791_{097}$ | $.362_{035}$ |
|  | EOT | $2.554_{527}$ | $1.396_{379}$ | $.820_{130}$ | $.778_{187}$ | $2.278_{356}$ | $2.533_{311}$ | $1.000_{149}$ | $.746_{089}$ | $.805_{093}$ | $.361_{038}$ |
| Hinge-IT non-ordered | MT | $2.436_{551}$ | $1.413_{404}$ | $.875_{132}$ | $.911_{232}$ | $2.314_{351}$ | $2.479_{315}$ | $1.046_{173}$ | $.742_{110}$ | $.826_{091}$ | $.414_{055}$ |
|  | ST | $2.444_{561}$ | $1.424_{402}$ | $.872_{132}$ | $.874_{238}$ | $2.304_{354}$ | $2.498_{316}$ | $1.040_{165}$ | $.742_{110}$ | $.826_{091}$ | $.406_{049}$ |
|  | EOT | $2.448_{537}$ | $1.433_{394}$ | $\mathbf{.817_{123}}$ | $.819_{195}$ | $2.277_{325}$ | $2.492_{329}$ | $1.008_{148}$ | $.733_{087}$ | $.814_{096}$ | $\mathbf{.374_{043}}$ |
| Hinge-IT ordered | MT,ST | $2.426_{534}$ | $1.456_{346}$ | $.874_{145}$ | $.830_{186}$ | $2.328_{307}$ | $2.566_{358}$ | $1.085_{165}$ | $.731_{089}$ | $.812_{084}$ | $.353_{043}$ |
|  | EOT | $2.542_{489}$ | $1.468_{375}$ | $.823_{133}$ | $.798_{178}$ | $2.259_{303}$ | $2.505_{272}$ | $\mathbf{1.025_{149}}$ | $.739_{075}$ | $.813_{087}$ | $.347_{038}$ |
| Hinge-AT ordered | MT,ST | $2.406_{499}$ | $1.427_{384}$ | $.844_{149}$ | $.742_{162}$ | $2.240_{314}$ | $2.541_{309}$ | $.989_{152}$ | $\mathbf{.702_{094}}$ | $.802_{097}$ | $.330_{039}$ |
|  | EOT | $2.476_{507}$ | $1.504_{380}$ | $.827_{142}$ | $.721_{161}$ | $2.219_{320}$ | $2.499_{336}$ | $.992_{140}$ | $.731_{082}$ | $.807_{092}$ | $.329_{035}$ |
| Logistic-IT non-ordered | MT | $2.356_{496}$ | $1.456_{411}$ | $.873_{152}$ | $.825_{208}$ | $2.303_{285}$ | $2.531_{408}$ | $1.042_{160}$ | $.737_{092}$ | $.824_{092}$ | $.384_{059}$ |
|  | ST | $2.368_{476}$ | $1.451_{407}$ | $.865_{149}$ | $.817_{211}$ | $2.295_{287}$ | $2.536_{397}$ | $1.033_{150}$ | $.737_{092}$ | $.824_{092}$ | $.383_{058}$ |
|  | EOT | $2.522_{543}$ | $1.447_{379}$ | $.818_{116}$ | $.783_{190}$ | $2.239_{319}$ | $2.489_{377}$ | $.995_{141}$ | $.743_{082}$ | $.810_{099}$ | $.365_{043}$ |
| Logistic-IT ordered | MT,ST | $\mathbf{2.330_{543}}$ | $1.453_{365}$ | $.838_{151}$ | $.841_{210}$ | $2.302_{294}$ | $2.606_{430}$ | $1.069_{182}$ | $.732_{084}$ | $.811_{088}$ | $.331_{032}$ |
|  | EOT | $2.530_{551}$ | $1.487_{405}$ | $.827_{124}$ | $.783_{198}$ | $2.234_{239}$ | $2.607_{381}$ | $1.026_{146}$ | $.735_{088}$ | $.813_{087}$ | $.335_{041}$ |
| Logistic-AT ordered | MT,ST,LB | $\mathbf{2.324_{473}}$ | $1.465_{419}$ | $.837_{130}$ | $.733_{171}$ | $2.288_{270}$ | $2.520_{350}$ | $.995_{149}$ | $.719_{093}$ | $.796_{106}$ | $.312_{034}$ |
|  | EOT | $2.526_{544}$ | $1.471_{412}$ | $.838_{131}$ | $.732_{187}$ | $2.233_{275}$ | $2.511_{323}$ | $1.002_{139}$ | $.727_{092}$ | $.799_{096}$ | $.315_{037}$ |
| OLR-NLL ordered | MT,ST,LB | $2.362_{465}$ | $1.465_{435}$ | $.842_{118}$ | $.744_{195}$ | $2.252_{290}$ | $2.508_{318}$ | $1.008_{145}$ | $.695_{087}$ | $.797_{088}$ | $.317_{034}$ |
|  | EOT | $2.474_{621}$ | $1.465_{408}$ | $.812_{129}$ | $.720_{185}$ | $2.206_{283}$ | $2.548_{381}$ | $1.004_{131}$ | $.717_{095}$ | $.805_{086}$ | $.314_{030}$ |

| Learning | Labeling | ABA | AI2 | KRA | CO1 | PU1 | BA1 | CO2 | PU2 | BA2 | EL2 |
|---|---|---|---|---|---|---|---|---|---|---|---|
| AD | NNT | $1.375_{040}$ | $1.149_{026}$ | $.599_{025}$ | $.606_{017}$ | $1.249_{027}$ | $.424_{014}$ | $.732_{022}$ | $.542_{026}$ | $1.379_{030}$ | $1.335_{025}$ |
|  | EOT | $1.373_{037}$ | $1.148_{025}$ | $.598_{021}$ | $.604_{015}$ | $1.251_{028}$ | $.425_{012}$ | $.730_{022}$ | $\mathbf{.532_{023}}$ | $1.380_{029}$ | $1.333_{024}$ |
| Hinge-IT non-ordered | MT | $1.404_{044}$ | $1.200_{032}$ | $.595_{020}$ | $.615_{017}$ | $1.272_{026}$ | $.426_{013}$ | $.746_{018}$ | $.541_{019}$ | $1.384_{031}$ | $1.377_{027}$ |
|  | ST | $1.403_{045}$ | $1.200_{032}$ | $.595_{020}$ | $.615_{017}$ | $1.272_{026}$ | $.426_{013}$ | $.746_{018}$ | $.541_{019}$ | $1.384_{031}$ | $1.377_{027}$ |
|  | EOT | $\mathbf{1.382_{041}}$ | $\mathbf{1.152_{024}}$ | $.581_{018}$ | $.611_{017}$ | $\mathbf{1.248_{024}}$ | $.419_{011}$ | $.736_{019}$ | $.532_{017}$ | $1.382_{027}$ | $\mathbf{1.334_{023}}$ |
| Hinge-IT ordered | MT,ST | $1.463_{053}$ | $1.218_{024}$ | $.593_{023}$ | $.606_{016}$ | $1.395_{041}$ | $.420_{012}$ | $.741_{021}$ | $.543_{020}$ | $1.415_{034}$ | $1.366_{026}$ |
|  | EOT | $\mathbf{1.394_{039}}$ | $\mathbf{1.163_{024}}$ | $.590_{018}$ | $.607_{018}$ | $\mathbf{1.253_{025}}$ | $.423_{014}$ | $\mathbf{.733_{018}}$ | $.539_{020}$ | $\mathbf{1.391_{030}}$ | $\mathbf{1.339_{026}}$ |
| Hinge-AT ordered | MT,ST | $1.366_{038}$ | $1.177_{026}$ | $.586_{018}$ | $.596_{015}$ | $1.258_{025}$ | $.420_{013}$ | $.720_{018}$ | $.521_{019}$ | $1.387_{032}$ | $1.365_{022}$ |
|  | EOT | $1.375_{039}$ | $\mathbf{1.149_{022}}$ | $.584_{020}$ | $.599_{013}$ | $\mathbf{1.248_{024}}$ | $.420_{011}$ | $.719_{020}$ | $.517_{017}$ | $1.382_{031}$ | $\mathbf{1.333_{025}}$ |
| Logistic-IT non-ordered | MT | $1.397_{044}$ | $1.226_{030}$ | $.588_{022}$ | $.611_{016}$ | $1.266_{026}$ | $.424_{013}$ | $.744_{021}$ | $.541_{023}$ | $1.386_{034}$ | $1.380_{028}$ |
|  | ST | $1.397_{044}$ | $1.226_{030}$ | $.588_{022}$ | $.611_{016}$ | $1.266_{026}$ | $.424_{013}$ | $.744_{021}$ | $.541_{023}$ | $1.386_{034}$ | $1.382_{028}$ |
|  | EOT | $\mathbf{1.383_{042}}$ | $\mathbf{1.151_{024}}$ | $.578_{021}$ | $.609_{014}$ | $\mathbf{1.249_{027}}$ | $.420_{012}$ | $.740_{020}$ | $\mathbf{.528_{017}}$ | $1.385_{032}$ | $\mathbf{1.333_{025}}$ |
| Logistic-IT ordered | MT,ST | $1.437_{054}$ | $1.244_{027}$ | $.596_{022}$ | $.603_{017}$ | $1.297_{025}$ | $.419_{013}$ | $.736_{025}$ | $.555_{018}$ | $1.402_{035}$ | $1.363_{027}$ |
|  | EOT | $\mathbf{1.398_{040}}$ | $\mathbf{1.169_{026}}$ | $.591_{022}$ | $.601_{017}$ | $\mathbf{1.249_{026}}$ | $.419_{012}$ | $.735_{020}$ | $.550_{017}$ | $\mathbf{1.387_{029}}$ | $\mathbf{1.335_{024}}$ |
| Logistic-AT ordered | MT,ST,LB | $1.363_{039}$ | $1.151_{025}$ | $.590_{018}$ | $.598_{014}$ | $1.251_{024}$ | $.418_{010}$ | $.722_{021}$ | $.520_{019}$ | $1.386_{029}$ | $1.339_{024}$ |
|  | EOT | $1.371_{039}$ | $1.148_{024}$ | $.587_{021}$ | $.598_{016}$ | $1.250_{027}$ | $.421_{013}$ | $.720_{020}$ | $.516_{019}$ | $1.385_{028}$ | $1.333_{024}$ |
| OLR-NLL ordered | MT,ST,LB | $\mathbf{1.362_{041}}$ | $1.150_{024}$ | $.577_{021}$ | $.594_{014}$ | $1.245_{026}$ | $.418_{013}$ | $.721_{020}$ | $.522_{018}$ | $1.383_{029}$ | $1.334_{027}$ |
|  | EOT | $1.374_{042}$ | $1.143_{022}$ | $.574_{022}$ | $.594_{016}$ | $1.248_{025}$ | $.418_{012}$ | $.717_{020}$ | $.523_{017}$ | $1.381_{033}$ | $1.331_{024}$ |

| Learning | Labeling | POT | AI1 | EL1 | CAL | CE1 | CE2 | 2DP | FRA | MVA |
|---|---|---|---|---|---|---|---|---|---|---|
| AD | NNT | $1.155_{025}$ | $.899_{014}$ | $1.196_{019}$ | $.915_{020}$ | $1.144_{016}$ | $1.084_{015}$ | $.496_{007}$ | $.458_{009}$ | $.249_{087}$ |
|  | EOT | $1.145_{022}$ | $.898_{015}$ | $1.191_{019}$ | $.913_{018}$ | $1.142_{017}$ | $1.082_{015}$ | $.495_{006}$ | $\mathbf{.454_{007}}$ | $\mathbf{.217_{082}}$ |
| Hinge-IT non-ordered | MT | $1.329_{041}$ | $.947_{015}$ | $1.234_{024}$ | $1.112_{017}$ | $1.323_{022}$ | $1.237_{020}$ | $.502_{006}$ | $.456_{006}$ | $.373_{094}$ |
|  | ST | $1.329_{041}$ | $.947_{015}$ | $1.234_{024}$ | $1.112_{017}$ | $1.323_{022}$ | $1.237_{020}$ | $.502_{006}$ | $.456_{006}$ | $.373_{094}$ |
|  | EOT | $\mathbf{1.197_{024}}$ | $.942_{015}$ | $\mathbf{1.177_{021}}$ | $\mathbf{1.094_{015}}$ | $\mathbf{1.294_{019}}$ | $\mathbf{1.210_{018}}$ | $.495_{007}$ | $\mathbf{.450_{005}}$ | $\mathbf{.295_{088}}$ |
| Hinge-IT ordered | MT,ST | $1.401_{055}$ | $.912_{015}$ | $1.266_{028}$ | $1.138_{021}$ | $1.421_{030}$ | $1.341_{032}$ | $.504_{007}$ | $.452_{007}$ | $.171_{075}$ |
|  | EOT | $\mathbf{1.121_{020}}$ | $\mathbf{.901_{013}}$ | $\mathbf{1.191_{022}}$ | $\mathbf{1.100_{014}}$ | $\mathbf{1.301_{019}}$ | $\mathbf{1.230_{019}}$ | $.495_{006}$ | $\mathbf{.448_{007}}$ | $\mathbf{.148_{062}}$ |
| Hinge-AT ordered | MT,ST | $1.125_{021}$ | $.942_{017}$ | $1.186_{021}$ | $1.105_{015}$ | $1.283_{020}$ | $1.210_{019}$ | $.500_{008}$ | $.449_{007}$ | $.210_{081}$ |
|  | EOT | $1.126_{020}$ | $.940_{014}$ | $1.185_{020}$ | $1.102_{016}$ | $1.277_{017}$ | $1.205_{020}$ | $.494_{006}$ | $.447_{007}$ | $\mathbf{.169_{048}}$ |
| Logistic-IT non-ordered | MT | $1.447_{035}$ | $.945_{014}$ | $1.237_{022}$ | $1.113_{017}$ | $1.314_{022}$ | $1.233_{021}$ | $.499_{007}$ | $.461_{010}$ | $.279_{105}$ |
|  | ST | $1.446_{035}$ | $.945_{014}$ | $1.237_{022}$ | $1.113_{017}$ | $1.314_{022}$ | $1.233_{021}$ | $.499_{007}$ | $.461_{010}$ | $.279_{105}$ |
|  | EOT | $\mathbf{1.189_{029}}$ | $.940_{016}$ | $\mathbf{1.183_{021}}$ | $\mathbf{1.095_{016}}$ | $\mathbf{1.291_{018}}$ | $\mathbf{1.212_{018}}$ | $.495_{006}$ | $\mathbf{.451_{007}}$ | $\mathbf{.232_{080}}$ |
| Logistic-IT ordered | MT,ST | $1.279_{032}$ | $.910_{015}$ | $1.256_{025}$ | $.943_{022}$ | $1.219_{020}$ | $1.124_{019}$ | $.496_{007}$ | $.451_{008}$ | $.153_{045}$ |
|  | EOT | $1.119_{021}$ | $.905_{015}$ | $\mathbf{1.214_{022}}$ | $\mathbf{.933_{020}}$ | $\mathbf{1.172_{018}}$ | $\mathbf{1.093_{014}}$ | $.494_{006}$ | $\mathbf{.448_{007}}$ | $.138_{043}$ |
| Logistic-AT ordered | MT,ST,LB | $1.132_{020}$ | $.942_{014}$ | $1.191_{021}$ | $1.107_{015}$ | $1.287_{019}$ | $1.221_{024}$ | $.498_{007}$ | $.450_{007}$ | $.227_{080}$ |
|  | EOT | $1.128_{020}$ | $.939_{016}$ | $1.188_{020}$ | $1.103_{015}$ | $\mathbf{1.280_{020}}$ | $1.214_{018}$ | $\mathbf{.494_{006}}$ | $.446_{006}$ | $\mathbf{.197_{071}}$ |
| OLR-NLL ordered | MT,ST,LB | $1.198_{023}$ | $.940_{015}$ | $1.183_{022}$ | $1.096_{016}$ | $1.287_{018}$ | $1.197_{019}$ | $.498_{007}$ | $.448_{006}$ | $.199_{087}$ |
|  | EOT | $\mathbf{1.167_{021}}$ | $.938_{017}$ | $1.175_{020}$ | $1.092_{014}$ | $1.284_{020}$ | $1.194_{015}$ | $.495_{006}$ | $.447_{006}$ | $.181_{083}$ |

Table 15: A counterpart of Table 2 regarding MAE for Task-A and EL3 datasets.

| Learning | Labeling | DIA | PYR | APR | SER | TRI | WBC | CPU | AMP | BOS | STO |
|---|---|---|---|---|---|---|---|---|---|---|---|
| AD | NNT | $\mathbf{.330_{140}}$ | $.184_{122}$ | $.138_{065}$ | $.073_{042}$ | $.352_{092}$ | $.631_{105}$ | $.048_{039}$ | $.201_{040}$ | $.226_{054}$ | $.041_{017}$ |
| | EOT | $.382_{141}$ | $.176_{103}$ | $.126_{065}$ | $.069_{047}$ | $.351_{084}$ | $.621_{116}$ | $.046_{038}$ | $\mathbf{.181_{043}}$ | $\mathbf{.210_{048}}$ | $.043_{016}$ |
| Hinge-IT non-ordered | MT | $.346_{147}$ | $.193_{130}$ | $.150_{062}$ | $.070_{053}$ | $.342_{086}$ | $.631_{093}$ | $.048_{037}$ | $.193_{048}$ | $.199_{041}$ | $.032_{012}$ |
| | ST | $.346_{147}$ | $.193_{130}$ | $.150_{062}$ | $.070_{053}$ | $.342_{086}$ | $.631_{093}$ | $.048_{037}$ | $.193_{048}$ | $.199_{041}$ | $.032_{012}$ |
| | EOT | $.382_{147}$ | $.189_{095}$ | $\mathbf{.124_{052}}$ | $.075_{047}$ | $.355_{084}$ | $.612_{103}$ | $.046_{038}$ | $\mathbf{.168_{042}}$ | $.193_{033}$ | $.036_{012}$ |
| Hinge-IT ordered | MT,ST | $\mathbf{.324_{156}}$ | $.185_{111}$ | $.146_{053}$ | $.073_{045}$ | $.336_{079}$ | $.643_{084}$ | $.050_{037}$ | $.197_{041}$ | $.201_{042}$ | $.031_{012}$ |
| | EOT | $.384_{155}$ | $.183_{115}$ | $\mathbf{.121_{060}}$ | $.072_{049}$ | $.348_{078}$ | $.620_{106}$ | $.049_{037}$ | $\mathbf{.170_{043}}$ | $.194_{036}$ | $.036_{013}$ |
| Hinge-AT ordered | MT,ST | $.340_{156}$ | $.179_{102}$ | $.147_{062}$ | $.080_{046}$ | $.329_{079}$ | $.623_{107}$ | $.050_{036}$ | $.194_{043}$ | $.207_{048}$ | $\mathbf{.030_{014}}$ |
| | EOT | $.374_{151}$ | $.187_{098}$ | $\mathbf{.121_{052}}$ | $.080_{047}$ | $.347_{073}$ | $.626_{111}$ | $.047_{039}$ | $\mathbf{.168_{044}}$ | $.205_{047}$ | $.034_{011}$ |
| Logistic-IT non-ordered | MT | $.392_{145}$ | $.223_{122}$ | $.151_{075}$ | $.068_{045}$ | $.365_{096}$ | $.649_{107}$ | $.056_{037}$ | $.168_{041}$ | $.186_{046}$ | $.029_{011}$ |
| | ST | $.392_{145}$ | $.223_{122}$ | $.151_{075}$ | $.068_{045}$ | $.365_{096}$ | $.649_{107}$ | $.056_{037}$ | $.168_{041}$ | $.186_{046}$ | $.029_{011}$ |
| | EOT | $.402_{154}$ | $.196_{109}$ | $\mathbf{.124_{068}}$ | $.078_{049}$ | $.352_{087}$ | $.625_{099}$ | $.053_{040}$ | $.161_{044}$ | $.184_{046}$ | $.030_{011}$ |
| Logistic-IT ordered | MT,ST | $.394_{141}$ | $.228_{140}$ | $.146_{063}$ | $.066_{045}$ | $.349_{074}$ | $.649_{097}$ | $.056_{032}$ | $.168_{039}$ | $.162_{035}$ | $.029_{011}$ |
| | EOT | $.428_{140}$ | $.212_{123}$ | $\mathbf{.122_{061}}$ | $.079_{051}$ | $.345_{086}$ | $.621_{105}$ | $.055_{035}$ | $.162_{042}$ | $.175_{045}$ | $.031_{012}$ |
| Logistic-AT ordered | MT,ST,LB | $.408_{128}$ | $.219_{124}$ | $.146_{061}$ | $.072_{043}$ | $.341_{096}$ | $.626_{106}$ | $.058_{035}$ | $.166_{043}$ | $.169_{039}$ | $.030_{009}$ |
| | EOT | $.406_{143}$ | $.199_{114}$ | $\mathbf{.121_{050}}$ | $.071_{043}$ | $.347_{096}$ | $.620_{101}$ | $.055_{036}$ | $.161_{040}$ | $.176_{038}$ | $.030_{010}$ |
| OLR-NLL ordered | MT,ST,LB | $.398_{135}$ | $.213_{129}$ | $.139_{064}$ | $.071_{047}$ | $.349_{096}$ | $.625_{109}$ | $.058_{036}$ | $.167_{039}$ | $.175_{046}$ | $.031_{012}$ |
| | EOT | $.418_{137}$ | $.207_{118}$ | $.119_{051}$ | $.079_{051}$ | $.349_{094}$ | $.612_{093}$ | $.054_{036}$ | $.165_{039}$ | $.183_{045}$ | $.030_{010}$ |

| Learning | Labeling | ABA | AI2 | KRA | CO1 | PU1 | BA1 | CO2 | PU2 | BA2 | EL2 |
|---|---|---|---|---|---|---|---|---|---|---|---|
| AD | NNT | $.237_{013}$ | $.030_{004}$ | $.198_{011}$ | $.012_{004}$ | $.287_{012}$ | $.065_{010}$ | $.015_{003}$ | $.126_{008}$ | $.092_{009}$ | $.062_{004}$ |
| | EOT | $.238_{013}$ | $.030_{005}$ | $\mathbf{.194_{014}}$ | $.011_{003}$ | $.285_{011}$ | $\mathbf{.055_{005}}$ | $.015_{003}$ | $\mathbf{.120_{007}}$ | $.084_{006}$ | $.061_{004}$ |
| Hinge-IT non-ordered | MT | $.236_{014}$ | $.030_{004}$ | $.150_{010}$ | $.010_{002}$ | $.273_{011}$ | $.056_{007}$ | $.013_{003}$ | $\mathbf{.110_{009}}$ | $.088_{006}$ | $.062_{004}$ |
| | ST | $.236_{014}$ | $.030_{004}$ | $.150_{010}$ | $.010_{002}$ | $.273_{011}$ | $.056_{007}$ | $.013_{003}$ | $\mathbf{.110_{009}}$ | $.088_{006}$ | $.062_{004}$ |
| | EOT | $.236_{014}$ | $.030_{004}$ | $.149_{011}$ | $.011_{003}$ | $.274_{011}$ | $.055_{006}$ | $.013_{003}$ | $.113_{010}$ | $\mathbf{.085_{006}}$ | $.061_{004}$ |
| Hinge-IT ordered | MT,ST | $.236_{015}$ | $.030_{004}$ | $.147_{012}$ | $.011_{002}$ | $.274_{010}$ | $.054_{006}$ | $.013_{002}$ | $.110_{008}$ | $.087_{006}$ | $.062_{004}$ |
| | EOT | $.236_{012}$ | $.030_{004}$ | $.145_{011}$ | $.011_{002}$ | $.274_{010}$ | $.054_{006}$ | $.013_{003}$ | $.114_{008}$ | $.086_{006}$ | $.061_{004}$ |
| Hinge-AT ordered | MT,ST | $.234_{011}$ | $.030_{004}$ | $.144_{009}$ | $.010_{003}$ | $.272_{011}$ | $.055_{006}$ | $.013_{003}$ | $.115_{010}$ | $.087_{005}$ | $.062_{004}$ |
| | EOT | $.235_{012}$ | $.030_{004}$ | $.143_{010}$ | $.010_{003}$ | $.273_{012}$ | $.054_{005}$ | $.013_{003}$ | $.117_{010}$ | $\mathbf{.085_{005}}$ | $.061_{004}$ |
| Logistic-IT non-ordered | MT | $.232_{013}$ | $.029_{004}$ | $.138_{008}$ | $.011_{003}$ | $.269_{010}$ | $.053_{005}$ | $.014_{003}$ | $.098_{008}$ | $.086_{007}$ | $.061_{004}$ |
| | ST | $.232_{013}$ | $.029_{004}$ | $.138_{008}$ | $.011_{003}$ | $.269_{010}$ | $.053_{005}$ | $.014_{003}$ | $.098_{008}$ | $.086_{007}$ | $.061_{004}$ |
| | EOT | $.231_{013}$ | $.030_{004}$ | $.138_{009}$ | $.011_{003}$ | $.270_{009}$ | $.054_{005}$ | $.013_{002}$ | $.098_{008}$ | $.086_{006}$ | $.061_{004}$ |
| Logistic-IT ordered | MT,ST | $.231_{013}$ | $.029_{004}$ | $.133_{008}$ | $.011_{002}$ | $.269_{010}$ | $.053_{004}$ | $.013_{003}$ | $.092_{007}$ | $.085_{006}$ | $.061_{004}$ |
| | EOT | $.229_{012}$ | $.030_{005}$ | $.131_{009}$ | $.010_{002}$ | $.270_{010}$ | $.054_{005}$ | $.013_{002}$ | $.091_{007}$ | $.086_{006}$ | $.061_{004}$ |
| Logistic-AT ordered | MT,ST,LB | $.231_{014}$ | $.029_{004}$ | $.133_{008}$ | $.011_{003}$ | $.269_{010}$ | $.053_{005}$ | $.014_{003}$ | $.092_{006}$ | $.086_{006}$ | $.061_{004}$ |
| | EOT | $.229_{014}$ | $.030_{004}$ | $.131_{008}$ | $.010_{002}$ | $.270_{009}$ | $.054_{005}$ | $.014_{003}$ | $.092_{006}$ | $.086_{007}$ | $.061_{004}$ |
| OLR-NLL ordered | MT,ST,LB | $.231_{013}$ | $.029_{004}$ | $.132_{007}$ | $.011_{003}$ | $.271_{010}$ | $.054_{005}$ | $.013_{002}$ | $.093_{007}$ | $.087_{007}$ | $.061_{004}$ |
| | EOT | $.231_{013}$ | $.030_{004}$ | $.132_{008}$ | $.011_{002}$ | $.270_{009}$ | $.053_{005}$ | $.013_{003}$ | $.093_{007}$ | $.086_{006}$ | $.061_{004}$ |

| Learning | Labeling | POT | AI1 | EL1 | CAL | CE1 | CE2 | 2DP | FRA | MVA |
|---|---|---|---|---|---|---|---|---|---|---|
| AD | NNT | $.077_{005}$ | $.068_{004}$ | $.019_{002}$ | $.229_{006}$ | $.035_{003}$ | $.044_{004}$ | $.094_{003}$ | $.125_{011}$ | $.002_{001}$ |
| | EOT | $.075_{006}$ | $.069_{005}$ | $\mathbf{.018_{002}}$ | $.228_{006}$ | $.035_{002}$ | $.043_{004}$ | $.095_{003}$ | $.123_{008}$ | $.001_{000}$ |
| Hinge-IT non-ordered | MT | $.087_{005}$ | $.069_{005}$ | $.019_{002}$ | $.219_{006}$ | $.033_{002}$ | $.044_{003}$ | $.095_{003}$ | $.094_{007}$ | $.002_{001}$ |
| | ST | $.087_{005}$ | $.069_{005}$ | $.019_{002}$ | $.219_{006}$ | $.033_{002}$ | $.044_{003}$ | $.095_{003}$ | $.094_{007}$ | $.002_{001}$ |
| | EOT | $\mathbf{.077_{005}}$ | $.069_{005}$ | $.018_{002}$ | $.218_{006}$ | $.033_{003}$ | $.044_{003}$ | $.095_{003}$ | $\mathbf{.092_{006}}$ | $\mathbf{.001_{000}}$ |
| Hinge-IT ordered | MT,ST | $.089_{005}$ | $.069_{005}$ | $.019_{002}$ | $.219_{006}$ | $.033_{003}$ | $.044_{004}$ | $.095_{003}$ | $.091_{005}$ | $.002_{001}$ |
| | EOT | $\mathbf{.077_{005}}$ | $.068_{005}$ | $\mathbf{.018_{002}}$ | $.218_{007}$ | $.033_{003}$ | $\mathbf{.038_{003}}$ | $.095_{003}$ | $.090_{006}$ | $.002_{001}$ |
| Hinge-AT ordered | MT,ST | $.085_{005}$ | $.068_{005}$ | $.019_{002}$ | $.222_{006}$ | $.033_{002}$ | $.037_{003}$ | $.094_{003}$ | $.091_{005}$ | $.002_{001}$ |
| | EOT | $\mathbf{.077_{005}}$ | $.069_{004}$ | $\mathbf{.018_{002}}$ | $.222_{007}$ | $.033_{003}$ | $.038_{003}$ | $.095_{003}$ | $.090_{005}$ | $\mathbf{.001_{000}}$ |
| Logistic-IT non-ordered | MT | $.052_{004}$ | $.071_{004}$ | $.019_{002}$ | $.204_{005}$ | $.033_{002}$ | $.043_{003}$ | $.095_{003}$ | $.084_{003}$ | $.002_{001}$ |
| | ST | $.052_{004}$ | $.071_{004}$ | $.019_{002}$ | $.204_{005}$ | $.033_{002}$ | $.043_{003}$ | $.095_{003}$ | $.084_{003}$ | $.002_{001}$ |
| | EOT | $\mathbf{.037_{004}}$ | $.069_{005}$ | $\mathbf{.018_{002}}$ | $.203_{006}$ | $.033_{002}$ | $\mathbf{.041_{003}}$ | $.095_{003}$ | $.083_{003}$ | $\mathbf{.001_{000}}$ |
| Logistic-IT ordered | MT,ST | $.054_{004}$ | $.070_{005}$ | $.018_{002}$ | $.204_{006}$ | $.033_{002}$ | $.038_{003}$ | $.094_{003}$ | $.081_{004}$ | $.002_{000}$ |
| | EOT | $\mathbf{.038_{005}}$ | $.069_{005}$ | $.017_{002}$ | $.203_{006}$ | $.033_{002}$ | $.038_{003}$ | $.095_{003}$ | $.081_{003}$ | $\mathbf{.001_{000}}$ |
| Logistic-AT ordered | MT,ST,LB | $.040_{005}$ | $.070_{005}$ | $.019_{002}$ | $.204_{005}$ | $.033_{002}$ | $.037_{003}$ | $.095_{003}$ | $.082_{003}$ | $.002_{000}$ |
| | EOT | $\mathbf{.036_{004}}$ | $.069_{004}$ | $\mathbf{.018_{002}}$ | $.203_{005}$ | $.033_{002}$ | $.038_{003}$ | $.095_{003}$ | $.081_{004}$ | $\mathbf{.001_{000}}$ |
| OLR-NLL ordered | MT,ST,LB | $.039_{004}$ | $.070_{005}$ | $.019_{002}$ | $.204_{006}$ | $.033_{002}$ | $.041_{003}$ | $.095_{003}$ | $.082_{003}$ | $.002_{000}$ |
| | EOT | $.035_{004}$ | $.069_{005}$ | $\mathbf{.018_{002}}$ | $.202_{006}$ | $.033_{002}$ | $.041_{003}$ | $.095_{003}$ | $.081_{003}$ | $\mathbf{.001_{000}}$ |

Table 16: A counterpart of Table 2 regarding MAE for Task-A and EL5 datasets.

| Learning | Labeling | DIA | PYR | APR | SER | TRI | WBC | CPU | AMP | BOS | STO |
|---|---|---|---|---|---|---|---|---|---|---|---|
| AD | NNT | $.698_{182}$ | $.520_{155}$ | $.203_{087}$ | $.152_{093}$ | $.730_{139}$ | $1.126_{158}$ | $.102_{054}$ | $.269_{053}$ | $.295_{049}$ | $.168_{025}$ |
| | EOT | $.654_{218}$ | $.548_{167}$ | $.213_{088}$ | $.150_{087}$ | $.757_{140}$ | $1.123_{170}$ | $.103_{059}$ | $.268_{055}$ | $.288_{048}$ | $.167_{030}$ |
| Hinge-IT non-ordered | MT | $.658_{270}$ | $.521_{221}$ | $.216_{092}$ | $.168_{090}$ | $.743_{147}$ | $1.109_{164}$ | $.114_{068}$ | $.283_{054}$ | $.321_{049}$ | $.208_{033}$ |
| | ST | $.658_{270}$ | $.512_{200}$ | $.219_{092}$ | $.168_{089}$ | $.739_{142}$ | $1.111_{164}$ | $.121_{073}$ | $.283_{054}$ | $.321_{049}$ | $.208_{033}$ |
| | EOT | $.656_{213}$ | $.532_{173}$ | $.218_{088}$ | $.186_{112}$ | $.742_{136}$ | $1.123_{179}$ | $.114_{060}$ | $.259_{050}$ | $.297_{042}$ | $.164_{024}$ |
| Hinge-IT ordered | MT,ST | $.670_{219}$ | $.531_{161}$ | $.210_{097}$ | $.174_{086}$ | $.744_{151}$ | $1.121_{135}$ | $.119_{069}$ | $.270_{047}$ | $.292_{046}$ | $.147_{025}$ |
| | EOT | $.668_{219}$ | $.529_{164}$ | $.205_{086}$ | $.176_{083}$ | $.759_{135}$ | $1.094_{142}$ | $.112_{066}$ | $.262_{049}$ | $.290_{042}$ | $.145_{025}$ |
| Hinge-AT ordered | MT,ST | $.722_{215}$ | $.500_{153}$ | $.195_{095}$ | $.168_{077}$ | $.732_{132}$ | $1.115_{161}$ | $.113_{063}$ | $.267_{046}$ | $.305_{058}$ | $.153_{030}$ |
| | EOT | $.644_{198}$ | $.527_{156}$ | $.205_{092}$ | $.168_{083}$ | $.749_{128}$ | $1.133_{178}$ | $.114_{060}$ | $.264_{055}$ | $.295_{044}$ | $.148_{025}$ |
| Logistic-IT non-ordered | MT | $.672_{203}$ | $.565_{214}$ | $.235_{097}$ | $.175_{076}$ | $.763_{145}$ | $1.152_{204}$ | $.135_{073}$ | $.267_{047}$ | $.318_{048}$ | $.145_{023}$ |
| | ST | $.670_{210}$ | $.540_{175}$ | $.236_{097}$ | $.174_{075}$ | $.757_{134}$ | $1.151_{204}$ | $.139_{081}$ | $.267_{047}$ | $.318_{048}$ | $.145_{023}$ |
| | EOT | $.660_{188}$ | $.525_{172}$ | $.219_{088}$ | $.181_{088}$ | $.734_{123}$ | $1.136_{194}$ | $.131_{067}$ | $.256_{052}$ | $.288_{040}$ | $.141_{027}$ |
| Logistic-IT ordered | MT,ST | $.676_{202}$ | $.531_{169}$ | $.237_{098}$ | $.168_{081}$ | $.741_{130}$ | $1.141_{165}$ | $.134_{068}$ | $.250_{045}$ | $.279_{043}$ | $.129_{021}$ |
| | EOT | $.676_{201}$ | $.517_{169}$ | $.216_{088}$ | $.185_{104}$ | $.737_{139}$ | $1.121_{188}$ | $.130_{067}$ | $.249_{053}$ | $.287_{047}$ | $.127_{023}$ |
| Logistic-AT ordered | MT,ST,LB | $.680_{196}$ | $.528_{163}$ | $.221_{093}$ | $.161_{077}$ | $.747_{127}$ | $1.114_{154}$ | $.132_{070}$ | $.252_{044}$ | $.279_{042}$ | $.130_{019}$ |
| | EOT | $.666_{214}$ | $.531_{151}$ | $.225_{087}$ | $.168_{077}$ | $.763_{126}$ | $1.119_{184}$ | $.127_{066}$ | $.259_{057}$ | $.291_{051}$ | $.130_{020}$ |
| OLR-NLL ordered | MT,ST,LB | $.658_{181}$ | $.553_{167}$ | $.225_{100}$ | $.162_{085}$ | $.745_{140}$ | $1.161_{161}$ | $.135_{071}$ | $.254_{044}$ | $.283_{042}$ | $.129_{021}$ |
| | EOT | $.652_{217}$ | $.515_{165}$ | $.215_{093}$ | $.168_{101}$ | $.764_{120}$ | $1.134_{194}$ | $.133_{064}$ | $.253_{050}$ | $.284_{042}$ | $.129_{019}$ |

| Learning | Labeling | ABA | AI2 | KRA | CO1 | PU1 | BA1 | CO2 | PU2 | BA2 | EL2 |
|---|---|---|---|---|---|---|---|---|---|---|---|
| AD | NNT | $.284_{028}$ | $.157_{009}$ | $.223_{012}$ | $.052_{006}$ | $.521_{014}$ | $.113_{009}$ | $.069_{006}$ | $.164_{009}$ | $.207_{009}$ | $.178_{009}$ |
| | EOT | $.241_{019}$ | $.140_{009}$ | $.221_{012}$ | $.050_{006}$ | $.519_{014}$ | $.110_{007}$ | $.067_{006}$ | $.160_{010}$ | $.204_{009}$ | $.168_{008}$ |
| Hinge-IT non-ordered | MT | $.268_{018}$ | $.154_{010}$ | $.234_{016}$ | $.054_{006}$ | $.565_{020}$ | $.111_{008}$ | $.069_{007}$ | $.202_{011}$ | $.227_{012}$ | $.178_{008}$ |
| | ST | $.268_{018}$ | $.154_{010}$ | $.234_{016}$ | $.054_{006}$ | $.565_{020}$ | $.111_{008}$ | $.069_{007}$ | $.202_{011}$ | $.227_{012}$ | $.178_{008}$ |
| | EOT | $.238_{015}$ | $.138_{009}$ | $.231_{014}$ | $.051_{005}$ | $.522_{012}$ | $.108_{008}$ | $.067_{006}$ | $.164_{011}$ | $.206_{011}$ | $.167_{008}$ |
| Hinge-IT ordered | MT,ST | $.268_{018}$ | $.146_{010}$ | $.221_{014}$ | $.052_{005}$ | $.544_{013}$ | $.111_{008}$ | $.069_{006}$ | $.169_{009}$ | $.226_{012}$ | $.177_{008}$ |
| | EOT | $.234_{015}$ | $.137_{009}$ | $.218_{015}$ | $.051_{005}$ | $.521_{013}$ | $.108_{009}$ | $.067_{006}$ | $.161_{009}$ | $.205_{010}$ | $.167_{008}$ |
| Hinge-AT ordered | MT,ST | $.260_{019}$ | $.147_{010}$ | $.228_{013}$ | $.053_{005}$ | $.525_{013}$ | $.111_{008}$ | $.068_{007}$ | $.164_{010}$ | $.213_{012}$ | $.176_{009}$ |
| | EOT | $.237_{015}$ | $.137_{009}$ | $.224_{014}$ | $.051_{005}$ | $.519_{013}$ | $.109_{008}$ | $.066_{007}$ | $.158_{009}$ | $.204_{008}$ | $.167_{008}$ |
| Logistic-IT non-ordered | MT | $.236_{017}$ | $.151_{009}$ | $.217_{014}$ | $.054_{005}$ | $.525_{015}$ | $.106_{008}$ | $.069_{006}$ | $.209_{010}$ | $.208_{011}$ | $.171_{009}$ |
| | ST | $.236_{017}$ | $.137_{009}$ | $.217_{014}$ | $.054_{005}$ | $.525_{015}$ | $.106_{008}$ | $.068_{006}$ | $.209_{010}$ | $.208_{011}$ | $.168_{009}$ |
| | EOT | $.233_{016}$ | $.138_{009}$ | $.213_{012}$ | $.050_{005}$ | $.520_{012}$ | $.107_{007}$ | $.064_{006}$ | $.160_{011}$ | $.204_{010}$ | $.167_{008}$ |
| Logistic-IT ordered | MT,ST | $.232_{016}$ | $.137_{010}$ | $.213_{014}$ | $.053_{005}$ | $.525_{014}$ | $.107_{007}$ | $.066_{006}$ | $.161_{012}$ | $.208_{010}$ | $.168_{008}$ |
| | EOT | $.232_{016}$ | $.137_{008}$ | $.210_{014}$ | $.050_{005}$ | $.522_{013}$ | $.107_{007}$ | $.063_{006}$ | $.157_{010}$ | $.204_{011}$ | $.167_{008}$ |
| Logistic-AT ordered | MT,ST,LB | $.230_{015}$ | $.137_{008}$ | $.219_{015}$ | $.052_{005}$ | $.524_{014}$ | $.108_{008}$ | $.064_{006}$ | $.160_{011}$ | $.205_{011}$ | $.167_{008}$ |
| | EOT | $.231_{014}$ | $.137_{009}$ | $.216_{013}$ | $.050_{005}$ | $.522_{013}$ | $.107_{008}$ | $.063_{007}$ | $.157_{011}$ | $.207_{012}$ | $.167_{008}$ |
| OLR-NLL ordered | MT,ST,LB | $.230_{016}$ | $.137_{010}$ | $.212_{014}$ | $.051_{005}$ | $.522_{013}$ | $.106_{007}$ | $.065_{007}$ | $.159_{010}$ | $.205_{010}$ | $.167_{009}$ |
| | EOT | $.231_{016}$ | $.137_{008}$ | $.209_{014}$ | $.050_{005}$ | $.521_{014}$ | $.106_{007}$ | $.063_{006}$ | $.158_{011}$ | $.207_{013}$ | $.167_{008}$ |

| Learning | Labeling | POT | AI1 | EL1 | CAL | CE1 | CE2 | 2DP | FRA | MVA |
|---|---|---|---|---|---|---|---|---|---|---|
| AD | NNT | $.049_{004}$ | $.174_{007}$ | $.045_{003}$ | $.387_{009}$ | $.106_{006}$ | $.140_{006}$ | $.172_{006}$ | $.142_{006}$ | $.004_{001}$ |
| | EOT | $.046_{004}$ | $.171_{006}$ | $.046_{004}$ | $.385_{009}$ | $.105_{006}$ | $.132_{006}$ | $.167_{005}$ | $.140_{004}$ | $.003_{001}$ |
| Hinge-IT non-ordered | MT | $.171_{017}$ | $.175_{006}$ | $.062_{007}$ | $.445_{011}$ | $.111_{008}$ | $.140_{006}$ | $.177_{003}$ | $.163_{009}$ | $.032_{005}$ |
| | ST | $.170_{017}$ | $.175_{006}$ | $.062_{007}$ | $.445_{011}$ | $.112_{007}$ | $.140_{006}$ | $.177_{003}$ | $.163_{009}$ | $.032_{005}$ |
| | EOT | $.150_{020}$ | $.170_{007}$ | $.051_{004}$ | $.421_{008}$ | $.103_{005}$ | $.124_{005}$ | $.177_{004}$ | $.152_{009}$ | $.006_{001}$ |
| Hinge-IT ordered | MT,ST | $.038_{004}$ | $.174_{006}$ | $.051_{004}$ | $.431_{009}$ | $.133_{014}$ | $.140_{006}$ | $.172_{006}$ | $.143_{007}$ | $.004_{001}$ |
| | EOT | $.036_{004}$ | $.170_{006}$ | $.048_{004}$ | $.420_{008}$ | $.103_{005}$ | $.125_{005}$ | $.165_{005}$ | $.137_{004}$ | $.003_{001}$ |
| Hinge-AT ordered | MT,ST | $.031_{004}$ | $.173_{006}$ | $.055_{005}$ | $.421_{009}$ | $.106_{005}$ | $.139_{006}$ | $.175_{005}$ | $.146_{006}$ | $.004_{001}$ |
| | EOT | $.030_{003}$ | $.170_{007}$ | $.049_{004}$ | $.420_{008}$ | $.103_{005}$ | $.118_{005}$ | $.167_{005}$ | $.138_{004}$ | $.003_{001}$ |
| Logistic-IT non-ordered | MT | $.100_{008}$ | $.177_{008}$ | $.057_{005}$ | $.442_{009}$ | $.105_{006}$ | $.112_{005}$ | $.172_{005}$ | $.144_{005}$ | $.043_{008}$ |
| | ST | $.100_{008}$ | $.174_{006}$ | $.057_{005}$ | $.442_{009}$ | $.103_{005}$ | $.112_{005}$ | $.172_{005}$ | $.144_{005}$ | $.043_{008}$ |
| | EOT | $.075_{007}$ | $.169_{007}$ | $.050_{004}$ | $.421_{007}$ | $.102_{004}$ | $.109_{005}$ | $.163_{003}$ | $.136_{004}$ | $.007_{002}$ |
| Logistic-IT ordered | MT,ST | $.030_{004}$ | $.167_{006}$ | $.046_{003}$ | $.433_{010}$ | $.104_{005}$ | $.112_{005}$ | $.163_{004}$ | $.137_{004}$ | $.003_{001}$ |
| | EOT | $.029_{004}$ | $.167_{006}$ | $.046_{003}$ | $.422_{008}$ | $.102_{005}$ | $.110_{005}$ | $.163_{004}$ | $.135_{004}$ | $.003_{001}$ |
| Logistic-AT ordered | MT,ST,LB | $.029_{003}$ | $.171_{006}$ | $.053_{004}$ | $.422_{009}$ | $.102_{005}$ | $.111_{005}$ | $.163_{004}$ | $.137_{004}$ | $.004_{001}$ |
| | EOT | $.028_{003}$ | $.169_{008}$ | $.049_{004}$ | $.422_{008}$ | $.102_{005}$ | $.111_{005}$ | $.162_{003}$ | $.135_{003}$ | $.003_{001}$ |
| OLR-NLL ordered | MT,ST,LB | $.029_{003}$ | $.171_{007}$ | $.050_{004}$ | $.420_{008}$ | $.101_{005}$ | $.109_{005}$ | $.163_{004}$ | $.136_{004}$ | $.010_{003}$ |
| | EOT | $.028_{004}$ | $.169_{007}$ | $.048_{003}$ | $.420_{007}$ | $.101_{005}$ | $.108_{005}$ | $.162_{004}$ | $.136_{004}$ | $.007_{001}$ |

Table 17: A counterpart of Table 2 regarding MAE for Task-A and EL10 datasets.

| Learning | Labeling | DIA | PYR | APR | SER | TRI | WBC | CPU | AMP | BOS | STO |
|---|---|---|---|---|---|---|---|---|---|---|---|
| AD | NNT | $1.348_{369}$ | $.852_{252}$ | $.523_{154}$ | $.391_{150}$ | $1.361_{224}$ | $2.412_{289}$ | $.261_{117}$ | $.558_{091}$ | $.547_{069}$ | $.286_{029}$ |
|  | EOT | $1.412_{376}$ | $.888_{261}$ | $.536_{149}$ | $.431_{193}$ | $1.351_{247}$ | $2.366_{257}$ | $.248_{107}$ | $.563_{087}$ | $.557_{077}$ | $.289_{027}$ |
| Hinge-IT non-ordered | MT | $1.354_{416}$ | $.881_{313}$ | $.553_{166}$ | $.561_{170}$ | $1.343_{216}$ | $2.369_{370}$ | $.272_{128}$ | $.579_{077}$ | $.583_{077}$ | $.382_{029}$ |
|  | ST | $1.354_{416}$ | $.869_{295}$ | $.564_{162}$ | $.523_{155}$ | $1.336_{216}$ | $2.378_{333}$ | $.278_{131}$ | $.585_{077}$ | $.582_{078}$ | $.382_{029}$ |
|  | EOT | $1.412_{374}$ | $.872_{268}$ | $.513_{157}$ | $\mathbf{.414_{165}}$ | $1.354_{229}$ | $2.349_{322}$ | $.265_{116}$ | $\mathbf{.542_{079}}$ | $\mathbf{.555_{077}}$ | $\mathbf{.295_{032}}$ |
| Hinge-IT ordered | MT,ST | $1.376_{422}$ | $.841_{271}$ | $.522_{174}$ | $.428_{171}$ | $1.343_{227}$ | $2.380_{359}$ | $.275_{121}$ | $.571_{086}$ | $.553_{081}$ | $.282_{031}$ |
|  | EOT | $1.366_{391}$ | $.851_{248}$ | $.515_{166}$ | $.452_{168}$ | $1.335_{260}$ | $2.341_{298}$ | $.259_{109}$ | $\mathbf{.537_{087}}$ | $.550_{094}$ | $.281_{030}$ |
| Hinge-AT ordered | MT,ST | $1.322_{363}$ | $.844_{224}$ | $.525_{147}$ | $.467_{188}$ | $1.334_{231}$ | $2.350_{312}$ | $.287_{129}$ | $.552_{083}$ | $.542_{072}$ | $.285_{028}$ |
|  | EOT | $1.358_{401}$ | $.868_{255}$ | $.493_{142}$ | $.456_{192}$ | $1.339_{239}$ | $2.365_{320}$ | $.265_{111}$ | $.549_{088}$ | $.555_{076}$ | $\mathbf{.276_{028}}$ |
| Logistic-IT non-ordered | MT | $1.366_{407}$ | $.924_{261}$ | $.578_{155}$ | $.583_{200}$ | $1.385_{230}$ | $2.368_{359}$ | $.294_{146}$ | $.559_{077}$ | $.552_{073}$ | $.307_{031}$ |
|  | ST | $1.352_{411}$ | $.897_{241}$ | $.572_{155}$ | $.538_{153}$ | $1.378_{255}$ | $2.354_{358}$ | $.318_{142}$ | $.561_{078}$ | $.551_{074}$ | $.307_{031}$ |
|  | EOT | $1.354_{396}$ | $.840_{256}$ | $\mathbf{.513_{153}}$ | $.526_{227}$ | $1.351_{259}$ | $2.394_{360}$ | $.269_{113}$ | $\mathbf{.535_{084}}$ | $.557_{079}$ | $\mathbf{.280_{025}}$ |
| Logistic-IT ordered | MT,ST | $1.350_{365}$ | $.876_{264}$ | $.537_{150}$ | $.378_{187}$ | $1.383_{203}$ | $2.383_{347}$ | $.277_{135}$ | $.552_{090}$ | $.557_{078}$ | $.274_{027}$ |
|  | EOT | $1.368_{364}$ | $.833_{244}$ | $.510_{150}$ | $.407_{208}$ | $1.372_{231}$ | $2.395_{361}$ | $.275_{140}$ | $.536_{086}$ | $.544_{080}$ | $.274_{026}$ |
| Logistic-AT ordered | MT,ST,LB | $1.300_{359}$ | $.841_{254}$ | $.530_{148}$ | $.426_{178}$ | $1.349_{223}$ | $2.298_{271}$ | $.275_{129}$ | $.542_{093}$ | $.546_{083}$ | $.267_{027}$ |
|  | EOT | $1.342_{372}$ | $.843_{230}$ | $.514_{141}$ | $.446_{207}$ | $1.345_{215}$ | $2.367_{289}$ | $.268_{121}$ | $.540_{078}$ | $.560_{074}$ | $.265_{025}$ |
| OLR-NLL ordered | MT,ST,LB | $1.362_{364}$ | $.827_{287}$ | $.518_{170}$ | $.368_{163}$ | $1.356_{209}$ | $2.377_{304}$ | $.269_{130}$ | $.544_{096}$ | $.544_{073}$ | $.263_{025}$ |
|  | EOT | $1.360_{389}$ | $.857_{230}$ | $.486_{135}$ | $.398_{214}$ | $1.361_{219}$ | $2.365_{349}$ | $.267_{118}$ | $.529_{085}$ | $.544_{066}$ | $.266_{027}$ |

| Learning | Labeling | ABA | AI2 | KRA | CO1 | PU1 | BA1 | CO2 | PU2 | BA2 | EL2 |
|---|---|---|---|---|---|---|---|---|---|---|---|
| AD | NNT | $.534_{026}$ | $.228_{012}$ | $.417_{017}$ | $.167_{009}$ | $.994_{020}$ | $.251_{010}$ | $.214_{011}$ | $.327_{014}$ | $.527_{024}$ | $.331_{012}$ |
|  | EOT | $.534_{021}$ | $\mathbf{.215_{012}}$ | $.417_{017}$ | $.167_{009}$ | $.995_{020}$ | $.253_{013}$ | $.212_{010}$ | $.322_{014}$ | $\mathbf{.516_{022}}$ | $\mathbf{.326_{011}}$ |
| Hinge-IT non-ordered | MT | $.616_{029}$ | $.229_{012}$ | $.435_{015}$ | $.169_{010}$ | $1.068_{030}$ | $.253_{013}$ | $.214_{012}$ | $.351_{015}$ | $.543_{022}$ | $.332_{012}$ |
|  | ST | $.607_{030}$ | $.229_{012}$ | $.435_{016}$ | $.168_{008}$ | $1.068_{030}$ | $.253_{013}$ | $.213_{012}$ | $.353_{017}$ | $.543_{022}$ | $.332_{012}$ |
|  | EOT | $\mathbf{.549_{023}}$ | $\mathbf{.213_{012}}$ | $.417_{020}$ | $\mathbf{.164_{009}}$ | $1.000_{020}$ | $\mathbf{.247_{011}}$ | $.207_{010}$ | $\mathbf{.329_{015}}$ | $\mathbf{.512_{021}}$ | $\mathbf{.326_{011}}$ |
| Hinge-IT ordered | MT,ST | $.582_{029}$ | $.228_{011}$ | $.436_{019}$ | $.161_{008}$ | $1.130_{031}$ | $.247_{012}$ | $.208_{010}$ | $.342_{013}$ | $.639_{026}$ | $.332_{012}$ |
|  | EOT | $\mathbf{.550_{021}}$ | $\mathbf{.210_{012}}$ | $.431_{020}$ | $.161_{009}$ | $\mathbf{1.003_{022}}$ | $.247_{012}$ | $.207_{011}$ | $.342_{014}$ | $\mathbf{.511_{020}}$ | $\mathbf{.325_{012}}$ |
| Hinge-AT ordered | MT,ST | $.557_{033}$ | $.224_{013}$ | $.427_{018}$ | $.163_{008}$ | $1.014_{025}$ | $.249_{011}$ | $.208_{012}$ | $.326_{012}$ | $.524_{019}$ | $.332_{011}$ |
|  | EOT | $\mathbf{.533_{022}}$ | $\mathbf{.211_{013}}$ | $.427_{019}$ | $.164_{010}$ | $\mathbf{.992_{023}}$ | $.248_{011}$ | $.207_{011}$ | $.324_{012}$ | $.510_{019}$ | $\mathbf{.323_{012}}$ |
| Logistic-IT non-ordered | MT | $.588_{031}$ | $.213_{012}$ | $.442_{019}$ | $.167_{009}$ | $1.016_{024}$ | $.247_{012}$ | $.208_{010}$ | $.352_{015}$ | $.521_{023}$ | $.325_{012}$ |
|  | ST | $.562_{029}$ | $.217_{012}$ | $.421_{019}$ | $.165_{007}$ | $1.016_{024}$ | $.247_{012}$ | $.208_{010}$ | $.355_{016}$ | $.521_{023}$ | $.323_{011}$ |
|  | EOT | $\mathbf{.536_{024}}$ | $\mathbf{.208_{011}}$ | $.410_{018}$ | $\mathbf{.160_{010}}$ | $.989_{021}$ | $.246_{011}$ | $\mathbf{.198_{010}}$ | $\mathbf{.327_{014}}$ | $\mathbf{.512_{020}}$ | $.321_{011}$ |
| Logistic-IT ordered | MT,ST | $.548_{027}$ | $.206_{012}$ | $.421_{018}$ | $.159_{009}$ | $1.028_{023}$ | $.245_{011}$ | $.199_{011}$ | $.341_{013}$ | $.547_{024}$ | $.321_{011}$ |
|  | EOT | $\mathbf{.540_{027}}$ | $.207_{011}$ | $.420_{019}$ | $.158_{007}$ | $\mathbf{.989_{021}}$ | $.243_{011}$ | $.200_{011}$ | $.336_{014}$ | $\mathbf{.520_{021}}$ | $.320_{011}$ |
| Logistic-AT ordered | MT,ST,LB | $.529_{020}$ | $.206_{011}$ | $.417_{021}$ | $.161_{007}$ | $.993_{022}$ | $.246_{011}$ | $.201_{010}$ | $.322_{012}$ | $.511_{021}$ | $.320_{012}$ |
|  | EOT | $.535_{026}$ | $.208_{011}$ | $.416_{020}$ | $.159_{008}$ | $.991_{021}$ | $.245_{012}$ | $.200_{010}$ | $.320_{014}$ | $.511_{022}$ | $.321_{011}$ |
| OLR-NLL ordered | MT,ST,LB | $.526_{020}$ | $.206_{010}$ | $.408_{020}$ | $.158_{008}$ | $.989_{020}$ | $.244_{011}$ | $.200_{012}$ | $.322_{014}$ | $.513_{021}$ | $.320_{011}$ |
|  | EOT | $.534_{022}$ | $.206_{011}$ | $.406_{019}$ | $.158_{008}$ | $.990_{021}$ | $.244_{011}$ | $.202_{016}$ | $.323_{011}$ | $.510_{021}$ | $.321_{011}$ |

| Learning | Labeling | POT | AI1 | EL1 | CAL | CE1 | CE2 | 2DP | FRA | MVA |
|---|---|---|---|---|---|---|---|---|---|---|
| AD | NNT | $.321_{035}$ | $.300_{008}$ | $.271_{006}$ | $.746_{015}$ | $.284_{010}$ | $.283_{010}$ | $.319_{005}$ | $.263_{005}$ | $.050_{004}$ |
|  | EOT | $.312_{037}$ | $.298_{008}$ | $.271_{005}$ | $\mathbf{.741_{014}}$ | $.285_{010}$ | $.283_{009}$ | $\mathbf{.316_{004}}$ | $.262_{005}$ | $\mathbf{.041_{004}}$ |
| Hinge-IT non-ordered | MT | $.211_{023}$ | $.329_{014}$ | $.327_{011}$ | $.867_{013}$ | $.314_{011}$ | $.348_{012}$ | $.348_{005}$ | $.308_{007}$ | $.080_{008}$ |
|  | ST | $.209_{022}$ | $.329_{014}$ | $.327_{011}$ | $.870_{011}$ | $.314_{011}$ | $.348_{012}$ | $.332_{006}$ | $.289_{006}$ | $.067_{007}$ |
|  | EOT | $\mathbf{.196_{023}}$ | $\mathbf{.300_{008}}$ | $\mathbf{.301_{008}}$ | $.855_{011}$ | $\mathbf{.300_{011}}$ | $.324_{011}$ | $\mathbf{.317_{004}}$ | $\mathbf{.269_{005}}$ | $\mathbf{.039_{003}}$ |
| Hinge-IT ordered | MT,ST | $.392_{018}$ | $.300_{009}$ | $.265_{005}$ | $.772_{015}$ | $.330_{013}$ | $.376_{038}$ | $.325_{006}$ | $.269_{004}$ | $.047_{005}$ |
|  | EOT | $\mathbf{.257_{017}}$ | $\mathbf{.298_{008}}$ | $.264_{006}$ | $.726_{012}$ | $\mathbf{.307_{011}}$ | $\mathbf{.339_{010}}$ | $\mathbf{.316_{005}}$ | $.268_{005}$ | $\mathbf{.038_{004}}$ |
| Hinge-AT ordered | MT,ST | $.195_{019}$ | $.300_{007}$ | $.263_{006}$ | $.737_{014}$ | $.312_{011}$ | $.324_{010}$ | $.326_{006}$ | $.270_{005}$ | $.051_{006}$ |
|  | EOT | $\mathbf{.163_{016}}$ | $\mathbf{.297_{007}}$ | $.264_{006}$ | $.733_{014}$ | $\mathbf{.295_{010}}$ | $\mathbf{.315_{010}}$ | $\mathbf{.316_{004}}$ | $.268_{005}$ | $\mathbf{.043_{004}}$ |
| Logistic-IT non-ordered | MT | $.153_{015}$ | $.311_{010}$ | $.316_{011}$ | $.862_{011}$ | $.302_{010}$ | $.324_{011}$ | $.345_{007}$ | $.309_{010}$ | $.079_{007}$ |
|  | ST | $.153_{015}$ | $.309_{009}$ | $.316_{011}$ | $.864_{011}$ | $.302_{010}$ | $.321_{011}$ | $.331_{006}$ | $.294_{006}$ | $.071_{006}$ |
|  | EOT | $\mathbf{.137_{013}}$ | $.298_{008}$ | $.292_{007}$ | $\mathbf{.853_{012}}$ | $.298_{011}$ | $.313_{011}$ | $\mathbf{.316_{004}}$ | $.268_{005}$ | $\mathbf{.038_{003}}$ |
| Logistic-IT ordered | MT,ST | $.235_{019}$ | $.292_{008}$ | $.262_{007}$ | $.756_{013}$ | $.283_{009}$ | $.342_{012}$ | $.319_{004}$ | $.260_{004}$ | $.038_{004}$ |
|  | EOT | $\mathbf{.165_{014}}$ | $.292_{007}$ | $.261_{007}$ | $\mathbf{.723_{010}}$ | $\mathbf{.272_{009}}$ | $\mathbf{.316_{011}}$ | $.317_{005}$ | $.260_{004}$ | $\mathbf{.033_{003}}$ |
| Logistic-AT ordered | MT,ST,LB | $.183_{014}$ | $.300_{008}$ | $.284_{007}$ | $.851_{011}$ | $.297_{010}$ | $.313_{010}$ | $.322_{005}$ | $.268_{005}$ | $.043_{004}$ |
|  | EOT | $\mathbf{.155_{013}}$ | $.297_{007}$ | $\mathbf{.275_{006}}$ | $.849_{010}$ | $.295_{010}$ | $.310_{010}$ | $.316_{005}$ | $.266_{005}$ | $\mathbf{.037_{003}}$ |
| OLR-NLL ordered | MT,ST,LB | $.151_{013}$ | $.298_{007}$ | $.281_{007}$ | $.848_{012}$ | $.295_{010}$ | $.307_{011}$ | $.321_{004}$ | $.267_{005}$ | $.040_{003}$ |
|  | EOT | $\mathbf{.135_{014}}$ | $.297_{008}$ | $\mathbf{.275_{007}}$ | $.846_{011}$ | $.295_{010}$ | $.304_{011}$ | $.316_{004}$ | $\mathbf{.266_{005}}$ | $\mathbf{.034_{003}}$ |

Table 18: A counterpart of Table 2 regarding RMSE for Task-S and EF3 datasets.

| Learning | Labeling | DIA | PYR | APR | SER | TRI | WBC | CPU | AMP | BOS | STO |
|---|---|---|---|---|---|---|---|---|---|---|---|
| AD | NNT | $.877_{174}$ | $.642_{121}$ | $.380_{075}$ | $.489_{098}$ | $.897_{100}$ | $.935_{111}$ | $.535_{076}$ | $.459_{053}$ | $.533_{047}$ | $.282_{036}$ |
|  | EOT | $.884_{152}$ | $.622_{135}$ | $.372_{085}$ | $.486_{106}$ | $.889_{108}$ | $.930_{111}$ | $.521_{072}$ | $\mathbf{.429_{045}}$ | $.532_{049}$ | $.279_{037}$ |
| Hinge-IT non-ordered | MT | $.887_{200}$ | $.639_{118}$ | $.391_{076}$ | $.470_{098}$ | $.896_{095}$ | $.937_{122}$ | $.528_{074}$ | $.459_{059}$ | $.520_{049}$ | $.282_{035}$ |
|  | ST | $.887_{200}$ | $.639_{118}$ | $.391_{076}$ | $.470_{098}$ | $.896_{095}$ | $.937_{122}$ | $.528_{074}$ | $.459_{059}$ | $.520_{049}$ | $.282_{035}$ |
|  | EOT | $.888_{126}$ | $.628_{106}$ | $.387_{065}$ | $.463_{102}$ | $.878_{112}$ | $.919_{103}$ | $.524_{070}$ | $\mathbf{.431_{050}}$ | $.517_{046}$ | $.273_{035}$ |
| Hinge-IT ordered | MT,ST | $.894_{190}$ | $.649_{119}$ | $.382_{080}$ | $.478_{099}$ | $.911_{112}$ | $.918_{110}$ | $.520_{071}$ | $.447_{055}$ | $.515_{050}$ | $.282_{036}$ |
|  | EOT | $.900_{157}$ | $.613_{109}$ | $.375_{085}$ | $.450_{078}$ | $.884_{114}$ | $.914_{097}$ | $.522_{065}$ | $.434_{044}$ | $.518_{044}$ | $.275_{036}$ |
| Hinge-AT ordered | MT,ST | $.908_{165}$ | $.639_{102}$ | $.381_{078}$ | $.483_{103}$ | $.900_{118}$ | $.915_{112}$ | $.510_{074}$ | $.450_{053}$ | $.517_{045}$ | $.277_{036}$ |
|  | EOT | $.874_{135}$ | $\mathbf{.598_{108}}$ | $.377_{070}$ | $.475_{087}$ | $.893_{105}$ | $.931_{119}$ | $.524_{075}$ | $.436_{048}$ | $.513_{048}$ | $.274_{036}$ |
| Logistic-IT non-ordered | MT | $.843_{183}$ | $.672_{141}$ | $.388_{086}$ | $.491_{088}$ | $.914_{097}$ | $.938_{118}$ | $.504_{073}$ | $.444_{058}$ | $.516_{043}$ | $.278_{035}$ |
|  | ST | $.843_{183}$ | $.672_{141}$ | $.388_{086}$ | $.491_{088}$ | $.914_{097}$ | $.938_{118}$ | $.504_{073}$ | $.444_{058}$ | $.516_{043}$ | $.278_{035}$ |
|  | EOT | $.871_{146}$ | $.631_{109}$ | $.377_{065}$ | $.484_{075}$ | $\mathbf{.872_{096}}$ | $.920_{115}$ | $.508_{072}$ | $\mathbf{.422_{045}}$ | $\mathbf{.502_{046}}$ | $.279_{035}$ |
| Logistic-IT ordered | MT,ST | $.849_{178}$ | $.671_{119}$ | $.379_{097}$ | $.496_{085}$ | $.896_{105}$ | $.920_{127}$ | $.519_{077}$ | $.441_{052}$ | $.502_{048}$ | $.269_{035}$ |
|  | EOT | $.876_{137}$ | $.636_{097}$ | $.358_{075}$ | $.473_{074}$ | $.878_{100}$ | $.933_{117}$ | $.515_{074}$ | $.425_{042}$ | $.500_{055}$ | $.271_{035}$ |
| Logistic-AT ordered | MT,ST | $.828_{130}$ | $.645_{116}$ | $.369_{095}$ | $.485_{084}$ | $.900_{099}$ | $.918_{107}$ | $.513_{075}$ | $.434_{046}$ | $.491_{049}$ | $.273_{037}$ |
|  | LB | $.855_{199}$ | $.654_{115}$ | $.373_{093}$ | $.485_{088}$ | $.903_{093}$ | $.919_{119}$ | $.516_{078}$ | $.441_{048}$ | $.493_{045}$ | $.272_{035}$ |
|  | EOT | $.880_{146}$ | $.633_{102}$ | $.360_{077}$ | $.476_{082}$ | $.883_{107}$ | $.925_{116}$ | $.517_{083}$ | $.427_{046}$ | $.496_{053}$ | $.275_{030}$ |
| OLR-NLL ordered | MT,ST | $.840_{129}$ | $.643_{116}$ | $.368_{092}$ | $.491_{090}$ | $.892_{094}$ | $.908_{099}$ | $.506_{084}$ | $.429_{037}$ | $.495_{049}$ | $.270_{036}$ |
|  | LB | $.867_{187}$ | $.663_{129}$ | $.370_{094}$ | $.498_{100}$ | $.892_{099}$ | $.906_{118}$ | $.510_{080}$ | $.444_{056}$ | $.493_{045}$ | $.270_{040}$ |
|  | EOT | $.878_{146}$ | $.633_{098}$ | $.359_{081}$ | $.475_{072}$ | $.882_{112}$ | $.912_{109}$ | $.520_{085}$ | $.422_{046}$ | $.493_{052}$ | $.275_{032}$ |

| Learning | Labeling | ABA | AI2 | KRA | CO1 | PU1 | BA1 | CO2 | PU2 | BA2 | EL2 |
|---|---|---|---|---|---|---|---|---|---|---|---|
| AD | NNT | $.651_{021}$ | $.613_{011}$ | $.426_{010}$ | $.419_{010}$ | $.597_{014}$ | $.326_{012}$ | $.456_{010}$ | $.409_{011}$ | $.630_{014}$ | $.627_{012}$ |
|  | EOT | $\mathbf{.640_{018}}$ | $.610_{011}$ | $.422_{011}$ | $.419_{010}$ | $.597_{012}$ | $.327_{012}$ | $.456_{012}$ | $.408_{010}$ | $\mathbf{.624_{014}}$ | $.623_{011}$ |
| Hinge-IT non-ordered | MT | $.660_{022}$ | $.631_{013}$ | $.397_{012}$ | $.416_{011}$ | $.593_{011}$ | $.319_{013}$ | $.456_{012}$ | $.404_{012}$ | $.640_{017}$ | $.629_{013}$ |
|  | ST | $.660_{022}$ | $.631_{013}$ | $.397_{012}$ | $.416_{011}$ | $.593_{011}$ | $.319_{013}$ | $.456_{012}$ | $.404_{012}$ | $.640_{017}$ | $.629_{013}$ |
|  | EOT | $\mathbf{.637_{016}}$ | $\mathbf{.610_{010}}$ | $.396_{012}$ | $.415_{011}$ | $.593_{012}$ | $.318_{013}$ | $.454_{011}$ | $\mathbf{.399_{010}}$ | $\mathbf{.624_{015}}$ | $\mathbf{.622_{010}}$ |
| Hinge-IT ordered | MT,ST | $.655_{023}$ | $.631_{014}$ | $.397_{013}$ | $.416_{010}$ | $.594_{011}$ | $.318_{012}$ | $.455_{011}$ | $.399_{011}$ | $.640_{014}$ | $.626_{013}$ |
|  | EOT | $\mathbf{.636_{018}}$ | $\mathbf{.611_{010}}$ | $.396_{011}$ | $.415_{010}$ | $.594_{011}$ | $.317_{013}$ | $.454_{011}$ | $.398_{012}$ | $\mathbf{.624_{014}}$ | $.623_{011}$ |
| Hinge-AT ordered | MT,ST | $.638_{018}$ | $.620_{014}$ | $.395_{013}$ | $.416_{011}$ | $.591_{011}$ | $.318_{013}$ | $.454_{010}$ | $.401_{010}$ | $.628_{013}$ | $.624_{011}$ |
|  | EOT | $.638_{018}$ | $\mathbf{.608_{010}}$ | $.398_{013}$ | $.416_{011}$ | $.592_{011}$ | $.316_{012}$ | $.453_{010}$ | $.398_{012}$ | $.623_{014}$ | $.623_{010}$ |
| Logistic-IT non-ordered | MT | $.643_{023}$ | $.608_{011}$ | $.393_{012}$ | $.413_{011}$ | $.593_{011}$ | $.318_{011}$ | $.448_{009}$ | $.418_{011}$ | $.636_{014}$ | $.625_{011}$ |
|  | ST | $.643_{023}$ | $.608_{011}$ | $.393_{012}$ | $.413_{011}$ | $.593_{010}$ | $.318_{011}$ | $.448_{009}$ | $.418_{011}$ | $.636_{014}$ | $.625_{011}$ |
|  | EOT | $\mathbf{.634_{016}}$ | $\mathbf{.604_{010}}$ | $.391_{011}$ | $.411_{011}$ | $.591_{010}$ | $.317_{012}$ | $.448_{010}$ | $\mathbf{.397_{010}}$ | $\mathbf{.623_{014}}$ | $.622_{011}$ |
| Logistic-IT ordered | MT,ST | $.640_{020}$ | $.607_{010}$ | $.391_{012}$ | $.413_{011}$ | $.593_{011}$ | $.318_{011}$ | $.449_{010}$ | $.401_{011}$ | $.635_{014}$ | $.626_{011}$ |
|  | EOT | $\mathbf{.633_{017}}$ | $.604_{010}$ | $.391_{010}$ | $.412_{011}$ | $.591_{009}$ | $.316_{012}$ | $.446_{010}$ | $.399_{011}$ | $\mathbf{.623_{013}}$ | $\mathbf{.621_{011}}$ |
| Logistic-AT ordered | MT,ST | $.632_{020}$ | $.604_{011}$ | $.393_{012}$ | $.413_{010}$ | $.588_{010}$ | $.317_{012}$ | $.450_{010}$ | $.397_{011}$ | $.624_{014}$ | $.622_{010}$ |
|  | LB | $.632_{018}$ | $.604_{010}$ | $.392_{012}$ | $.413_{010}$ | $.589_{012}$ | $.317_{012}$ | $.449_{011}$ | $.398_{012}$ | $.627_{015}$ | $.624_{011}$ |
|  | EOT | $.632_{015}$ | $.602_{010}$ | $.392_{013}$ | $.413_{012}$ | $.589_{011}$ | $.316_{012}$ | $.447_{009}$ | $.395_{011}$ | $.623_{014}$ | $.622_{010}$ |
| OLR-NLL ordered | MT,ST | $.632_{019}$ | $.604_{012}$ | $.390_{012}$ | $.413_{011}$ | $.590_{010}$ | $.318_{012}$ | $.448_{010}$ | $.396_{011}$ | $.623_{013}$ | $.621_{011}$ |
|  | LB | $.633_{018}$ | $.605_{010}$ | $.390_{012}$ | $.412_{010}$ | $.590_{011}$ | $.317_{011}$ | $.448_{009}$ | $.397_{010}$ | $.627_{014}$ | $.622_{011}$ |
|  | EOT | $.633_{017}$ | $.603_{009}$ | $.390_{012}$ | $.412_{011}$ | $.591_{011}$ | $.316_{013}$ | $.447_{010}$ | $.395_{012}$ | $.623_{015}$ | $.622_{011}$ |

| Learning | Labeling | POT | AI1 | EL1 | CAL | CE1 | CE2 | 2DP | FRA | MVA |
|---|---|---|---|---|---|---|---|---|---|---|
| AD | NNT | $.564_{007}$ | $.501_{008}$ | $.576_{009}$ | $.537_{007}$ | $.583_{009}$ | $.559_{007}$ | $.366_{005}$ | $.345_{005}$ | $.095_{024}$ |
|  | EOT | $.564_{008}$ | $.500_{009}$ | $.575_{008}$ | $.535_{008}$ | $.580_{008}$ | $.557_{007}$ | $.367_{005}$ | $.344_{005}$ | $.093_{024}$ |
| Hinge-IT non-ordered | MT | $.671_{038}$ | $.497_{009}$ | $.578_{008}$ | $.526_{010}$ | $.630_{009}$ | $.609_{010}$ | $.386_{011}$ | $.348_{006}$ | $.084_{007}$ |
|  | ST | $.671_{038}$ | $.497_{009}$ | $.578_{008}$ | $.526_{010}$ | $.630_{009}$ | $.609_{010}$ | $.386_{011}$ | $.348_{006}$ | $.084_{007}$ |
|  | EOT | $\mathbf{.607_{011}}$ | $.497_{009}$ | $\mathbf{.575_{008}}$ | $.524_{010}$ | $\mathbf{.604_{008}}$ | $\mathbf{.588_{009}}$ | $\mathbf{.370_{004}}$ | $\mathbf{.346_{006}}$ | $\mathbf{.080_{007}}$ |
| Hinge-IT ordered | MT,ST | $.560_{008}$ | $.497_{009}$ | $.578_{008}$ | $.526_{010}$ | $.576_{009}$ | $.557_{008}$ | $.366_{005}$ | $.347_{006}$ | $.086_{008}$ |
|  | EOT | $.559_{007}$ | $.497_{009}$ | $\mathbf{.575_{009}}$ | $.524_{009}$ | $.573_{008}$ | $.555_{008}$ | $.365_{004}$ | $\mathbf{.345_{006}}$ | $\mathbf{.082_{007}}$ |
| Hinge-AT ordered | MT,ST | $.562_{007}$ | $.496_{008}$ | $.576_{010}$ | $.525_{008}$ | $.571_{009}$ | $.555_{008}$ | $.366_{004}$ | $.347_{006}$ | $.090_{025}$ |
|  | EOT | $.561_{007}$ | $.496_{009}$ | $.575_{009}$ | $.523_{008}$ | $.569_{008}$ | $.553_{009}$ | $.365_{005}$ | $.345_{005}$ | $.086_{024}$ |
| Logistic-IT non-ordered | MT | $.561_{008}$ | $.493_{008}$ | $.575_{010}$ | $.507_{008}$ | $.562_{008}$ | $.582_{008}$ | $.365_{005}$ | $.348_{006}$ | $.076_{007}$ |
|  | ST | $.561_{008}$ | $.493_{008}$ | $.575_{010}$ | $.507_{008}$ | $.562_{008}$ | $.582_{008}$ | $.365_{005}$ | $.348_{006}$ | $.076_{007}$ |
|  | EOT | $.560_{008}$ | $.494_{009}$ | $\mathbf{.570_{009}}$ | $.505_{008}$ | $\mathbf{.558_{007}}$ | $\mathbf{.569_{008}}$ | $.365_{004}$ | $\mathbf{.345_{006}}$ | $\mathbf{.073_{007}}$ |
| Logistic-IT ordered | MT,ST | $.560_{007}$ | $.493_{009}$ | $.575_{009}$ | $.510_{009}$ | $.561_{008}$ | $.547_{008}$ | $.365_{005}$ | $.341_{005}$ | $.086_{035}$ |
|  | EOT | $.560_{007}$ | $.493_{009}$ | $\mathbf{.570_{009}}$ | $\mathbf{.506_{008}}$ | $\mathbf{.558_{007}}$ | $.545_{007}$ | $.365_{004}$ | $.341_{005}$ | $\mathbf{.083_{036}}$ |
| Logistic-AT ordered | MT,ST | $.560_{008}$ | $.494_{009}$ | $.570_{009}$ | $.508_{009}$ | $.558_{007}$ | $.579_{007}$ | $.365_{004}$ | $.345_{006}$ | $.125_{075}$ |
|  | LB | $.560_{007}$ | $.493_{008}$ | $.573_{009}$ | $.509_{009}$ | $.558_{008}$ | $.581_{009}$ | $.365_{004}$ | $.346_{006}$ | $.125_{075}$ |
|  | EOT | $.559_{007}$ | $.493_{009}$ | $.570_{008}$ | $.507_{008}$ | $.557_{008}$ | $\mathbf{.576_{008}}$ | $.365_{004}$ | $.344_{006}$ | $.122_{074}$ |
| OLR-NLL ordered | MT,ST | $.560_{007}$ | $.493_{009}$ | $.571_{007}$ | $.508_{010}$ | $.557_{007}$ | $.573_{008}$ | $.366_{004}$ | $.346_{006}$ | $.113_{068}$ |
|  | LB | $.560_{007}$ | $.493_{009}$ | $.574_{010}$ | $.509_{009}$ | $.559_{007}$ | $.575_{008}$ | $.366_{004}$ | $.345_{006}$ | $.113_{068}$ |
|  | EOT | $.559_{007}$ | $.493_{009}$ | $.572_{008}$ | $.507_{009}$ | $.556_{007}$ | $.570_{008}$ | $.365_{004}$ | $.344_{005}$ | $\mathbf{.108_{067}}$ |

Table 19: A counterpart of Table 2 regarding RMSE for Task-S and EF5 datasets.

| Learning | Labeling | DIA | PYR | APR | SER | TRI | WBC | CPU | AMP | BOS | STO |
|---|---|---|---|---|---|---|---|---|---|---|---|
| AD | NNT | $1.435_{284}$ | $1.051_{275}$ | $.664_{098}$ | $.763_{133}$ | $1.407_{196}$ | $1.454_{181}$ | $.769_{098}$ | $.594_{055}$ | $.714_{085}$ | $.425_{028}$ |
| | EOT | $1.399_{257}$ | $1.021_{268}$ | $.657_{095}$ | $.776_{131}$ | $1.385_{215}$ | $1.445_{147}$ | $\mathbf{.728_{088}}$ | $.604_{049}$ | $.712_{073}$ | $.419_{031}$ |
| Hinge-IT non-ordered | MT | $1.444_{291}$ | $1.073_{272}$ | $.702_{103}$ | $.825_{150}$ | $1.457_{245}$ | $1.539_{243}$ | $.802_{104}$ | $.583_{054}$ | $.702_{068}$ | $.454_{043}$ |
| | ST | $1.431_{291}$ | $1.073_{272}$ | $.702_{103}$ | $.825_{150}$ | $1.457_{245}$ | $1.539_{243}$ | $.802_{104}$ | $.583_{054}$ | $.702_{068}$ | $.454_{043}$ |
| | EOT | $1.339_{250}$ | $1.068_{306}$ | $\mathbf{.662_{090}}$ | $.793_{128}$ | $1.362_{194}$ | $\mathbf{1.444_{186}}$ | $.716_{089}$ | $.592_{055}$ | $.695_{065}$ | $\mathbf{.397_{033}}$ |
| Hinge-IT ordered | MT,ST | $1.426_{289}$ | $1.059_{298}$ | $.685_{087}$ | $.826_{179}$ | $1.467_{221}$ | $1.532_{220}$ | $.803_{118}$ | $.580_{057}$ | $.687_{071}$ | $.393_{035}$ |
| | EOT | $1.377_{269}$ | $1.069_{304}$ | $.651_{089}$ | $\mathbf{.771_{127}}$ | $1.393_{203}$ | $1.464_{181}$ | $\mathbf{.736_{089}}$ | $.579_{059}$ | $.693_{064}$ | $.392_{035}$ |
| Hinge-AT ordered | MT,ST | $1.430_{298}$ | $1.027_{262}$ | $.691_{100}$ | $.758_{147}$ | $1.425_{192}$ | $1.442_{160}$ | $.750_{093}$ | $.580_{057}$ | $.680_{068}$ | $.395_{032}$ |
| | EOT | $1.415_{271}$ | $1.060_{326}$ | $.666_{084}$ | $.743_{133}$ | $1.373_{175}$ | $1.419_{185}$ | $.724_{094}$ | $.590_{060}$ | $.692_{062}$ | $.390_{037}$ |
| Logistic-IT non-ordered | MT | $1.420_{281}$ | $1.089_{294}$ | $.722_{102}$ | $.829_{190}$ | $1.465_{205}$ | $1.480_{251}$ | $.779_{115}$ | $.602_{061}$ | $.715_{083}$ | $.406_{039}$ |
| | ST | $1.413_{277}$ | $1.089_{294}$ | $.722_{102}$ | $.829_{190}$ | $1.465_{205}$ | $1.480_{251}$ | $.779_{115}$ | $.602_{061}$ | $.715_{083}$ | $.406_{039}$ |
| | EOT | $1.378_{256}$ | $1.025_{252}$ | $\mathbf{.659_{087}}$ | $.774_{132}$ | $\mathbf{1.391_{214}}$ | $1.458_{172}$ | $\mathbf{.733_{083}}$ | $.580_{059}$ | $.697_{064}$ | $\mathbf{.390_{033}}$ |
| Logistic-IT ordered | MT,ST | $1.439_{283}$ | $1.100_{294}$ | $.692_{110}$ | $.730_{161}$ | $1.471_{198}$ | $1.518_{217}$ | $.774_{110}$ | $.573_{062}$ | $.675_{074}$ | $.369_{035}$ |
| | EOT | $1.371_{249}$ | $1.031_{269}$ | $.663_{103}$ | $.737_{137}$ | $1.377_{190}$ | $\mathbf{1.441_{173}}$ | $.742_{090}$ | $.585_{056}$ | $.677_{061}$ | $.369_{035}$ |
| Logistic-AT ordered | MT,ST | $1.339_{267}$ | $1.008_{310}$ | $.703_{087}$ | $.706_{104}$ | $1.386_{183}$ | $1.415_{145}$ | $.730_{088}$ | $.577_{056}$ | $.670_{065}$ | $.369_{032}$ |
| | LB | $1.374_{273}$ | $1.089_{299}$ | $.715_{102}$ | $.697_{139}$ | $1.426_{193}$ | $1.447_{204}$ | $.746_{098}$ | $.567_{054}$ | $.665_{067}$ | $.369_{033}$ |
| | EOT | $1.421_{253}$ | $1.054_{292}$ | $.680_{100}$ | $.715_{152}$ | $1.370_{183}$ | $1.450_{168}$ | $.716_{088}$ | $.578_{058}$ | $.681_{067}$ | $.373_{036}$ |
| OLR-NLL ordered | MT,ST | $1.312_{247}$ | $1.041_{315}$ | $.700_{104}$ | $.714_{129}$ | $\mathbf{1.387_{193}}$ | $1.440_{141}$ | $.738_{070}$ | $.575_{054}$ | $.676_{065}$ | $.369_{033}$ |
| | LB | $1.385_{263}$ | $1.028_{247}$ | $.703_{112}$ | $.722_{143}$ | $1.452_{195}$ | $1.452_{192}$ | $.755_{094}$ | $.574_{056}$ | $.666_{060}$ | $.368_{031}$ |
| | EOT | $1.400_{281}$ | $1.012_{283}$ | $\mathbf{.665_{100}}$ | $.737_{135}$ | $\mathbf{1.378_{209}}$ | $1.436_{163}$ | $.729_{079}$ | $.587_{054}$ | $.675_{064}$ | $.368_{034}$ |

| Learning | Labeling | ABA | AI2 | KRA | CO1 | PU1 | BA1 | CO2 | PU2 | BA2 | EL2 |
|---|---|---|---|---|---|---|---|---|---|---|---|
| AD | NNT | $.982_{026}$ | $.843_{018}$ | $.570_{013}$ | $.579_{011}$ | $.906_{017}$ | $.455_{011}$ | $.647_{014}$ | $.532_{014}$ | $.975_{021}$ | $.945_{017}$ |
| | EOT | $\mathbf{.969_{024}}$ | $\mathbf{.835_{016}}$ | $.566_{015}$ | $\mathbf{.573_{011}}$ | $\mathbf{.894_{017}}$ | $.453_{013}$ | $.648_{013}$ | $.533_{014}$ | $\mathbf{.967_{019}}$ | $\mathbf{.923_{014}}$ |
| Hinge-IT non-ordered | MT | $1.040_{036}$ | $.849_{019}$ | $.549_{013}$ | $.576_{012}$ | $.952_{023}$ | $.453_{010}$ | $.651_{014}$ | $.538_{011}$ | $1.011_{019}$ | $1.023_{023}$ |
| | ST | $1.040_{036}$ | $.849_{019}$ | $.549_{013}$ | $.576_{012}$ | $.952_{023}$ | $.453_{010}$ | $.651_{014}$ | $.538_{011}$ | $1.011_{019}$ | $1.023_{023}$ |
| | EOT | $\mathbf{.966_{024}}$ | $\mathbf{.832_{016}}$ | $.545_{012}$ | $\mathbf{.572_{013}}$ | $\mathbf{.889_{017}}$ | $.452_{010}$ | $\mathbf{.640_{012}}$ | $\mathbf{.522_{012}}$ | $\mathbf{.965_{017}}$ | $\mathbf{.921_{012}}$ |
| Hinge-IT ordered | MT,ST | $1.043_{039}$ | $.847_{018}$ | $.549_{013}$ | $.574_{011}$ | $.946_{021}$ | $.451_{011}$ | $.646_{014}$ | $.530_{012}$ | $1.013_{019}$ | $1.044_{022}$ |
| | EOT | $\mathbf{.964_{025}}$ | $\mathbf{.831_{016}}$ | $.548_{013}$ | $.570_{010}$ | $\mathbf{.891_{015}}$ | $.453_{009}$ | $\mathbf{.639_{013}}$ | $.528_{013}$ | $\mathbf{.964_{017}}$ | $\mathbf{.922_{013}}$ |
| Hinge-AT ordered | MT,ST | $.968_{024}$ | $.837_{017}$ | $.548_{012}$ | $.570_{012}$ | $.901_{017}$ | $.453_{010}$ | $.639_{014}$ | $.524_{012}$ | $.968_{022}$ | $.935_{017}$ |
| | EOT | $.961_{021}$ | $.833_{016}$ | $.546_{013}$ | $.569_{010}$ | $\mathbf{.889_{016}}$ | $.452_{010}$ | $.637_{013}$ | $.524_{011}$ | $.963_{019}$ | $\mathbf{.920_{012}}$ |
| Logistic-IT non-ordered | MT | $1.026_{031}$ | $.883_{022}$ | $.553_{014}$ | $.570_{011}$ | $.913_{019}$ | $.452_{011}$ | $.641_{015}$ | $.570_{014}$ | $.993_{022}$ | $.988_{020}$ |
| | ST | $1.026_{031}$ | $.883_{022}$ | $.553_{014}$ | $.570_{012}$ | $.913_{019}$ | $.452_{011}$ | $.641_{015}$ | $.570_{014}$ | $.993_{022}$ | $.988_{020}$ |
| | EOT | $\mathbf{.962_{025}}$ | $\mathbf{.828_{016}}$ | $\mathbf{.540_{014}}$ | $.568_{012}$ | $\mathbf{.889_{017}}$ | $.451_{011}$ | $\mathbf{.636_{012}}$ | $\mathbf{.529_{012}}$ | $\mathbf{.963_{019}}$ | $\mathbf{.920_{013}}$ |
| Logistic-IT ordered | MT,ST | $1.030_{032}$ | $.875_{018}$ | $.541_{012}$ | $.567_{013}$ | $.910_{018}$ | $.451_{011}$ | $.638_{015}$ | $.533_{016}$ | $.995_{023}$ | $.986_{016}$ |
| | EOT | $\mathbf{.965_{021}}$ | $\mathbf{.829_{014}}$ | $.538_{012}$ | $.565_{012}$ | $\mathbf{.889_{016}}$ | $.451_{011}$ | $\mathbf{.633_{012}}$ | $.532_{014}$ | $\mathbf{.963_{018}}$ | $.919_{012}$ |
| Logistic-AT ordered | MT,ST | $\mathbf{.959_{021}}$ | $.832_{016}$ | $.542_{012}$ | $.568_{013}$ | $.890_{016}$ | $.452_{010}$ | $.633_{013}$ | $.515_{011}$ | $\mathbf{.959_{017}}$ | $\mathbf{.920_{013}}$ |
| | LB | $.970_{024}$ | $.840_{018}$ | $.541_{014}$ | $.566_{012}$ | $.894_{017}$ | $.451_{010}$ | $.635_{014}$ | $.518_{011}$ | $.968_{019}$ | $.936_{016}$ |
| | EOT | $\mathbf{.959_{024}}$ | $\mathbf{.827_{016}}$ | $.540_{012}$ | $.566_{011}$ | $.889_{015}$ | $.451_{009}$ | $\mathbf{.633_{014}}$ | $.516_{012}$ | $\mathbf{.960_{018}}$ | $\mathbf{.918_{013}}$ |
| OLR-NLL ordered | MT,ST | $\mathbf{.960_{020}}$ | $\mathbf{.832_{015}}$ | $.537_{013}$ | $.565_{011}$ | $.891_{016}$ | $.451_{011}$ | $.634_{015}$ | $.519_{012}$ | $\mathbf{.962_{017}}$ | $\mathbf{.922_{012}}$ |
| | LB | $.971_{027}$ | $.840_{017}$ | $.538_{015}$ | $.566_{011}$ | $.896_{019}$ | $.450_{010}$ | $.636_{014}$ | $.520_{013}$ | $.970_{020}$ | $.938_{016}$ |
| | EOT | $\mathbf{.961_{023}}$ | $\mathbf{.827_{015}}$ | $.536_{013}$ | $.566_{010}$ | $.888_{016}$ | $.451_{010}$ | $.634_{013}$ | $.520_{014}$ | $\mathbf{.963_{018}}$ | $\mathbf{.920_{014}}$ |

| Learning | Labeling | POT | AI1 | EL1 | CAL | CE1 | CE2 | 2DP | FRA | MVA |
|---|---|---|---|---|---|---|---|---|---|---|
| AD | NNT | $.840_{015}$ | $.728_{013}$ | $.883_{012}$ | $.768_{013}$ | $.937_{014}$ | $.921_{014}$ | $.493_{005}$ | $.471_{006}$ | $.158_{017}$ |
| | EOT | $\mathbf{.823_{015}}$ | $\mathbf{.721_{011}}$ | $\mathbf{.876_{012}}$ | $.762_{012}$ | $\mathbf{.929_{012}}$ | $\mathbf{.907_{011}}$ | $.492_{005}$ | $\mathbf{.469_{006}}$ | $\mathbf{.148_{019}}$ |
| Hinge-IT non-ordered | MT | $.910_{028}$ | $.759_{014}$ | $.946_{017}$ | $.868_{014}$ | $.983_{019}$ | $.942_{016}$ | $.498_{006}$ | $.467_{005}$ | $.334_{028}$ |
| | ST | $.910_{028}$ | $.759_{014}$ | $.946_{017}$ | $.868_{014}$ | $.983_{019}$ | $.942_{016}$ | $.498_{006}$ | $.467_{005}$ | $.334_{028}$ |
| | EOT | $\mathbf{.800_{015}}$ | $\mathbf{.732_{012}}$ | $\mathbf{.868_{012}}$ | $\mathbf{.841_{010}}$ | $\mathbf{.927_{012}}$ | $\mathbf{.891_{011}}$ | $\mathbf{.492_{005}}$ | $\mathbf{.464_{005}}$ | $\mathbf{.316_{023}}$ |
| Hinge-IT ordered | MT,ST | $.956_{020}$ | $.730_{012}$ | $.959_{017}$ | $.861_{013}$ | $.981_{017}$ | $.934_{015}$ | $.499_{005}$ | $.463_{005}$ | $.131_{016}$ |
| | EOT | $\mathbf{.788_{013}}$ | $\mathbf{.719_{012}}$ | $\mathbf{.867_{013}}$ | $\mathbf{.840_{010}}$ | $\mathbf{.926_{012}}$ | $\mathbf{.890_{012}}$ | $.492_{004}$ | $.462_{004}$ | $\mathbf{.126_{014}}$ |
| Hinge-AT ordered | MT,ST | $.837_{023}$ | $.736_{013}$ | $.883_{012}$ | $.844_{012}$ | $.929_{013}$ | $.896_{015}$ | $.498_{005}$ | $.465_{005}$ | $.273_{035}$ |
| | EOT | $\mathbf{.821_{019}}$ | $\mathbf{.732_{011}}$ | $\mathbf{.869_{013}}$ | $\mathbf{.840_{010}}$ | $\mathbf{.920_{011}}$ | $\mathbf{.885_{010}}$ | $.492_{005}$ | $\mathbf{.464_{004}}$ | $\mathbf{.262_{034}}$ |
| Logistic-IT non-ordered | MT | $.955_{020}$ | $.755_{014}$ | $.936_{016}$ | $.859_{014}$ | $.973_{016}$ | $.922_{017}$ | $.497_{006}$ | $.468_{007}$ | $.285_{015}$ |
| | ST | $.955_{020}$ | $.755_{014}$ | $.936_{016}$ | $.859_{014}$ | $.973_{016}$ | $.922_{017}$ | $.497_{006}$ | $.468_{007}$ | $.285_{015}$ |
| | EOT | $\mathbf{.788_{012}}$ | $\mathbf{.734_{011}}$ | $\mathbf{.871_{012}}$ | $\mathbf{.838_{010}}$ | $\mathbf{.923_{011}}$ | $\mathbf{.881_{010}}$ | $\mathbf{.493_{004}}$ | $\mathbf{.460_{005}}$ | $\mathbf{.267_{011}}$ |
| Logistic-IT ordered | MT,ST | $.967_{021}$ | $.723_{012}$ | $.957_{017}$ | $.854_{013}$ | $.974_{016}$ | $.913_{015}$ | $.496_{004}$ | $.461_{005}$ | $.129_{013}$ |
| | EOT | $\mathbf{.784_{012}}$ | $\mathbf{.715_{011}}$ | $\mathbf{.870_{012}}$ | $\mathbf{.837_{011}}$ | $\mathbf{.922_{011}}$ | $\mathbf{.877_{010}}$ | $\mathbf{.493_{005}}$ | $.460_{004}$ | $\mathbf{.123_{012}}$ |
| Logistic-AT ordered | MT,ST | $.791_{012}$ | $\mathbf{.731_{012}}$ | $\mathbf{.872_{012}}$ | $.841_{011}$ | $\mathbf{.917_{011}}$ | $\mathbf{.886_{013}}$ | $.496_{005}$ | $.461_{006}$ | $.295_{069}$ |
| | LB | $.790_{013}$ | $.737_{013}$ | $.888_{013}$ | $.843_{012}$ | $.925_{014}$ | $.896_{014}$ | $.496_{005}$ | $.461_{006}$ | $.295_{069}$ |
| | EOT | $\mathbf{.776_{011}}$ | $\mathbf{.730_{011}}$ | $\mathbf{.870_{012}}$ | $\mathbf{.840_{010}}$ | $.915_{011}$ | $\mathbf{.883_{010}}$ | $\mathbf{.493_{005}}$ | $.460_{006}$ | $\mathbf{.285_{071}}$ |
| OLR-NLL ordered | MT,ST | $.789_{012}$ | $\mathbf{.730_{011}}$ | $\mathbf{.868_{012}}$ | $.838_{011}$ | $\mathbf{.922_{012}}$ | $\mathbf{.879_{011}}$ | $.496_{005}$ | $.461_{006}$ | $.269_{060}$ |
| | LB | $.792_{015}$ | $.737_{013}$ | $.884_{014}$ | $.841_{012}$ | $.929_{014}$ | $.889_{013}$ | $.496_{005}$ | $.461_{005}$ | $.268_{059}$ |
| | EOT | $.774_{011}$ | $\mathbf{.732_{012}}$ | $\mathbf{.867_{012}}$ | $.839_{011}$ | $\mathbf{.920_{013}}$ | $.876_{011}$ | $\mathbf{.492_{005}}$ | $.460_{006}$ | $\mathbf{.261_{058}}$ |

Table 20: A counterpart of Table 2 regarding RMSE for Task-S and EF10 datasets.

| Learning | Labeling | DIA | PYR | APR | SER | TRI | WBC | CPU | AMP | BOS | STO |
|---|---|---|---|---|---|---|---|---|---|---|---|
| AD | NNT | $2.899_{576}$ | $1.882_{521}$ | $1.152_{147}$ | $1.339_{289}$ | $2.900_{403}$ | $2.983_{356}$ | $1.379_{179}$ | $1.030_{107}$ | $1.280_{163}$ | $.640_{037}$ |
|  | EOT | $2.922_{529}$ | $1.872_{499}$ | $1.102_{148}$ | $1.327_{247}$ | $2.816_{352}$ | $2.945_{320}$ | $1.378_{189}$ | $1.055_{102}$ | $1.287_{150}$ | $.649_{050}$ |
| Hinge-IT non-ordered | MT | $2.861_{563}$ | $1.919_{600}$ | $1.182_{134}$ | $1.433_{309}$ | $2.956_{434}$ | $2.961_{373}$ | $1.480_{173}$ | $1.044_{107}$ | $1.306_{161}$ | $.707_{073}$ |
|  | ST | $2.871_{569}$ | $1.878_{548}$ | $1.188_{135}$ | $1.413_{328}$ | $2.956_{433}$ | $2.945_{381}$ | $1.468_{162}$ | $1.044_{107}$ | $1.306_{161}$ | $.695_{060}$ |
|  | EOT | $2.811_{581}$ | $\color{red}\mathbf{1.860_{455}}$ | $\color{red}\mathbf{1.073_{145}}$ | $1.386_{349}$ | $2.853_{426}$ | $2.936_{362}$ | $\mathbf{1.396_{188}}$ | $1.067_{113}$ | $1.263_{156}$ | $\mathbf{.657_{049}}$ |
| Hinge-IT ordered | MT,ST | $2.977_{551}$ | $1.961_{543}$ | $1.170_{136}$ | $1.354_{304}$ | $2.941_{417}$ | $2.934_{356}$ | $1.531_{186}$ | $1.045_{107}$ | $1.285_{137}$ | $.629_{050}$ |
|  | EOT | $2.904_{544}$ | $1.920_{500}$ | $\mathbf{1.099_{149}}$ | $1.345_{261}$ | $2.876_{374}$ | $2.923_{366}$ | $\mathbf{1.398_{165}}$ | $1.054_{108}$ | $1.277_{140}$ | $.624_{044}$ |
| Hinge-AT ordered | MT,ST | $2.845_{521}$ | $1.923_{563}$ | $1.141_{171}$ | $1.260_{243}$ | $2.922_{352}$ | $2.887_{318}$ | $1.390_{171}$ | $\mathbf{1.014_{109}}$ | $1.224_{123}$ | $.598_{042}$ |
|  | EOT | $2.782_{535}$ | $1.879_{522}$ | $1.112_{159}$ | $1.258_{252}$ | $\mathbf{2.791_{333}}$ | $2.910_{357}$ | $\color{red}1.358_{182}$ | $1.052_{107}$ | $1.220_{130}$ | $.598_{044}$ |
| Logistic-IT non-ordered | MT | $2.767_{495}$ | $2.014_{618}$ | $1.163_{155}$ | $1.366_{295}$ | $2.944_{376}$ | $2.925_{355}$ | $1.458_{203}$ | $1.035_{102}$ | $1.259_{138}$ | $.678_{084}$ |
|  | ST | $2.770_{499}$ | $2.028_{623}$ | $1.151_{161}$ | $1.348_{300}$ | $2.940_{372}$ | $2.934_{403}$ | $1.447_{199}$ | $1.035_{102}$ | $1.259_{138}$ | $.676_{082}$ |
|  | EOT | $2.843_{582}$ | $1.886_{530}$ | $1.104_{124}$ | $1.319_{320}$ | $2.874_{404}$ | $2.925_{404}$ | $\mathbf{1.381_{157}}$ | $1.048_{102}$ | $1.242_{120}$ | $.648_{052}$ |
| Logistic-IT ordered | MT,ST | $2.853_{513}$ | $1.961_{503}$ | $1.158_{146}$ | $1.390_{244}$ | $2.972_{444}$ | $2.947_{365}$ | $1.503_{191}$ | $1.031_{094}$ | $1.262_{142}$ | $.604_{038}$ |
|  | EOT | $2.899_{568}$ | $1.911_{515}$ | $\mathbf{1.107_{148}}$ | $1.317_{247}$ | $2.849_{378}$ | $2.937_{403}$ | $\mathbf{1.412_{174}}$ | $1.034_{106}$ | $1.238_{118}$ | $.597_{035}$ |
| Logistic-AT ordered | MT,ST | $\color{red}2.740_{477}$ | $1.918_{562}$ | $1.143_{133}$ | $1.267_{259}$ | $2.848_{331}$ | $2.894_{342}$ | $1.373_{175}$ | $1.009_{096}$ | $\color{red}1.212_{119}$ | $\color{red}.577_{037}$ |
|  | LB | $2.802_{495}$ | $1.981_{587}$ | $1.147_{154}$ | $1.286_{251}$ | $2.906_{341}$ | $2.890_{359}$ | $1.400_{174}$ | $1.017_{096}$ | $1.216_{123}$ | $.584_{043}$ |
|  | EOT | $2.889_{567}$ | $1.931_{517}$ | $1.108_{145}$ | $1.266_{246}$ | $\color{red}\mathbf{2.789_{360}}$ | $2.911_{340}$ | $1.369_{167}$ | $1.033_{093}$ | $1.218_{128}$ | $.580_{038}$ |
| OLR-NLL ordered | MT,ST | $2.792_{472}$ | $1.888_{561}$ | $1.159_{112}$ | $1.241_{269}$ | $2.876_{353}$ | $\color{red}\mathbf{2.881_{294}}$ | $1.400_{201}$ | $1.006_{097}$ | $1.227_{126}$ | $.587_{033}$ |
|  | LB | $2.812_{494}$ | $1.963_{627}$ | $1.150_{146}$ | $\color{red}1.231_{249}$ | $2.896_{365}$ | $2.888_{332}$ | $1.413_{210}$ | $\color{red}1.002_{103}$ | $1.233_{137}$ | $.584_{032}$ |
|  | EOT | $2.848_{567}$ | $1.959_{546}$ | $1.129_{157}$ | $1.277_{271}$ | $2.864_{404}$ | $2.913_{352}$ | $1.368_{168}$ | $1.027_{103}$ | $1.224_{129}$ | $.582_{036}$ |

| Learning | Labeling | ABA | AI2 | KRA | CO1 | PU1 | BA1 | CO2 | PU2 | BA2 | EL2 |
|---|---|---|---|---|---|---|---|---|---|---|---|
| AD | NNT | $1.857_{056}$ | $1.558_{032}$ | $.926_{031}$ | $.979_{019}$ | $1.698_{034}$ | $.688_{015}$ | $1.124_{026}$ | $.842_{026}$ | $1.865_{038}$ | $1.738_{031}$ |
|  | EOT | $\mathbf{1.838_{045}}$ | $\mathbf{1.535_{028}}$ | $\mathbf{.917_{026}}$ | $\mathbf{.970_{019}}$ | $1.690_{031}$ | $.688_{013}$ | $1.119_{027}$ | $.838_{028}$ | $1.857_{037}$ | $\mathbf{1.723_{027}}$ |
| Hinge-IT non-ordered | MT | $1.915_{057}$ | $1.626_{028}$ | $.917_{026}$ | $.993_{021}$ | $1.739_{033}$ | $.687_{013}$ | $1.148_{026}$ | $.854_{020}$ | $1.877_{035}$ | $1.766_{031}$ |
|  | ST | $1.915_{057}$ | $1.626_{028}$ | $.917_{026}$ | $.993_{021}$ | $1.739_{033}$ | $.687_{013}$ | $1.148_{026}$ | $.854_{020}$ | $1.877_{035}$ | $1.766_{031}$ |
|  | EOT | $\mathbf{1.849_{048}}$ | $\mathbf{1.533_{029}}$ | $\mathbf{.898_{023}}$ | $\mathbf{.975_{017}}$ | $1.685_{033}$ | $\mathbf{.682_{012}}$ | $1.121_{025}$ | $.837_{020}$ | $\mathbf{1.854_{038}}$ | $1.721_{029}$ |
| Hinge-IT ordered | MT,ST | $1.973_{063}$ | $1.641_{030}$ | $.920_{030}$ | $.987_{020}$ | $1.912_{053}$ | $.683_{012}$ | $1.145_{030}$ | $.859_{023}$ | $1.929_{040}$ | $1.784_{031}$ |
|  | EOT | $\mathbf{1.861_{045}}$ | $\mathbf{1.547_{026}}$ | $\mathbf{.906_{026}}$ | $\mathbf{.971_{020}}$ | $1.695_{034}$ | $.681_{013}$ | $\mathbf{1.117_{026}}$ | $\mathbf{.850_{022}}$ | $1.856_{036}$ | $\mathbf{1.725_{026}}$ |
| Hinge-AT ordered | MT,ST | $1.844_{057}$ | $1.570_{028}$ | $.905_{023}$ | $.951_{018}$ | $1.709_{035}$ | $.683_{013}$ | $1.102_{027}$ | $.821_{020}$ | $1.860_{038}$ | $1.747_{028}$ |
|  | EOT | $1.826_{046}$ | $\mathbf{1.532_{028}}$ | $.904_{022}$ | $.951_{016}$ | $\mathbf{1.685_{035}}$ | $.680_{011}$ | $1.099_{024}$ | $.819_{018}$ | $1.851_{036}$ | $\mathbf{1.720_{029}}$ |
| Logistic-IT non-ordered | MT | $1.893_{063}$ | $1.619_{029}$ | $.905_{026}$ | $.981_{018}$ | $1.726_{035}$ | $.685_{014}$ | $1.137_{030}$ | $.856_{026}$ | $1.869_{038}$ | $1.761_{034}$ |
|  | ST | $1.893_{063}$ | $1.619_{029}$ | $.905_{026}$ | $.981_{018}$ | $1.726_{035}$ | $.685_{014}$ | $1.137_{030}$ | $.856_{026}$ | $1.869_{038}$ | $1.761_{034}$ |
|  | EOT | $\mathbf{1.847_{047}}$ | $\mathbf{1.535_{026}}$ | $\color{red}.890_{029}$ | $\mathbf{.967_{017}}$ | $1.686_{036}$ | $.681_{013}$ | $\mathbf{1.118_{025}}$ | $\mathbf{.833_{018}}$ | $1.853_{035}$ | $\mathbf{1.720_{028}}$ |
| Logistic-IT ordered | MT,ST | $1.934_{063}$ | $1.629_{029}$ | $.920_{028}$ | $.975_{018}$ | $1.787_{037}$ | $.680_{014}$ | $1.129_{031}$ | $.876_{021}$ | $1.892_{040}$ | $1.769_{033}$ |
|  | EOT | $\mathbf{1.863_{044}}$ | $\mathbf{1.552_{030}}$ | $\mathbf{.909_{028}}$ | $\mathbf{.960_{018}}$ | $1.688_{032}$ | $\color{red}.679_{013}$ | $1.116_{026}$ | $\mathbf{.860_{017}}$ | $1.856_{038}$ | $\mathbf{1.723_{027}}$ |
| Logistic-AT ordered | MT,ST | $\color{red}\mathbf{1.824_{048}}$ | $\mathbf{1.533_{026}}$ | $.903_{021}$ | $.951_{016}$ | $1.683_{032}$ | $.680_{013}$ | $1.095_{023}$ | $.820_{020}$ | $1.850_{034}$ | $\mathbf{1.721_{027}}$ |
|  | LB | $1.844_{051}$ | $1.545_{028}$ | $.905_{022}$ | $.949_{017}$ | $1.693_{033}$ | $.680_{012}$ | $1.097_{025}$ | $.823_{022}$ | $1.862_{038}$ | $1.732_{030}$ |
|  | EOT | $\mathbf{1.826_{044}}$ | $\mathbf{1.531_{028}}$ | $.901_{024}$ | $\color{red}.947_{017}$ | $1.683_{033}$ | $.680_{012}$ | $\color{red}1.094_{022}$ | $\color{red}.817_{019}$ | $\mathbf{1.849_{036}}$ | $\mathbf{1.720_{029}}$ |
| OLR-NLL ordered | MT,ST | $1.831_{050}$ | $\mathbf{1.532_{029}}$ | $.893_{030}$ | $.953_{018}$ | $1.684_{033}$ | $.681_{014}$ | $1.097_{026}$ | $.826_{019}$ | $\mathbf{1.851_{034}}$ | $\mathbf{1.720_{027}}$ |
|  | LB | $1.848_{054}$ | $1.549_{031}$ | $.893_{030}$ | $.957_{017}$ | $1.693_{034}$ | $.681_{013}$ | $1.102_{024}$ | $.827_{020}$ | $1.864_{039}$ | $1.734_{031}$ |
|  | EOT | $1.833_{047}$ | $\color{red}\mathbf{1.529_{027}}$ | $.891_{030}$ | $.949_{014}$ | $\color{red}1.682_{032}$ | $.680_{013}$ | $1.098_{025}$ | $.827_{020}$ | $\mathbf{1.851_{037}}$ | $\mathbf{1.718_{029}}$ |

| Learning | Labeling | POT | AI1 | EL1 | CAL | CE1 | CE2 | 2DP | FRA | MVA |
|---|---|---|---|---|---|---|---|---|---|---|
| AD | NNT | $1.569_{024}$ | $1.291_{020}$ | $1.669_{024}$ | $1.386_{025}$ | $1.620_{025}$ | $1.574_{017}$ | $.770_{007}$ | $.727_{008}$ | $.632_{104}$ |
|  | EOT | $\mathbf{1.557_{021}}$ | $1.287_{021}$ | $\mathbf{1.649_{019}}$ | $\color{red}1.378_{024}$ | $\color{red}\mathbf{1.608_{020}}$ | $\color{red}\mathbf{1.561_{020}}$ | $\mathbf{.766_{006}}$ | $\mathbf{.724_{007}}$ | $\mathbf{.601_{105}}$ |
| Hinge-IT non-ordered | MT | $1.813_{044}$ | $1.348_{024}$ | $1.723_{029}$ | $1.611_{024}$ | $1.865_{034}$ | $1.758_{028}$ | $.779_{007}$ | $.727_{006}$ | $.739_{110}$ |
|  | ST | $1.813_{044}$ | $1.348_{024}$ | $1.723_{029}$ | $1.611_{024}$ | $1.865_{034}$ | $1.758_{028}$ | $.779_{007}$ | $.727_{006}$ | $.739_{110}$ |
|  | EOT | $\mathbf{1.605_{026}}$ | $\mathbf{1.322_{022}}$ | $\mathbf{1.641_{022}}$ | $\mathbf{1.564_{019}}$ | $\mathbf{1.785_{024}}$ | $\mathbf{1.692_{022}}$ | $\mathbf{.767_{006}}$ | $\mathbf{.720_{005}}$ | $\mathbf{.665_{099}}$ |
| Hinge-IT ordered | MT,ST | $1.868_{045}$ | $1.314_{017}$ | $1.740_{031}$ | $1.657_{029}$ | $2.000_{040}$ | $1.904_{040}$ | $.780_{008}$ | $.723_{007}$ | $.487_{106}$ |
|  | EOT | $\mathbf{1.533_{020}}$ | $\color{red}\mathbf{1.286_{020}}$ | $\mathbf{1.653_{022}}$ | $\mathbf{1.566_{018}}$ | $\mathbf{1.783_{021}}$ | $\mathbf{1.709_{024}}$ | $\mathbf{.767_{007}}$ | $\mathbf{.718_{007}}$ | $.464_{096}$ |
| Hinge-AT ordered | MT,ST | $1.540_{020}$ | $1.329_{022}$ | $1.655_{024}$ | $1.576_{021}$ | $1.763_{024}$ | $1.688_{024}$ | $.776_{007}$ | $.719_{007}$ | $.522_{094}$ |
|  | EOT | $1.538_{020}$ | $\mathbf{1.316_{020}}$ | $\mathbf{1.639_{025}}$ | $\mathbf{1.565_{020}}$ | $\mathbf{1.746_{021}}$ | $\mathbf{1.668_{023}}$ | $\mathbf{.767_{006}}$ | $\color{red}.715_{007}$ | $\mathbf{.480_{065}}$ |
| Logistic-IT non-ordered | MT | $1.893_{031}$ | $1.341_{024}$ | $1.711_{026}$ | $1.600_{024}$ | $1.847_{030}$ | $1.748_{029}$ | $.775_{007}$ | $.732_{010}$ | $.635_{139}$ |
|  | ST | $1.892_{031}$ | $1.341_{024}$ | $1.711_{026}$ | $1.600_{024}$ | $1.847_{030}$ | $1.748_{029}$ | $.775_{007}$ | $.732_{010}$ | $.635_{139}$ |
|  | EOT | $\mathbf{1.597_{030}}$ | $\mathbf{1.319_{019}}$ | $\mathbf{1.641_{021}}$ | $\mathbf{1.564_{019}}$ | $\mathbf{1.778_{023}}$ | $\mathbf{1.686_{025}}$ | $\mathbf{.767_{006}}$ | $\mathbf{.719_{006}}$ | $\mathbf{.576_{102}}$ |
| Logistic-IT ordered | MT,ST | $1.824_{044}$ | $1.308_{021}$ | $1.718_{026}$ | $1.416_{031}$ | $1.743_{031}$ | $1.653_{026}$ | $.771_{007}$ | $.722_{009}$ | $.456_{074}$ |
|  | EOT | $\color{red}\mathbf{1.529_{019}}$ | $\mathbf{1.288_{019}}$ | $\mathbf{1.670_{022}}$ | $1.391_{024}$ | $\mathbf{1.635_{022}}$ | $\mathbf{1.572_{020}}$ | $\mathbf{.767_{007}}$ | $\mathbf{.716_{007}}$ | $\color{red}\mathbf{.437_{069}}$ |
| Logistic-AT ordered | MT,ST | $1.541_{020}$ | $\mathbf{1.318_{019}}$ | $\mathbf{1.643_{023}}$ | $\mathbf{1.568_{021}}$ | $\mathbf{1.751_{021}}$ | $\mathbf{1.677_{023}}$ | $.773_{008}$ | $.719_{007}$ | $.541_{101}$ |
|  | LB | $1.545_{021}$ | $1.327_{021}$ | $1.664_{024}$ | $1.577_{023}$ | $1.767_{025}$ | $1.698_{026}$ | $.772_{008}$ | $.720_{008}$ | $.538_{101}$ |
|  | EOT | $1.541_{022}$ | $\mathbf{1.316_{021}}$ | $\mathbf{1.638_{021}}$ | $\mathbf{1.563_{020}}$ | $\mathbf{1.747_{020}}$ | $\mathbf{1.669_{022}}$ | $\mathbf{.767_{006}}$ | $\mathbf{.716_{006}}$ | $\mathbf{.514_{100}}$ |
| OLR-NLL ordered | MT,ST | $\mathbf{1.596_{023}}$ | $\mathbf{1.319_{021}}$ | $\mathbf{1.639_{023}}$ | $\mathbf{1.562_{021}}$ | $\mathbf{1.767_{022}}$ | $\mathbf{1.668_{023}}$ | $.772_{007}$ | $.717_{006}$ | $.515_{114}$ |
|  | LB | $1.620_{026}$ | $1.328_{023}$ | $1.667_{028}$ | $1.572_{024}$ | $1.782_{027}$ | $1.687_{024}$ | $.772_{007}$ | $.718_{007}$ | $.511_{114}$ |
|  | EOT | $\mathbf{1.575_{020}}$ | $\mathbf{1.316_{021}}$ | $\color{red}1.634_{022}$ | $\mathbf{1.560_{020}}$ | $\mathbf{1.766_{022}}$ | $\mathbf{1.665_{022}}$ | $\color{red}\mathbf{.766_{007}}$ | $.716_{007}$ | $.492_{112}$ |

Table 21: A counterpart of Table 2 regarding RMSE for Task-S and EL3 datasets.

| Learning | Labeling | DIA | PYR | APR | SER | TRI | WBC | CPU | AMP | BOS | STO |
|---|---|---|---|---|---|---|---|---|---|---|---|
| AD | NNT | $.560_{126}$ | $.407_{155}$ | $.362_{083}$ | $.259_{108}$ | $.629_{102}$ | $.859_{113}$ | $.205_{104}$ | $.452_{046}$ | $.481_{058}$ | $.198_{040}$ |
|  | EOT | $.607_{119}$ | $.400_{131}$ | $.352_{096}$ | $.258_{118}$ | $.634_{093}$ | $.863_{111}$ | $.197_{104}$ | $\mathbf{.433_{055}}$ | $.468_{056}$ | $.203_{038}$ |
| Hinge-IT non-ordered | MT | $.574_{129}$ | $.421_{180}$ | $.384_{085}$ | $.248_{120}$ | $.625_{100}$ | $.922_{106}$ | $.199_{105}$ | $.443_{058}$ | $.449_{049}$ | $.176_{032}$ |
|  | ST | $.574_{129}$ | $.421_{180}$ | $.384_{085}$ | $.248_{120}$ | $.625_{100}$ | $.922_{106}$ | $.199_{105}$ | $.443_{058}$ | $.449_{049}$ | $.176_{032}$ |
|  | EOT | $.608_{130}$ | $.432_{138}$ | $\mathbf{.347_{082}}$ | $.264_{106}$ | $.640_{095}$ | $\mathbf{.853_{106}}$ | $.198_{115}$ | $\mathbf{.416_{053}}$ | $.442_{042}$ | $.186_{033}$ |
| Hinge-IT ordered | MT,ST | $\mathbf{.552_{137}}$ | $.414_{156}$ | $.380_{075}$ | $.263_{098}$ | $.618_{092}$ | $.936_{102}$ | $.207_{106}$ | $.447_{048}$ | $.448_{051}$ | $\mathbf{.172_{034}}$ |
|  | EOT | $.610_{140}$ | $.405_{167}$ | $\mathbf{.344_{094}}$ | $.268_{138}$ | $.633_{088}$ | $\mathbf{.870_{113}}$ | $.206_{114}$ | $\mathbf{.418_{056}}$ | $.444_{047}$ | $.186_{032}$ |
| Hinge-AT ordered | MT,ST | $.567_{137}$ | $.407_{164}$ | $.379_{086}$ | $.276_{098}$ | $.610_{088}$ | $.882_{119}$ | $.208_{103}$ | $.445_{048}$ | $.458_{054}$ | $\mathbf{.167_{044}}$ |
|  | EOT | $.597_{133}$ | $.430_{142}$ | $\mathbf{.343_{083}}$ | $.287_{103}$ | $.625_{093}$ | $.861_{112}$ | $.199_{114}$ | $\mathbf{.410_{060}}$ | $.454_{050}$ | $.183_{030}$ |
| Logistic-IT non-ordered | MT | $.617_{126}$ | $.473_{159}$ | $.392_{113}$ | $.245_{115}$ | $.664_{110}$ | $.887_{112}$ | $.232_{105}$ | $.408_{053}$ | $.430_{055}$ | $.168_{033}$ |
|  | ST | $.617_{126}$ | $.473_{159}$ | $.392_{113}$ | $.245_{115}$ | $.664_{110}$ | $.887_{112}$ | $.232_{105}$ | $.408_{053}$ | $.430_{055}$ | $.168_{033}$ |
|  | EOT | $.621_{130}$ | $\mathbf{.449_{169}}$ | $.348_{099}$ | $.266_{114}$ | $.631_{096}$ | $\mathbf{.833_{103}}$ | $.218_{121}$ | $.401_{057}$ | $.427_{055}$ | $.170_{033}$ |
| Logistic-IT ordered | MT,ST | $.618_{127}$ | $.473_{177}$ | $.380_{094}$ | $.244_{111}$ | $.647_{097}$ | $.895_{106}$ | $.231_{093}$ | $.409_{050}$ | $.401_{044}$ | $.168_{032}$ |
|  | EOT | $.646_{119}$ | $.462_{183}$ | $\mathbf{.345_{096}}$ | $.269_{122}$ | $.632_{094}$ | $\mathbf{.835_{106}}$ | $.228_{104}$ | $.401_{053}$ | $.415_{055}$ | $.173_{033}$ |
| Logistic-AT ordered | MT,ST | $.604_{140}$ | $.456_{171}$ | $.381_{090}$ | $.263_{098}$ | $.630_{098}$ | $.801_{088}$ | $.232_{099}$ | $.404_{050}$ | $.417_{051}$ | $.169_{029}$ |
|  | LB | $.629_{109}$ | $.467_{164}$ | $.380_{088}$ | $.262_{099}$ | $.630_{092}$ | $.864_{120}$ | $.234_{098}$ | $.406_{050}$ | $.410_{048}$ | $.170_{028}$ |
|  | EOT | $.626_{118}$ | $.446_{155}$ | $\mathbf{.346_{082}}$ | $.271_{111}$ | $.637_{104}$ | $.837_{105}$ | $.225_{103}$ | $.400_{049}$ | $.419_{047}$ | $.170_{030}$ |
| OLR-NLL ordered | MT,ST | $.618_{119}$ | $.455_{161}$ | $.366_{087}$ | $.256_{108}$ | $.640_{108}$ | $\mathbf{.792_{083}}$ | $.227_{111}$ | $.407_{052}$ | $.405_{045}$ | $.171_{028}$ |
|  | LB | $.623_{120}$ | $.459_{172}$ | $.372_{099}$ | $.254_{108}$ | $.633_{103}$ | $.859_{103}$ | $.236_{100}$ | $.408_{050}$ | $.416_{055}$ | $.172_{033}$ |
|  | EOT | $.636_{113}$ | $.461_{173}$ | $.344_{086}$ | $.263_{122}$ | $.631_{103}$ | $\mathbf{.822_{092}}$ | $.224_{104}$ | $.405_{049}$ | $.425_{056}$ | $.170_{029}$ |

| Learning | Labeling | ABA | AI2 | KRA | CO1 | PU1 | BA1 | CO2 | PU2 | BA2 | EL2 |
|---|---|---|---|---|---|---|---|---|---|---|---|
| AD | NNT | $.488_{014}$ | $.172_{012}$ | $.445_{013}$ | $.107_{016}$ | $.554_{015}$ | $.255_{019}$ | $.120_{014}$ | $.355_{011}$ | $.317_{021}$ | $.248_{008}$ |
|  | EOT | $.489_{014}$ | $.173_{013}$ | $.440_{016}$ | $.106_{014}$ | $\mathbf{.539_{012}}$ | $\mathbf{.233_{011}}$ | $.121_{014}$ | $\mathbf{.346_{010}}$ | $\mathbf{.297_{012}}$ | $.247_{008}$ |
| Hinge-IT non-ordered | MT | $.486_{012}$ | $.172_{012}$ | $.387_{013}$ | $.101_{011}$ | $.530_{012}$ | $.235_{014}$ | $.115_{012}$ | $\mathbf{.331_{013}}$ | $.305_{012}$ | $.248_{008}$ |
|  | ST | $.486_{012}$ | $.172_{012}$ | $.387_{013}$ | $.101_{011}$ | $.530_{012}$ | $.235_{014}$ | $.115_{012}$ | $\mathbf{.331_{013}}$ | $.305_{012}$ | $.248_{008}$ |
|  | EOT | $.487_{015}$ | $.172_{013}$ | $.385_{014}$ | $.102_{012}$ | $.530_{011}$ | $.233_{013}$ | $.113_{011}$ | $.336_{015}$ | $.301_{014}$ | $.246_{007}$ |
| Hinge-IT ordered | MT,ST | $.485_{012}$ | $.172_{012}$ | $.383_{015}$ | $.102_{010}$ | $.530_{011}$ | $.232_{012}$ | $.114_{011}$ | $\mathbf{.331_{013}}$ | $.305_{011}$ | $.248_{008}$ |
|  | EOT | $.486_{013}$ | $.173_{012}$ | $.381_{015}$ | $.102_{011}$ | $.530_{012}$ | $.233_{013}$ | $.114_{011}$ | $.337_{012}$ | $\mathbf{.299_{013}}$ | $.247_{008}$ |
| Hinge-AT ordered | MT,ST | $.486_{012}$ | $.172_{012}$ | $.379_{012}$ | $.101_{013}$ | $.529_{011}$ | $.234_{012}$ | $.115_{010}$ | $.338_{014}$ | $.304_{011}$ | $.248_{008}$ |
|  | EOT | $.486_{013}$ | $.172_{012}$ | $.378_{013}$ | $.101_{013}$ | $.530_{012}$ | $.233_{011}$ | $.114_{011}$ | $.342_{014}$ | $\mathbf{.299_{012}}$ | $.247_{008}$ |
| Logistic-IT non-ordered | MT | $.483_{015}$ | $.169_{012}$ | $.372_{013}$ | $.108_{014}$ | $.526_{010}$ | $.231_{011}$ | $.117_{013}$ | $.313_{013}$ | $.301_{013}$ | $.246_{008}$ |
|  | ST | $.483_{015}$ | $.169_{012}$ | $.372_{010}$ | $.108_{014}$ | $.526_{010}$ | $.231_{011}$ | $.117_{013}$ | $.313_{013}$ | $.301_{013}$ | $.246_{008}$ |
|  | EOT | $.481_{013}$ | $.173_{012}$ | $.372_{012}$ | $\mathbf{.102_{012}}$ | $.526_{010}$ | $.231_{010}$ | $.114_{011}$ | $.313_{013}$ | $.301_{012}$ | $.246_{008}$ |
| Logistic-IT ordered | MT,ST | $.481_{013}$ | $.170_{013}$ | $.364_{011}$ | $.104_{012}$ | $.526_{011}$ | $.230_{010}$ | $.115_{011}$ | $.303_{012}$ | $.300_{013}$ | $.246_{008}$ |
|  | EOT | $.480_{013}$ | $.172_{014}$ | $.362_{012}$ | $.102_{012}$ | $.527_{011}$ | $.231_{011}$ | $.114_{011}$ | $.301_{012}$ | $.300_{012}$ | $.247_{007}$ |
| Logistic-AT ordered | MT,ST | $.481_{014}$ | $.171_{012}$ | $.364_{012}$ | $.103_{013}$ | $.526_{010}$ | $.231_{010}$ | $.115_{012}$ | $.304_{010}$ | $.300_{014}$ | $.247_{008}$ |
|  | LB | $.481_{014}$ | $.170_{012}$ | $.364_{012}$ | $.103_{012}$ | $.526_{010}$ | $.230_{010}$ | $.116_{012}$ | $.304_{010}$ | $.300_{013}$ | $.247_{008}$ |
|  | EOT | $.478_{014}$ | $.173_{012}$ | $.362_{012}$ | $.101_{012}$ | $.526_{009}$ | $.231_{011}$ | $.116_{012}$ | $.303_{010}$ | $.301_{014}$ | $.247_{008}$ |
| OLR-NLL ordered | MT,ST | $.480_{014}$ | $.170_{013}$ | $.363_{010}$ | $.101_{011}$ | $.527_{010}$ | $.230_{011}$ | $.115_{010}$ | $.305_{011}$ | $.301_{013}$ | $.246_{008}$ |
|  | LB | $.481_{014}$ | $.170_{012}$ | $.363_{010}$ | $.104_{012}$ | $.529_{012}$ | $.231_{011}$ | $.115_{010}$ | $.306_{011}$ | $.301_{013}$ | $.246_{008}$ |
|  | EOT | $.481_{014}$ | $.173_{013}$ | $.363_{011}$ | $.102_{011}$ | $.527_{011}$ | $.230_{011}$ | $.115_{011}$ | $.306_{012}$ | $.300_{013}$ | $.247_{007}$ |

| Learning | Labeling | POT | AI1 | EL1 | CAL | CE1 | CE2 | 2DP | FRA | MVA |
|---|---|---|---|---|---|---|---|---|---|---|
| AD | NNT | $.327_{015}$ | $.262_{009}$ | $.138_{007}$ | $.489_{006}$ | $.197_{008}$ | $.234_{013}$ | $.307_{005}$ | $.353_{016}$ | $.040_{007}$ |
|  | EOT | $.323_{015}$ | $.262_{009}$ | $\mathbf{.136_{008}}$ | $.488_{007}$ | $.197_{009}$ | $\mathbf{.224_{011}}$ | $.308_{005}$ | $.351_{012}$ | $.038_{006}$ |
| Hinge-IT non-ordered | MT | $.351_{012}$ | $.262_{009}$ | $.138_{007}$ | $.478_{007}$ | $.189_{007}$ | $.235_{010}$ | $.308_{005}$ | $.306_{010}$ | $.041_{007}$ |
|  | ST | $.351_{012}$ | $.262_{009}$ | $.138_{007}$ | $.478_{007}$ | $.189_{007}$ | $.235_{010}$ | $.308_{005}$ | $.306_{010}$ | $.041_{007}$ |
|  | EOT | $\mathbf{.325_{014}}$ | $.262_{008}$ | $.135_{009}$ | $.477_{007}$ | $.189_{008}$ | $\mathbf{.225_{010}}$ | $.308_{005}$ | $\mathbf{.303_{010}}$ | $\mathbf{.038_{006}}$ |
| Hinge-IT ordered | MT,ST | $.353_{012}$ | $.262_{009}$ | $.138_{008}$ | $.477_{007}$ | $.191_{008}$ | $.232_{013}$ | $.307_{005}$ | $.302_{009}$ | $.041_{007}$ |
|  | EOT | $\mathbf{.327_{014}}$ | $.262_{009}$ | $\mathbf{.134_{007}}$ | $.476_{008}$ | $.189_{007}$ | $\mathbf{.210_{009}}$ | $.308_{004}$ | $.300_{009}$ | $.039_{007}$ |
| Hinge-AT ordered | MT,ST | $.345_{013}$ | $.262_{009}$ | $.139_{008}$ | $.481_{007}$ | $.190_{008}$ | $.208_{009}$ | $.307_{005}$ | $.302_{009}$ | $.042_{006}$ |
|  | EOT | $\mathbf{.326_{014}}$ | $.263_{008}$ | $\mathbf{.135_{008}}$ | $.481_{008}$ | $.190_{007}$ | $.208_{008}$ | $.308_{005}$ | $.300_{008}$ | $\mathbf{.038_{005}}$ |
| Logistic-IT non-ordered | MT | $.242_{013}$ | $.266_{008}$ | $.138_{008}$ | $.459_{007}$ | $.189_{007}$ | $.226_{010}$ | $.308_{005}$ | $.290_{006}$ | $.039_{007}$ |
|  | ST | $.242_{013}$ | $.266_{008}$ | $.138_{008}$ | $.459_{007}$ | $.189_{007}$ | $.226_{010}$ | $.308_{005}$ | $.290_{006}$ | $.039_{007}$ |
|  | EOT | $\mathbf{.202_{013}}$ | $.263_{009}$ | $\mathbf{.133_{008}}$ | $.459_{007}$ | $.190_{008}$ | $\mathbf{.217_{009}}$ | $.308_{005}$ | $.289_{006}$ | $\mathbf{.035_{006}}$ |
| Logistic-IT ordered | MT,ST | $.246_{011}$ | $.265_{009}$ | $.132_{008}$ | $.460_{008}$ | $.189_{007}$ | $.207_{009}$ | $.307_{005}$ | $.285_{007}$ | $.039_{006}$ |
|  | EOT | $\mathbf{.201_{014}}$ | $.263_{009}$ | $.131_{008}$ | $.459_{008}$ | $.189_{007}$ | $.208_{009}$ | $.308_{005}$ | $.284_{006}$ | $\mathbf{.036_{006}}$ |
| Logistic-AT ordered | MT,ST | $.203_{013}$ | $.264_{009}$ | $.137_{007}$ | $.459_{007}$ | $.188_{007}$ | $.206_{009}$ | $.308_{005}$ | $.286_{006}$ | $.039_{006}$ |
|  | LB | $.210_{013}$ | $.264_{009}$ | $.137_{008}$ | $.458_{006}$ | $.188_{008}$ | $.207_{009}$ | $.308_{005}$ | $.286_{006}$ | $.039_{006}$ |
|  | EOT | $\mathbf{.199_{012}}$ | $.263_{008}$ | $\mathbf{.133_{008}}$ | $.458_{007}$ | $.189_{007}$ | $.208_{009}$ | $.308_{005}$ | $.284_{006}$ | $\mathbf{.037_{006}}$ |
| OLR-NLL ordered | MT,ST | $\mathbf{.199_{013}}$ | $.264_{010}$ | $.137_{008}$ | $.460_{007}$ | $.189_{008}$ | $.216_{008}$ | $.308_{005}$ | $.287_{006}$ | $.040_{006}$ |
|  | LB | $.207_{013}$ | $.264_{009}$ | $.137_{008}$ | $.460_{006}$ | $.188_{007}$ | $.217_{009}$ | $.308_{005}$ | $.287_{006}$ | $.040_{006}$ |
|  | EOT | $.197_{013}$ | $.263_{010}$ | $\mathbf{.133_{008}}$ | $.459_{007}$ | $.188_{007}$ | $.216_{009}$ | $.308_{004}$ | $.285_{006}$ | $\mathbf{.037_{005}}$ |

Table 22: A counterpart of Table 2 regarding RMSE for Task-S and EL5 datasets.

| Learning | Labeling | DIA | PYR | APR | SER | TRI | WBC | CPU | AMP | BOS | STO |
|---|---|---|---|---|---|---|---|---|---|---|---|
| AD | NNT | $.975_{218}$ | $.798_{203}$ | $.498_{168}$ | $.481_{235}$ | $1.089_{186}$ | $1.375_{170}$ | $.343_{130}$ | $.525_{054}$ | $.603_{059}$ | $.408_{031}$ |
| | EOT | $.901_{191}$ | $.820_{205}$ | $.502_{152}$ | $.498_{240}$ | $1.084_{153}$ | $1.362_{138}$ | $.343_{136}$ | $.524_{059}$ | $.605_{062}$ | $.408_{039}$ |
| Hinge-IT non-ordered | MT | $.991_{308}$ | $.789_{263}$ | $.514_{168}$ | $.520_{226}$ | $1.115_{172}$ | $1.389_{160}$ | $.371_{159}$ | $.537_{057}$ | $.631_{061}$ | $.455_{035}$ |
| | ST | $.991_{308}$ | $.775_{237}$ | $.524_{168}$ | $.520_{225}$ | $1.108_{168}$ | $1.390_{162}$ | $.386_{164}$ | $.537_{057}$ | $.631_{061}$ | $.455_{035}$ |
| | EOT | $.867_{192}$ | $.814_{215}$ | $.505_{158}$ | $.524_{235}$ | $1.107_{159}$ | $1.367_{149}$ | $.365_{131}$ | $.514_{053}$ | $.599_{060}$ | $.405_{028}$ |
| Hinge-IT ordered | MT,ST | $.965_{231}$ | $.813_{218}$ | $.502_{167}$ | $.524_{217}$ | $1.093_{175}$ | $1.408_{159}$ | $.384_{157}$ | $.529_{055}$ | $.600_{056}$ | $.383_{032}$ |
| | EOT | $.898_{194}$ | $.811_{206}$ | $.491_{164}$ | $.508_{210}$ | $1.086_{157}$ | $1.375_{160}$ | $.354_{138}$ | $.526_{060}$ | $.604_{060}$ | $.381_{033}$ |
| Hinge-AT ordered | MT,ST | $.998_{245}$ | $.787_{215}$ | $.486_{178}$ | $.514_{194}$ | $1.077_{160}$ | $1.364_{179}$ | $.364_{152}$ | $.523_{051}$ | $.601_{076}$ | $.390_{039}$ |
| | EOT | $.871_{185}$ | $.802_{204}$ | $.488_{165}$ | $.513_{209}$ | $1.069_{156}$ | $1.356_{174}$ | $.363_{124}$ | $.524_{051}$ | $.605_{062}$ | $.384_{032}$ |
| Logistic-IT non-ordered | MT | $.973_{223}$ | $.856_{268}$ | $.528_{160}$ | $.525_{181}$ | $1.123_{187}$ | $1.400_{224}$ | $.424_{186}$ | $.525_{050}$ | $.624_{059}$ | $.380_{030}$ |
| | ST | $.969_{226}$ | $.825_{215}$ | $.540_{163}$ | $.522_{180}$ | $1.115_{167}$ | $1.399_{223}$ | $.433_{198}$ | $.525_{050}$ | $.624_{059}$ | $.380_{030}$ |
| | EOT | $.875_{199}$ | $.808_{226}$ | $.499_{153}$ | $.529_{186}$ | $1.074_{156}$ | $1.370_{173}$ | $.400_{171}$ | $.510_{052}$ | $.581_{062}$ | $.374_{036}$ |
| Logistic-IT ordered | MT,ST | $.947_{189}$ | $.815_{216}$ | $.541_{157}$ | $.505_{190}$ | $1.101_{160}$ | $1.452_{212}$ | $.418_{175}$ | $.506_{050}$ | $.581_{065}$ | $.358_{030}$ |
| | EOT | $.903_{196}$ | $.798_{219}$ | $.501_{153}$ | $.511_{183}$ | $1.085_{151}$ | $1.387_{187}$ | $.401_{167}$ | $.512_{060}$ | $.586_{065}$ | $.356_{033}$ |
| Logistic-AT ordered | MT,ST | $.920_{227}$ | $.812_{214}$ | $.492_{160}$ | $.484_{182}$ | $1.085_{146}$ | $1.362_{168}$ | $.399_{160}$ | $.517_{051}$ | $.573_{057}$ | $.359_{027}$ |
| | LB | $.929_{211}$ | $.819_{232}$ | $.510_{161}$ | $.481_{176}$ | $1.090_{152}$ | $1.365_{163}$ | $.408_{171}$ | $.513_{051}$ | $.579_{064}$ | $.360_{027}$ |
| | EOT | $.883_{211}$ | $.812_{216}$ | $.498_{142}$ | $.496_{201}$ | $1.080_{148}$ | $1.392_{196}$ | $.384_{166}$ | $.515_{053}$ | $.586_{065}$ | $.359_{027}$ |
| OLR-NLL ordered | MT,ST | $.934_{174}$ | $.821_{214}$ | $.512_{159}$ | $.476_{184}$ | $1.078_{163}$ | $1.369_{185}$ | $.421_{179}$ | $.509_{045}$ | $.579_{061}$ | $.356_{031}$ |
| | LB | $.931_{233}$ | $.837_{224}$ | $.515_{162}$ | $.491_{199}$ | $1.108_{163}$ | $1.399_{201}$ | $.420_{179}$ | $.514_{048}$ | $.584_{060}$ | $.358_{030}$ |
| | EOT | $.883_{191}$ | $.800_{225}$ | $.490_{151}$ | $.502_{207}$ | $1.084_{158}$ | $1.393_{190}$ | $.403_{156}$ | $.517_{053}$ | $.588_{066}$ | $.359_{028}$ |

| Learning | Labeling | ABA | AI2 | KRA | CO1 | PU1 | BA1 | CO2 | PU2 | BA2 | EL2 |
|---|---|---|---|---|---|---|---|---|---|---|---|
| AD | NNT | $.596_{032}$ | $.404_{012}$ | $.474_{013}$ | $.228_{013}$ | $.791_{016}$ | $.336_{013}$ | $.262_{012}$ | $.405_{011}$ | $.522_{018}$ | $.430_{011}$ |
| | EOT | $.524_{023}$ | $.376_{012}$ | $.471_{013}$ | $.224_{013}$ | $.785_{013}$ | $.331_{011}$ | $.260_{012}$ | $.400_{012}$ | $.505_{016}$ | $.413_{010}$ |
| Hinge-IT non-ordered | MT | $.583_{027}$ | $.398_{014}$ | $.486_{017}$ | $.238_{015}$ | $.843_{020}$ | $.333_{012}$ | $.264_{013}$ | $.450_{013}$ | $.580_{020}$ | $.430_{011}$ |
| | ST | $.583_{027}$ | $.398_{014}$ | $.486_{017}$ | $.235_{013}$ | $.843_{020}$ | $.333_{012}$ | $.263_{012}$ | $.450_{013}$ | $.580_{020}$ | $.430_{011}$ |
| | EOT | $.521_{021}$ | $.371_{012}$ | $.483_{015}$ | $.228_{014}$ | $.785_{013}$ | $.329_{012}$ | $.261_{011}$ | $.405_{013}$ | $.507_{016}$ | $.412_{010}$ |
| Hinge-IT ordered | MT,ST | $.583_{027}$ | $.385_{014}$ | $.471_{016}$ | $.230_{012}$ | $.820_{015}$ | $.333_{012}$ | $.263_{011}$ | $.411_{011}$ | $.574_{023}$ | $.429_{011}$ |
| | EOT | $.512_{022}$ | $.371_{012}$ | $.468_{016}$ | $.227_{012}$ | $.786_{013}$ | $.329_{013}$ | $.261_{011}$ | $.401_{011}$ | $.501_{017}$ | $.412_{010}$ |
| Hinge-AT ordered | MT,ST | $.567_{029}$ | $.387_{014}$ | $.480_{014}$ | $.231_{011}$ | $.793_{014}$ | $.333_{012}$ | $.262_{012}$ | $.405_{012}$ | $.536_{022}$ | $.428_{011}$ |
| | EOT | $.511_{021}$ | $.371_{011}$ | $.477_{014}$ | $.226_{011}$ | $.783_{014}$ | $.330_{012}$ | $.259_{013}$ | $.397_{010}$ | $.503_{017}$ | $.412_{011}$ |
| Logistic-IT non-ordered | MT | $.522_{026}$ | $.394_{014}$ | $.468_{016}$ | $.236_{013}$ | $.794_{015}$ | $.326_{012}$ | $.265_{013}$ | $.458_{011}$ | $.524_{019}$ | $.420_{012}$ |
| | ST | $.522_{026}$ | $.371_{012}$ | $.468_{016}$ | $.236_{013}$ | $.794_{015}$ | $.326_{012}$ | $.262_{012}$ | $.458_{011}$ | $.524_{019}$ | $.412_{011}$ |
| | EOT | $.509_{018}$ | $.372_{011}$ | $.463_{014}$ | $.226_{012}$ | $.785_{012}$ | $.327_{011}$ | $.255_{012}$ | $.400_{013}$ | $.502_{016}$ | $.412_{010}$ |
| Logistic-IT ordered | MT,ST | $.513_{023}$ | $.370_{014}$ | $.463_{016}$ | $.233_{011}$ | $.795_{014}$ | $.328_{010}$ | $.259_{013}$ | $.401_{015}$ | $.529_{018}$ | $.413_{010}$ |
| | EOT | $.510_{019}$ | $.371_{012}$ | $.461_{016}$ | $.224_{012}$ | $.786_{013}$ | $.326_{011}$ | $.252_{012}$ | $.396_{012}$ | $.501_{016}$ | $.412_{010}$ |
| Logistic-AT ordered | MT,ST | $.507_{020}$ | $.370_{012}$ | $.470_{017}$ | $.227_{011}$ | $.787_{014}$ | $.328_{011}$ | $.254_{012}$ | $.399_{015}$ | $.505_{017}$ | $.413_{010}$ |
| | LB | $.510_{022}$ | $.370_{011}$ | $.469_{016}$ | $.229_{012}$ | $.791_{016}$ | $.328_{011}$ | $.256_{013}$ | $.400_{013}$ | $.508_{016}$ | $.412_{010}$ |
| | EOT | $.508_{019}$ | $.370_{012}$ | $.468_{016}$ | $.224_{011}$ | $.788_{013}$ | $.327_{012}$ | $.252_{013}$ | $.397_{013}$ | $.505_{018}$ | $.412_{009}$ |
| OLR-NLL ordered | MT,ST | $.509_{019}$ | $.370_{012}$ | $.462_{016}$ | $.226_{012}$ | $.787_{012}$ | $.326_{011}$ | $.256_{012}$ | $.399_{013}$ | $.505_{014}$ | $.412_{010}$ |
| | LB | $.508_{023}$ | $.370_{014}$ | $.462_{016}$ | $.229_{013}$ | $.789_{014}$ | $.326_{011}$ | $.256_{014}$ | $.399_{013}$ | $.508_{018}$ | $.412_{011}$ |
| | EOT | $.508_{018}$ | $.371_{011}$ | $.459_{015}$ | $.224_{011}$ | $.786_{013}$ | $.325_{011}$ | $.252_{011}$ | $.397_{013}$ | $.500_{018}$ | $.412_{010}$ |

| Learning | Labeling | POT | AI1 | EL1 | CAL | CE1 | CE2 | 2DP | FRA | MVA |
|---|---|---|---|---|---|---|---|---|---|---|
| AD | NNT | $.250_{015}$ | $.419_{009}$ | $.213_{007}$ | $.688_{010}$ | $.384_{017}$ | $.492_{017}$ | $.415_{007}$ | $.376_{008}$ | $.065_{008}$ |
| | EOT | $.241_{015}$ | $.415_{007}$ | $.214_{009}$ | $.683_{009}$ | $.373_{014}$ | $.438_{014}$ | $.409_{006}$ | $.374_{006}$ | $.057_{008}$ |
| Hinge-IT non-ordered | MT | $.592_{048}$ | $.421_{008}$ | $.252_{015}$ | $.763_{013}$ | $.397_{024}$ | $.492_{017}$ | $.421_{004}$ | $.404_{011}$ | $.177_{014}$ |
| | ST | $.589_{047}$ | $.421_{008}$ | $.252_{015}$ | $.763_{013}$ | $.401_{022}$ | $.492_{017}$ | $.421_{004}$ | $.404_{011}$ | $.177_{014}$ |
| | EOT | $.534_{057}$ | $.415_{010}$ | $.226_{009}$ | $.718_{009}$ | $.369_{011}$ | $.423_{014}$ | $.421_{004}$ | $.390_{011}$ | $.076_{008}$ |
| Hinge-IT ordered | MT,ST | $.207_{014}$ | $.420_{008}$ | $.229_{008}$ | $.741_{010}$ | $.468_{042}$ | $.492_{017}$ | $.415_{007}$ | $.378_{009}$ | $.065_{007}$ |
| | EOT | $.203_{014}$ | $.414_{007}$ | $.219_{009}$ | $.718_{009}$ | $.369_{012}$ | $.422_{012}$ | $.407_{006}$ | $.370_{005}$ | $.056_{006}$ |
| Hinge-AT ordered | MT,ST | $.180_{012}$ | $.419_{008}$ | $.237_{010}$ | $.722_{008}$ | $.373_{013}$ | $.489_{018}$ | $.418_{006}$ | $.381_{008}$ | $.066_{009}$ |
| | EOT | $.177_{012}$ | $.414_{009}$ | $.223_{010}$ | $.715_{010}$ | $.361_{012}$ | $.410_{013}$ | $.409_{006}$ | $.372_{005}$ | $.058_{007}$ |
| Logistic-IT non-ordered | MT | $.370_{021}$ | $.424_{009}$ | $.242_{012}$ | $.762_{015}$ | $.380_{015}$ | $.405_{014}$ | $.414_{006}$ | $.380_{006}$ | $.206_{019}$ |
| | ST | $.370_{021}$ | $.419_{008}$ | $.242_{012}$ | $.762_{012}$ | $.372_{014}$ | $.402_{014}$ | $.414_{006}$ | $.380_{006}$ | $.206_{019}$ |
| | EOT | $.308_{020}$ | $.414_{010}$ | $.225_{009}$ | $.718_{010}$ | $.364_{012}$ | $.392_{013}$ | $.403_{004}$ | $.369_{005}$ | $.083_{010}$ |
| Logistic-IT ordered | MT,ST | $.176_{011}$ | $.412_{008}$ | $.214_{006}$ | $.744_{010}$ | $.381_{013}$ | $.407_{014}$ | $.404_{005}$ | $.370_{006}$ | $.059_{007}$ |
| | EOT | $.172_{012}$ | $.411_{008}$ | $.214_{008}$ | $.718_{008}$ | $.365_{012}$ | $.392_{014}$ | $.403_{005}$ | $.368_{006}$ | $.051_{007}$ |
| Logistic-AT ordered | MT,ST | $.174_{011}$ | $.413_{008}$ | $.235_{010}$ | $.718_{009}$ | $.357_{012}$ | $.391_{014}$ | $.404_{005}$ | $.370_{005}$ | $.060_{007}$ |
| | LB | $.174_{012}$ | $.416_{008}$ | $.233_{008}$ | $.721_{009}$ | $.360_{014}$ | $.393_{013}$ | $.404_{005}$ | $.370_{005}$ | $.060_{007}$ |
| | EOT | $.171_{010}$ | $.413_{009}$ | $.222_{008}$ | $.716_{008}$ | $.357_{011}$ | $.391_{012}$ | $.402_{004}$ | $.367_{005}$ | $.053_{007}$ |
| OLR-NLL ordered | MT,ST | $.173_{012}$ | $.414_{008}$ | $.223_{009}$ | $.717_{009}$ | $.359_{013}$ | $.390_{014}$ | $.403_{005}$ | $.369_{006}$ | $.098_{013}$ |
| | LB | $.173_{012}$ | $.417_{008}$ | $.224_{008}$ | $.720_{009}$ | $.361_{012}$ | $.393_{013}$ | $.403_{005}$ | $.369_{006}$ | $.098_{013}$ |
| | EOT | $.171_{013}$ | $.414_{009}$ | $.219_{008}$ | $.715_{008}$ | $.357_{012}$ | $.389_{014}$ | $.403_{004}$ | $.368_{005}$ | $.082_{009}$ |

Table 23: A counterpart of Table 2 regarding RMSE for Task-S and EL10 datasets.

| Learning | Labeling | DIA | PYR | APR | SER | TRI | WBC | CPU | AMP | BOS | STO |
|---|---|---|---|---|---|---|---|---|---|---|---|
| AD | NNT | $1.741_{419}$ | $1.276_{544}$ | $.961_{300}$ | $.935_{411}$ | $1.943_{315}$ | $2.858_{296}$ | $.674_{286}$ | $.867_{101}$ | $.983_{135}$ | $.539_{028}$ |
| | EOT | $1.736_{397}$ | $1.309_{534}$ | $.898_{241}$ | $1.034_{439}$ | $1.927_{315}$ | $2.817_{347}$ | $.633_{277}$ | $.878_{098}$ | $.997_{137}$ | $.544_{028}$ |
| Hinge-IT non-ordered | MT | $1.831_{450}$ | $1.310_{587}$ | $.984_{299}$ | $1.192_{373}$ | $1.941_{289}$ | $2.856_{424}$ | $.699_{306}$ | $.897_{092}$ | $1.005_{137}$ | $.627_{028}$ |
| | ST | $1.831_{450}$ | $1.312_{569}$ | $1.020_{301}$ | $1.107_{361}$ | $1.936_{304}$ | $2.764_{330}$ | $.714_{320}$ | $.904_{089}$ | $1.007_{137}$ | $.627_{028}$ |
| | EOT | $1.731_{400}$ | $1.276_{497}$ | $.916_{266}$ | $1.019_{328}$ | $1.940_{309}$ | $2.813_{380}$ | $.691_{331}$ | $.873_{116}$ | $.970_{135}$ | $\mathbf{.550_{034}}$ |
| Hinge-IT ordered | MT,ST | $1.801_{469}$ | $1.282_{515}$ | $.958_{306}$ | $1.057_{400}$ | $1.939_{314}$ | $2.821_{450}$ | $.718_{304}$ | $.884_{105}$ | $.981_{148}$ | $.535_{033}$ |
| | EOT | $1.705_{367}$ | $1.301_{473}$ | $.896_{272}$ | $1.055_{372}$ | $1.915_{317}$ | $2.761_{317}$ | $.692_{285}$ | $.853_{093}$ | $.963_{140}$ | $.533_{029}$ |
| Hinge-AT ordered | MT,ST | $1.690_{418}$ | $1.285_{514}$ | $.950_{293}$ | $1.032_{400}$ | $1.895_{299}$ | $2.769_{289}$ | $.731_{351}$ | $.873_{099}$ | $.966_{135}$ | $.535_{028}$ |
| | EOT | $1.703_{345}$ | $1.277_{487}$ | $.888_{263}$ | $1.120_{421}$ | $1.898_{295}$ | $2.808_{385}$ | $.671_{289}$ | $.871_{111}$ | $.948_{135}$ | $.531_{030}$ |
| Logistic-IT non-ordered | MT | $1.713_{467}$ | $1.325_{539}$ | $1.033_{290}$ | $1.237_{389}$ | $1.944_{293}$ | $2.770_{348}$ | $.750_{376}$ | $.868_{095}$ | $.973_{144}$ | $.557_{030}$ |
| | ST | $1.708_{460}$ | $1.310_{528}$ | $1.042_{300}$ | $1.114_{304}$ | $1.920_{303}$ | $2.785_{362}$ | $.801_{374}$ | $.867_{097}$ | $.972_{144}$ | $.557_{030}$ |
| | EOT | $1.732_{427}$ | $1.284_{468}$ | $\mathbf{.881_{256}}$ | $1.099_{301}$ | $1.907_{284}$ | $2.825_{331}$ | $.672_{278}$ | $.855_{106}$ | $.971_{131}$ | $\mathbf{.533_{026}}$ |
| Logistic-IT ordered | MT,ST | $1.758_{411}$ | $1.312_{527}$ | $.970_{278}$ | $.942_{381}$ | $1.959_{328}$ | $2.806_{376}$ | $.715_{380}$ | $.854_{106}$ | $.963_{131}$ | $.525_{029}$ |
| | EOT | $1.733_{352}$ | $1.276_{486}$ | $\mathbf{.890_{259}}$ | $.959_{349}$ | $1.927_{318}$ | $2.780_{374}$ | $.706_{361}$ | $.856_{123}$ | $.947_{141}$ | $.526_{024}$ |
| Logistic-AT ordered | MT,ST | $1.679_{400}$ | $1.242_{494}$ | $.935_{256}$ | $.995_{368}$ | $1.898_{277}$ | $2.745_{283}$ | $.707_{365}$ | $.858_{098}$ | $.922_{134}$ | $.519_{028}$ |
| | LB | $1.658_{412}$ | $1.252_{525}$ | $.940_{257}$ | $1.020_{390}$ | $1.930_{284}$ | $2.715_{269}$ | $.699_{366}$ | $.863_{108}$ | $.938_{143}$ | $.519_{027}$ |
| | EOT | $1.715_{392}$ | $1.328_{509}$ | $.910_{242}$ | $1.089_{384}$ | $1.911_{279}$ | $2.793_{326}$ | $.661_{300}$ | $.864_{113}$ | $.956_{140}$ | $.518_{029}$ |
| OLR-NLL ordered | MT,ST | $1.678_{368}$ | $1.213_{495}$ | $.914_{274}$ | $.890_{321}$ | $1.914_{277}$ | $2.745_{298}$ | $.705_{341}$ | $.842_{104}$ | $.928_{128}$ | $.516_{029}$ |
| | LB | $1.704_{438}$ | $1.234_{556}$ | $.919_{278}$ | $.912_{349}$ | $1.915_{298}$ | $2.781_{348}$ | $.700_{351}$ | $.855_{109}$ | $.943_{142}$ | $.515_{026}$ |
| | EOT | $1.720_{401}$ | $1.284_{479}$ | $.863_{233}$ | $.957_{379}$ | $1.907_{278}$ | $2.794_{332}$ | $.648_{287}$ | $.842_{101}$ | $.946_{133}$ | $.519_{032}$ |

| Learning | Labeling | ABA | AI2 | KRA | CO1 | PU1 | BA1 | CO2 | PU2 | BA2 | EL2 |
|---|---|---|---|---|---|---|---|---|---|---|---|
| AD | NNT | $.886_{037}$ | $.512_{019}$ | $.679_{019}$ | $.412_{012}$ | $1.392_{026}$ | $.509_{011}$ | $.469_{013}$ | $.575_{013}$ | $1.064_{036}$ | $.639_{017}$ |
| | EOT | $.887_{030}$ | $\mathbf{.488_{018}}$ | $.676_{019}$ | $.412_{011}$ | $1.383_{025}$ | $.509_{012}$ | $.468_{013}$ | $.571_{013}$ | $\mathbf{1.004_{033}}$ | $.621_{016}$ |
| Hinge-IT non-ordered | MT | $1.003_{048}$ | $.517_{017}$ | $.703_{014}$ | $.418_{017}$ | $1.503_{039}$ | $.511_{015}$ | $.477_{018}$ | $.600_{015}$ | $1.112_{039}$ | $.640_{018}$ |
| | ST | $.981_{047}$ | $.517_{017}$ | $.701_{015}$ | $.416_{017}$ | $1.503_{039}$ | $.511_{015}$ | $.472_{016}$ | $.600_{016}$ | $1.112_{039}$ | $.640_{018}$ |
| | EOT | $\mathbf{.913_{036}}$ | $\mathbf{.482_{016}}$ | $\mathbf{.679_{023}}$ | $.413_{016}$ | $\mathbf{1.389_{025}}$ | $\mathbf{.502_{012}}$ | $\mathbf{.463_{013}}$ | $\mathbf{.578_{013}}$ | $\mathbf{.999_{033}}$ | $\mathbf{.622_{015}}$ |
| Hinge-IT ordered | MT,ST | $1.005_{047}$ | $.515_{018}$ | $.700_{020}$ | $.408_{014}$ | $1.598_{038}$ | $.501_{012}$ | $.462_{014}$ | $.590_{013}$ | $1.349_{042}$ | $.641_{019}$ |
| | EOT | $\mathbf{.915_{033}}$ | $\mathbf{.478_{015}}$ | $.694_{022}$ | $.407_{013}$ | $\mathbf{1.390_{028}}$ | $.500_{012}$ | $.463_{015}$ | $.590_{012}$ | $\mathbf{.994_{029}}$ | $\mathbf{.621_{014}}$ |
| Hinge-AT ordered | MT,ST | $.939_{053}$ | $.507_{019}$ | $.692_{022}$ | $.410_{012}$ | $1.413_{030}$ | $.504_{012}$ | $.463_{014}$ | $.577_{012}$ | $1.060_{036}$ | $.643_{018}$ |
| | EOT | $\mathbf{.883_{033}}$ | $\mathbf{.478_{016}}$ | $.689_{023}$ | $.409_{012}$ | $\mathbf{1.379_{024}}$ | $.503_{011}$ | $.462_{013}$ | $.574_{012}$ | $\mathbf{.993_{030}}$ | $\mathbf{.616_{015}}$ |
| Logistic-IT non-ordered | MT | $.982_{044}$ | $.493_{020}$ | $.715_{020}$ | $.413_{016}$ | $1.422_{029}$ | $.502_{013}$ | $.466_{014}$ | $.601_{015}$ | $1.043_{035}$ | $.631_{021}$ |
| | ST | $.913_{040}$ | $.491_{017}$ | $.687_{021}$ | $.411_{013}$ | $1.422_{029}$ | $.502_{013}$ | $.463_{014}$ | $.601_{015}$ | $1.043_{035}$ | $.624_{016}$ |
| | EOT | $\mathbf{.881_{031}}$ | $\mathbf{.475_{014}}$ | $\mathbf{.670_{022}}$ | $\mathbf{.406_{015}}$ | $\mathbf{1.375_{024}}$ | $.501_{011}$ | $.454_{012}$ | $\mathbf{.578_{014}}$ | $\mathbf{1.000_{027}}$ | $\mathbf{.613_{014}}$ |
| Logistic-IT ordered | MT,ST | $.932_{036}$ | $.476_{016}$ | $.689_{023}$ | $.402_{011}$ | $1.450_{027}$ | $.498_{012}$ | $.454_{014}$ | $.588_{013}$ | $1.120_{044}$ | $.622_{016}$ |
| | EOT | $\mathbf{.884_{033}}$ | $.473_{013}$ | $.685_{025}$ | $.403_{012}$ | $\mathbf{1.377_{022}}$ | $.497_{012}$ | $.455_{015}$ | $\mathbf{.584_{012}}$ | $\mathbf{1.012_{029}}$ | $\mathbf{.613_{015}}$ |
| Logistic-AT ordered | MT,ST | $.866_{027}$ | $.473_{015}$ | $.680_{026}$ | $.404_{010}$ | $1.377_{024}$ | $.500_{012}$ | $.456_{013}$ | $.571_{012}$ | $1.001_{027}$ | $.612_{015}$ |
| | LB | $.875_{031}$ | $.473_{014}$ | $.680_{025}$ | $.404_{009}$ | $1.384_{026}$ | $.500_{011}$ | $.455_{013}$ | $.572_{012}$ | $1.019_{032}$ | $.612_{015}$ |
| | EOT | $.873_{029}$ | $.474_{014}$ | $.677_{024}$ | $.403_{011}$ | $1.378_{023}$ | $.499_{012}$ | $.455_{013}$ | $.571_{014}$ | $\mathbf{.991_{029}}$ | $.614_{015}$ |
| OLR-NLL ordered | MT,ST | $.871_{030}$ | $.474_{014}$ | $.668_{026}$ | $.404_{011}$ | $1.378_{023}$ | $.499_{014}$ | $.456_{013}$ | $.571_{012}$ | $1.000_{030}$ | $.614_{015}$ |
| | LB | $.874_{030}$ | $.475_{015}$ | $.670_{024}$ | $.401_{011}$ | $1.384_{026}$ | $.498_{012}$ | $.456_{015}$ | $.572_{013}$ | $1.015_{032}$ | $.613_{015}$ |
| | EOT | $.877_{029}$ | $.473_{014}$ | $.668_{025}$ | $.403_{012}$ | $1.375_{025}$ | $.498_{012}$ | $.457_{012}$ | $.572_{011}$ | $\mathbf{.997_{030}}$ | $.613_{014}$ |

| Learning | Labeling | POT | AI1 | EL1 | CAL | CE1 | CE2 | 2DP | FRA | MVA |
|---|---|---|---|---|---|---|---|---|---|---|
| AD | NNT | $1.084_{112}$ | $.575_{010}$ | $.528_{007}$ | $1.174_{017}$ | $.710_{020}$ | $.748_{030}$ | $.567_{005}$ | $.513_{005}$ | $.223_{009}$ |
| | EOT | $\mathbf{1.046_{102}}$ | $.574_{011}$ | $.529_{007}$ | $1.169_{016}$ | $\mathbf{.702_{020}}$ | $.746_{030}$ | $\mathbf{.564_{004}}$ | $.512_{005}$ | $\mathbf{.202_{010}}$ |
| Hinge-IT non-ordered | MT | $.750_{075}$ | $.620_{023}$ | $.607_{014}$ | $1.324_{018}$ | $.787_{024}$ | $.910_{031}$ | $.597_{005}$ | $.562_{008}$ | $.286_{015}$ |
| | ST | $.745_{073}$ | $.620_{023}$ | $.607_{014}$ | $1.324_{019}$ | $.787_{024}$ | $.910_{031}$ | $.580_{006}$ | $.539_{006}$ | $.257_{012}$ |
| | EOT | $\mathbf{.698_{063}}$ | $\mathbf{.577_{010}}$ | $\mathbf{.572_{011}}$ | $\mathbf{1.288_{015}}$ | $\mathbf{.739_{025}}$ | $\mathbf{.827_{030}}$ | $\mathbf{.565_{004}}$ | $\mathbf{.519_{005}}$ | $\mathbf{.197_{008}}$ |
| Hinge-IT ordered | MT,ST | $1.194_{047}$ | $.577_{011}$ | $.523_{006}$ | $1.259_{021}$ | $.833_{029}$ | $.977_{063}$ | $.572_{005}$ | $.519_{004}$ | $.216_{012}$ |
| | EOT | $\mathbf{.838_{049}}$ | $.574_{010}$ | $.521_{007}$ | $\mathbf{1.155_{015}}$ | $.757_{026}$ | $.845_{029}$ | $\mathbf{.564_{004}}$ | $.518_{004}$ | $\mathbf{.196_{011}}$ |
| Hinge-AT ordered | MT,ST | $.679_{052}$ | $.575_{011}$ | $.521_{006}$ | $1.164_{019}$ | $.765_{026}$ | $.817_{027}$ | $.573_{005}$ | $.520_{005}$ | $.225_{013}$ |
| | EOT | $\mathbf{.604_{041}}$ | $.574_{010}$ | $.519_{006}$ | $1.150_{017}$ | $.715_{026}$ | $\mathbf{.789_{025}}$ | $\mathbf{.564_{004}}$ | $.519_{005}$ | $\mathbf{.207_{009}}$ |
| Logistic-IT non-ordered | MT | $.558_{041}$ | $.593_{011}$ | $.586_{012}$ | $1.312_{019}$ | $.745_{024}$ | $.836_{032}$ | $.592_{006}$ | $.563_{012}$ | $.283_{012}$ |
| | ST | $.558_{041}$ | $.590_{012}$ | $.586_{012}$ | $1.312_{019}$ | $.745_{024}$ | $.825_{030}$ | $.578_{006}$ | $.545_{008}$ | $.267_{011}$ |
| | EOT | $\mathbf{.514_{042}}$ | $\mathbf{.574_{011}}$ | $\mathbf{.558_{010}}$ | $\mathbf{1.284_{017}}$ | $\mathbf{.726_{027}}$ | $\mathbf{.794_{028}}$ | $\mathbf{.565_{004}}$ | $\mathbf{.518_{005}}$ | $\mathbf{.194_{008}}$ |
| Logistic-IT ordered | MT,ST | $.744_{050}$ | $.569_{010}$ | $.518_{007}$ | $1.245_{017}$ | $.752_{028}$ | $.899_{036}$ | $.567_{004}$ | $.510_{004}$ | $.194_{011}$ |
| | EOT | $.567_{041}$ | $.568_{010}$ | $.518_{007}$ | $1.151_{016}$ | $.688_{022}$ | $.804_{031}$ | $\mathbf{.565_{004}}$ | $.510_{004}$ | $\mathbf{.181_{008}}$ |
| Logistic-AT ordered | MT,ST | $\mathbf{.567_{038}}$ | $.574_{010}$ | $.545_{008}$ | $\mathbf{1.276_{017}}$ | $\mathbf{.712_{024}}$ | $\mathbf{.774_{028}}$ | $.569_{004}$ | $.519_{005}$ | $.210_{009}$ |
| | LB | $.616_{038}$ | $.575_{011}$ | $.547_{009}$ | $1.289_{017}$ | $.724_{026}$ | $.787_{029}$ | $.569_{004}$ | $.519_{005}$ | $.208_{011}$ |
| | EOT | $\mathbf{.555_{034}}$ | $.574_{010}$ | $\mathbf{.532_{007}}$ | $1.275_{018}$ | $\mathbf{.708_{023}}$ | $\mathbf{.773_{026}}$ | $.564_{004}$ | $\mathbf{.516_{005}}$ | $\mathbf{.192_{008}}$ |
| OLR-NLL ordered | MT,ST | $.502_{042}$ | $.574_{010}$ | $.545_{009}$ | $\mathbf{1.278_{018}}$ | $\mathbf{.722_{024}}$ | $\mathbf{.780_{029}}$ | $.569_{004}$ | $.518_{005}$ | $.200_{008}$ |
| | LB | $.529_{043}$ | $.574_{010}$ | $.544_{008}$ | $1.288_{018}$ | $.732_{026}$ | $.791_{029}$ | $.568_{004}$ | $.518_{005}$ | $.199_{008}$ |
| | EOT | $.488_{040}$ | $.573_{011}$ | $\mathbf{.531_{007}}$ | $1.275_{017}$ | $.721_{025}$ | $\mathbf{.780_{028}}$ | $\mathbf{.564_{004}}$ | $.516_{005}$ | $\mathbf{.183_{008}}$ |

Table 24: The number $N_Z$ (resp., $N_A, N_S$) of times that each method was best for Task-Z (resp., Task-A, S) in each dataset group in the form '$N_Z, N_A, N_S$', and their summation in the column 'SUM'. The larger the number, the better than method is for that dataset group and that task.

| Learning | Labeling | RW | EF3 | EF5 | EF10 | EL3 | EL5 | EL10 | SUM |
|---|---|---|---|---|---|---|---|---|---|
| AD | NNT | 2,1,1 | 0,0,0 | 0,0,0 | 1,2,0 | 1,1,1 | 4,3,1 | 0,1,0 | 8,8,3 |
|  | EOT | 0,2,0 | 1,0,0 | 4,1,1 | 9,5,3 | 1,2,3 | 3,3,3 | 3,1,3 | 21,14,13 |
| Hinge-IT non-ordered | MT | 0,0,0 | 1,0,0 | 0,0,0 | 0,1,0 | 2,1,1 | 1,0,0 | 0,0,0 | 4,2,1 |
|  | ST | 0,0,0 | 1,0,0 | 0,1,0 | 0,0,0 | 2,1,1 | 1,0,1 | 0,0,0 | 4,2,2 |
|  | EOT | 3,3,1 | 0,0,0 | 2,0,2 | 1,0,2 | 1,3,1 | 0,0,1 | 0,0,0 | 7,6,7 |
| Hinge-IT ordered | MT,ST | 0,0,0 | 0,0,0 | 0,0,0 | 1,0,0 | 2,2,1 | 0,0,0 | 1,0,0 | 4,2,1 |
|  | EOT | 2,2,1 | 1,1,1 | 2,3,3 | 1,0,1 | 1,0,0 | 0,1,0 | 2,0,0 | 9,7,6 |
| Hinge-AT ordered | MT,ST | 0,1,1 | 0,1,0 | 0,0,0 | 0,1,0 | 3,3,3 | 2,2,1 | 1,2,1 | 6,10,6 |
|  | EOT | 2,3,4 | 1,2,1 | 2,5,0 | 2,1,2 | 0,0,1 | 1,1,3 | 0,1,1 | 8,13,12 |
| Logistic-IT non-ordered | MT | 0,0,0 | 2,4,1 | 0,0,0 | 0,0,0 | 1,2,1 | 0,0,0 | 0,0,0 | 3,6,2 |
|  | ST | 0,0,0 | 2,4,1 | 0,0,0 | 0,0,0 | 1,2,1 | 0,0,0 | 0,0,0 | 3,6,2 |
|  | EOT | 2,3,4 | 5,4,5 | 0,0,0 | 0,0,1 | 2,2,2 | 1,1,1 | 2,2,1 | 12,12,14 |
| Logistic-IT ordered | MT,ST | 1,0,0 | 1,1,1 | 1,0,0 | 2,0,0 | 3,5,2 | 1,0,1 | 3,1,0 | 12,7,4 |
|  | EOT | 1,0,0 | 8,7,6 | 4,4,3 | 2,2,3 | 5,5,4 | 4,6,4 | 7,8,6 | 31,32,26 |
| Logistic-AT ordered | MT,ST | 0,0,1 | 1,2,3 | 3,4,5 | 1,3,4 | 0,0,1 | 0,1,3 | 3,3,3 | 8,13,20 |
|  | LB | 0,0,0 | 0,2,1 | 2,4,3 | 2,3,0 | 1,0,1 | 0,1,0 | 0,3,3 | 5,13,8 |
|  | EOT | 2,4,5 | 3,2,5 | 4,2,3 | 1,2,5 | 2,1,2 | 4,4,4 | 2,2,2 | 18,17,26 |
| OLR-NLL ordered | MT,ST | 1,0,0 | 1,2,2 | 1,3,1 | 2,4,1 | 1,0,1 | 1,1,1 | 2,4,3 | 9,14,9 |
|  | LB | 1,0,1 | 2,2,1 | 1,3,1 | 2,4,2 | 0,0,1 | 1,1,0 | 2,4,2 | 9,14,8 |
|  | EOT | 4,4,0 | 2,3,2 | 3,6,7 | 2,8,5 | 3,4,3 | 6,6,5 | 1,4,4 | 21,35,26 |

