# OpenReview forum: "Optimal Threshold Labeling for Ordinal Regression Methods"
_TMLR — Accepted by TMLR_

### Review · Reviewer_UiW4 · 2022-12-04

**Summary Of Contributions:**

This paper tackles the ordinal regression problem which consists in affecting an input to a certain discrete category. Further, all categories are assumed to be ordered: for instance, the 3 categories "good", "medium" and "bad". The 1-dimensional transformation (1DT) approach relies on two components. First, a mapping $g$ transforming any multidimensional input $x \in \mathbb{R}^d$ to a scalar value. Second component, a labeling function $h$ mapping any real number to a discrete category ranging from 1 to K. A natural and popular labeling approach consists in using threshold to discretize the real line.
Both components ($g$ and $h$) can be learned from an i.i.d. dataset of input-target pairs $(x_i,y_i)$ valued in $\mathbb{R}^d \times [K ]$ so as to minimize an empirical loss $\frac{1}{N}\sum_{i=1}^n loss(h\circ g(x_i), y_i)$.
This paper focuses on the learning of the second component, namely the labeling function $h$, given some 1DT $g$.
After reviewing and providing theoretical guarantees for existing labeling methods (minimum threshold, summation threshold, nearest-neighbor threshold, labeling based on likelihood ), the authors describe their own method consisting in choosing the threshold parameters based on an empirical risk minimization procedure. From a computational point of view, the authors recall that these empirical optimal thresholds can be efficiently computed via dynamic programming.
Numerical experiments on several datasets (MORPH-2, CACD,AFAD ) illustrate the efficiency of the proposed method.

**Audience:**

Yes

**Broader Impact Concerns:**



**Claims And Evidence:**

Yes

**Requested Changes:**

minor concern: lack of discussion about the full joint optimization problem, over both components $g$ (1DT) and $h$ (labeling function)

**Strengths And Weaknesses:**

- The paper is well written and draws a broad picture of the ordinal regression problem
- related work is well discussed and theoretical guarantees are also given for already existing methods
- method based on a computationally efficient computation of the empirical optimal thresholds (quasi-linear time complexity of O(n*log(n)) with logarithmic factor due to a sorting step)

---

> ### Comment · Action_Editors · 2022-12-08
> **review needs updating**
>
> Hi Reviewer UiW4,
>
> Thanks for reviewing this TMLR paper. Somehow it is to my attention that your review is almost the same as the one to the earlier submission of this paper, while the authors have changed their contents significantly (e.g. there is no IOT anymore but your review still says IOT). So your review may not be based on the correct version of the paper. Can you read the latest version again and make sure that your review is based on the current version? Thank you!

---

### Review · Reviewer_Z9Q8 · 2022-12-09

**Summary Of Contributions:**

The submission has been changed to focus on an (existing) dynamic programming algorithm that finds the optimum thresholds for the discretization problem considered in the paper. The previously proposed algorithm has been removed.

The empirical results indicate that by applying the dynamic programming algorithm, it is possible to obtain small improvements in zero-one loss, absolute error, and squared error, for some datasets and some ordinal regression methods considered.

There are also some theoretical results (already present in the previous version) showing that certain types of likelihood-based ordinal regression methods can be interpreted in the threshold-based framework.

**Audience:**

Yes

**Claims And Evidence:**

No

**Requested Changes:**

The submission seems close to being ready for publication. However, I am disappointed to see that the significance level for the Wilcoxon rank sum test has been changed back to 0.1 (after this had been changed to 0.05 in the previous version following my concerns). Moreover, and I hadn't spotted this previously, a one-sided test is used instead of a two-sided one, which makes Type I error even more likely! Please use a 0.05 level with a two-sided test.

The dynamic programming algorithm (Algorithm 1) appears to depend on the fact that the loss function can be decomposed into a sum over subsets of instances. What about target losses such as the quadratic weighted kappa, which has been used in competitions involving ordinal regression problems (https://www.kaggle.com/c/prudential-life-insurance-assessment/overview/evaluation)?

The impact of K on the runtime of Algorithm 1 should be made clear. The algorithm appears to be quadratic in K, which could be relevant for problems with a large number target values. The min operation in Line 8 is linear in k.

Why are the results for SVOR no longer included in the table with results (Table 2 in the new version)?

Typos, etc.:

"imagine that" -> "because, generally," ?

"replacing ... to" -> "replacing ... by"

"can predict it as" -> "one can predict it as"

"minimization of sum" -> "minimization of the sum"

Theorem 3(i): "such as" -> "such as it is for" ?

Theorem 3(ii): Clarify what $\sigma^\prime$ is? Also, "such as" -> "such as it is for" ?

"is non-positive and non-negative" -> "non-positive (resp. non-negative)"

"a threshold labeling ... have"

"we changed a learning rate" -> "we changed the learning rate" ?

"we adopted a model at the timing with the smallest error" -> "we adopted the model at the time step with the smallest error"

"the one-sided Wilcoxon rank sum test with p-value 0.1" - why is the p-value back at 0.1? Also, and I didn't spot this last time, why is the one-sided version of the test used, which makes "significant" differences even more likely?

"as its design" -> "as its designed to do"

"were tie each other" -> "were tied with each other"

"able to take" -> "able to undertake"

"the EOT labeling will serve as the other ... components"

"contradicts to the equation (24)" -> "contradicts equation (24)"

"for a some point"

"j >= as well"


**Strengths And Weaknesses:**

The submission no longer presents a new algorithm, but focusing on the existing optimal algorithm instead seems a good choice. It may also be useful to have this existing algorithm spelled out in detail.

There is a nice discussion of all the various approaches from the literature that fit into the threshold-based framework, and it is interesting to see that some likelihood-based methods can also be covered.

The submission includes experiments on a fairly extensive collection of real-world ordinal regression datasets.

The observed improvements in predictive performance are rather small.

---

> ### Author Response · Authors · 2022-12-09
> **1st reply to Reviewer Z9Q8**
>
> We greatly appreciate the quick and constructive review. We here reply pointed Requested Changes.
>
> ---
>
> [1] **Change of $p$-value from 0.05 to 0.1**
>
> For the experimental settings, we referred to those of the previous study (Cao et al., 2020) and (Gutierrez et al., 2015).
> (Cao et al., 2020) performed only 3 trials, and the comparison is not good.
> (Gutierrez et al., 2015) performed Wilcoxon test with $p=0.1$;
> see the last paragraph of Section 4.1 of (Gutierrez et al., 2015).
> Exactly, it was the one-sided Wilcoxon signed rank test.
> A one-sided test is required in order for the test to yield win, tie, and lose (not two-sided test).
> We used $p=0.1$ in this submission because we thought it would be better to follow the settings of existing studies as much as possible (although we were worried about whether to set it to 0.05 or 0.1.).
> However, since the one-sided Wilcoxon signed rank test requires too-strong assumption (symmetry around the median), we changed the test to the one-sided Wilcoxon rank sum test.
> We believe that this change is reasonable.
>
>
> [2] **Quadratic weighted kappa**
>
> For the quadratic weighted kappa,
> $$
> \text{empirical task risk}
> =1-\frac{\sum_{j,k\in[K]} \ell(j, k) \frac{\sum_{i\in[n]}\mathbb{I}(f(x_{i})=j\text{ and }y_{i}=k)}{n}}{\sum_{j,k\in[K]} \ell(j, k) \frac{\sum_{i\in[n]}\mathbb{I}(f(x_{i})=j)}{n} \frac{\mathbb{I}(y_{i}=k)}{n}}
> =1-\frac{\sum_{i\in[n]} \ell(f(x_{i}),y_{i})}{\sum_{j,k\in[K]} \ell(j, k) \frac{\sum_{i\in[n]}\mathbb{I}(f(x_{i})=j)}{n} \frac{\mathbb{I}(y_{i}=k)}{n}},
> $$
> which cannot be decomposed into a sum over subsets of instances.
> We will note at the end of Section 2.1 that our discussion is not applicable to such criteria.
>
>
> [3] **impact of K on the runtime of Algorithm 1**
>
> Current Algorithm 1 has the computational load of order $\mathcal{O}(K^2)$ in $K$ due to line 8.
> However, $\min$ in line 8 can be made more efficient.
> We will modify it such that the algorithm has the computational load of order $\mathcal{O}(K)$ in $K$.
> Thank you greatly constructive review.
>
> [4] **SVOR in Experiment**
>
> The focus of this paper is the selection of the labeling function $h$.
> We noticed the danger that the experimental results with loss functions of different properties mislead the reader's focus from the selection of the labeling function to the selection of the loss function.
> Also, in the previous submission, Logistic-AT (and Hinge-IT) was used, but Logistic-IT (and Hinge-AT) was not used.
> We thought this was more unnatural, so we attempt to set up more natural experimental settings.
>
> [5] **Typos, etc.**
> Thank you for careful reading.
> I think that the usage of "such as" is not incorrect, but we will revise the text to be more readable.
>
> ---
>
> Additionally,
>
> Although our contribution is (appropriately) smaller than our previous submission, we believe that the discussion in Section 3.3 is completely new and meaningful.
> Thank you for your attention to the discussion in section 3.3.

---

> > ### Comment · Action_Editors · 2022-12-10
> > **SVOR**
> >
> > Adding my two cents about [4]: Given that SVOR was a very representative approach (STOA for a long time) in the OR family, including the results on SVOR help understand the relative strength and difference of the proposed approach and should strengthen this paper.

---

> > > ### Author Response · Authors · 2022-12-10
> > > **Reply for SVOR**
> > >
> > > Thank you for the advice.
> > >
> > > I will add the experiments with Hinge-IT, non-ordered; Hinge-IT, ordered; Hinge-AT, ordered.
> > >
> > > The experiment will take about a week, so it will take some time to revise the paper.

---

### Review · Reviewer_RVhG · 2022-12-16

**Summary Of Contributions:**

The authors consider the setting of ordinal regression/classification (OR), where the task is to learn a mapping from X to Y, which is a set of classes with ordinal relations. The author focuses on the group of one-dimensional transformation (1DT)-based methods that first map X to a continuous score and then apply the labeling function that uses thresholds to decide on the final class by checking to which interval between thresholds the score belongs. The authors discuss a few methods for learning a mapping from X to score by using different surrogate losses (since task losses considered in OR have a discreet nature) and different labeling functions for mapping scores to Ys under unified notation. The authors demonstrate that discussed 1DT methods can become sub-optimal for the minimization of task risk. Because of that the authors propose to apply Empirical Optimal Threshold (EOR) labeling procedure that aims to find a proper threshold by minimizing empirical task loss on available data. It is done by finding optimal thresholds turning the problem into a combinatorial problem, and solving it using dynamic programming (an algorithm proposed by Lin & Li (2006)). The proposed approach is evaluated for different methods for learning scores and compared with other labeling methods on eight datasets. The proposed approach improves mean test task loss in most of the cases, mainly being worst only on the WQR dataset.

**Audience:**

Yes

**Broader Impact Concerns:**

I have no concerns about the ethical implications of this work.

**Claims And Evidence:**

Yes

**Requested Changes:**

I believe that description of the EOT method and Algorithm 1 would be a nice addition that would make the paper more complete.

**Strengths And Weaknesses:**

Strengths:
- The paper nicely describes different 1DT methods under the same notation. I find the rewritten Sections 2 and 3 to be much easier to understand than in the previous version.
- The proposed EOT approach beats competing methods in most cases.
- Not any expert in the area, but related works seem to be sufficiently covered and discussed, the previous comments of the reviewers regarding this aspect seem to be correctly addressed by the authors in this revision.
- The paper is sound and well-written. I see significant improvement over the previous version. I didn't notice any mistakes.

Weaknesses:
- The EOT method could be better explained in the text, as in the previous version IOT was explained. Since proposition of using EOT still seems to be one of the main contributions of this work among theoretical results on sub-optimality of some methods and unification of notation. At the moment, Algorithm 1 is not described in the text, the reader needs to refer to Lin & Li (2006).
- Because of the lack of explanation, it is not that clear what is authors' contribution and what is the contribution of Lin & Li (2006) is in terms of the EOT approach. Is it only the task under consideration, as explained in Section 5?
- If I remember correctly, the previous version of the manuscript was criticized for using a significance level
of 0.1, the authors changed it to 0.05 again in this version using 0.1.

---

> ### Author Response · Authors · 2022-12-23
> **1st reply to Reviewer RVhG**
>
> We greatly appreciate the quick and constructive review. We here reply pointed Requested Changes.
>
> ---
>
> [1] More explanation of Algorithm 1.
>
> See the 2nd paragraph in Section 4.
> We emphasize the recurrence relation (46).
>
> ---
>
> [2] Clarify difference of (Lin & Li, 2006)'s and our contributions on the EOT labeling.
>
> See the added last sentence in Section 5.
> Our contribution is to expand the scope of applicability of the EOT labeling, and to consider the relation to the task under consideration.
> (Lin & Li, 2006)'s contribution is development of Algorithm 1 as a method for the minimization of
> the empirical task risk with a given task loss function.
> As we written the reason in the 3rd paragraph in Section 4, many regression-based, threshold, and statistical methods have not applied the EOT labeling.
> We believe that expansion of the scope of applicability of the EOT labeling is also important in the OR research.
>
> ---
>
> [3] We changed $p$-value from 0.1 to 0.05.
>
> We are sorry that we have confused the reviewers.
> We turn to consider that it is necessary to stick to the settings of existing works.

---

### Author Response · Authors · 2022-12-23
**We submit the revision**

Thank you for reading this again and again.

We have carefully considered the reviewers' and AE's comments and have made the following major modifications (**these include modification of our previous replies**):

- We changed $p$-value from 0.1 to 0.05.

We are sorry that we have confused the reviewers.
We turn to consider that it is necessary to stick to the settings of existing works.

- Expand the experiments.

See Tables 1--24 and Figure 1 in Section 6 and Appendix D.
The summary is Table 24 which indicates the usefulness of the EOT labeling.
The experiments in Appendix D is designed following (Frank & Hall, 2001) and (Chu & Ghahramani, 2005).
We also tried Hinge-IT and Hinge-AT losses.

- Correct RMSE.

We had forgotten the root operation for the RMSE.
Sorry, we modify it (superiority or inferiority does not change).

- On other-type criteria such as quadratic weighted kappa.

See the last paragraph in Section 2.1.

- More explanation of Algorithm 1.

See the 2nd paragraph in Section 4.
We emphasize the recurrence relation (46).

- Modification of Algorithm 1.

See Lines 8 and 9.
This is an efficient implementation of min operation, and reduces computational complexity regarding the number $K$ of classes from $O(K^2)$ to $O(K)$.

- Clarify difference of (Lin & Li, 2006)'s and our contributions on the EOT labeling.

See the added last sentence in Section 5.
Our contribution is to expand the scope of applicability of the EOT labeling, and to consider the relation to the task under consideration.
(Lin & Li, 2006)'s contribution is development of Algorithm 1 as a method for the minimization of
the empirical task risk with a given task loss function.
As we written the reason in the 3rd paragraph in Section 4, many regression-based, threshold, and statistical methods have not applied the EOT labeling.
We believe that expansion of the scope of applicability of the EOT labeling is also important in the OR research.

---

If you remain some question, please ask us here.

---

### Decision · Action_Editors · 2023-01-14

**Recommendation:** Accept as is

**Comment:**

The author studies the one-dimensional transform (1DT) methodology to solve the ordinal regression problem, which projects the data by some scoring function and then quantizes the scores with respect to some loss function of interest. The author discusses the methodology with a decent review of the field, solve the optimization problem with a pre-existing dynamic programming method (that was not deeply studied), and demonstrates that the proposed method achieves superior performance on several data sets.

All reviewers agree that the paper is significantly improved over the previous version and should be accepted. The contribution is on revisiting a simple solution for the task, gluing existing work together with the 1DT perspective, and sufficient experiments to back the solution. The author is encouraged to address all remaining issues pointed out by reviewers in the camera-ready version.

While one reviewer recommended the survey certification, the AE checked the paper and believed that the survey part was insightful but arguably not broad enough for this mature field. Hence the survey certification is not recommended.

**Audience:**

Yes!

**Claims And Evidence:**

Yes!

---

> ### Author Response · Authors · 2023-01-15
> **I submit the camera-ready version.**
>
> I submit the camera-ready version.
>
> Thank you for reading this again and again.
> Thanks to constructive advices by the AE and reviewers, this paper becomes a marked improvement over the first submitted version.
> Thank you for this as well.
>
> I disclose author name, affiliation, acknowledgments, and github repository that are anonymized in the review process.
> I believe that this camera-ready version addresses all issues pointed out by reviewers.
>
> Also, I am satisfied with AE's decision regarding "the survey certification".

---

> > ### Comment · Action_Editors · 2023-01-18
> > **EF5 or EL5**
> >
> > Hi author,
> >
> > Could you double-check if this comment has been addressed?
> >
> > ``In the caption of Table 10, "EF5" should presumably read "EL5" ''
> >
> > Thanks.
> >
> > AC

---

> > > ### Author Response · Authors · 2023-01-18
> > > **It is EL5.**
> > >
> > > Thank you.
> > >
> > > "EF5" in the caption of Table 10 is "EL5".
> > >
> > > I re-submit the camera-review.